# Learning quadratic neural networks in high dimensions: SGD dynamics and scaling laws

**Gérard Ben Arous**[1], **Murat A. Erdogdu**[2,3], **Nuri Mert Vural**[2,3], **Denny Wu**[1,4]

[1]New York University, [2]University of Toronto, [3]Vector Institute, [4]Flatiron Institute
gba@cims.nyu.edu, {erdogdu,vural}@cs.toronto.edu, dennywu@nyu.edu

## Abstract

We study the optimization and sample complexity of gradient-based training of a two-layer neural network with quadratic activation function in the high-dimensional regime, where the data is generated as $f_*(\boldsymbol{x}) \propto \sum_{j=1}^{r} \lambda_j \sigma\left(\langle \boldsymbol{\theta_j}, \boldsymbol{x} \rangle\right), \boldsymbol{x} \sim \mathcal{N}(0, \boldsymbol{I}_d)$, $\sigma$ is the 2nd Hermite polynomial, and $\{\boldsymbol{\theta}_j\}_{j=1}^{r} \subset \mathbb{R}^d$ are orthonormal signal directions. We consider the extensive-width regime $r \asymp d^\beta$ for $\beta \in [0, 1)$, and assume a power-law decay on the (non-negative) second-layer coefficients $\lambda_j \asymp j^{-\alpha}$ for $\alpha \geq 0$. We present a sharp analysis of the SGD dynamics in the feature learning regime, for both the population limit and the finite-sample (online) discretization, and derive scaling laws for the prediction risk that highlight the power-law dependencies on the optimization time, sample size, and model width. Our analysis combines a precise characterization of the associated matrix Riccati differential equation with novel matrix monotonicity arguments to establish convergence guarantees for the infinite-dimensional effective dynamics.

## 1 Introduction

We study the problem of learning a two-layer neural network (NN) with quadratic activation on isotropic Gaussian data. The target function (or the "teacher" model) is defined as

$$y = f_*(\boldsymbol{x}) + \epsilon \text{ with } f_*(\boldsymbol{x}) = \tfrac{1}{\|\boldsymbol{\Lambda}\|_{\mathrm{F}}} \sum_{j=1}^{r} \lambda_j \sigma\left(\langle \boldsymbol{\theta_j}, \boldsymbol{x} \rangle\right) \text{ and } \boldsymbol{x} \sim \mathcal{N}(0, \boldsymbol{I}_d), \qquad (1.1)$$

where $\sigma(z) = z^2 - 1$ is the 2nd Hermite polynomial; $\epsilon$ is zero-mean, independent sub-Gaussian noise; $\{\boldsymbol{\theta}_j\}_{j=1}^{r} \subset \mathbb{R}^d$ are unknown signal directions (index features) which we assume to be orthonormal; $\lambda_1 > \lambda_2 > \cdots > \lambda_r > 0$ are their respective contributions; and $\boldsymbol{\Lambda} = \mathrm{diag}(\lambda_1, \cdots, \lambda_r)$ collects the second-layer coefficients. Our goal is to learn this target network using a "student" two-layer neural network with quadratic activation and $r_s$ neurons, trained via a gradient-based optimization algorithm. This setting encompasses several well-known problems:

- *Phase retrieval ($r = 1$).* The learning of one quadratic neuron has been studied extensively [Fie82, CC15, TV23]. The quadratic $\sigma$ has information exponent $k = 2$ (defined as the index of the lowest non-zero Hermite coefficient [DH18, BAGJ21]). This entails that randomly initialized parameters are close to a saddle point in high dimensions; hence the SGD dynamics exhibit a plateau ("search" phase) of length $\log d$ before the loss decreases sharply ("descent" phase).

- *Multi-spike PCA ($r = \Theta_d(1)$).* The target (1.1) is a subclass of Gaussian multi-index models, for which various algorithms have been proposed for the finite-rank case $r_s = \Theta_d(1)$ [CM20, DLS22, BBPV23]. The setting also closely relates to the multi-spike PCA problem, for which online SGD [AGP24] and other streaming algorithms has been studied [OK85, JJK+16, AZL17].

- *Linear-width quadratic NN ($r \asymp d$).* The regime where the teacher width $r$ grows proportionally with dimensionality $d$ has also been studied, typically in the well-conditioned setting (e.g., identical $\lambda_j$'s). Recent works characterized the landscape [SJL18, DL18, VBB19, GKZ19, GMMM19], optimization dynamics [MVEZ20, MBB23], and statistical efficiency [MTM+24, ETZK25, BCN+25].

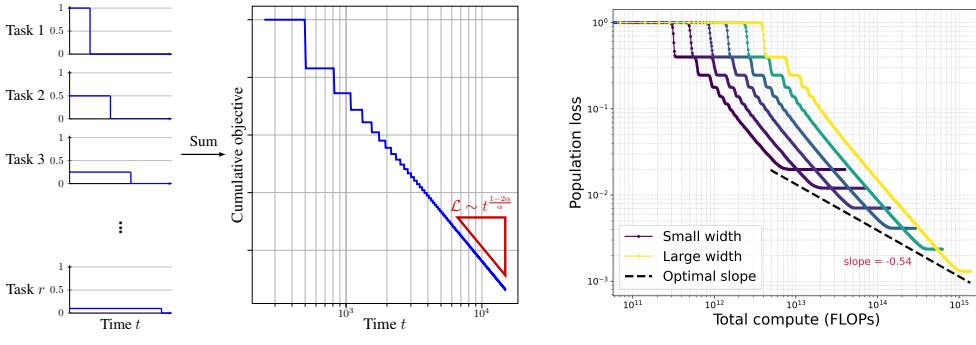

(a) Additive model hypothesis for scaling laws.  (b) SGD risk curves for quadratic NN.

Figure 1: $(a)$ Illustration of the additive model hypothesis, i.e., sum of emergent learning curves at different timescales yields a power law cumulative loss. $(b)$ Population loss vs. compute for two-layer quadratic NNs trained with online SGD with batch size $d$ on MSE loss. We set $d = 3200$, and for the teacher $r = 2400$, $\alpha = 1$.

In this work we focus on the "extensive-rank" regime where $r \asymp d^\beta$ for $\beta \in (0,1)$ and $r_s \asymp d^\gamma$ for $\gamma \in [0,1)$, and place a power-law assumption on the second-layer coefficients: $\lambda_j \asymp j^{-\alpha}$ for $\alpha \geq 0$. Our setting is motivated by the following lines of research.

**Neural scaling laws & emergence.**  Recent empirical studies on large language models (LLMs) reveal that increasing the model or training data size often results in a predictable, power-law decrease in the loss known as *neural scaling laws* [HNA+17, KMH+20, HBM+22]. While such scaling of generalization error has been derived for sketched linear models [MRS22, BAP24, PPXP24, LWK+24, DLM24], these analyses assume random projection with no *feature learning*, and hence cannot capture the NN's ability to learn useful features [GDDM14, DCLT18] that adapt to the underlying data structure. We aim to investigate a setting where the training of a nonlinear NN beyond the "lazy" regime exhibits a nontrivial scaling law.

Feature learning in neural networks is often studied theoretically through the learning of *multi-index models*, where the target function depends on a small number of latent directions (see [BH25] and references therein). For these low-dimensional targets, it is known that the training dynamics typically exhibit *emergent* (or staircase-like) behavior — long plateaus followed by sharp drops in loss [BAGJ22, AAM23]. To reconcile this emergent loss curve with smooth power-law decay, recent works hypothesized that the pretraining objective can be decomposed into a sum of losses on individual tasks [MLGT24, NFLL24], the learning of each exhibits a sharp transition, and the superposition of numerous emergent risk curves at different timescales yields a power-law scaling of the cumulative loss (see Figure 1(a)). In this context, the two-layer network (1.1) can be viewed as a sum of single-index phase retrieval tasks, where the length of each $\sim \log d$ plateau in the risk trajectory can be modulated by the second-layer coefficient $\lambda_j$. This motivates the following question:

> **Q1:** *Does gradient-based training of a two-layer quadratic network yield power-law loss scaling, when the target function is an additive model with varying second-layer coefficients $\{\lambda_j\}_{j=1}^r$?*

In Figure 1(b) we empirically observe the affirmative: when the target function has smoothly decaying second-layer weights, online SGD training yields a power-law risk curve that resembles the scaling laws in [KMH+20, HBM+22]. The goal of this work is to rigorously establish such scaling laws.

**Learning extensive-width neural networks.**  Prior works on multi-index models have shown that when $r = \Theta_d(1)$, gradient-based training succeeds with polynomial sample complexity depending on properties of the link function [AAM22, DLS22, BBSS22]. The "extensive-rank" regime where $r \asymp d^\beta$ for $\beta > 0$ is relatively under-explored (except for the linear width regime $r \asymp d$ [MBB23, MTM+24]); this setting is arguably closer to the practical neural network training (compared to the narrow-width setting), and also bears connections to several observations in the LLM literature such as *superposition* [EHO+22] and *skill localization* [DDH+21, WWZ+22, PSZA23], where the model simultaneously acquires a large number of "skills" during pretraining (see e.g., [OSSW24]).

The learning dynamics of (1.1) with divergingly many neurons is challenging to analyze primarily due to the fact that the effective dynamics may not be captured by a finite set of *summary statistics* [BAGJ22] (as in the finite-$r$ case). Recent works [OSSW24, SBH24] addressed this challenge by assuming that the activation $\sigma$ has information exponent $k \geq 3$, which allows the learning dynamics

| Algorithm | Decay rate ($\lambda_j$) | Risk scaling law | Result |
|---|---|---|---|
| Gradient flow | $\alpha > 0.5$ | $\bar{t}^{-\frac{2\alpha-1}{\alpha}} + r_s^{-(2\alpha-1)}$ | Theorem 1 |
| | $\alpha < 0.5$ | $(1 - \bar{t}^{\frac{1-2\alpha}{\alpha}})_+ + (1 - (r_s/r)^{1-2\alpha})_+$ | |
| Online SGD (Stiefel) | $\alpha > 0.5$ | $(\eta\bar{t})^{-\frac{2\alpha-1}{\alpha}} + r_s^{-(2\alpha-1)}$ | Theorem 2 |
| | $\alpha < 0.5$ | $(1 - (\eta\bar{t})^{\frac{1-2\alpha}{\alpha}})_+ + (1 - (r_s/r)^{1-2\alpha})_+$ | |

Table 1: Scaling laws for learning quadratic neural network (1.1) using population gradient flow and its online SGD discretization. We omit constant factors in the risk scaling for ease of presentation.

- In $\alpha > 0.5$, for population gradient flow, $\bar{t} \sim t \cdot \log d$ is the rescaled time; for online SGD, $\bar{t} \sim t \cdot \log d$ where $t$ is the number of gradient steps, which is equal to the sample size, and $\eta \sim 1/(d \operatorname{polylog}(d))$ is the step size.
- In $\alpha < 0.5$, for population gradient flow, $\bar{t} \sim t \cdot r \log d$ is the rescaled time; for online SGD, $\bar{t} \sim t \cdot r \log d$ where $t$ is the number of gradient steps and $\eta \sim 1/(dr^\alpha \operatorname{polylog}(d))$ is the step size.

to decouple across feature directions. However, the case $k \leq 2$, which includes the quadratic activation studied in this work, remained open: existing analyses either assumed "isotropic" feature contributions ($\lambda_1 = \lambda_r$) [RL24, SBH24], or established a computational complexity for SGD that scales with $d^{\Theta(\lambda_1/\lambda_r)}$ [LMZ20], which leads to pessimistic *exponential* dimension dependency in the power-law setting we consider. We therefore ask the following question.

> **Q2:** *Can we establish optimization and sample complexity of learning an extensive-width quadratic neural network* (1.1) *with anisotropic, power-decaying feature contributions?*

Finally, although our problem setup does not directly encompass commonly used activation functions such as ReLU, for SGD on multi-index models it is known that the Hermite-2 component is the first harmonic capturing multi-dimensional structure in the target function [DLS22, DKL$^+$23b]. Consequently, our quadratic setting represents the first nontrivial mode of feature learning in SGD dynamics. Consistent with this view, Figure 3 shows that the risk trajectory of ReLU networks closely follows our theoretical characterization of quadratic networks over a substantial portion of training.

## 1.1 Our Contributions

We analyze the risk trajectory of learning (1.1) with both gradient flow on the mean squared error (MSE) loss and its online SGD discretization on Stiefel manifold, covering the extensive-width and power-law settings. We derive scaling laws for feature recovery and population risk as a function of teacher and student network widths $r_s, r$, the decay exponent $\alpha$, the optimization time, and the sample size (for the discretized dynamics). Our contributions are summarized as follow (see Table 1).

1. In Section 3, we analyze the population gradient flow and tightly characterize the loss decay with respect to time and the student width $r_s$. We show that signal directions are recovered sequentially, and the population MSE follows a smooth power law specified by the decay rate $\alpha > 0$.

2. In Section 4, we consider the online stochastic gradient descent (SGD) dynamics on the Stiefel manifold and derive scaling laws with respect to sample size. When specializing to the isotropic setting $\alpha = 0$, our sample complexity improves upon [RL24] in the extensive-width setting and matches the information theoretic limit (in terms of $d, r$ dependence) up to polylogarithmic factors.

The following technical challenges in the extensive-width regime are central to our analysis:

- *Coupled population dynamics.* As $r, r_s \to \infty$, we must track infinitely many overlapping student and teacher neurons. [OSSW24, SBH24] assumed high information exponent $k > 2$, to decouple the dynamics into $r$ independent single-index models, but such property does not hold in our quadratic case ($k = 2$). We address this by leveraging the closed-form solution of the quadratic problem [MBB23], which satisfies a *Matrix Riccati ODE*. A key ingredient in our analysis is its *monotonicity with respect to its initialization*, illustrated in Figures 4(a), which enables sharp risk bounds via comparisons to decoupled models.

- *Operator norm discretization error.* Prior works [BAGJ21, BBPV23, AGP24] focused on finite-$r$ settings, where Frobenius norm control of the SGD noise was sufficient and natural: it allows bounding error direction-wise without incurring additional dimension dependence. However, in the extensive-width regime, such bounds become pessimistic and lead to suboptimal $r$-dependent rates. Hence we need to establish *operator norm* concentration around the population dynamics.

- *Matrix-monotone comparison framework.* To control discretization error in operator norm, we extend the monotonicity-based argument to discrete time and introduce a novel comparison-based discretization technique. Our approach constructs matrix-valued reference sequences corresponding to decoupled dynamics that tightly bound the discrete evolution from above and below. This yields sharp operator norm control even when the true trajectories are non-monotone (see Figure 4), as the analysis avoids relying on the trajectory itself by comparing against simpler bounding sequences.

## 1.2 Additional Related Works

**Learning multi-index models with SGD.** When $r = 1$, the target is a *single-index model* with quadratic link function. The SGD learning of single-index models has been extensively studied in the feature learning literature [BAGJ21, BES+22, MHPG+22, BES+23, MHWSE23, MLHD23, MZD+23, BMZ24, DNGL24, GWB25]; while this model has $d$ parameters to be estimated, the quadratic link (with information exponent $k = 2$) incurs an additional $\log d$ factor in the complexity of online SGD. More generally, the setting where $r = \Theta_d(1)$ is covered by recent analyses of *multi-index models* [AAM22, AAM23, BBPV23, DKL+23a, CWPPS23, AGP24, VE24, MHWE24]; however, these learning guarantees for multi-index models typically yield superpolynomial complexity when the target function is rank-extensive. The sample complexity of gradient-based learning is also connected to statistical query lower bounds [DPVLB24, DTA+24, LOSW24, ADK+24].

**Quadratic NNs and additive models.** Prior theoretical works on learning two-layer neural network with quadratic activation function have studied the loss landscape [SJL18, DL18, VBB19, GKZ19, GMMM19] and the optimization dynamics [MVEZ20, AKLS23, MBB23, RL24]. While existing optimization and statistical guarantees may cover the extensive-width regime (see e.g., [DL18, MBB23, RL24]), to our knowledge, precise scaling laws have not been established in our extensive-rank and power-law setting. (1.1) is also an instance of the *additive model* [Sto85, HT87, Bac17] where the individual functions are given as (orthogonal) single-index models with *unknown* index features. For this model, [OSSW24, SBH24] established learning guarantees in the well-conditioned regime, under the assumption that the link function $\sigma$ has information exponent $k > 2$.

## 2 Background and Problem Setting

### 2.1 Student-teacher Setting

**Teacher Network.** We consider the task of learning a teacher network with a quadratic (second-order Hermite) activation function written as

$$y = f_*(\boldsymbol{x}) + \epsilon \ \text{ with } \ f_*(\boldsymbol{x}) := \frac{1}{\|\boldsymbol{\Lambda}\|_{\mathrm{F}}} \sum_{j=1}^{r} \lambda_j \big( \langle \boldsymbol{\theta}_j, \boldsymbol{x} \rangle^2 - 1 \big) \text{ and } \boldsymbol{x} \sim \mathcal{N}(0, \boldsymbol{I}_d), \qquad (2.1)$$

where $\boldsymbol{x} \in \mathbb{R}^d$ is the input; $\epsilon$ is zero-mean, bounded-variance sub-Gaussian noise; $r$ is the teacher network width; and $\{\boldsymbol{\theta}_j\}_{j=1}^{r} \subset \mathbb{R}^d$ is an orthonormal set of unknown signal vectors. We collect these as columns of the matrix $\boldsymbol{\Theta} \in \mathbb{R}^{d \times r}$. The contributions of these vectors are determined by the unknown second-layer coefficients $\lambda_1 > \lambda_2 > \cdots > \lambda_r > 0$ with a power-law decay $\lambda_j \asymp j^{-\alpha}$ for $\alpha \geq 0$, and $\boldsymbol{\Lambda}$ is a diagonal matrix whose $j$-th diagonal entry is $\lambda_j$. The normalization in front of summation ensures $\mathbb{E}[y^2]$ is constant. We focus on the regime where $r \asymp d^{\beta}$ for $\beta \in (0, 1)$.

**Remark 1.** *The orthogonality of $\{\boldsymbol{\theta}_j\}_{j=1}^{r}$ can be assumed without loss of generality. Specifically, consider (2.1) with arbitrary first-layer weights $\boldsymbol{\Theta}$ and normalization $\mathbb{E}[y] = 0$, the output can be written as $y \propto \mathrm{Tr}(\boldsymbol{x}\boldsymbol{x}^\top \boldsymbol{\Theta} \boldsymbol{\Lambda} \boldsymbol{\Theta}^\top) + \text{constant} + \text{noise}$; hence we may redefine $(\lambda_j, \boldsymbol{\theta}_j)$ via the spectral decomposition.*

**Student Network.** We learn the target model with a quadratic student network defined as

$$\hat{y}(\boldsymbol{x}, \boldsymbol{W}) = \frac{1}{\sqrt{r_s}} \sum_{j=1}^{r_s} \langle \boldsymbol{w}_j, \boldsymbol{x} \rangle^2 - \|\boldsymbol{w}_j\|_2^2, \qquad (2.2)$$

where $r_s$ is the width of the student network, and $\{\boldsymbol{w}_j\}_{j=1}^{r_s} \subset \mathbb{R}^d$ denotes the set of trainable weights. We collect these weights as the columns of the matrix $\boldsymbol{W} \in \mathbb{R}^{d \times r_s}$, and omit the dependence on $\boldsymbol{x}$

in $\hat{y}(\boldsymbol{x}, \boldsymbol{W})$ when clear from the context. Note that the norm subtraction ensures $\mathbb{E}_{\boldsymbol{x}}[\hat{y}(\boldsymbol{x}, \boldsymbol{W})] = 0$. We may equivalently write the student network as $\hat{y}(\boldsymbol{x}, \boldsymbol{W}) = \frac{1}{\sqrt{r_s}} \sum_{j=1}^{r_s} \|\boldsymbol{w}_j\|_2^2 \cdot (\langle \bar{\boldsymbol{w}}_j, \boldsymbol{x} \rangle^2 - 1)$ where $\bar{\boldsymbol{w}}_j$ is unit-norm; since our student does not have trainable second-layer, the norm component $\|\boldsymbol{w}_j\|_2^2$ allows the model to adapt to the target second-layer $\lambda_j$; this homogeneous parameterization has been studied in prior works [CB20, GRWZ21].

## 2.2 Training Objective

Training constitutes to minimizing the squared loss; we define the instantaneous loss on $(\boldsymbol{x}, y)$ as

$$\mathcal{L}(\boldsymbol{W}; (\boldsymbol{x}, y)) := \tfrac{1}{16} (y - \hat{y}(\boldsymbol{x}, \boldsymbol{W}))^2,$$

where the prefactor is included for notational convenience in the gradient computation. We omit the dependence on $(\boldsymbol{x}, y)$ when clear from context. The population risk can be written as

$$R(\boldsymbol{W}) := \mathbb{E}_{(\boldsymbol{x}, y)}[\mathcal{L}(\boldsymbol{W})] = \tfrac{1}{8} \left\| \tfrac{1}{\sqrt{r_s}} \boldsymbol{W}\boldsymbol{W}^\top - \tfrac{1}{\|\boldsymbol{\Lambda}\|_\mathrm{F}} \boldsymbol{\Theta}\boldsymbol{\Lambda}\boldsymbol{\Theta}^\top \right\|_\mathrm{F}^2 + \tfrac{1}{16} \mathbb{E}[\epsilon^2]. \tag{2.3}$$

**Alignment.** Observe that the student network is invariant to right-multiplication of its weight matrix by an orthonormal matrix, i.e., $\hat{y}(\boldsymbol{x}, \boldsymbol{W}) = \hat{y}(\boldsymbol{x}, \boldsymbol{W}\boldsymbol{O})$ for any $\boldsymbol{O} \in \mathbb{R}^{r_s \times r_s}$ with $\boldsymbol{O}^\top \boldsymbol{O} = \boldsymbol{I}$. Consequently, any notion of alignment that depends on individual directions in $\boldsymbol{W}$ may not be informative. To capture directional learning in a way that respects this symmetry, we define alignment in terms of the subspace spanned by student weights. We formalize this using the polar decomposition:

$$\boldsymbol{W} := \boldsymbol{U}\boldsymbol{Q}^{1/2}, \quad \text{where} \quad \boldsymbol{Q} := \boldsymbol{W}^\top \boldsymbol{W} \quad \text{and} \quad \boldsymbol{U}^\top \boldsymbol{U} = \boldsymbol{I}_{r_s}. \tag{2.4}$$

Here, $\boldsymbol{Q}$ denotes the radial component of the student weights, while $\boldsymbol{U}$ is an orthonormal matrix that encodes their directional component. We quantify the alignment between the student network and the $j$th teacher feature by the squared norm of the projection of $\boldsymbol{\theta}_j$ onto the column space of $\boldsymbol{W}$:

$$\mathrm{Alignment}(\boldsymbol{W}, \boldsymbol{\theta}_j) := \|\boldsymbol{U}^\top \boldsymbol{\theta}_j\|_2^2. \tag{2.5}$$

$\mathrm{Alignment}(\boldsymbol{W}, \boldsymbol{\theta}_j)$ takes values in the interval $[0, 1]$; it is 0 if $\boldsymbol{\theta}_j$ is orthogonal to $\boldsymbol{W}$ (no alignment), while it is 1 if $\boldsymbol{\theta}_j$ is in the column space of $\boldsymbol{W}$ (perfect alignment)[1].

# 3 Continuous Dynamics: Population Gradient Flow

We first analyze the continuous-time population gradient flow dynamics for (2.3), given as

$$\partial_t \boldsymbol{W}_t = -\nabla R(\boldsymbol{W}_t), \quad \text{where } \boldsymbol{W}_0 \in \mathbb{R}^{d \times r_s}, \ \boldsymbol{W}_{0,ij} \sim_{iid} \mathcal{N}(0, 1/d), \tag{GF}$$

and the population gradient reads

$$\nabla R(\boldsymbol{W}_t) = -\frac{1}{2\sqrt{r_s}\|\boldsymbol{\Lambda}\|_\mathrm{F}} \left( \boldsymbol{\Theta}\boldsymbol{\Lambda}\boldsymbol{\Theta}^\top - \frac{\|\boldsymbol{\Lambda}\|_\mathrm{F}}{\sqrt{r_s}} \boldsymbol{W}_t \boldsymbol{W}_t^\top \right) \boldsymbol{W}_t.$$

For notational convenience, we define

$$\mathcal{R}(t) := \left\| \tfrac{1}{\sqrt{r_s}} \boldsymbol{W}_t \boldsymbol{W}_t^\top - \tfrac{1}{\|\boldsymbol{\Lambda}\|_\mathrm{F}} \boldsymbol{\Theta}\boldsymbol{\Lambda}\boldsymbol{\Theta}^\top \right\|_\mathrm{F}^2 \quad \text{and} \quad \mathcal{A}(t, \boldsymbol{\theta}_j) := \mathrm{Alignment}(\boldsymbol{W}_t, \boldsymbol{\theta}_j).$$

The following theorem sharply characterizes the timescale for alignment and the limiting risk curve.

**Theorem 1.** *Let* $\lambda_j = j^{-\alpha}$ *and* $r \asymp d^\beta$ *for some* $\alpha \geq 0$ *and* $\beta \in (0, 1)$. *Consider the regime*

$$\begin{cases} \frac{r_s}{r} \to \varphi \in (0, \infty) & \text{and } d \geq \Omega_{\alpha, \beta, \varphi}(1), & \text{if } \alpha \in [0, 0.5), \\ r_s \asymp 1, & \text{and } d \geq \Omega_{\alpha, r_s}(1), & \text{if } \alpha > 0.5. \end{cases} \tag{3.1}$$

*Define the effective student width and effective timescale as*

$$r_{\mathrm{eff}} := \begin{cases} \lfloor r_s(1 - \log^{-1/8} d) \wedge r \rfloor, & \text{if } \alpha \in [0, 0.5) \\ r_s, & \text{if } \alpha > 0.5. \end{cases} \quad \text{and} \quad \mathsf{T}_{\mathrm{eff}} := \sqrt{r_s}\|\boldsymbol{\Lambda}\|_\mathrm{F} \log{}^{d/r_s}.$$

*Then, the population* (GF) *dynamics satisfy the following with probability* $1 - o(1/d^2) - \Omega(1/r_s^2)$:

---

[1] The definition in (2.5) may fail to converge to 1 when $\alpha = 0$ and $r_s < r$, due to rotational symmetry in the teacher network. In this case, a more suitable notion of alignment can be defined using the principal angles between the subspaces spanned by $\boldsymbol{W}$ and $\boldsymbol{\Theta}$, which provides a rotation-invariant characterization of directional overlap. Specifically, for $\alpha = 0$, we define $\mathrm{Alignment}(\boldsymbol{W}, \boldsymbol{\theta}_j)$ as the $j$th largest eigenvalue of $\boldsymbol{\Theta}^\top \boldsymbol{U}\boldsymbol{U}^\top \boldsymbol{\Theta}$.

1. **Alignment:** *For $j \leq r_{\text{eff}}$ and $t > 0$ satisfying $t \asymp r^\alpha$ when $\alpha \in [0, 0.5)$ and $t \asymp 1$ when $\alpha > 0.5$,*

$$\mathcal{A}(t\mathsf{T}_{\text{eff}}, \boldsymbol{\theta}_j) = \mathbb{1}\{t \geq \tfrac{1}{\lambda_j}\} + o_d(1). \tag{3.2}$$

2. **Risk curve:** *Under the same time scaling,*

$$\mathcal{R}(t\mathsf{T}_{\text{eff}}) = 1 - \frac{1}{\|\boldsymbol{\Lambda}\|_{\text{F}}^2} \sum_{j=1}^{r_{\text{eff}}} \lambda_j^2 \mathbb{1}\{t \geq \tfrac{1}{\lambda_j}\} + o_d(1). \tag{3.3}$$

**Remark 2.** *We make the following remarks about our result in Theorem 1:*

- *The spectral decay $\alpha$ determines both the choice of student width $r_s$ and the learning timescale in Theorem 1. Specifically, when $\alpha > 1/2$ (i.e., light-tailed regime), the coefficients $\{\lambda_j\}_{j=1}^r$ are square-summable, making the teacher model effectively finite-dimensional. Hence only finitely many directions need to be learned to achieve small loss, and a timescale of order $\log d$ and finite-width student suffices. In contrast, for the heavy-tailed regime $\alpha < 1/2$, we need to recover linear-in-$r$ directions, which require proportional student width $r_s/r \to \varphi$ and a longer timescale $r \log d$. This difference will be made explicit in Corollary 1.*

- *Theorem 1 verifies the additive model hypothesis [MLGT24] for quadratic neural networks in the feature learning regime; specifically, (3.2) identifies sharp transition time in alignment between student weights and the $j$-th teacher direction, and (3.3) suggests that the cumulative loss can be decomposed into individual emergent risk curves where the timescale is decided by the signal strength $\lambda_j$.*

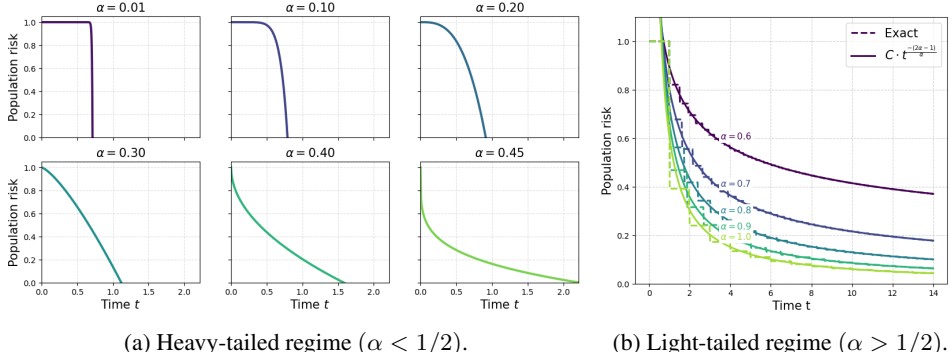

(a) Heavy-tailed regime ($\alpha < 1/2$).      (b) Light-tailed regime ($\alpha > 1/2$).

Figure 2: Illustration of the limiting risk trajectories and scaling behavior given in Corollary 1.

**Neural scaling laws.** As a corollary of Theorem 1, we obtain the following risk characterization.

**Corollary 1.** *By Theorem 1, the asymptotic risk of (GF) is given as follows:*

- *Heavy-tailed regime ($\alpha \in [0, 0.5)$): Almost surely, for all $t > 0$*

$$\mathcal{R}(tr \log d) \xrightarrow{d \to \infty} \left(1 - Ct^{\frac{1-2\alpha}{\alpha}}\right)_+ \vee \left(1 - \varphi^{1-2\alpha}\right)_+.$$

- *Light-tailed regime ($\alpha > 0.5$): With probability $1 - \Omega(1/r_s)$, for all $t > 0$, the risk $\mathcal{R}(t \log d)$ converges as $d \to \infty$ to a deterministic limit satisfying*

$$\mathcal{R}(t \log d) \xrightarrow{d \to \infty} \Theta\left(t^{-\frac{2\alpha-1}{\alpha}} + r_s^{-(2\alpha-1)}\right).$$

Corollary 1 shows that, over appropriate timescales, the cumulative effect of these emergent transitions yields a smoothly decaying risk curve. Intuitively speaking, the power-law exponent arises from the Riemann integral approximation of the infinite sum (3.3) – see Appendix D.5 for details.

The asymptotic risk behavior in Corollary 1 is visualized in Figure 2 (see also Figure 1(b) for empirical simulation). The figure illustrates how the sharp, step-like emergent curve at $\alpha = 0$ (as observed in earlier works on multi-index learning [BAGJ21, AAM23]) gradually transitions into a smooth curve as $\alpha$ increases. Notably, in the light-tailed regime $\alpha > 1/2$, our risk curve resembles the neural scaling laws in [KMH+20, HBM+22] which takes the form of $\mathcal{R} \sim 1/(\text{Data size})^a + 1/(\text{Model size})^b$, where the data size can be connected to optimization time under the one-pass discretization, which we analyze in the ensuing section.

# 4 Discrete Dynamics: Online Stochastic Gradient Descent

Now we analyze the finite-sample, discrete-time counterpart of the population dynamics (GF) and establish computational and statistical guarantees. We first discretize the directional component of the dynamics via online SGD with Stiefel constraint (see Proposition 2), and then introduce a fine-tuning step with negligible statistical and computational cost to fit the radial component; this mirrors the layer-wise training paradigm commonly used in theoretical analyses of gradient-based feature learning [AAM22, DLS22, BES+22, BEG+22]. The procedure is summarized in Algorithm 1.

---

**Algorithm 1** Online Stochastic Gradient Descent (Stiefel)

---

1: **for** $t = 1, 2, \ldots$ **do**
2: $\quad \widetilde{W}_t = W_{t-1} - \eta \nabla_{\mathrm{St}} \mathcal{L}(W_{t-1})$
3: $\quad W_t = \widetilde{W}_t \left( \widetilde{W}_t^\top \widetilde{W}_t \right)^{-1/2}$ $\qquad\qquad\qquad\qquad\qquad$ ▷ Feature learning
4: **end for**
5: $W_t^{\mathrm{final}} = W_t \Omega_*$ where $\Omega_* = \underset{\Omega \in \mathbb{R}^{r_s \times r_s}}{\arg\min} \sum_{j=1}^{N_{\mathrm{Ft}}} \mathcal{L}\big(W_t \Omega; (x_{t+j}, y_{t+j})\big)$ $\qquad$ ▷ Fine-tuning

---

In the feature learning step, we update the first-layer weights $W_t$ to recover the subspace spanned by the teacher directions. To this end, we use online SGD on Stiefel manifold [AGP24] with polar retraction. The Riemannian gradient on the Stiefel manifold is given by:

$$\nabla_{\mathrm{St}} \mathcal{L}(W_{t-1}) := \nabla \mathcal{L}(W_{t-1}) - \tfrac{1}{2} W_{t-1} \left( W_{t-1}^\top \nabla \mathcal{L}(W_{t-1}) + \nabla \mathcal{L}(W_{t-1})^\top W_{t-1} \right),$$

where the instantaneous loss is defined for sample $(x_t, y_t)$. Since the goal is to ensure subspace alignment (2.5), the overlap of individual student-teacher weights is not relevant during this phase.

After the feature learning phase, we perform a fine-tuning step to rotate $W_t$ so that each $w_j$ aligns with the corresponding teacher direction $\theta_j$. This is achieved by solving an empirical risk minimization problem over $N_{\mathrm{Ft}}$ fresh samples. The optimal fine-tuning matrix $\Omega_*$ admits a closed-form solution that is also numerically easy to compute. Importantly, the computational and statistical complexity of this step scales only quadratically with the student width $r_s$, which is negligible compared to the cost of feature learning. The complexity analysis for this phase is provided in Appendix E.

**Remark 3.** *Recall that the stage-wise training procedure is not required in our continuous analysis in Section 3. This is because we employ a Stiefel gradient similar to [BBPV23, AGP24] – which cannot fit the radial component – to simplify the discretization analysis. We conjecture that standard Euclidean discretization can achieve the same risk scaling; see Figure 1(b) for empirical evidence.*

We define the population risk of the output of Algorithm 1, the alignment with a teacher direction $\theta_j$, and the optimal risk achievable by a student neural network with width $r_s$ respectively as

$$\mathcal{R}(t) := R(W_t^{\mathrm{final}}), \quad \mathcal{A}(t, \theta_j) := \mathrm{Alignment}(W_t, \theta_j), \quad \mathcal{R}_{\mathrm{opt}} := \tfrac{1}{\|\Lambda\|_{\mathrm{F}}^2} \sum_{j=(r_s \wedge r)+1}^r \lambda_j^2.$$

Intuitively, $\mathcal{R}_{\mathrm{opt}}$ is the risk achieved by exactly fitting the top $r_s \leq r$ components of the teacher model. Note that the alignment $\mathcal{A}(t, \theta_j)$ depends only on the directional component of $W_t$; thus, this quantity remains unchanged during fine-tuning. The following theorem characterizes the alignment and risk curve for the discrete-time Algorithm 1.

**Theorem 2.** *Let the parameters $\{\lambda_j\}_{j=1}^r$, $r, r_s$, $r_{\mathrm{eff}}$ and $\mathsf{T}_{\mathrm{eff}}$, and the scaling regime (3.1) be as in Theorem 1. Suppose the student weights are initialized uniformly on the Stiefel manifold, and that the step size $\eta$ and fine-tuning sample size $N_{\mathrm{Ft}}$ satisfy*

$$\eta \asymp \frac{1}{d} \begin{cases} \frac{1}{r^\alpha \log^{C_\alpha}(1+d/r_s)}, & \alpha \in [0, 0.5) \\ \frac{1}{\log^{C_\alpha} d}, & \alpha > 0.5 \end{cases} \quad \text{and} \quad N_{Ft} \asymp r_s^2 \log^5 d,$$

*for some constant $C_\alpha > 0$ depending only on $\alpha$. Then with probability $1 - o_d(1/d^2) - \Omega(1/r_s^2)$,*

1. **Runtime and sample complexity:** *If*

$$T \geq \begin{cases} dr^{1+\alpha} \log^{C_\alpha+1}(1+d/r_s), & \alpha \in [0, 0.5) \\ d \log^{C_\alpha+1} d, & \alpha > 0.5. \end{cases} \tag{4.1}$$

*we have $\mathcal{R}(T) = \mathcal{R}_{\mathrm{opt}} + o_d(1)$.*

2. **Alignment & Risk curve:** *For $t > 0$ satisfying $t \asymp r^\alpha/\eta$ for $\alpha \in [0, 0.5)$ and $t \asymp 1/\eta$ for $\alpha > 0.5$,*

$$\bullet \; \mathcal{A}(t\mathsf{T}_{\mathrm{eff}}, \boldsymbol{\theta}_j) = \mathbb{1}\{\eta t \geq \tfrac{1}{\lambda_j}\} + o_d(1). \quad \bullet \; \mathcal{R}(t\mathsf{T}_{\mathrm{eff}}) = 1 - \frac{1}{\|\boldsymbol{\Lambda}\|_{\mathrm{F}}^2} \sum_{j=1}^{r_{\mathrm{eff}}} \lambda_j^2 \, \mathbb{1}\{\eta t \geq \tfrac{1}{\lambda_j}\} + o_d(1).$$

**Remark 4.** *We make the following remarks on the sample complexity.*

- *The bound in* (4.1) *implies a complexity of $n \asymp T \simeq dr^{1+\alpha} \operatorname{polylog}(1 + d/r_s)$ in the heavy-tailed case, and $T \simeq d \operatorname{polylog}(d)$ in the light-tailed case. Note that due to the one-pass nature of the algorithm, the runtime and sample complexity are identical (up to the negligible fine-tuning step).*

- *In the light-tailed regime $(\alpha > 1/2)$, the required sample size $n \simeq d \operatorname{polylog}(d)$ is information theoretically optimal up to logarithmic factors. Note that kernel methods and neural networks in the lazy regime [JGH18, COB19] requires $n \gtrsim d^2$ samples to learn a quadratic target function; thus our sample complexity bound illustrates the benefit of feature learning.*

- *In the heavy-tailed regime $(\alpha < 1/2)$, we obtain (nearly) information theoretically optimal sample complexity when $\alpha = 0$. For the intermediate regime $\alpha \in (0, 1/2)$, we conjecture that the optimal sample complexity is $T \simeq dr$, which implies our current bound is suboptimal by $r^\alpha$.*

**Isotropic Setting** $(\alpha = 0)$. In the isotropic case, where the goal is to estimate the $r$-dimensional subspace spanned by the teacher weights, the above theorem yields a sample and runtime complexity $n \asymp T \asymp dr \operatorname{polylog}(1 + d/r_s)$. This interpolates between the $n \simeq d \operatorname{polylog}(d)$ rate for phase retrieval $r = 1$ [TV23, BAGJ21], and $n \simeq d^2$ as $r \to d$, which matches the sample complexity in the linear-width regime [MTM$^+$24, ETZK25]. Notably, our $r$-dependence improves upon the recent work of [RL24], which established a sufficient sample size of $n \gtrsim d \operatorname{poly}(r)$ for a similar quadratic setting. We expect our result to be optimal up to polylogarithmic factors due to the intrinsic $dr$-dimensional nature of the subspace recovery problem.

**Scaling laws in discrete time.** As indicated by the alignment and risk expressions in Theorem 2, a sufficiently small learning rate $\eta$ ensures that running online SGD for $t$ steps closely tracks the population gradient flow trajectory (GF) at time $\eta t$, exhibiting the same scaling behavior. The following corollary formalizes the discrete-time counterpart of Corollary 1.

**Corollary 2.** *We consider $\eta t \xrightarrow{d \to \infty} t_{\mathrm{c}} > 0$. By Theorem 2, we have*

- *Heavy-tailed case $(\alpha \in [0, 0.5))$: Almost surely,*

$$\mathcal{R}(tr \log d) \xrightarrow{d \to \infty} \left(1 - C t_c^{\frac{1-2\alpha}{\alpha}}\right) \vee (1 - \varphi^{1-2\alpha})_+.$$

- *Light-tailed case $(\alpha > 0.5)$: With probability $1 - \Omega(1/r_s)$, $\mathcal{R}(t \log d)$ has an asymptotic limit*

$$\mathcal{R}(t \log d) \xrightarrow{d \to \infty} \Theta\left(t_c^{-\frac{2\alpha-1}{\alpha}} + r_s^{-(2\alpha-1)}\right).$$

## 5 Overview of Proof Techniques

To avoid notational confusion between discrete-time and continuous-time dynamics, we adopt the following convention throughout this section. Subscripts (e.g., $\boldsymbol{W}_t$) denote discrete-time quantities, while parentheses (e.g., $\boldsymbol{W}(t)$) denote continuous-time trajectories. Specifically, $\{\boldsymbol{W}_t\}_{t \in \mathbb{N}}$ refers to the iterates of online SGD; $\{\boldsymbol{W}(t)\}_{t \geq 0}$ denotes the continuous-time gradient flow governed by (GF). Since our proof strategy heavily relies on the matrix (Loewner) order for symmetric matrices, we introduce the following notations. For symmetric matrices $\boldsymbol{G}_1, \boldsymbol{G}_2 \in \mathbb{R}^{d \times d}$, we write $\boldsymbol{G}_1 \prec \boldsymbol{G}_2$ if $\boldsymbol{G}_2 - \boldsymbol{G}_1$ is positive definite . The reverse relations are denoted by $\boldsymbol{G}_1 \succ \boldsymbol{G}_2$ and $\boldsymbol{G}_1 \succeq \boldsymbol{G}_2$.

### 5.1 Proof Sketch of Theorem 1

We first observe that both the population risk $R(\boldsymbol{W}(t))$ and the alignment $\mathrm{Alignment}(\boldsymbol{W}(t), \boldsymbol{\theta}_j)$ depend on $\boldsymbol{W}(t)$ through two Gram matrices: the weight Gram matrix $\boldsymbol{G}_W(t) := \boldsymbol{W}(t)\boldsymbol{W}(t)^\top$, and the alignment Gram matrix $\boldsymbol{G}_U(t) := \boldsymbol{\Theta}^\top \boldsymbol{U}(t)\boldsymbol{U}(t)^\top \boldsymbol{\Theta}$, where $\{\boldsymbol{U}(t)\}_{t \geq 0}$ denotes the directional component of $\boldsymbol{W}(t)$, as defined in (2.4). The proof proceeds by analyzing the evolution of these matrices, each governed by an autonomous ODE; in particular, a matrix Riccati differential equation.

**Proposition 1.** *The Gram matrices defined above satisfy the following matrix Riccati ODEs:*

- **Weight Gram matrix:** $\partial_t \boldsymbol{G}_W(t) = \frac{0.5}{\|\boldsymbol{\Lambda}\|_{\mathrm{F}}\sqrt{r_s}}\left(\boldsymbol{\Theta}\boldsymbol{\Lambda}\boldsymbol{\Theta}^\top \boldsymbol{G}_W(t) + \boldsymbol{G}_W(t)\boldsymbol{\Theta}\boldsymbol{\Lambda}\boldsymbol{\Theta}^\top - \frac{2\|\boldsymbol{\Lambda}\|_{\mathrm{F}}}{\sqrt{r_s}}\boldsymbol{G}_W^2(t)\right).$

- **Alignment Gram matrix:** $\partial_t \boldsymbol{G}_U(t) = \frac{0.5}{\|\boldsymbol{\Lambda}\|_{\mathrm{F}}\sqrt{r_s}}\left(\boldsymbol{\Lambda}\boldsymbol{G}_U(t) + \boldsymbol{G}_U(t)\boldsymbol{\Lambda} - 2\boldsymbol{G}_U(t)\boldsymbol{\Lambda}\boldsymbol{G}_U(t)\right).$

Both equations in Proposition 1 take the form of matrix Riccati ODEs [BLW91], whose structural properties play a central role in the proof. To illustrate the core idea, we focus on the alignment dynamics. For simplicity, we write $\boldsymbol{G}(t) := \boldsymbol{G}_U(t)$ and consider

$$\partial_t \boldsymbol{G}(t) = \frac{0.5}{\|\boldsymbol{\Lambda}\|_{\mathrm{F}}\sqrt{r_s}}\left(\boldsymbol{\Lambda}\boldsymbol{G}(t) + \boldsymbol{G}(t)\boldsymbol{\Lambda} - 2\boldsymbol{G}(t)\boldsymbol{\Lambda}\boldsymbol{G}(t)\right). \tag{5.1}$$

Note that $\mathrm{Alignment}\left(\boldsymbol{W}(t), \boldsymbol{\theta}_j\right)$ corresponds to the $j^{\text{th}}$ diagonal entry of $\boldsymbol{G}(t)$. To characterize its trajectory, we leverage the monotonicity of the matrix Riccati flow with respect to its initialization, i.e., if $\boldsymbol{G}_0^+ \succeq \boldsymbol{G}_0^-$, the corresponding solutions satisfy $\boldsymbol{G}(t, \boldsymbol{G}_0^+) \succeq \boldsymbol{G}(t, \boldsymbol{G}_0^-)$ for all $t \geq 0$, where $\boldsymbol{G}(t, \boldsymbol{G}_0)$ denotes the solution to (5.1) with initial condition $\boldsymbol{G}_0$. Our proof strategy builds on this monotonicity and proceeds as follows:

1. **Diagonalization & decoupling.** If $\boldsymbol{G}_0$ is diagonal, the solution $\{\boldsymbol{G}(t)\}_{t \geq 0}$ remains diagonal under (5.1), reducing the dynamics to independent scalar ODEs that govern each diagonal entry. Moreover, each scalar ODE admits a closed-form solution.

2. **Asymptotic characterization.** For general $\boldsymbol{G}_0$, we construct diagonal matrices $\boldsymbol{G}_0^+ \succeq \boldsymbol{G}_0 \succeq \boldsymbol{G}_0^-$. By monotonicity, the corresponding trajectories upper and lower bound $\{\boldsymbol{G}(t)\}_{t \geq 0}$. These bounding systems are diagonal and decoupled, and as $d \to \infty$, they converge to the same limit.

We apply this strategy in Appendix D.3 to derive the exact asymptotics stated in Theorem 1.

**Remark 5.** *We note that while the Riccati flow is monotone with respect to its initialization, this does not imply that its solution is monotone in time. That is, the trajectory $\boldsymbol{G}(t)$ may not evolve monotonically in matrix order, even though a larger initialization yields a trajectory that remains above that of a smaller one for all $t \geq 0$. This distinction is illustrated in Figure 4.*

### 5.2 Proof Sketch of Theorem 2

**Extending Monotonicity Arguments to Discrete Dynamics**

We begin by observing that online SGD on the Stiefel manifold approximates the directional component of the continuous-time gradient flow, with stochastic gradients arising from online sampling. The proposition below formalizes the idea that online SGD approximates the directional dynamics of the continuous gradient flow at the population level. For the statement, recall that $\boldsymbol{U}(t)$ denotes the directional component of the gradient flow solution $\boldsymbol{W}(t)$ from (GF), as defined in (2.4).

**Proposition 2.** *Let $\widehat{\boldsymbol{W}}(t)$ be the solution to the continuous-time gradient flow on the Stiefel manifold, initialized with $\boldsymbol{U}(0)$. Then for all $t \geq 0$, the column spaces of $\widehat{\boldsymbol{W}}(t)$ and $\boldsymbol{U}(t)$ coincide.*

This result justifies studying the online SGD on Stiefel manifold via the directional dynamics of (GF). To this end, we introduce the discrete analog of $\boldsymbol{G}(t)$ above as $\boldsymbol{G}_t = \boldsymbol{\Theta}^\top \boldsymbol{W}_t \boldsymbol{W}_t^\top \boldsymbol{\Theta}$. Extending the analysis to discrete time is non-trivial due to the *loss of monotonicity* in the Euler discretization of the Riccati dynamics (5.1). In particular, the update

$$\boldsymbol{G}_t = \underbrace{\boldsymbol{G}_{t-1} + \frac{0.5\eta}{\|\boldsymbol{\Lambda}\|_{\mathrm{F}}\sqrt{r_s}}\left(\boldsymbol{\Lambda}\boldsymbol{G}_{t-1} + \boldsymbol{G}_{t-1}\boldsymbol{\Lambda} - 2\boldsymbol{G}_{t-1}\boldsymbol{\Lambda}\boldsymbol{G}_{t-1}\right)}_{\text{non-monotone dynamics}} + (\text{2nd-order terms and noise}) \tag{5.2}$$

no longer preserves the matrix order structure crucial to the continuous-time argument.

To overcome this, we construct an auxiliary discrete system that approximates (5.2) up to second-order terms while preserving monotonicity. Specifically, we define the map

$$\boldsymbol{G}(\boldsymbol{G}_t, \eta) := \boldsymbol{G}_t - \frac{\eta}{2}(2\boldsymbol{G}_t - \boldsymbol{I}_r)\boldsymbol{\Lambda}(2\boldsymbol{G}_t - \boldsymbol{I}_r)\left(\boldsymbol{I}_r + \eta\boldsymbol{\Lambda}(2\boldsymbol{G}_t - \boldsymbol{I}_r)\right)^{-1} + \eta\boldsymbol{\Lambda} \tag{5.3}$$

which matches (5.2) up to second-order terms. Indeed, expanding the inverse term gives

$$\boldsymbol{G}(\boldsymbol{G}_t, \eta) = \underbrace{\boldsymbol{G}_t - \frac{\eta}{2}(2\boldsymbol{G}_t - \boldsymbol{I}_r)\boldsymbol{\Lambda}(2\boldsymbol{G}_t - \boldsymbol{I}_r) + \eta\boldsymbol{\Lambda}}_{=\boldsymbol{G}_t + \eta(\boldsymbol{\Lambda}\boldsymbol{G}_t + \boldsymbol{G}_t\boldsymbol{\Lambda} - 2\boldsymbol{G}_t\boldsymbol{\Lambda}\boldsymbol{G}_t)} + \text{2nd-order terms.}$$

The key advantage of the iteration (5.3) is that it preserves matrix order:

**Proposition 3.** *For $\eta > 0$, if $\boldsymbol{G}_t^+ \succeq \boldsymbol{G}_t^- \succeq 0$, we have $\boldsymbol{G}(\boldsymbol{G}_t^+, \eta) \succeq \boldsymbol{G}(\boldsymbol{G}_t^-, \eta)$.*

We use this to bound the non-monotone dynamics (5.2) via monotone iterates. Roughly, we show that for small enough step size $\eta$, the following holds:

$$\boldsymbol{G}\big(\boldsymbol{G}_{t-1}, (1+\varepsilon)\eta\big) + \text{Noise} \succeq \boldsymbol{G}_t \succeq \boldsymbol{G}\big(\boldsymbol{G}_{t-1}, (1-\varepsilon)\eta\big) + \text{Noise}$$

for some $\varepsilon = o_d(1)$, where we denote the effective learning rate in (5.2) with $\eta = \frac{\eta}{\sqrt{r_s}\|\boldsymbol{\Lambda}\|_F}$. We then follow the same bounding argument used in the continuous case by defining

$$\boldsymbol{G}_t^\pm = \boldsymbol{G}\big(\boldsymbol{G}_{t-1}^\pm, (1 \pm \varepsilon)\eta\big) + \text{Noise}, \quad \text{where} \quad \boldsymbol{G}_0^+ \succeq \boldsymbol{G}_0 \succeq \boldsymbol{G}_0^- \succeq 0,$$

and show that $\boldsymbol{G}_t^+ \succeq \boldsymbol{G}_t \succeq \boldsymbol{G}_t^-$ for all $t \in \mathbb{N}$. Finally, by choosing $\boldsymbol{G}_0^\pm$ to be diagonal, the bounding dynamics reduce to decoupled scalar recursions, which can be analyzed explicitly. This allows us to establish concentration of the original iterates $\{\boldsymbol{G}_t\}_{t\in\mathbb{N}}$ around the bounding sequences, leading to operator-norm convergence of the discrete-time dynamics to their continuous-time counterparts.

# 6 Conclusion

In this work, we presented a comprehensive theoretical analysis of gradient-based learning in high-dimensional, extensive-width two-layer neural networks with quadratic activation. We established precise scaling laws that characterize both the population gradient flow and its empirical, discrete-time approximation. These results demonstrate how anisotropic signal strengths in the target function fundamentally shapes the convergence behavior and sample efficiency of gradient-based learning.

**Beyond quadratic activations.** An immediate direction for future research is to extend our analysis to more general activation functions. Link functions with higher information exponent is studied in a companion work [RNWL25], where the precise risk scaling is established by exploiting a decoupling structure that is unique to the information exponent $k > 2$ setting. Importantly, many commonly-used activation functions (ReLU, GeLU, etc.) have information exponent $k = 1$ and also contain a nonzero $\text{He}_2$ component. For such nonlinearities, we conjecture that SGD dynamics exhibits a multi-phase risk curve (analogous to the incremental learning phenomenon in [AAM23, BBPV23]), where the higher Hermite modes affects the learning dynamics after the low-order terms are learned. In Figure 3 we report the SGD risk curves for ReLU networks, in which we observe $(i)$ an initial loss drop driven by the $\text{He}_1$ component (which finds a degenerate rank-1 subspace), followed by $(ii)$ a power-law decay phase driven by the quadratic $\text{He}_2$ component where the empirical scaling exponent align closely with our theoretical predictions, and finally $(iii)$ a slope change late in training likely due to higher Hermite terms (in Figure 5 we confirm that this "late" phase is absent if we remove these higher-order components). Understanding this complex multi-phase learning dynamics remains an interesting challenge for future work.

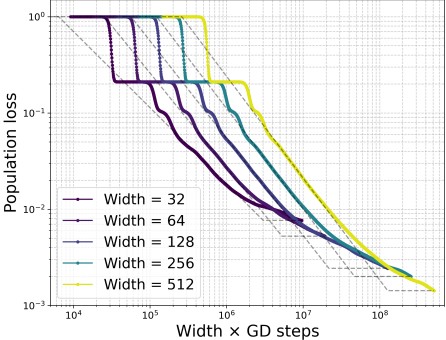 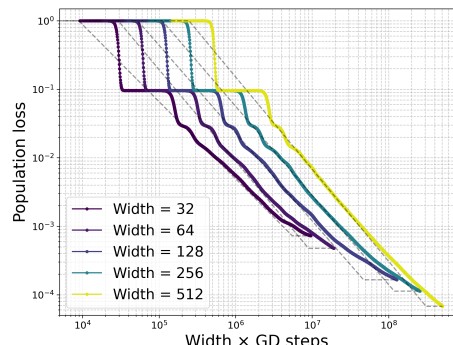

(a) $\alpha = 1$. Ideal exponent: $-1$; empirical: $-1.01$.      (b) $\alpha = \frac{3}{2}$. Ideal exponent; $-\frac{4}{3}$, empirical:: $-1.26$.

Figure 3: Population loss vs. compute for two-layer ReLU network (power-law second-layer with exponent $\alpha$) trained with population gradient descent. The student network adopts the 2-homogeneous parameterization as in (2.2). Observe that after the initial loss drop due to the $\text{He}_1$ component, the risk curves follow a power-law scaling where the exponent (dashed) nearly matches our theoretical prediction for the quadratic setting $\frac{1-2\alpha}{\alpha}$.

**Acknowledgment**

The authors thank Florent Krzakala, Jason D. Lee, and Lenka Zdeborová for discussion and feedback. The research of GBA was supported in part by NSF grant 2134216. MAE was partially supported by the NSERC Grant [2019-06167], the CIFAR AI Chairs program, and the CIFAR Catalyst grant. Part of this work was completed when NMV interned at the Flatiron Institute.

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

# Contents

# A Additional Figures

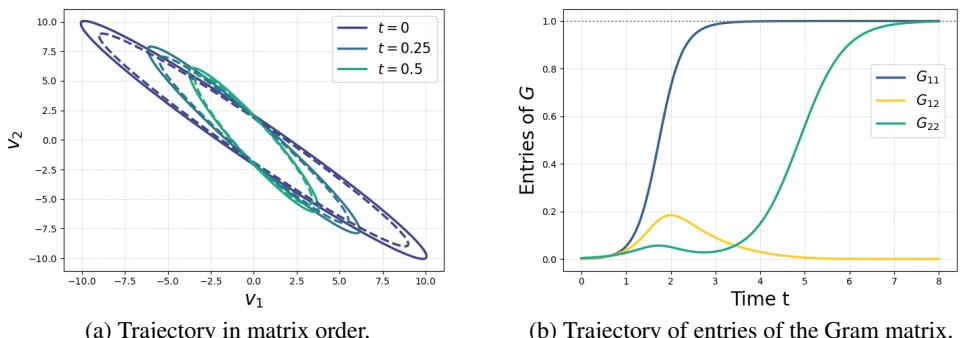

(a) Trajectory in matrix order.

(b) Trajectory of entries of the Gram matrix.

Figure 4: Solutions of the matrix Riccati ODE in (5.1) with $\lambda_1 = 2$, $\lambda_2 = 1$, $r_s = 2$. *(a)* To visualize the dynamics under matrix order, we plot the level sets of $\boldsymbol{G}(t)$ at times $t \in \{0, 0.25, 0.5\}$ for two initializations: $\boldsymbol{G}(0)$ (solid) and a scaled version $1.25\,\boldsymbol{G}(0)$ (dashed). The dashed ellipses remain enclosed within the solid ones at all times, illustrating monotonicity of the Riccati flow *with respect to initialization*. However, note that $\boldsymbol{G}(t)$ is not monotone in Loewner order over time, as seen from the lack of nesting among the solid ellipses. *(b)* Entry-wise evolution of $\boldsymbol{G}(t)$ under a random initialization with $d = 1024$. The diagonal entry $\boldsymbol{G}_{22}(t)$ exhibits non-monotonic behavior, illustrating that the solution trajectory $\boldsymbol{G}(t)$ need not be monotone in time.

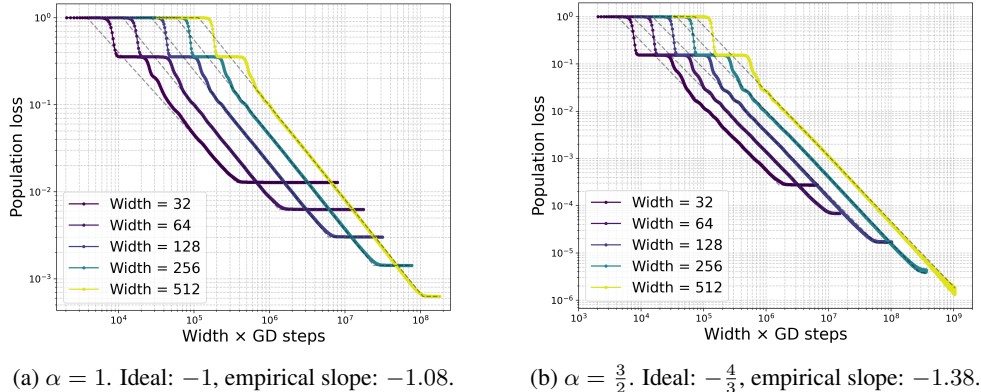

(a) $\alpha = 1$. Ideal: $-1$, empirical slope: $-1.08$.

(b) $\alpha = \frac{3}{2}$. Ideal: $-\frac{4}{3}$, empirical slope: $-1.38$.

Figure 5: Population loss vs. compute for two-layer neural network with activation function $\sigma \propto \mathrm{He}_1 + \mathrm{He}_2$, trained with population gradient descent. The student network adopts the 2-homogeneous parameterization as in (2.2). Observe that after the initial loss drop due to the $\mathrm{He}_1$ component, the risk curves exhibit a power-law scaling where the exponent (dashed lines) nearly matches our theoretical prediction for the quadratic setting $\frac{1-2\alpha}{\alpha}$; and unlike the ReLU setting (Figure 3), the loss immediately plateaus after the power-law phase.

**Experiment Setting.** In Figures 3 and 5, we plot the mean squared error loss for gradient descent with a constant step size on the population loss, using activations $\sigma = \mathrm{ReLU}$ and $\sigma = \mathrm{He}_1 + \mathrm{He}_2$. The teacher model has orthogonal first-layer neurons and power-law decay in the second-layer coefficients with $\alpha \in \{1.0, 1.5\}$. Both teacher and student networks use the same activation function, which we normalize to have zero-mean an unit $L^2$ norm. The student network uses the 2-homogeneous parameterization:

$$\hat{y}(\boldsymbol{W}) = \frac{1}{\sqrt{r_s}} \sum_{i=1}^{r_s} \|\boldsymbol{w}_i\|_2^2 \cdot \sigma(\langle \boldsymbol{w}_i, \boldsymbol{x} \rangle) \ \text{ where } \ \sigma \in \left\{ \tfrac{\mathrm{ReLU} - 1/\sqrt{2\pi}}{0.5}, \tfrac{\mathrm{He}_1 + \mathrm{He}_2}{3} \right\}.$$

We set dimension $d = 5000$, number of teacher neurons $r = 2400$, student widths $r_s \in \{32, 64, 128, 256, 512\}$, and learning rate $\eta = 0.5/\sqrt{r}$. To estimate the scaling exponents, we first identify the range of compute exhibiting a linear trend by visual inspection, and then fit the exponent via least squares. The dashed lines in the plots correspond to these fitted lines, and the reported empirical exponents represent the median values across different student widths.

# B Preliminaries for Proofs

**Proof organization.** Section B introduces the notations and definitions used throughout the paper. In Section C, we provide a brief review of matrix Riccati ODEs and difference equations, along with the necessary supporting statements. The main results are proved in Section D. In Section E we discuss the fine-tuning phase for the discretized algorithm. Additional proofs related to online SGD and auxiliary lemmas are deferred to Sections F and G, respectively.

**Notation and Definitions.** We use $[n] := \{1, 2, \ldots, n\}$ to denote the first $n$ natural numbers. The Euclidean inner product and norm are denoted by $\langle \cdot, \cdot \rangle$ and $\|\cdot\|_2$, respectively. For matrices, $\|\cdot\|_2$ and $\|\cdot\|_F$ denote the operator norm and Frobenius norm. The positive part is denoted by $(x)_+ := \max\{x, 0\}$. We write $f_d = o_d(1)$ if $f_d \to 0$ as $d \to \infty$, and $f_d \ll g_d$ if $f_d/g_d \to 0$. We use $O(\cdot)$ or $\Omega(\cdot)$ to suppress constants in upper and lower bounds respectively, and we use subscript to indicate parameter dependence, e.g., $O_\alpha(\cdot)$.

The symmetric part of a square matrix $M \in \mathbb{R}^{d \times d}$ is given by $\mathrm{Sym}(M) := \frac{1}{2}(M + M^\top)$. For symmetric matrices $A, B \in \mathbb{R}^{d \times d}$, we write $A \prec B$ (or $A \preceq B$) if $B - A$ is positive definite (or positive semidefinite). Moreover, if $A$ and $B$ are mutually diagonalizable, we write $AB^{-1} = \frac{A}{B}$.

We follow the convention that subscripts (e.g., $W_t$) refer to discrete-time quantities, and parentheses (e.g., $W(t)$) refer to continuous-time quantities. The overlap matrices of interest are defined as

$$\underbrace{G_W(t) := W(t)W(t)^\top}_{\text{Weight Gram matrix}}, \quad \underbrace{G_U(t) := \Theta^\top U(t)U(t)^\top \Theta}_{\text{Alignment Gram matrix}}, \quad \underbrace{G_t := \Theta^\top W_t W_t^\top \Theta}_{\text{Discrete alignment Gram matrix}}.$$

Let $Z \in \mathbb{R}^{d \times r_s}$ be a Gaussian matrix with i.i.d. entries distributed as $\mathcal{N}(0, 1/d)$. We define $Z_{1:m} \in \mathbb{R}^{m \times r_s}$ as the submatrix formed by the first $m$ rows:

$$Z = \begin{bmatrix} Z_{1:m} \\ Z_{\text{rest}} \end{bmatrix}.$$

Without loss of generality, we assume the teacher directions coincide with the standard basis vectors, i.e., $\theta_j = e_j$. With this, the initialization satisfies:

$$G_W(0) = ZZ^\top, \qquad G_U(0) = G_0 = Z_{1:r}(Z^\top Z)^{-1} Z_{1:r}^\top. \tag{B.1}$$

We start with characterizing "good events" for initial matrices given by the following lemma:

**Lemma 1.**

***Both cases*** ($\alpha \in [0, 0.5) \cup (0.5, \infty)$)***.*** *For $d \geq \Omega(1)$, the following holds:*

(E.1) $\frac{1}{1.05} \leq \lambda_{\min}(Z^\top Z) \leq \lambda_{\max}(Z^\top Z) \leq 1.05$.

(E.2) $1.05 Z_{1:r} Z_{1:r}^\top \succeq G_U(0) = G_0 \succeq \frac{1}{1.05} Z_{1:r} Z_{1:r}^\top$.

***Heavy-tailed case*** ($\alpha \in [0, 0.5)$)***.*** *For $d \geq \Omega_\varphi(1)$, the following holds:*

(H.1) *For $m \leq r_s(1 - \log^{-1/2} d) \wedge r$ uniformly, we consider $\lambda_{\min}(Z_{1:m} Z_{1:m}^\top) \geq \frac{r_s}{5d}(1 - \frac{m}{r_s})^2$.*

(H.2) *For all $m \leq r_s(1 - \log^{-1/2} d) \wedge r$ uniformly, $\lambda_m(Z_{1:r} Z_{1:r}^\top) \geq \frac{r_s}{5d}(1 - \frac{m}{r_s})^2$.*

(H.3) $\lambda_{\max}(Z_{1:r} Z_{1:r}^\top) \leq \frac{2r_s}{d}(1 + \frac{1}{\sqrt{\varphi}})^2$.

***Light-tailed case*** ($\alpha \in [0, 0.5)$)***.*** *For $d \geq \Omega(1)$, the following holds:*

(L.1) $\frac{1}{r_s^5 d} \leq \lambda_{\min}(Z_{1:r_s} Z_{1:r_s}^\top)$

(L.2) *For $m \in \{1, 2, \cdots, 5r_s, \lceil \log^{2.5} d \rceil, \lceil \log^6 d \rceil, r\}$ uniformly, $\lambda_{\max}(Z_{1:m} Z_{1:m}^\top) \leq \frac{5(r_s \vee m)}{d}$*

*We define*

$$\mathcal{G}_{init} \equiv \begin{cases} (E.1) \cap (E.2) \cap (H.1) \cap (H.2) \cap (H.3), & \alpha \in [0, 0.5) \\ (E.1) \cap (E.2) \cap (L.1) \cap (L.2), & \alpha \in > 0.5. \end{cases}$$

*We have*

$$\mathbb{P}[\mathcal{G}_{init}] \geq \begin{cases} 1 - 3r_s \exp\left(\frac{-r_s}{2\log^2 d}\right), & \alpha \in [0, 0.5) \\ 1 - \Omega(1/r_s^2), & \alpha > 0.5. \end{cases}$$

*Proof.* We will use the following:

(S.1) By [Ver10, Corollary 5.35], for $m \leq r_s$ and $\sqrt{r_s} - \sqrt{m} \geq t > 0$

$$\mathbb{P}\left[\lambda_{\min}\left(\boldsymbol{Z}_{1:m}\boldsymbol{Z}_{1:m}^{\top}\right) \geq \frac{r_s}{d}\left(1 - \sqrt{\frac{m}{r_s}} - \frac{t}{\sqrt{r_s}}\right)^2\right] \geq 1 - 2e^{-t^2},$$

and for $m \geq r_s$ and $\sqrt{m} - \sqrt{r_s} \geq t > 0$

$$\mathbb{P}\left[\lambda_{\min}\left(\boldsymbol{Z}_{1:m}^{\top}\boldsymbol{Z}_{1:m}\right) \geq \frac{m}{d}\left(1 - \sqrt{\frac{r_s}{m}} - \frac{t}{\sqrt{m}}\right)^2\right] \geq 1 - 2e^{-t^2}.$$

(S.2) By [Ver10, Corollary 5.35], for any fixed $m$

$$\mathbb{P}\left[\frac{m}{d}\left(1 + \sqrt{\frac{r_s}{m}} + \frac{t}{\sqrt{m}}\right)^2 \geq \lambda_{\max}\left(\boldsymbol{Z}_{1:m}^{\top}\boldsymbol{Z}_{1:m}\right)\right] \geq 1 - 2e^{-t^2}.$$

(S.3) By [Ver10, Theorem 5.38], there exists $C, c > 0$ such that

$$\mathbb{P}\left[\lambda_{\min}\left(\boldsymbol{Z}_{1:r_s}\boldsymbol{Z}_{1:r_s}^{\top}\right) \geq \frac{\varepsilon^2}{4dr_s}\right] \geq 1 - C\varepsilon - e^{-cr_s}.$$

For the heavy tailed case, we consider $d$ is large enough to guarantee $|\frac{r_s}{r} - \varphi| \leq \frac{\varphi}{2}$. We have

- By using (S.1) and (S.2) with $m = d$, $t = \sqrt{\frac{d}{\log d}}$, we can show that $\mathbb{P}[(E.1)] \geq 1 - e^{\frac{-d}{\log d}}$ for $d \geq \Omega(1)$.

- By (B.1) and (E.1) , (E.2) follows.

- For (H.1), by using (S.1) with $t = \frac{\sqrt{r_s} - \sqrt{m}}{\sqrt{\log d}} \geq \sqrt{\frac{r_s}{2\log^2 d}}$, we have with probability $1 - 2r_s \exp\left(\frac{-r_s}{2\log^2 d}\right)$, for $m \leq r_s(1 - \log^{-1/2} d) \wedge r$ uniformly:

$$\lambda_{\min}\left(\boldsymbol{Z}_{1:m}\boldsymbol{Z}_{1:m}^{\top}\right) > \frac{r_s}{d}\left(1 - \frac{1}{\sqrt{\log d}}\right)^2\left(1 - \sqrt{\frac{m}{r_s}}\right)^2 \geq \frac{r_s}{5d}\left(1 - \frac{m}{r_s}\right)^2.$$

Therefore, $\mathbb{P}[(H.1)] \geq 1 - 2r_s \exp\left(\frac{-r_s}{2\log d}\right)$.

- By Cauchy's eigenvalue interlacing theorem, $\lambda_m(\boldsymbol{Z}_{1:r}\boldsymbol{Z}_{1:r}^{\top}) \geq \lambda_{\min}\left(\boldsymbol{Z}_{1:m}\boldsymbol{Z}_{1:m}^{\top}\right)$. Therefore, by (H.1) , (H.2) follows.

- For (H.3) , by using $m = r$ and $t = 0.4\sqrt{r_s}$ in (S.2), we have $\mathbb{P}[(H.3)] \geq 1 - 2e^{-0.16r_s}$.

- For (L.1), by using (S.3) with $\varepsilon = \frac{2}{r_s^2}$, we have $\mathbb{P}[(L.1)] \geq 1 - \Omega(1/r_s^2)$.

- For (L.2), by using (S.2) with $t = 0.4\sqrt{r_s}$, we have with probability $1 - (10r_s + 6)e^{-0.16r_s}$ for $m \in [5r_s] \cup \{\lceil \log^{2.5} d \rceil, \lceil \log^6 d \rceil, r\}$ uniformly:

$$\lambda_{\max}(\boldsymbol{Z}_{1:m}\boldsymbol{Z}_{1:m}^{\top}) \leq \frac{r_s}{d}\left(1.4 + \sqrt{\frac{m}{r_s}}\right)^2 \leq \frac{5(r_s \vee m)}{d}.$$

By union bound, we have the result. $\qquad\square$

## C Background: Matrix Riccati Dynamical Systems

We begin by reviewing Riccati dynamical systems in both continuous and discrete time, establishing the necessary background for the arguments that follow. For the following, we define

$$\boldsymbol{\Lambda}_e := \begin{bmatrix} \boldsymbol{\Lambda} & 0 \\ 0 & 0 \end{bmatrix} \text{ and } \widetilde{\boldsymbol{\Lambda}} := \frac{\sqrt{r_s}}{\|\boldsymbol{\Lambda}\|_F} \boldsymbol{\Lambda}_e.$$

For notational convenience, we adapt the abuse of notation:

$$\frac{\boldsymbol{\Lambda}_e}{\boldsymbol{I}_d - \exp(-t\boldsymbol{\Lambda}_e)} = \lim_{\varepsilon \to 0} \frac{(\boldsymbol{\Lambda}_e + \varepsilon \boldsymbol{I}_d)}{\boldsymbol{I}_d - \exp(-t(\boldsymbol{\Lambda}_e + \varepsilon \boldsymbol{I}_d))} = \begin{bmatrix} \frac{\boldsymbol{\Lambda}}{\boldsymbol{I}_r - \exp(-t\boldsymbol{\Lambda})} & 0 \\ 0 & \frac{1}{t}\boldsymbol{I}_{d-r} \end{bmatrix}.$$

### C.1 Continous-time Matrix Riccati ODE

In this paper, we study continuous-time matrix Riccati differential equations of the following form:

- Weight Gram matrix: For $T_W = r_s$

$$\partial_t \boldsymbol{G}_W(t) = \frac{0.5}{T_W} \left( \widetilde{\boldsymbol{\Lambda}} \boldsymbol{G}_W(t) + \boldsymbol{G}_W(t) \widetilde{\boldsymbol{\Lambda}} - 2\boldsymbol{G}_W^2(t) \right). \tag{C.1}$$

- Alignment Gram matrix: For $T_U = \|\boldsymbol{\Lambda}\|_F \sqrt{r_s}$,

$$\partial_t \boldsymbol{G}_U(t) = \frac{0.5}{T_U} \left( \boldsymbol{\Lambda} \boldsymbol{G}_U(t) + \boldsymbol{G}_U(t) \boldsymbol{\Lambda} - 2\boldsymbol{G}_U(t) \boldsymbol{\Lambda} \boldsymbol{G}_U(t) \right). \tag{C.2}$$

For $\alpha = 0$, we assume that the ODEs are expressed in the eigenbasis of $\boldsymbol{G}_W(0)$ or $\boldsymbol{G}_U(0)$, ensuring that the trajectories remain diagonal. The solutions of these ODEs are characterized in the following statement:

**Lemma 2.** (C.1) *and* (C.2) *admit the following solutions:*

$$\boldsymbol{G}_W(t) = \frac{\widetilde{\boldsymbol{\Lambda}}}{\boldsymbol{I}_d - \exp(-t\widetilde{\boldsymbol{\Lambda}}/T_W)}$$
$$- \frac{\widetilde{\boldsymbol{\Lambda}}\exp(-0.5t\widetilde{\boldsymbol{\Lambda}}/T_W)}{\boldsymbol{I}_d - \exp(-t\widetilde{\boldsymbol{\Lambda}}/T_W)} \left( \boldsymbol{G}_W(0) + \frac{\widetilde{\boldsymbol{\Lambda}}\exp(-t\widetilde{\boldsymbol{\Lambda}}/T_W)}{\boldsymbol{I}_d - \exp(-t\widetilde{\boldsymbol{\Lambda}}/T_W)} \right)^{-1} \frac{\widetilde{\boldsymbol{\Lambda}}\exp(-0.5t\widetilde{\boldsymbol{\Lambda}}/T_W)}{\boldsymbol{I}_d - \exp(-t\widetilde{\boldsymbol{\Lambda}}/T_W)}$$

$$\boldsymbol{G}_U(t) = \frac{\boldsymbol{I}_r}{\boldsymbol{I}_r - \exp(-t\boldsymbol{\Lambda}/T_U)}$$
$$- \frac{\exp(-0.5t\boldsymbol{\Lambda}/T_U)}{\boldsymbol{I}_r - \exp(-t\boldsymbol{\Lambda}/T_U)} \left( \boldsymbol{G}_U(0) + \frac{\exp(-t\boldsymbol{\Lambda}/T_U)}{\boldsymbol{I}_r - \exp(-t\boldsymbol{\Lambda}/T_U)} \right)^{-1} \frac{\exp(-0.5t\boldsymbol{\Lambda}/T_U)}{\boldsymbol{I}_r - \exp(-t\boldsymbol{\Lambda}/T_U)}$$

*Moreover,* $(\boldsymbol{G}_W(t))_{t \geq 0}$ *and* $(\boldsymbol{G}_U(t))_{t \geq 0}$ *are monotone with respect to* $\boldsymbol{G}_W(0) \succeq 0$ *and* $\boldsymbol{G}_U(0) \succeq 0$ *respectively.*

*Proof.* One can check by direct differentiation that the given closed-form expressions satisfy the ODEs above. The uniqueness of the solutions follow the local Lipschitzness of the drifts. Monotonicity is a consequence of Proposition 25. $\square$

### C.2 Discrete-time Matrix Riccati Difference Equations

In this section, we will study a particular discretization of Alignment Gram matrix ODE, given as

$$\boldsymbol{G}_{t+1} = \boldsymbol{G}_t - \frac{\eta}{2}(2\boldsymbol{G}_t - \boldsymbol{I}_r)\boldsymbol{\Lambda}(2\boldsymbol{G}_t - \boldsymbol{I}_r)\left(\boldsymbol{I}_r + \eta\boldsymbol{\Lambda}(2\boldsymbol{G}_t - \boldsymbol{I}_r)\right)^{-1} + \eta\boldsymbol{\Lambda}. \tag{C.3}$$

For convenience, we will make a change of variable and define $\boldsymbol{V}_t := 2\boldsymbol{\Lambda}^{\frac{1}{2}}\boldsymbol{G}_t\boldsymbol{\Lambda}^{\frac{1}{2}} - \boldsymbol{\Lambda}$. We write (C.3) in terms of $\boldsymbol{V}_t$ as follows:

$$\boldsymbol{V}_{t+1} = \boldsymbol{V}_t - \eta\boldsymbol{V}_t^2\left(\boldsymbol{I}_r + \eta\boldsymbol{V}_t\right)^{-1} + \eta\boldsymbol{\Lambda}^2.$$

We characterize the dynamics of $(\boldsymbol{V}_t)_{t \in \mathbb{N}}$ as follows:

**Lemma 3.** *We consider*

$$\begin{bmatrix} \boldsymbol{X}_{t+1,1} \\ \boldsymbol{X}_{t+1,2} \end{bmatrix} = \begin{bmatrix} \boldsymbol{X}_{t,1} \\ \boldsymbol{X}_{t,2} \end{bmatrix} + \eta \boldsymbol{H} \begin{bmatrix} \boldsymbol{X}_{t,1} \\ \boldsymbol{X}_{t,2} \end{bmatrix} \quad \text{where} \quad \begin{bmatrix} \boldsymbol{X}_{0,1} \\ \boldsymbol{X}_{0,2} \end{bmatrix} = \begin{bmatrix} \boldsymbol{I}_r \\ \boldsymbol{V}_0 \end{bmatrix} \text{ and } \boldsymbol{H} := \begin{bmatrix} 0 & \boldsymbol{I}_r \\ \boldsymbol{\Lambda}^2 & \eta \boldsymbol{\Lambda}^2 \end{bmatrix}.$$

*The following hold for all $n \in \mathbb{N}$:*

*(R.1) We have*

$$\begin{bmatrix} \boldsymbol{A}_{t,11} & \boldsymbol{\Lambda}^{-1} \boldsymbol{A}_{t,12} \\ \boldsymbol{\Lambda} \boldsymbol{A}_{t,12} & \boldsymbol{A}_{t,22} \end{bmatrix} := (\boldsymbol{I}_{2r} + \eta \boldsymbol{H})^t. \tag{C.4}$$

*where $\boldsymbol{A}_{t,11}$, $\boldsymbol{A}_{t,12}$, $\boldsymbol{A}_{t,22}$ are positive definite diagonal matrices.*

*(R.2) $\boldsymbol{A}_{t,11} + \eta \boldsymbol{\Lambda} \boldsymbol{A}_{t,12} = \boldsymbol{A}_{t,22}$ and $\boldsymbol{A}_{t,22} \boldsymbol{A}_{t,11} - \boldsymbol{A}_{t,12}^2 = \boldsymbol{I}_r$.*

*(R.3) For $\eta \leq 1$, we have $\boldsymbol{A}_{t,11} \boldsymbol{A}_{t,12}^{-1} \succ \left(\boldsymbol{I}_r + \frac{\eta^2}{4}\boldsymbol{\Lambda}^2\right)^{1/2} - \frac{\eta}{2}\boldsymbol{\Lambda}$ and $\boldsymbol{A}_{t,22} \boldsymbol{A}_{t,12}^{-1} \succ \left(\boldsymbol{I}_r + \frac{\eta^2}{4}\boldsymbol{\Lambda}^2\right)^{1/2} + \frac{\eta}{2}\boldsymbol{\Lambda}$.*

*(R.4) If $\|\eta \boldsymbol{\Lambda}\|_2 < 1$,*

$$\boldsymbol{A}_{t,22} \boldsymbol{A}_{t,12}^{-1} \succ \frac{(\boldsymbol{I}_r + \eta \boldsymbol{\Lambda})^t + (\boldsymbol{I}_r - \eta \boldsymbol{\Lambda})^t}{(\boldsymbol{I}_r + \eta \boldsymbol{\Lambda})^t - (\boldsymbol{I}_r - \eta \boldsymbol{\Lambda})^t} \succeq \boldsymbol{A}_{t,11} \boldsymbol{A}_{t,12}^{-1}.$$

*Moreover, if $\boldsymbol{X}_{t,1}$ and $\boldsymbol{X}_{t+1,1}$ are invertible:*

*(R.7) For $\boldsymbol{V}_{t+1} := \boldsymbol{X}_{t+1,2} \boldsymbol{X}_{t+1,1}^{-1}$, and $\boldsymbol{V}_t := \boldsymbol{X}_{2,t} \boldsymbol{X}_{t,1}^{-1}$, we have*

$$\boldsymbol{V}_{t+1} = \boldsymbol{V}_t - \eta \boldsymbol{V}_t^2 \left(\boldsymbol{I}_r + \eta \boldsymbol{V}_t\right)^{-1} + \eta \boldsymbol{\Lambda}^2.$$

*Proof of Lemma 3.* We have

$$\boldsymbol{I}_{2r} + \eta \boldsymbol{H} = \begin{bmatrix} \boldsymbol{I}_r & \eta \boldsymbol{I}_r \\ \eta \boldsymbol{\Lambda}^2 & \boldsymbol{I}_r + \eta^2 \boldsymbol{\Lambda}^2 \end{bmatrix}. \tag{C.5}$$

Let

$$(\boldsymbol{I}_{2r} + \eta \boldsymbol{H})^t =: \begin{bmatrix} \tilde{\boldsymbol{A}}_{t,11} & \tilde{\boldsymbol{A}}_{t,12} \\ \tilde{\boldsymbol{A}}_{21,t} & \tilde{\boldsymbol{A}}_{t,12} \end{bmatrix}.$$

Since each submatrix in (C.5) is diagonal positive definite, the matrices in (C.4) are also diagonal positive definite. To prove (R.1) and the first part of (R.2) we use proof by induction. We assume $\tilde{\boldsymbol{A}}_{t,11} + \eta \tilde{\boldsymbol{A}}_{21,t} = \tilde{\boldsymbol{A}}_{t,12}$ and $\tilde{\boldsymbol{A}}_{21,t} \tilde{\boldsymbol{A}}_{12,t}^{-1} = \boldsymbol{\Lambda}^2$. We have

$$\tilde{\boldsymbol{A}}_{12,t+1} = \tilde{\boldsymbol{A}}_{t,12} + \eta \tilde{\boldsymbol{A}}_{t,12} \overset{(a)}{=} \tilde{\boldsymbol{A}}_{t,12} + \eta(\tilde{\boldsymbol{A}}_{t,11} + \eta \tilde{\boldsymbol{A}}_{21,t}) \overset{(b)}{=} \eta \tilde{\boldsymbol{A}}_{t,11} + \left(\boldsymbol{I}_r + \eta^2 \boldsymbol{\Lambda}^2\right) \tilde{\boldsymbol{A}}_{t,12}$$

$$\overset{(c)}{=} \eta \tilde{\boldsymbol{A}}_{t,11} + \boldsymbol{\Lambda}^{-2}\left(\boldsymbol{I}_r + \eta^2 \boldsymbol{\Lambda}^2\right) \tilde{\boldsymbol{A}}_{21,t}$$

$$= \boldsymbol{\Lambda}^{-2} \tilde{\boldsymbol{A}}_{21,t+1}.$$

where (a) follows the first assumption, (b) and (c) follow the second assumption. Moreover,

$$\tilde{\boldsymbol{A}}_{11,t+1} + \eta \tilde{\boldsymbol{A}}_{21,t+1} = \tilde{\boldsymbol{A}}_{t,11} + \eta \tilde{\boldsymbol{A}}_{21,t} + \eta^2 \boldsymbol{\Lambda}^2 \tilde{\boldsymbol{A}}_{t,11} + \eta(\boldsymbol{I}_r + \eta^2 \boldsymbol{\Lambda}^2)\tilde{\boldsymbol{A}}_{21,t}$$

$$= (\boldsymbol{I}_r + \eta^2 \boldsymbol{\Lambda}^2)(\tilde{\boldsymbol{A}}_{t,11} + \eta \tilde{\boldsymbol{A}}_{21,t}) + \eta \tilde{\boldsymbol{A}}_{21,t}$$

$$\overset{(d)}{=} (\boldsymbol{I}_r + \eta^2 \boldsymbol{\Lambda}^2)\tilde{\boldsymbol{A}}_{t,12} + \eta \boldsymbol{\Lambda}^2 \tilde{\boldsymbol{A}}_{t,12} = \tilde{\boldsymbol{A}}_{22,t+1}.$$

where (d) follows the first and second assumptions. For the second part of (R.2) we again use proof by induction. We assume $\boldsymbol{A}_{t,22} \boldsymbol{A}_{t,11} - \boldsymbol{A}_{t,12}^2 = \boldsymbol{I}_r$. We have

$$\tilde{\boldsymbol{A}}_{11,t+1} \tilde{\boldsymbol{A}}_{22,t+1} - \tilde{\boldsymbol{A}}_{12,n+1} \tilde{\boldsymbol{A}}_{21,t+1} = \left(\tilde{\boldsymbol{A}}_{t,11} + \eta \tilde{\boldsymbol{A}}_{21,t}\right)\left(\eta \boldsymbol{\Lambda}^2 \tilde{\boldsymbol{A}}_{t,12} + \left(\boldsymbol{I}_r + \eta^2 \boldsymbol{\Lambda}^2\right) \tilde{\boldsymbol{A}}_{t,12}\right)$$

$$- \left(\eta \boldsymbol{\Lambda}^2 \tilde{\boldsymbol{A}}_{t,11} + \left(\boldsymbol{I}_r + \eta^2 \boldsymbol{\Lambda}^2\right) \tilde{\boldsymbol{A}}_{21,t}\right)\left(\tilde{\boldsymbol{A}}_{t,12} + \eta \tilde{\boldsymbol{A}}_{t,12}\right)$$

$$= \tilde{\boldsymbol{A}}_{t,11}\tilde{\boldsymbol{A}}_{t,12} - \tilde{\boldsymbol{A}}_{t,12}\tilde{\boldsymbol{A}}_{21,t} = \boldsymbol{I}_r.$$

For (R.3), by using (R.2), we have

$$\boldsymbol{A}_{t,11}\left(\boldsymbol{A}_{t,11} + \eta\boldsymbol{\Lambda}\boldsymbol{A}_{t,12}\right) - \boldsymbol{A}_{t,12}^2 = \boldsymbol{I}_r \Rightarrow \left(\boldsymbol{A}_{t,11}\boldsymbol{A}_{t,12}^{-1}\right)^2 + \eta\boldsymbol{\Lambda}\left(\boldsymbol{A}_{t,11}\boldsymbol{A}_{t,12}^{-1}\right) - \boldsymbol{I}_r \succ 0$$

$$\Rightarrow \boldsymbol{A}_{t,11}\boldsymbol{A}_{t,12}^{-1} \succ \left(\boldsymbol{I}_r + \frac{\eta^2}{4}\boldsymbol{\Lambda}\right)^{1/2} - \frac{\eta}{2}\boldsymbol{\Lambda}.$$

The second part follows (R.2). For (R.4), we recall that

$$\boldsymbol{A}_{t+1,12} = \boldsymbol{A}_{t,12} + \eta\boldsymbol{\Lambda}\boldsymbol{A}_{t,22} = (\boldsymbol{I}_r + \eta^2\boldsymbol{\Lambda}^2)\boldsymbol{A}_{t,12} + \eta\boldsymbol{\Lambda}\boldsymbol{A}_{t,11} \tag{C.6}$$

$$\boldsymbol{A}_{t+1,22} = \eta\boldsymbol{\Lambda}\boldsymbol{A}_{t,12} + (\boldsymbol{I}_r + \eta^2\boldsymbol{\Lambda}^2)\boldsymbol{A}_{t,22} \succ \eta\boldsymbol{\Lambda}\boldsymbol{A}_{t,12} + \boldsymbol{A}_{t,22}. \tag{C.7}$$

We use proof by induction. Suppose the lower bound for $\frac{\boldsymbol{A}_{t,22}}{\boldsymbol{A}_{t,12}}$ holds. We have

$$\frac{\boldsymbol{A}_{t+1,22}}{\boldsymbol{A}_{t+1,12}} \overset{(e)}{\succ} \frac{\eta\boldsymbol{\Lambda}\boldsymbol{A}_{t,12} + \boldsymbol{A}_{t,22}}{\boldsymbol{A}_{t,12} + \eta\boldsymbol{\Lambda}\boldsymbol{A}_{t,22}}$$

$$\overset{(f)}{\succ} \left(\eta\boldsymbol{\Lambda} + \frac{(\boldsymbol{I}_r + \eta\boldsymbol{\Lambda})^t + (\boldsymbol{I}_r - \eta\boldsymbol{\Lambda})^t}{(\boldsymbol{I}_r + \eta\boldsymbol{\Lambda})^t - (\boldsymbol{I}_r - \eta\boldsymbol{\Lambda})^t}\right) \left(\boldsymbol{I}_r + \eta\boldsymbol{\Lambda}\frac{(\boldsymbol{I}_r + \eta\boldsymbol{\Lambda})^t + (\boldsymbol{I}_r - \eta\boldsymbol{\Lambda})^t}{(\boldsymbol{I}_r + \eta\boldsymbol{\Lambda})^t - (\boldsymbol{I}_r - \eta\boldsymbol{\Lambda})^t}\right)^{-1}$$

$$= \frac{(\boldsymbol{I}_r + \eta\boldsymbol{\Lambda})^{t+1} + (\boldsymbol{I}_r - \eta\boldsymbol{\Lambda})^{t+1}}{(\boldsymbol{I}_r + \eta\boldsymbol{\Lambda})^{t+1} - (\boldsymbol{I}_r - \eta\boldsymbol{\Lambda})^{t+1}}.$$

where (e) follows (C.7) and (f) follows the induction hypothesis with that $x \to \frac{x+\eta\lambda}{1+\eta\lambda x}$ is monotonic increasing for $\eta\lambda < 1$. For the upper bound, suppose the lower bound for $\frac{\boldsymbol{A}_{t,11}}{\boldsymbol{A}_{t,12}}$ holds. We have

$$\frac{\boldsymbol{A}_{t+1,11}}{\boldsymbol{A}_{t+1,12}} \overset{(g)}{\preceq} \frac{\boldsymbol{A}_{t,11} + \eta\boldsymbol{\Lambda}\boldsymbol{A}_{t,12}}{\boldsymbol{A}_{t,12} + \eta\boldsymbol{\Lambda}\boldsymbol{A}_{t,11}}$$

$$\overset{(h)}{\preceq} \left(\eta\boldsymbol{\Lambda} + \frac{(\boldsymbol{I}_r + \eta\boldsymbol{\Lambda})^t + (\boldsymbol{I}_r - \eta\boldsymbol{\Lambda})^t}{(\boldsymbol{I}_r + \eta\boldsymbol{\Lambda})^t - (\boldsymbol{I}_r - \eta\boldsymbol{\Lambda})^t}\right) \left(\boldsymbol{I}_r + \eta\boldsymbol{\Lambda}\frac{(\boldsymbol{I}_r + \eta\boldsymbol{\Lambda})^t + (\boldsymbol{I}_r - \eta\boldsymbol{\Lambda})^t}{(\boldsymbol{I}_r + \eta\boldsymbol{\Lambda})^t - (\boldsymbol{I}_r - \eta\boldsymbol{\Lambda})^t}\right)^{-1}$$

$$= \frac{(\boldsymbol{I}_r + \eta\boldsymbol{\Lambda})^{t+1} + (\boldsymbol{I}_r - \eta\boldsymbol{\Lambda})^{t+1}}{(\boldsymbol{I}_r + \eta\boldsymbol{\Lambda})^{t+1} - (\boldsymbol{I}_r - \eta\boldsymbol{\Lambda})^{t+1}}.$$

where (g) follows (C.6), and (h) follows the induction hypothesis.

Lastly if $\boldsymbol{X}_{t,1}$ and $\boldsymbol{X}_{t+1,1}$ are invertible,

$$\boldsymbol{X}_{t+1,2}\boldsymbol{X}_{t+1,1}^{-1} = \left(\boldsymbol{X}_{t,2}\boldsymbol{X}_{t,1}^{-1} + \eta^2\boldsymbol{\Lambda}^2\boldsymbol{X}_{t,2}\boldsymbol{X}_{t,1}^{-1} + \eta\boldsymbol{\Lambda}^2\right)\left(\boldsymbol{I}_r + \eta\boldsymbol{X}_{t,2}\boldsymbol{X}_{t,2}^{-1}\right)^{-1}$$

$$= \boldsymbol{X}_{t,2}\boldsymbol{X}_{t,2}^{-1}\left(\boldsymbol{I}_r + \eta\boldsymbol{X}_{t,2}\boldsymbol{X}_{t,2}^{-1}\right)^{-1} + \eta\boldsymbol{\Lambda}^2$$

$$= \boldsymbol{X}_{t,2}\boldsymbol{X}_{t,2}^{-1} - \eta\boldsymbol{X}_{t,2}\boldsymbol{X}_{t,1}^{-1}\boldsymbol{X}_{t,2}\boldsymbol{X}_{t,1}^{-1}\left(\boldsymbol{I}_r + \eta\boldsymbol{X}_{t,2}\boldsymbol{X}_{t,1}^{-1}\right)^{-1} + \eta\boldsymbol{\Lambda}^2.$$

$\square$

**Corollary 3.** *For $\boldsymbol{V}_0 = 2\boldsymbol{\Lambda}_{\frac{1}{2}}^{\frac{1}{2}}\boldsymbol{G}_0\boldsymbol{\Lambda}_{\frac{1}{2}}^{\frac{1}{2}} - \boldsymbol{\Lambda}_1$, we define*

$$\boldsymbol{V}_{t+1} = \boldsymbol{V}_t - \eta\boldsymbol{V}_t^2\left(\boldsymbol{I}_r + \eta\boldsymbol{V}_t\right)^{-1} + \eta\hat{\boldsymbol{\Lambda}}^2.$$

*If $\boldsymbol{\Lambda}_1$, $\boldsymbol{\Lambda}_2$ and $\hat{\boldsymbol{\Lambda}}$ are mutually diagonalizable, and $\boldsymbol{X}_{1,t}$ is invertible for $t \leq t^* \in \mathbb{N}$, we have for $t \leq t^*$*

$$\boldsymbol{G}_t = \frac{\frac{\boldsymbol{\Lambda}_1}{\boldsymbol{\Lambda}_2} + \frac{\boldsymbol{A}_{t,22}}{\boldsymbol{A}_{t,12}}\frac{\hat{\boldsymbol{\Lambda}}}{\boldsymbol{\Lambda}_2}}{2} - \frac{1}{4}\frac{\boldsymbol{A}_{t,12}^{-1}\hat{\boldsymbol{\Lambda}}}{\boldsymbol{\Lambda}_2}\left(\frac{\frac{\hat{\boldsymbol{\Lambda}}}{\boldsymbol{\Lambda}_2}\frac{\boldsymbol{A}_{t,11}}{\boldsymbol{A}_{t,12}} - \frac{\boldsymbol{\Lambda}_1}{\boldsymbol{\Lambda}_2}}{2} + \boldsymbol{G}_0\right)^{-1}\frac{\hat{\boldsymbol{\Lambda}}\boldsymbol{A}_{t,12}^{-1}}{\boldsymbol{\Lambda}_2},$$

*where $\boldsymbol{A}_{11t}$, $\boldsymbol{A}_{t,12}$, $\boldsymbol{A}_{t,22}$ are defined with $\hat{\boldsymbol{\Lambda}}$.*

*Proof.* By using (C.4), we can write that

$$
\begin{aligned}
\boldsymbol{V}_t &= \left( \hat{\boldsymbol{\Lambda}} \boldsymbol{A}_{t,12} + \boldsymbol{A}_{t,22} \boldsymbol{V}_0 \right) \left( \hat{\boldsymbol{\Lambda}} \boldsymbol{A}_{t,12}^{-1} \boldsymbol{A}_{t,11} + \boldsymbol{V}_0 \right)^{-1} \hat{\boldsymbol{\Lambda}} \boldsymbol{A}_{t,12}^{-1} \\
&= \hat{\boldsymbol{\Lambda}} \boldsymbol{A}_{t,12} \left( \hat{\boldsymbol{\Lambda}} \boldsymbol{A}_{t,12}^{-1} \boldsymbol{A}_{t,11} + \boldsymbol{V}_0 \right)^{-1} \hat{\boldsymbol{\Lambda}} \boldsymbol{A}_{t,12}^{-1} \\
&\quad + \boldsymbol{A}_{t,22} \left( \boldsymbol{I}_r - \boldsymbol{A}_{t,11} \boldsymbol{A}_{t,12}^{-1} \hat{\boldsymbol{\Lambda}} \left( \hat{\boldsymbol{\Lambda}} \boldsymbol{A}_{t,12}^{-1} \boldsymbol{A}_{t,11} + \boldsymbol{V}_0 \right)^{-1} \right) \hat{\boldsymbol{\Lambda}} \boldsymbol{A}_{t,12}^{-1} \\
&= \boldsymbol{A}_{t,22} \boldsymbol{A}_{t,12}^{-1} \hat{\boldsymbol{\Lambda}} - \boldsymbol{A}_{t,12}^{-1} \hat{\boldsymbol{\Lambda}} \left( \hat{\boldsymbol{\Lambda}} \boldsymbol{A}_{t,12}^{-1} \boldsymbol{A}_{t,11} + \boldsymbol{V}_0 \right)^{-1} \hat{\boldsymbol{\Lambda}} \boldsymbol{A}_{t,12}^{-1}.
\end{aligned}
$$

Therefore,

$$
\boldsymbol{G}_t = \frac{\frac{\boldsymbol{\Lambda}_1}{\boldsymbol{\Lambda}_2} + \frac{\boldsymbol{A}_{t,22}}{\boldsymbol{A}_{t,12}} \frac{\hat{\boldsymbol{\Lambda}}}{\boldsymbol{\Lambda}_2}}{2} - \frac{1}{4} \frac{\boldsymbol{A}_{t,12}^{-1} \hat{\boldsymbol{\Lambda}}}{\boldsymbol{\Lambda}_2} \left( \frac{\frac{\hat{\boldsymbol{\Lambda}}}{\boldsymbol{\Lambda}_2} \frac{\boldsymbol{A}_{t,11}}{\boldsymbol{A}_{t,12}} - \frac{\boldsymbol{\Lambda}_1}{\boldsymbol{\Lambda}_2}}{2} + \boldsymbol{G}_0 \right)^{-1} \frac{\hat{\boldsymbol{\Lambda}} \boldsymbol{A}_{t,12}^{-1}}{\boldsymbol{\Lambda}_2}.
$$

$\square$

**Proposition 4.** *For some symmetric matrix $\boldsymbol{S}$, we consider*

$$
\boldsymbol{V}_1 = \boldsymbol{V}_0 + \eta \boldsymbol{S} - \eta \boldsymbol{V}_0^2 \left( \boldsymbol{I}_r + \eta \boldsymbol{V}_0 \right)^{-1}. \tag{C.8}
$$

*If $\boldsymbol{V}_0^+ \succeq \boldsymbol{V}_0 \succ \frac{-1}{\eta} \boldsymbol{I}_r$, we have $\boldsymbol{V}_1^+ \succeq \boldsymbol{V}_1$, where $\boldsymbol{V}_1^+$ is the next iterate if we use $\boldsymbol{V}_0^+$ in (C.8).*

*Proof.* We have

$$
\boldsymbol{V}_1 = \frac{1}{\eta} \left( \boldsymbol{I}_r - \left( \boldsymbol{I}_r + \eta \boldsymbol{V}_0 \right)^{-1} \right) + \eta \boldsymbol{S}.
$$

The statement follows by Proposition 25. $\square$

# D  Proofs for Main Results

## D.1  Proof of Propositions 2 and 3

For Proposition 2, we observe that

$$
\widehat{\boldsymbol{G}}(t) := \widehat{\boldsymbol{W}}(t) \widehat{\boldsymbol{W}}(t)^\top \quad \text{and} \quad \widetilde{\boldsymbol{G}}(t) := \boldsymbol{U}(t) \boldsymbol{U}(t)^\top = \boldsymbol{W}(t) (\boldsymbol{W}(t)^\top \boldsymbol{W}(t))^{-1} \boldsymbol{W}(t)^\top
$$

have the exact same dynamics. Therefore the statement follows. Proposition 3 follows Proposition 25.

## D.2  Decomposition of the population risk

The fine-tuning step relies on the following decomposition of the population risk:

**Proposition 5.** *For any $\boldsymbol{\Omega} \in \mathbb{R}^{r_s \times r_s}$, the population risk defined in (2.3) can be written as:*

$$
R(\boldsymbol{W}_t \boldsymbol{\Omega}) = \frac{1}{r_s} \left\| \boldsymbol{\Omega} \boldsymbol{\Omega}^\top - \frac{\sqrt{r_s}}{\|\boldsymbol{\Lambda}\|_{\mathrm{F}}} \boldsymbol{W}_t^\top \boldsymbol{\Theta} \boldsymbol{\Lambda} \boldsymbol{\Theta}^\top \boldsymbol{W}_t \right\|_F^2 + \frac{1}{\|\boldsymbol{\Lambda}\|_{\mathrm{F}}^2} \left( \|\boldsymbol{\Lambda}\|_{\mathrm{F}}^2 - \|\boldsymbol{\Lambda}^{\frac{1}{2}} \boldsymbol{G}_t \boldsymbol{\Lambda}^{\frac{1}{2}}\|_F^2 \right),
$$

*where $\boldsymbol{G}_t = \boldsymbol{\Theta}^\top \boldsymbol{W}_t \boldsymbol{W}_t^\top \boldsymbol{\Theta}$ is the discrete alignment Gram matrix defined in the previous part.*

We observe that both the second term and the matrix $\boldsymbol{W}_t^\top \boldsymbol{\Theta} \boldsymbol{\Lambda} \boldsymbol{\Theta}^\top \boldsymbol{W}_t$ are independent of $\boldsymbol{\Omega}$. Hence, the fine-tuning step reduces to a least squares problem in the matrix $\boldsymbol{\Omega} \boldsymbol{\Omega}^\top$ in population, which is approximated via empirical risk minimization over a fresh batch of samples. By standard concentration arguments, a sample size of $N_{\mathrm{Ft}} \geq r_s^2 \mathrm{polylog} d$ suffices to ensure that the empirical minimizer approximates the population solution with high probability.

*Proof.* We begin by noting that $\boldsymbol{W}_t$ is an orthonormal matrix. Using this, we can express the population risk as:

$$
R(\boldsymbol{W}_t \boldsymbol{\Omega}) = \left\| \frac{1}{\sqrt{r_s}} \boldsymbol{W}_t \boldsymbol{\Omega} \boldsymbol{\Omega}^\top \boldsymbol{W}_t^\top - \frac{1}{\|\boldsymbol{\Lambda}\|_{\mathrm{F}}} \boldsymbol{\Theta} \boldsymbol{\Lambda} \boldsymbol{\Theta}^\top \right\|_{\mathrm{F}}^2
$$

$$= \frac{1}{\|\boldsymbol{\Lambda}\|_{\mathrm{F}}^2}\left(\|\boldsymbol{\Lambda}\|_{\mathrm{F}}^2 + \frac{\|\boldsymbol{\Lambda}\|_{\mathrm{F}}^2}{r_s}\|\boldsymbol{\Omega}\boldsymbol{\Omega}^\top\|_F^2 - \frac{2\|\boldsymbol{\Lambda}\|_{\mathrm{F}}}{\sqrt{r_s}}\mathrm{Tr}(\boldsymbol{\Omega}\boldsymbol{\Omega}^\top \boldsymbol{W}_t^\top \boldsymbol{\Theta}\boldsymbol{\Lambda}\boldsymbol{\Theta}^\top \boldsymbol{W}_t) \pm \|\boldsymbol{W}_t^\top \boldsymbol{\Theta}\boldsymbol{\Lambda}\boldsymbol{\Theta}^\top \boldsymbol{W}_t\|_F^2\right)$$

$$= \frac{1}{\|\boldsymbol{\Lambda}\|_{\mathrm{F}}^2}\left(\|\boldsymbol{\Lambda}\|_{\mathrm{F}}^2 - \|\boldsymbol{W}_t^\top \boldsymbol{\Theta}\boldsymbol{\Lambda}\boldsymbol{\Theta}^\top \boldsymbol{W}_t\|_F^2 + \left\|\frac{\|\boldsymbol{\Lambda}\|_{\mathrm{F}}}{\sqrt{r_s}}\boldsymbol{\Omega}\boldsymbol{\Omega}^\top - \boldsymbol{W}_t^\top \boldsymbol{\Theta}\boldsymbol{\Lambda}\boldsymbol{\Theta}^\top \boldsymbol{W}_t\right\|_F^2\right)$$

By observing that $\|\boldsymbol{W}^\top \boldsymbol{\Theta}\boldsymbol{\Lambda}\boldsymbol{\Theta}^\top \boldsymbol{W}\|_F^2 = \|\boldsymbol{\Lambda}^{\frac{1}{2}} \boldsymbol{G}_t \boldsymbol{\Lambda}^{\frac{1}{2}}\|_F^2$, we have the statement. $\qquad\square$

### D.3  Proof of Theorem 1

We let

$$t_{\mathrm{sc}} := t\sqrt{r_s}\|\boldsymbol{\Lambda}\|_{\mathrm{F}}, \quad \kappa_{\mathrm{eff}} := \begin{cases} r^\alpha, & \alpha \in [0,0.5) \\ 1, & \alpha > 0.5 \end{cases}, \qquad \mathsf{T}_{\mathrm{eff}} := \kappa_{\mathrm{eff}}\sqrt{r_s}\|\boldsymbol{\Lambda}\|_{\mathrm{F}} \log {}^d/r_s.$$

and

$$r_u := \begin{cases} r, & \alpha \in [0,0.5) \\ \lceil \log^{2.5} d\rceil, & \alpha > 0.5 \end{cases} \qquad r_{u_\star} := \begin{cases} \lfloor r_s(1 - \log^{-1/8} d) \wedge r\rfloor, & \alpha \in [0,0.5) \\ r_s, & \alpha > 0.5. \end{cases}$$

In the following part, we will establish the high-dimensional limit of the risk curve and the alignment.

#### D.3.1  High-dimensional limit for the alignment

By Lemma 2, we have

$$\boldsymbol{G}_U(t_{\mathrm{sc}}) = \frac{\boldsymbol{I}_r}{\boldsymbol{I}_r - \exp(-t\boldsymbol{\Lambda})} - \frac{\exp(-0.5t\boldsymbol{\Lambda})}{\boldsymbol{I}_r - \exp(-t\boldsymbol{\Lambda})}\left(\boldsymbol{G}_U(0) + \frac{\exp(-t\boldsymbol{\Lambda})}{\boldsymbol{I}_r - \exp(-t\boldsymbol{\Lambda})}\right)^{-1}\frac{\exp(-0.5t\boldsymbol{\Lambda})}{\boldsymbol{I}_r - \exp(-t\boldsymbol{\Lambda})}.$$

We define the block matrix forms

$$\boldsymbol{G}_U(t) =: \begin{bmatrix} \boldsymbol{G}_{U,11}(t) & \boldsymbol{G}_{U,12}(t) \\ \boldsymbol{G}_{U,12}^\top(t) & \boldsymbol{G}_{U,22}(t) \end{bmatrix}, \quad \boldsymbol{\Lambda} = \begin{bmatrix} \boldsymbol{\Lambda}_{\mathrm{eff}} & 0 \\ 0 & \boldsymbol{\Lambda}_{22} \end{bmatrix}, \quad \boldsymbol{\Lambda}_{\mathrm{e},11} := \boldsymbol{\Lambda}_{\mathrm{eff}}, \quad \boldsymbol{\Lambda}_{\mathrm{e},22} := \begin{bmatrix} \boldsymbol{\Lambda}_{22} & 0 \\ 0 & 0 \end{bmatrix},$$

where $\boldsymbol{G}_{U,11}(t), \boldsymbol{\Lambda}_{\mathrm{eff}} \in \mathbb{R}^{r_{u_\star} \times r_{u_\star}}$. The following statement characterizes the time-scales for the alignment terms.

**Proposition 6.** $\mathcal{G}_{init}$ implies that $\mathcal{A}(t\mathsf{T}_{\mathrm{eff}}, \boldsymbol{\theta}_j) = \mathbb{1}\{t\kappa_{\mathrm{eff}} \geq \frac{1}{\lambda_j}\} + o_d(1)$ for $t \neq \lim_{d\to\infty}\frac{1}{\lambda_j \kappa_{\mathrm{eff}}}$ and $j \leq r_{u_\star}$.

*Proof.* For $\alpha = 0$, since the trajectory stays diagonal and the diagonal entries are monotonically increasing, by using the events (E.2) and (H.2) with Lemma 10 we have the result.

In the following, we will prove the result for $\alpha > 0$. By using Proposition 22 with (L.2)

$$\boldsymbol{G}_U(0) \preceq \begin{bmatrix} 2.1\boldsymbol{Z}_{1:r_{u_\star}}\boldsymbol{Z}_{1:r_{u_\star}}^\top & 0 \\ 0 & 2.1\boldsymbol{Z}_2\boldsymbol{Z}_2^\top \end{bmatrix},$$

where

$$2.1\lambda_{\max}(\boldsymbol{Z}_{1:r_{u_\star}}\boldsymbol{Z}_{1:r_{u_\star}}^\top) \leq \begin{cases} 5(1 + \frac{1}{\sqrt{\varphi}})^2, & \alpha \in [0,0.5) \\ 15, & \alpha > 0.5. \end{cases}$$

Therefore,

$$\boldsymbol{G}_{U,11}(t_{\mathrm{sc}}) \preceq \frac{\boldsymbol{I}_{r_{u_\star}}}{\boldsymbol{I}_{r_{u_\star}} - \exp(-t\boldsymbol{\Lambda}_{\mathrm{eff}})}$$

$$- \frac{\exp(-0.5t\boldsymbol{\Lambda}_{\mathrm{eff}})}{\boldsymbol{I}_{r_{u_\star}} - \exp(-t\boldsymbol{\Lambda}_{\mathrm{eff}})}\left(\frac{O(r_s)}{d}\boldsymbol{I}_{r_{u_\star}} + \frac{\exp(-t\boldsymbol{\Lambda}_{\mathrm{eff}})}{\boldsymbol{I}_{r_{u_\star}} - \exp(-t\boldsymbol{\Lambda}_{\mathrm{eff}})}\right)^{-1}\frac{\exp(-0.5t\boldsymbol{\Lambda}_{\mathrm{eff}})}{\boldsymbol{I}_{r_{u_\star}} - \exp(-t\boldsymbol{\Lambda}_{\mathrm{eff}})}.$$

Therefore, by Proposition 36, for $j \leq r_{u_\star}$,

$$\mathcal{A}(t\mathsf{T}_{\mathrm{eff}}, \boldsymbol{\theta}_j) \leq \frac{1}{1 + \left(\frac{d}{r_s}\frac{1}{\log^3 d} - 1\right)\frac{d}{r_s}^{-t\kappa_{\mathrm{eff}}j^{-\alpha}}} = \mathbb{1}\{t\kappa_{\mathrm{eff}} \geq \frac{1}{\lambda_j}\} + o_d(1).$$

Moreover, for $t \leq (r_{u_\star} + 1)^\alpha \log \frac{d}{r_s}$, by using the events (H.3) and (L.2), we have

$$\boldsymbol{Z}_2^\top \exp(t\boldsymbol{\Lambda}_{22})\boldsymbol{Z}_2 \preceq \begin{cases} O_\varphi(1)\boldsymbol{I}_{r_s}, & \alpha \in (0, 0.5) \\ O(\log^{2.5} d)\boldsymbol{I}_{r_s}, & \alpha > 0.5. \end{cases}$$

Therefore, for $t \leq (r_{u_\star} + 1)^\alpha \log \frac{d}{r_s}$, we have $\boldsymbol{G}_{U,11}(t_{\mathrm{sc}}) \succeq \underline{\boldsymbol{G}}(t)$ where

$$\underline{\boldsymbol{G}}(t) := \frac{\boldsymbol{I}_{r_{u_\star}}}{\boldsymbol{I}_{r_{u_\star}} - \exp(-t\boldsymbol{\Lambda}_{\mathrm{eff}})}$$

$$- \frac{\exp(-0.5t\boldsymbol{\Lambda}_{\mathrm{eff}})}{\boldsymbol{I}_{r_{u_\star}} - \exp(-t\boldsymbol{\Lambda}_{\mathrm{eff}})} \left( \frac{r_s}{d} \frac{O(1)}{\log^4 d} \boldsymbol{I}_{r_{u_\star}} + \frac{\exp(-t\boldsymbol{\Lambda}_{\mathrm{eff}})}{\boldsymbol{I}_{r_{u_\star}} - \exp(-t\boldsymbol{\Lambda}_{\mathrm{eff}})} \right)^{-1} \frac{\exp(-0.5t\boldsymbol{\Lambda}_{\mathrm{eff}})}{\boldsymbol{I}_{r_{u_\star}} - \exp(-t\boldsymbol{\Lambda}_{\mathrm{eff}})},$$

which implies that for $t < (r_{u_\star} + 1)^\alpha \log \frac{d}{r_s}$

$$\mathcal{A}(t\mathsf{T}_{\mathrm{eff}}, \boldsymbol{\theta}_j) \geq \frac{1}{1 + O(\log^4 d)\frac{d}{r_s}^{1 - t\kappa_{\mathrm{eff}}j^{-\alpha}}} = \mathbb{1}\{t\kappa_{\mathrm{eff}} \geq \frac{1}{\lambda_j}\} + o_d(1).$$

To extend the lower bound for $t > (r_{u_\star} + 1)^\alpha \log \frac{d}{r_s}$ , let us define

$$t_0 := (r_{u_\star} + 1)^\alpha \log \frac{d}{r_s} \quad \text{and} \quad \boldsymbol{\Lambda}_{\mathrm{eff}}^- := \boldsymbol{\Lambda}_{\mathrm{eff}} - (r_{u_\star} + 1)^{-\alpha}\boldsymbol{I}_{r_{u_\star}}.$$

We have for $t > t_0$,

$$\partial_t \boldsymbol{G}_{U,11}(t) = \frac{0.5}{T_U}\left( \boldsymbol{\Lambda}_{\mathrm{eff}}^- \boldsymbol{G}_{U,11}(t) + \boldsymbol{G}_{U,11}(t)\boldsymbol{\Lambda}_{\mathrm{eff}}^- - 2\boldsymbol{G}_{U,11}(t)\boldsymbol{\Lambda}_{\mathrm{eff}}^- \boldsymbol{G}_{U,11}(t) \right)$$

$$+ \underbrace{\frac{1}{T_U}\left( (r_u + 1)^{-\alpha}\boldsymbol{G}_{U,11}(t)(\boldsymbol{I}_{r_{u_\star}} - \boldsymbol{G}_{U,11}(t)) - \boldsymbol{G}_{U,12}(t)\boldsymbol{\Lambda}_{22}\boldsymbol{G}_{U,12}^\top(t) \right)}_{\succeq \frac{(r_u+1)^{-\alpha}}{T_U}\left( \boldsymbol{G}_{U,11}(t)\left(\boldsymbol{I}_{r_{u_\star}} - \boldsymbol{G}_{U,11}(t)\right) - \boldsymbol{G}_{U,12}(t)\boldsymbol{G}_{U,12}^\top(t)\right) \succeq 0}.$$

Therefore, for $t > t_0$, by monotonicity and [BR14, Theorem 38], $\boldsymbol{G}_{U,11}(t_{\mathrm{sc}}) \succeq \underline{\boldsymbol{G}}(t) \succeq \underline{\boldsymbol{G}}(t_0)$, where

$$\underline{\boldsymbol{G}}(t) = \frac{\boldsymbol{I}_{r_{u_\star}}}{\boldsymbol{I}_{r_{u_\star}} - \exp(-(t - t_0)\boldsymbol{\Lambda}_{\mathrm{eff}}^-)}$$

$$- \frac{\exp(-0.5(t - t_0)\boldsymbol{\Lambda}_{\mathrm{eff}}^-)}{\boldsymbol{I}_{r_{u_\star}} - \exp(-(t - t_0)\boldsymbol{\Lambda}_{\mathrm{eff}}^-)} \left( \underline{\boldsymbol{G}}(t_0) + \frac{\exp(-(t - t_0)\boldsymbol{\Lambda}_{\mathrm{eff}}^-)}{\boldsymbol{I}_{r_{u_\star}} - \exp(-(t - t_0)\boldsymbol{\Lambda}_{\mathrm{eff}}^-)} \right)^{-1} \frac{\exp(-0.5(t - t_0)\boldsymbol{\Lambda}_{\mathrm{eff}}^-)}{\boldsymbol{I}_{r_{u_\star}} - \exp(-(t - t_0)\boldsymbol{\Lambda}_{\mathrm{eff}}^-)}.$$

Therefore, the result extends to $t > t_0$ as well. $\qquad\square$

### D.3.2    High-dimensional limit for the risk curve

For $\mathsf{Err}(t) := \|\boldsymbol{\Lambda}\|_{\mathrm{F}}\left( \frac{\boldsymbol{\Lambda}_e}{\|\boldsymbol{\Lambda}\|_{\mathrm{F}}} - \frac{\boldsymbol{G}_W(t)}{\sqrt{r_s}} \right)$, by Lemma 2, we have

$$\mathsf{Err}(t_{\mathrm{sc}}) = \frac{-\boldsymbol{\Lambda}_e \exp(-t\boldsymbol{\Lambda}_e)}{\boldsymbol{I}_d - \exp(-t\boldsymbol{\Lambda}_e)}$$

$$+ \frac{\boldsymbol{\Lambda}_e \exp(-0.5t\boldsymbol{\Lambda}_e)}{\boldsymbol{I}_d - \exp(-t\boldsymbol{\Lambda}_e)} \left( \frac{\|\boldsymbol{\Lambda}\|_{\mathrm{F}}}{\sqrt{r_s}}\boldsymbol{G}_W(0) + \frac{\boldsymbol{\Lambda}_e \exp(-t\boldsymbol{\Lambda}_e)}{\boldsymbol{I}_d - \exp(-t\boldsymbol{\Lambda}_e)} \right)^{-1} \frac{\boldsymbol{\Lambda}_e \exp(-0.5t\boldsymbol{\Lambda}_e)}{\boldsymbol{I}_d - \exp(-t\boldsymbol{\Lambda}_e)}, \text{(D.1)}$$

We define the block matrix forms

$$\boldsymbol{G}_W(t) = \begin{bmatrix} \boldsymbol{G}_{W,11}(t) & \boldsymbol{G}_{W,12}(t) \\ \boldsymbol{G}_{W,12}^\top(t) & \boldsymbol{G}_{W,22}(t) \end{bmatrix}, \quad \boldsymbol{\Lambda} = \begin{bmatrix} \boldsymbol{\Lambda}_{\mathrm{eff}} & 0 \\ 0 & \boldsymbol{\Lambda}_{22} \end{bmatrix}, \quad \boldsymbol{\Lambda}_{e,11} := \boldsymbol{\Lambda}_{\mathrm{eff}}, \quad \boldsymbol{\Lambda}_{e,22} := \begin{bmatrix} \boldsymbol{\Lambda}_{22} & 0 \\ 0 & 0 \end{bmatrix},$$

where $\boldsymbol{G}_{W,11}(t), \boldsymbol{\Lambda}_{\mathrm{eff}} \in \mathbb{R}^{r_u \times r_u}$. Our proof strategy is as follows: In Proposition 7, we show that the off-diagonal and lower-right terms in (D.1) does not contribute to the high-dimensional limit. Then, in Proposition 8, we characterize the limit of the left-top terms. Finally, in Proposition 9, we prove the asymptotic behaviour of the risk curve.

**Proposition 7.** $\mathcal{G}_{init}$ implies that $\|\boldsymbol{G}_{W,12}(t\mathsf{T}_{\mathrm{eff}})\|_F^2 = o_d(r_s)$ and $\|\boldsymbol{G}_{W,22}(t\mathsf{T}_{\mathrm{eff}})\|_F^2 = o_d(r_s)$.

*Proof.* We let

$$\boldsymbol{D}_1 := \frac{\boldsymbol{\Lambda}_{\mathrm{e},11}\exp(-t\boldsymbol{\Lambda}_{\mathrm{e},11})}{\boldsymbol{I}_{r_u} - \exp(-t\boldsymbol{\Lambda}_{\mathrm{e},11})}, \quad \boldsymbol{D}_2 := \frac{\boldsymbol{\Lambda}_{\mathrm{e},22}\exp(-t\boldsymbol{\Lambda}_{\mathrm{e},22})}{\boldsymbol{I}_{d-r_u} - \exp(-t\boldsymbol{\Lambda}_{\mathrm{e},22})}, \quad \boldsymbol{Z} := \begin{bmatrix} \boldsymbol{Z}_{1:r_u} \\ \boldsymbol{Z}_2 \end{bmatrix}.$$

and

$$\begin{bmatrix} \boldsymbol{S}_{11} & \boldsymbol{S}_{12} \\ \boldsymbol{S}_{12}^\top & \boldsymbol{S}_{22} \end{bmatrix} := \left( \begin{bmatrix} \frac{\|\boldsymbol{\Lambda}\|_{\mathrm{F}}}{\sqrt{r_s}}\boldsymbol{Z}_{1:r_u}\boldsymbol{Z}_{1:r_u}^\top + \boldsymbol{D}_1 & \frac{\|\boldsymbol{\Lambda}\|_{\mathrm{F}}}{\sqrt{r_s}}\boldsymbol{Z}_{1:r_u}\boldsymbol{Z}_2^\top \\ \frac{\|\boldsymbol{\Lambda}\|_{\mathrm{F}}}{\sqrt{r_s}}\boldsymbol{Z}_2\boldsymbol{Z}_{1:r_u}^\top & \frac{\|\boldsymbol{\Lambda}\|_{\mathrm{F}}}{\sqrt{r_s}}\boldsymbol{Z}_2\boldsymbol{Z}_2^\top + \boldsymbol{D}_2 \end{bmatrix} \right)^{-1}.$$

where

$$\boldsymbol{S}_{11} = \left( \boldsymbol{D}_1 + \boldsymbol{Z}_{1:r_u}\left( \frac{\sqrt{r_s}}{\|\boldsymbol{\Lambda}\|_{\mathrm{F}}}\boldsymbol{I}_{r_s} + \boldsymbol{Z}_2^\top \boldsymbol{D}_2^{-1}\boldsymbol{Z}_2 \right)^{-1}\boldsymbol{Z}_{1:r_u}^\top \right)^{-1},$$

$$\boldsymbol{S}_{12} = -\left( \boldsymbol{D}_1 + \boldsymbol{Z}_{1:r_u}\left( \frac{\sqrt{r_s}}{\|\boldsymbol{\Lambda}\|_{\mathrm{F}}}\boldsymbol{I}_{r_s} + \boldsymbol{Z}_2^\top \boldsymbol{D}_2^{-1}\boldsymbol{Z}_2 \right)^{-1}\boldsymbol{Z}_{1:r_u}^\top \right)^{-1}\boldsymbol{Z}_{1:r_u}\boldsymbol{Z}_2^\top \left( \boldsymbol{Z}_2\boldsymbol{Z}_2^\top + \boldsymbol{D}_2 \right)^{-1},$$

$$\boldsymbol{S}_{22} = \left( \boldsymbol{D}_2 + \boldsymbol{Z}_2\left( \frac{\sqrt{r_s}}{\|\boldsymbol{\Lambda}\|_{\mathrm{F}}}\boldsymbol{I}_{r_s} + \boldsymbol{Z}_{1:r_u}^\top \boldsymbol{D}_1^{-1}\boldsymbol{Z}_{1:r_u} \right)^{-1}\boldsymbol{Z}_2^\top \right)^{-1}.$$

**Off-diagonal terms:** By Proposition 26

$$\tilde{\boldsymbol{G}}_{W,12}(t) := \frac{\boldsymbol{\Lambda}_{\mathrm{e},11}\exp(-0.5t\boldsymbol{\Lambda}_{\mathrm{e},11})}{\boldsymbol{I}_{r_u} - \exp(-t\boldsymbol{\Lambda}_{\mathrm{e},11})}\boldsymbol{S}_{12}\frac{\boldsymbol{\Lambda}_{\mathrm{e},22}\exp(-0.5t\boldsymbol{\Lambda}_{\mathrm{e},11})}{\boldsymbol{I}_{d-r_u} - \exp(-t\boldsymbol{\Lambda}_{\mathrm{e},22})}$$

$$= \exp(0.5t\boldsymbol{\Lambda}_{\mathrm{e},11})\boldsymbol{Z}_{1:r_u}\left( \frac{\sqrt{r_s}}{\|\boldsymbol{\Lambda}\|_{\mathrm{F}}}\boldsymbol{I}_{r_s} + \boldsymbol{Z}_2^\top \boldsymbol{D}_2^{-1}\boldsymbol{Z}_2 + \boldsymbol{Z}_{1:r_u}^\top \boldsymbol{D}_1^{-1}\boldsymbol{Z}_{1:r_u} \right)^{-1}\boldsymbol{Z}_2^\top \exp(0.5t\boldsymbol{\Lambda}_{\mathrm{e},22}).$$

We observe that $\lambda_{\max}(\boldsymbol{D}_2) \preceq \frac{1}{t}$ and $\frac{\sqrt{r_s}}{\|\boldsymbol{\Lambda}\|_{\mathrm{F}}} \asymp \kappa_{\mathrm{eff}}$. By using Proposition 27 with $\mathcal{G}_{init}$ and $\tilde{t} := t\kappa_{\mathrm{eff}}\log\frac{d}{r_s}$, we write

$$\frac{1}{r_s}\|\boldsymbol{G}_{W,12}(t\mathsf{T}_{\mathrm{eff}})\|_F^2 = \frac{1}{\|\boldsymbol{\Lambda}\|_{\mathrm{F}}^2}\|\tilde{\boldsymbol{G}}_{W,12}(\tilde{t})\|_F^2$$

$$\leq \frac{1}{\|\boldsymbol{\Lambda}\|_{\mathrm{F}}^2}\frac{O(1)}{(\kappa_{\mathrm{eff}} + \tilde{t})^2}\sum_{i=1}^{r_u \wedge r_s}\left( \lambda_{\max}(\boldsymbol{Z}_{1:r_u}\boldsymbol{Z}_{1:r_u}^\top)\exp(\tilde{t}\lambda_i) \wedge (\kappa_{\mathrm{eff}} + \tilde{t})\frac{\lambda_i\exp(\tilde{t}\lambda_i)}{\exp(\tilde{t}\lambda_i) - 1} \right). \quad \text{(D.2)}$$

For the heavy-tailed case ($\alpha \in [0, 0.5)$),

$$\text{(D.2)} \leq \frac{O_{\alpha,\varphi,\beta}(1)}{r\log^2 d}\sum_{i\leq r}\mathbb{1}\{\tilde{t}\lambda_i \leq \log\frac{d}{r_s}\} + \frac{O_{\alpha,\varphi,\beta}(1)}{r^{1-\alpha}\log d}\sum_{i\leq r}\lambda_i\mathbb{1}\{\tilde{t}\lambda_i > \log\frac{d}{r_s}\} = o_d(1).$$

For the light-tailed case ($\alpha > 0.5$),

$$\text{(D.2)} \leq \frac{O_{\alpha,r_s,\beta}(1)}{\log^2 d}\sum_{i\leq r_s}\mathbb{1}\{\tilde{t}\lambda_i \leq \log\frac{d}{r_u r_s}\} + \frac{O_{\alpha,\varphi,\beta}(1)}{\log d}\sum_{i\leq r_s}\lambda_i\mathbb{1}\{\tilde{t}\lambda_i > \log\frac{d}{r_s r_u}\} = o_d(1).$$

**Lower-right terms:** By using Matrix-Inversion lemma, we have

$$-\boldsymbol{D}_2 + \boldsymbol{D}_2\boldsymbol{S}_{22}\boldsymbol{D}_2 = -\boldsymbol{Z}_2\left( \frac{\sqrt{r_s}}{\|\boldsymbol{\Lambda}\|_{\mathrm{F}}}\boldsymbol{I}_{r_s} + \boldsymbol{Z}_{1:r_u}^\top \boldsymbol{D}_1^{-1}\boldsymbol{Z}_{1:r_u} + \boldsymbol{Z}_2^\top \boldsymbol{D}_2^{-1}\boldsymbol{Z}_2 \right)^{-1}\boldsymbol{Z}_2^\top.$$

We observe that $\lambda_{\max}(\boldsymbol{D}_2) \preceq \frac{1}{t}$ and $\frac{\sqrt{r_s}}{\|\boldsymbol{\Lambda}\|_{\mathrm{F}}} \asymp \kappa_{\mathrm{eff}}$. By using $\tilde{t} := t\kappa_{\mathrm{eff}}\log\frac{d}{r_s}$, we have

$$\frac{1}{\sqrt{r_s}}\boldsymbol{G}_{W,22}(t\mathsf{T}_{\mathrm{eff}}) = \frac{1}{\|\boldsymbol{\Lambda}\|_{\mathrm{F}}}\boldsymbol{Z}_2\left( \frac{\sqrt{r_s}}{\|\boldsymbol{\Lambda}\|_{\mathrm{F}}}\boldsymbol{I}_{r_s} + \boldsymbol{Z}_{1:r_u}^\top \boldsymbol{D}_1^{-1}\boldsymbol{Z}_{1:r_u} + \boldsymbol{Z}_2^\top \boldsymbol{D}_2^{-1}\boldsymbol{Z}_2 \right)^{-1}\boldsymbol{Z}_2^\top$$

$$\preceq \frac{O(1)}{\|\boldsymbol{\Lambda}\|_{\mathrm{F}}} \boldsymbol{Z}_2 \left( \kappa_{\mathrm{eff}} \boldsymbol{I}_{r_s} + \tilde{t} \boldsymbol{Z}_2^\top \boldsymbol{Z}_2 \right)^{-1} \boldsymbol{Z}_2^\top. \tag{D.3}$$

For the heavy-tailed case ($\alpha \in [0, 0.5)$),

$$\|(\text{D.3})\|_F^2 \leq \frac{O_{\alpha,\varphi,\beta}(1)}{r^{-2\alpha} \log^2 d} \frac{1}{t^2 r^{2\alpha} \log^2 d} = o_d(1).$$

For the light-tailed case ($\alpha > 0.5$),

$$\|(\text{D.3})\|_F^2 \leq O_{\alpha,r_s,\beta}(1) \frac{1}{t^2 \log^2 d} = o_d(1).$$

$\square$

**Proposition 8.** *For some $c > 0$, let $\boldsymbol{G}(0) := \frac{c}{t} \boldsymbol{Z}_{1:r_u} \boldsymbol{Z}_{1:r_u}^\top$ and*

$$t \in \begin{cases} \left(0, \frac{(r_{u_\star}+1)^\alpha}{\kappa_{\mathit{eff}}}\right), & \alpha \in [0, 0.5) \\ \left(0, \frac{(r_{u_\star}+1)^\alpha}{\kappa_{\mathit{eff}}}\right) \setminus \{j^\alpha : j \in \mathbb{N}\}, & \alpha > 0.5, \end{cases} \qquad d \geq \begin{cases} \Omega_{\varphi,\alpha}(1) & \alpha \in [0, 0.5) \\ \Omega_{r_s,\alpha}(1), & \alpha > 0.5. \end{cases}$$

*We define*

$$\mathsf{Err}_{r_u}(t_{\mathrm{sc}}) := \frac{-\boldsymbol{\Lambda}_{\mathrm{eff}} \exp(-t\boldsymbol{\Lambda}_{\mathrm{eff}})}{\boldsymbol{I}_{r_u} - \exp(-t\boldsymbol{\Lambda}_{\mathrm{eff}})}$$

$$+ \frac{\boldsymbol{\Lambda}_{\mathrm{eff}} \exp(-0.5t\boldsymbol{\Lambda}_{\mathrm{eff}})}{\boldsymbol{I}_{r_u} - \exp(-t\boldsymbol{\Lambda}_{\mathrm{eff}})} \left( \frac{\boldsymbol{\Lambda}_{\mathrm{eff}} \exp(-t\boldsymbol{\Lambda}_{\mathrm{eff}})}{\boldsymbol{I}_{r_u} - \exp(-t\boldsymbol{\Lambda}_{\mathrm{eff}})} + \boldsymbol{G}(0) \right)^{-1} \frac{\boldsymbol{\Lambda}_{\mathrm{eff}} \exp(-0.5t\boldsymbol{\Lambda}_{\mathrm{eff}})}{\boldsymbol{I}_{r_u} - \exp(-t\boldsymbol{\Lambda}_{\mathrm{eff}})}.$$

$\mathcal{G}_{\mathit{init}}$ *implies that*

$$\frac{\|\mathsf{Err}_{r_u}(t\mathsf{T}_{\mathrm{eff}})\|_F^2}{\|\boldsymbol{\Lambda}\|_{\mathrm{F}}^2} = 1 - \frac{1}{\|\boldsymbol{\Lambda}\|_{\mathrm{F}}^2} \sum_{j=1}^{r_{u_\star}} \lambda_j^2 \mathbb{1}\{t\kappa_{\mathrm{eff}} \geq \tfrac{1}{\lambda_j}\} + o_d(1).$$

*Proof.* We let

$$\boldsymbol{Z}_{1:r_u} \boldsymbol{Z}_{1:r_u}^\top := \begin{bmatrix} \boldsymbol{Z}_{1:r_{u_\star}} \boldsymbol{Z}_{1:r_{u_\star}}^\top & \boldsymbol{Z}_1 \boldsymbol{Z}_2^\top \\ \boldsymbol{Z}_2 \boldsymbol{Z}_1^\top & \boldsymbol{Z}_2 \boldsymbol{Z}_2^\top \end{bmatrix}, \qquad \boldsymbol{\Lambda}_{\mathrm{eff}} := \begin{bmatrix} \boldsymbol{\Lambda}_{\mathrm{eff},11} & 0 \\ 0 & \boldsymbol{\Lambda}_{\mathrm{eff},22} \end{bmatrix},$$

where $\boldsymbol{\Lambda}_{\mathrm{eff},11} \in \mathbb{R}^{r_{u_\star} \times r_{u_\star}}$, $\boldsymbol{Z}_2 \in \mathbb{R}^{(r_u - r_{u_\star}) \times r_s}$. Let

$$\boldsymbol{\Gamma}(t_{\mathrm{sc}}) := \left( \frac{\boldsymbol{\Lambda}_{\mathrm{eff}} \exp(-t\boldsymbol{\Lambda}_{\mathrm{eff}})}{\boldsymbol{I}_{r_u} - \exp(-t\boldsymbol{\Lambda}_{\mathrm{eff}})} + \frac{c}{t} \boldsymbol{Z}_{1:r_u} \boldsymbol{Z}_{1:r_u}^\top \right)^{-1} \quad \text{and} \quad \mathsf{Err}(t_{\mathrm{sc}}, \boldsymbol{\Gamma}) := \mathsf{Err}_{r_u}(t_{\mathrm{sc}}).$$

By using Proposition 22 and the events (H.3) and (L.2),

$$\boldsymbol{\Gamma}(t_{\mathrm{sc}}) \succeq \underline{\boldsymbol{\Gamma}}(t_{\mathrm{sc}}) := \begin{bmatrix} \dfrac{10c}{t} \dfrac{r_s}{d} \boldsymbol{I}_{r_{u_\star}} + \dfrac{\boldsymbol{\Lambda}_{\mathrm{eff},11} \exp(-t\boldsymbol{\Lambda}_{\mathrm{eff},11})}{\boldsymbol{I}_{r_{u_\star}} - \exp(-t\boldsymbol{\Lambda}_{\mathrm{eff},11})} & 0 \\[3ex] 0 & \dfrac{2c}{t} \dfrac{r_s \log^{2.5} d}{d} \boldsymbol{I}_{r_u - r_{u_\star}} + \dfrac{\boldsymbol{\Lambda}_{\mathrm{eff},22} \exp(-t\boldsymbol{\Lambda}_{\mathrm{eff},22})}{\boldsymbol{I}_{r_u - r_{u_\star}} - \exp(-t\boldsymbol{\Lambda}_{\mathrm{eff},22})} \end{bmatrix}^{-1}.$$

For the upper bound, by using $\varepsilon = \frac{1}{c \log^3 d}$ in Proposition 24, and the events (H.1), (L.1) and (H.3), (L.2), we have for $t \leq (r_{u_\star} + 1)^\alpha \log \frac{d}{r_s}$,

$$\boldsymbol{\Gamma}(t_{\mathrm{sc}}) \preceq \overline{\boldsymbol{\Gamma}}(t_{\mathrm{sc}}) := \begin{bmatrix} \dfrac{0.2/t}{\log^4 d} \dfrac{r_s}{d} \boldsymbol{I}_{r_{u_\star}} + \dfrac{\boldsymbol{\Lambda}_{\mathrm{eff},11} \exp(-t\boldsymbol{\Lambda}_{\mathrm{eff},11})}{\boldsymbol{I}_{r_{u_\star}} - \exp(-t\boldsymbol{\Lambda}_{\mathrm{eff},11})} & 0 \\[3ex] 0 & \dfrac{-1.1/t}{\log^{1/2} d} \dfrac{r_s}{d} \boldsymbol{I}_{r_u - r_{u_\star}} + \dfrac{\boldsymbol{\Lambda}_{\mathrm{eff},22} \exp(-t\boldsymbol{\Lambda}_{\mathrm{eff},22})}{\boldsymbol{I}_{r_u - r_{u_\star}} - \exp(-t\boldsymbol{\Lambda}_{\mathrm{eff},22})} \end{bmatrix}^{-1}.$$

Therefore, for $t < \frac{(r_{u_\star}+1)^\alpha}{\kappa_{\mathrm{eff}}}$, by Corollary 8, we have

$$\frac{\|\mathsf{Err}(t\mathsf{T}_{\mathrm{eff}},\boldsymbol{\Gamma})\|_F^2}{\|\boldsymbol{\Lambda}\|_{\mathrm{F}}^2} \geq \frac{\|\mathsf{Err}(t\mathsf{T}_{\mathrm{eff}},\underline{\boldsymbol{\Gamma}})\|_F^2}{\|\boldsymbol{\Lambda}\|_{\mathrm{F}}^2} = \frac{1}{\|\boldsymbol{\Lambda}\|_{\mathrm{F}}^2}\sum_{j=1}^{r_u}\lambda_j^2\mathbb{1}\{\tfrac{1}{\lambda_j} > t\kappa_{\mathrm{eff}}\} + o_d(1)$$

$$= 1 - \frac{1}{\|\boldsymbol{\Lambda}\|_{\mathrm{F}}^2}\sum_{j=1}^{r_{u_\star}}\lambda_j^2\mathbb{1}\{t\kappa_{\mathrm{eff}} \geq \tfrac{1}{\lambda_j}\} + o_d(1).$$

On the other hand, by Corollary 8, we have

$$\frac{\|\mathsf{Err}(t\mathsf{T}_{\mathrm{eff}},\boldsymbol{\Gamma})\|_F^2}{\|\boldsymbol{\Lambda}\|_{\mathrm{F}}^2} \leq \frac{\|\mathsf{Err}(t\mathsf{T}_{\mathrm{eff}},\overline{\boldsymbol{\Gamma}})\|_F^2}{\|\boldsymbol{\Lambda}\|_{\mathrm{F}}^2}$$

$$= \frac{1}{\|\boldsymbol{\Lambda}\|_{\mathrm{F}}^2}\sum_{j=1}^{r_{u_\star}}\lambda_j^2\mathbb{1}\{\tfrac{1}{\lambda_j} > t\kappa_{\mathrm{eff}}\} + \frac{1}{\|\boldsymbol{\Lambda}\|_{\mathrm{F}}^2}\sum_{j=r_{u_\star}+1}^{r_u}\lambda_j^2 + o_d(1)$$

$$= 1 - \frac{1}{\|\boldsymbol{\Lambda}\|_{\mathrm{F}}^2}\sum_{j=1}^{r_{u_\star}}\lambda_j^2\mathbb{1}\{t\kappa_{\mathrm{eff}} \geq \tfrac{1}{\lambda_j}\} + o_d(1).$$

$\square$

**Proposition 9.** *$\mathcal{G}_{init}$ implies that*

$$\frac{\|\mathsf{Err}\,(t\mathsf{T}_{\mathrm{eff}})\|_F^2}{\|\boldsymbol{\Lambda}\|_{\mathrm{F}}^2} = 1 - \frac{1}{\|\boldsymbol{\Lambda}\|_{\mathrm{F}}^2}\sum_{j=1}^{r_{u_\star}}\lambda_j^2\mathbb{1}\{t\kappa_{\mathrm{eff}} \geq \tfrac{1}{\lambda_j}\} + o_d(1),$$

*for*

$$t \in \begin{cases} (0,\infty), & \alpha \in [0,0.5) \\ (0,\infty)\setminus\{j^\alpha : j \in \mathbb{N}\}, & \alpha > 0.5, \end{cases} \quad d \geq \begin{cases} \Omega_{\varphi,\alpha}(1) & \alpha \in [0,0.5) \\ \Omega_{r_s,\alpha}(1), & \alpha > 0.5. \end{cases}$$

*Proof.* We recall that

$$\boldsymbol{D}_1 := \frac{\boldsymbol{\Lambda}_{\mathrm{e},11}\exp(-t\boldsymbol{\Lambda}_{\mathrm{e},11})}{\boldsymbol{I}_{r_u} - \exp(-t\boldsymbol{\Lambda}_{\mathrm{e},11})}, \quad \boldsymbol{D}_2 := \frac{\boldsymbol{\Lambda}_{\mathrm{e},22}\exp(-t\boldsymbol{\Lambda}_{\mathrm{e},22})}{\boldsymbol{I}_{d-r_u} - \exp(-t\boldsymbol{\Lambda}_{\mathrm{e},22})}, \quad \boldsymbol{Z} := \begin{bmatrix} \boldsymbol{Z}_{1:r_u} \\ \boldsymbol{Z}_2 \end{bmatrix},$$

and

$$\mathsf{Err}(t_{\mathrm{sc}}) = \frac{-\boldsymbol{\Lambda}_{\mathrm{e}}\exp(-t\boldsymbol{\Lambda}_{\mathrm{e}})}{\boldsymbol{I}_d - \exp(-t\boldsymbol{\Lambda}_{\mathrm{e}})}$$

$$+ \frac{\boldsymbol{\Lambda}_{\mathrm{e}}\exp(-0.5t\boldsymbol{\Lambda}_{\mathrm{e}})}{\boldsymbol{I}_d - \exp(-t\boldsymbol{\Lambda}_{\mathrm{e}})}\left(\frac{\|\boldsymbol{\Lambda}\|_{\mathrm{F}}}{\sqrt{r_s}}\boldsymbol{Z}\boldsymbol{Z}^\top + \frac{\boldsymbol{\Lambda}_{\mathrm{e}}\exp(-t\boldsymbol{\Lambda}_{\mathrm{e}})}{\boldsymbol{I}_d - \exp(-t\boldsymbol{\Lambda}_{\mathrm{e}})}\right)^{-1}\frac{\boldsymbol{\Lambda}_{\mathrm{e}}\exp(-0.5t\boldsymbol{\Lambda}_{\mathrm{e}})}{\boldsymbol{I}_d - \exp(-t\boldsymbol{\Lambda}_{\mathrm{e}})}$$

$$= \begin{bmatrix} \mathsf{Err}_{r_u}(t_{\mathrm{sc}}) & \frac{-1}{\sqrt{r_s}}\boldsymbol{G}_{W,12}(t_{\mathrm{sc}}) \\ \frac{-1}{\sqrt{r_s}}\boldsymbol{G}_{W,12}(t_{\mathrm{sc}}) & \frac{-1}{\sqrt{r_s}}\boldsymbol{G}_{W,22}(t_{\mathrm{sc}}) \end{bmatrix}.$$

Note that by Proposition 35, in the time scale we consider we have $\frac{1-o_d(1)}{t\mathsf{T}_{\mathrm{eff}}} \leq \lambda_{\min}(\boldsymbol{D}_2) \leq \lambda_{\max}(\boldsymbol{D}_2) \leq \frac{1}{t\mathsf{T}_{\mathrm{eff}}}$. The by using (E.1),

$$\mathsf{Err}_{r_u}(t_{\mathrm{sc}}) = \frac{-\boldsymbol{\Lambda}_{\mathrm{eff}}\exp(-t\boldsymbol{\Lambda}_{\mathrm{eff}})}{\boldsymbol{I}_{r_u} - \exp(-t\boldsymbol{\Lambda}_{\mathrm{eff}})}$$

$$+ \frac{\boldsymbol{\Lambda}_{\mathrm{eff}}\exp(-0.5t\boldsymbol{\Lambda}_{\mathrm{eff}})}{\boldsymbol{I}_{r_u} - \exp(-t\boldsymbol{\Lambda}_{\mathrm{eff}})}\left(\frac{\Theta(1)}{\frac{\|\boldsymbol{\Lambda}\|_{\mathrm{F}}}{\sqrt{r_s}}+t}\boldsymbol{Z}_{1:r_u}\boldsymbol{Z}_{1:r_u}^\top + \frac{\boldsymbol{\Lambda}_{\mathrm{e}}\exp(-t\boldsymbol{\Lambda}_{\mathrm{eff}})}{\boldsymbol{I}_{r_u} - \exp(-t\boldsymbol{\Lambda}_{\mathrm{eff}})}\right)^{-1}\frac{\boldsymbol{\Lambda}_{\mathrm{e}}\exp(-0.5t\boldsymbol{\Lambda}_{\mathrm{eff}})}{\boldsymbol{I}_{r_u} - \exp(-t\boldsymbol{\Lambda}_{\mathrm{eff}})}.$$

By Propositions 7 and 8, we have

$$\frac{\|\mathsf{Err}\,(t\mathsf{T}_{\mathrm{eff}})\|_F^2}{\|\boldsymbol{\Lambda}\|_{\mathrm{F}}^2} = 1 - \frac{1}{\|\boldsymbol{\Lambda}\|_{\mathrm{F}}^2}\sum_{j=1}^{(r_s\wedge r)}\lambda_j^2\mathbb{1}\{t\kappa_{\mathrm{eff}} \geq \tfrac{1}{\lambda_j}\} + o_d(1),$$

for

$$t \in \begin{cases} \left(0, \frac{(r_{u_\star}+1)^\alpha}{\kappa_{\text{eff}}}\right), & \alpha \in [0, 0.5) \\ \left(0, \frac{(r_{u_\star}+1)^\alpha}{\kappa_{\text{eff}}}\right) \setminus \{j^\alpha : j \in \mathbb{N}\}, & \alpha > 0.5. \end{cases} \tag{D.4}$$

To extend the limit for $t > \frac{(r_{u_\star}+1)^\alpha}{\kappa_{\text{eff}}}$, we observe that

- $\|\text{Err}\,(t)\|_F^2$ non increasing since it corresponds to the objective under (GF).

- The global optimum of (GF) and the previous item with (D.4) guarantees that for $t > \frac{(r_{u_\star}+1)^\alpha}{\kappa_{\text{eff}}}$,

$$1 - \frac{1}{\|\mathbf{\Lambda}\|_F^2} \sum_{j=1}^{(r_s \wedge r)} \lambda_j^2 \mathbb{1}\{t\kappa_{\text{eff}} \geq \tfrac{1}{\lambda_j}\} \leq \frac{\|\text{Err}\,(t\mathsf{T}_{\text{eff}})\|_F^2}{\|\mathbf{\Lambda}\|_F^2} \leq 1 - \frac{1}{\|\mathbf{\Lambda}\|_F^2} \sum_{j=1}^{(r_s \wedge r)} \lambda_j^2 \mathbb{1}\{t\kappa_{\text{eff}} \geq \tfrac{1}{\lambda_j}\} + o_d(1).$$

Therefore, the statement extends to $t > \frac{(r_{u_\star}+1)^\alpha}{\kappa_{\text{eff}}}$. $\qquad \square$

### D.4  Proof of Theorem 2

We redefine the time-scale and effective-width as:

$$t_{\text{sc}} = t\sqrt{r_s}\|\mathbf{\Lambda}\|_{\text{F}}, \quad \kappa_{\text{eff}} = \begin{cases} r^\alpha/\eta, & \alpha \in [0, 0.5) \\ 1/\eta, & \alpha > 0.5 \end{cases}, \quad \mathsf{T}_{\text{eff}} = \kappa_{\text{eff}}\sqrt{r_s}\|\mathbf{\Lambda}\|_{\text{F}} \log{d}/r_s.$$

and

$$r_u = \begin{cases} r, & \alpha \in [0, 0.5) \\ \lceil \log^{2.5} d \rceil, & \alpha > 0.5 \end{cases}, \quad r_{u_\star} := \begin{cases} \lfloor r_s(1 - \log^{-1/8} d) \wedge r \rfloor, & \alpha \in [0, 0.5) \\ r_s, & \alpha > 0.5. \end{cases}$$

We consider the learning rate and fine-tuning sample size given as

$$\eta \asymp \frac{1}{d} \begin{cases} \frac{1}{r^\alpha \log^{20}(1+d/r_s)}, & \alpha \in [0, 0.5) \\ \frac{1}{r_u^{4\alpha+3} \log^{18} d}, & \alpha > 0.5 \end{cases} \quad \text{and} \quad N_{\text{Ft}} \asymp r_s^2 \log^5 d.$$

We define the effective learning rate $\eta$ and the hitting time $\mathcal{T}_{\text{hit}}$ as follows:

$$\eta := \frac{\eta/2}{\|\mathbf{\Lambda}\|_{\text{F}}\sqrt{r_s}}, \quad \mathcal{T}_{\text{hit}} := \left\{ t \geq 0 \;\Big|\; 1 - \frac{\|\mathbf{\Lambda}^{\frac{1}{2}}\boldsymbol{G}_t\mathbf{\Lambda}^{\frac{1}{2}}\|_F^2}{\|\mathbf{\Lambda}\|_F^2} \leq \frac{1}{\|\mathbf{\Lambda}\|_F^2} \sum_{j=(r_s \wedge 1)+1}^{r} \lambda_j^2 + \frac{10}{\log^{\frac{1}{8}} d} \right\}.$$

We note that bounding $\mathcal{T}_{\text{hit}}$ suffices to derive sample complexity since by Proposition 11, we have

$$R(\boldsymbol{W}_t^{\text{final}}) \leq 1 - \frac{\|\mathbf{\Lambda}^{\frac{1}{2}}\boldsymbol{G}_t\mathbf{\Lambda}^{\frac{1}{2}}\|_F^2}{\|\mathbf{\Lambda}\|_F^2} + \frac{O(1)}{\log d}.$$

The main statement of this part is as follows:

**Proposition 10.** *The intersection of the following events hold with probability* $1 - o_d(1/d^2) - \Omega(1/r_s^2)$:

*1. We have*

$$\mathcal{T}_{\text{hit}} \leq \begin{cases} \frac{1}{2\eta}\left(r_s(1 - \log^{-1/8} d) \wedge r\right)^\alpha \log\left(\frac{20 d \log^{\frac{3}{4}}(1+d/r_s)}{r_s}\right), & \alpha \in [0, 0.5) \\ \frac{1}{2\eta} r_s^\alpha \log\left(20 \frac{d \log^{3/4} d}{r_s}\right), & \alpha > 0.5. \end{cases}$$

*2. For $t > 0$,*

- $\mathcal{A}(t\mathsf{T}_{\text{eff}}, \boldsymbol{\theta}_j) = \mathbb{1}\{\eta t\kappa_{\text{eff}} \geq \tfrac{1}{\lambda_j}\} + o_d(1)$ *for* $t \neq \lim_{d\to\infty} \frac{1}{\eta\kappa_{\text{eff}}\lambda_j}$ *and* $j \leq r_{u_\star}$.
- $\|\mathbf{\Lambda}\|_F^2 - \|\mathbf{\Lambda}^{\frac{1}{2}}\boldsymbol{G}_{t\mathsf{T}_{\text{eff}}}\mathbf{\Lambda}^{\frac{1}{2}}\|_F^2 = 1 - \sum_{j=1}^{r_{u_\star}} \lambda_j^2 \mathbb{1}\{\eta t\kappa_{\text{eff}} \geq \tfrac{1}{\lambda_j}\} + o_d(\|\mathbf{\Lambda}\|_F^2)$.

*Proof.* By using Lemma 1 and Corollary 5, we have with probability $1 - o_d(1/d^2) - \Omega(1/r_s^2)$:

$$\mathcal{T}_{\text{bad}} \geq \begin{cases} \frac{1}{2\eta}\left(r_s\left(1 - \log^{\frac{-1}{2}}d\right) \wedge r\right)^\alpha \log\left(\frac{d\log^{1.5}d}{r_s}\right), & \alpha \in [0, 0.5) \\ \frac{1}{2\eta}r_s^\alpha \log\left(\frac{d\log^{1.5}d}{r_s}\right), & \alpha > 0.5, \end{cases}$$

where $\mathcal{T}_{\text{bad}}$ is defined in (F.14). Given the lower bound, by Proposition 14, and the third item of Proposition 15, we have the first item.

For the second item, by Proposition 14, and Proposition 15 (for the lower bound) and Proposition 16 (for the upper bound), we have for $r_{u_\star} \times r_{u_\star}$ dimensional top left submatrices $\boldsymbol{G}_{t,11}$ and $\boldsymbol{\Lambda}_{11}$,

$$\frac{1}{\frac{1.2}{C_{\text{lb}}}\frac{d}{r_s}\exp\left(-2t\eta\boldsymbol{\Lambda}_{11}\right) + 1} - o_d(1) \preceq \boldsymbol{G}_{t,11} \preceq \left(\frac{C_{\text{ub}}r_s}{d}\exp\left(2\eta t\boldsymbol{\Lambda}_{11}\right) \wedge 1\right) + o_d(1), \quad \text{(D.5)}$$

and

$$\|\boldsymbol{\Lambda}\|_F^2 - \|\boldsymbol{\Lambda}^{\frac{1}{2}}\boldsymbol{G}_t\boldsymbol{\Lambda}^{\frac{1}{2}}\|_F^2 \geq \sum_{i=1}^{r_u}\lambda_i^2\left(1 - \frac{C_{\text{ub}}r_s}{d}\exp\left(2\eta t\lambda_i\right)\right)_+ - o_d(\|\boldsymbol{\Lambda}\|_F^2)$$

$$\|\boldsymbol{\Lambda}\|_F^2 - \|\boldsymbol{\Lambda}^{\frac{1}{2}}\boldsymbol{G}_t\boldsymbol{\Lambda}^{\frac{1}{2}}\|_F^2 \leq \sum_{i=(r_{u_\star}\wedge r)+1}^{r}\lambda_i^2 + \sum_{i=1}^{r_{u_\star}}\lambda_i^2\left(1 - \frac{1}{\frac{1.2}{C_{\text{lb}}}\frac{d}{r_s}\exp\left(-2t\eta\lambda_i\right) + 1}\right)^2 + o_d(\|\boldsymbol{\Lambda}\|_F^2) \quad \text{(D.6)}$$

for

$$t \leq \begin{cases} \frac{1}{2\eta}\left(r_s\left(1 - \log^{-1/8}d\right) \wedge r\right)^\alpha \log\left(\frac{d\log^{1.5}d}{r_s}\right), & \alpha \in [0, 0.5) \\ \frac{1}{2\eta}r_s^\alpha \log\left(\frac{d\log^{1.5}d}{r_s}\right), & \alpha > 0.5. \end{cases} \quad \text{(D.7)}$$

where

$$C_{\text{ub}} = \begin{cases} 2.5\left(1 + \frac{1}{\sqrt{\varphi}}\right)^2, & \alpha \in [0, 0.5) \\ 15, & \alpha > 0.5, \end{cases} \qquad C_{\text{lb}} = \frac{1}{15}\begin{cases} \log^{-1/2}d, & \alpha \in [0, 0.5) \\ r_s^{-6}, & \alpha > 0.5. \end{cases}$$

The high-dimensional limits of the alignment and risk up to the time horizon in (D.7) follow from (D.5) (for the alignment), and from (D.6) (for the risk) by Proposition 36. Proposition 21 then allows us to extend these results beyond the time limit in (D.7), yielding the full statement. $\square$

### D.5 Proof of Corollary 1 and Corollary 2

Finally, we derive the scaling of prediction risk under power-law second-layer coefficients. Since Corollary 2 is a rescaled version of Corollary 1, we will only consider the latter.

*Proof of Corollary 1.* We will prove heavy and light-tailed cases separately.

**Heavy-tailed case ($\alpha \in [0, 0.5)$):** We define $C := \left(\frac{(1-\beta)\sqrt{\varphi}}{\sqrt{1-2\alpha}}\right)^{\frac{1}{\alpha}}$. We first fix a $(C\varphi)^\alpha > t > 0$. By Proposition 9, for any $d \geq \Omega_{\varphi,\alpha}(1)$, we have with probability at least $1 - o(1/d^2)$

$$\mathcal{R}(tr\log d) = \underbrace{1 - \frac{1}{\|\boldsymbol{\Lambda}\|_F^2}\sum_{i=1}^{r_s}\lambda_j^2\mathbb{1}\{\frac{tr^\alpha}{C^\alpha} \geq \frac{1\pm o_d(1)}{\lambda_j}\}}_{:=\mathcal{R}_d((Ct)^{\frac{1}{\alpha}})} + o_d(1)$$

where we define $\mathcal{R}_d((Ct)^{\frac{1}{\alpha}})$ to isolate the main term and make the dependence on the ambient dimension explicit. By using $\lambda_j = j^{-\alpha}$ in the indicator function, we can rewrite

$$\mathcal{R}_d(t) = 1 - \frac{1}{\|\boldsymbol{\Lambda}\|_F^2}\sum_{i=1}^{r_s}\lambda_j^2\mathbb{1}\{(1 \pm o_d(1))t \geq \frac{j}{r}\}$$

We define a sequence of measures supported on $\{j/r : j \in [r]\}$, where $\mu_d\{\frac{j}{r}\} \propto j^{-2\alpha}$ for $j = 1, \cdots, r$. We observe the following:

- $\mu_d$ converges weakly weakly to a limiting probability measure $\mu$ supported on $[0,1]$, with cumulative distribution function

$$\mu\{[0,c)\} = \begin{cases} c^{1-2\alpha}, & c < 1 \\ 1 & x \geq 1 \end{cases}$$

- Moreover, the risk can be expressed as

$$\mathcal{R}_d(t) = 1 - (1 \pm o_d(1))\mathbb{E}_{X \sim \mu_d}[\mathbb{1}\{(1 \pm o_d(1))t \wedge \varphi \geq X\}]$$

By the Portmanteau theorem [Dur93], it follows that for any fixed $t \in (0, \varphi)$,

$$\mathcal{R}_d(t) \to 1 - t^{1-2\alpha}.$$

almost surely as $d \to \infty$. The almost sure convergence follows from the Borel-Cantelli lemma [Dur93] applied to the failure probabilities.

To extend this result to $t \geq \varphi$, we observe that by (GF), $\mathcal{R}_d(t)$ is non-increasing and $\inf_{t \geq 0} \mathcal{R}_d(t) \geq (1 - \varphi^{1-2\alpha})_+ - o_d(1)$. Hence, for all $t > 0$, we obtain

$$\mathcal{R}_d(t) = (1 - t^{1-2\alpha})_+ \vee (1 - \varphi^{1-2\alpha})_+$$

The desired result for a fixed $t > 0$ follows by a change of variable. Finally, since the risk curves are continuous in $t$, the almost sure convergence extends to all $t > 0$ pointwise.

**Light-tailed case ($\alpha > 0.5$):** For this part, we consider the probability space conditioned on $\mathcal{G}_{init}$ which holds with probability at least $1 - o(1/r_s^2)$. We define

$$\mathcal{Z} := \sum_{j=1}^{\infty} j^{-2\alpha}, \quad C := (r_s \mathcal{Z})^{\frac{1}{2\alpha}}.$$

We first fix a $t \in (0, (Cr_s)^{\alpha}) \setminus \{j^{\alpha} : j \in \mathbb{N}\}$. By Proposition 9, for any $d \geq \Omega_{r_s,\alpha}(1)$, we have ,

$$\mathcal{R}(t \log d) = \underbrace{1 - \frac{1}{\|\mathbf{\Lambda}\|_{\mathrm{F}}^2} \sum_{i=1}^{r_s} \lambda_j^2 \mathbb{1}\{\tfrac{t}{C^{\alpha}} \geq \tfrac{1 \pm o_d(1)}{\lambda_j}\}}_{:= \mathcal{R}_d((Ct)^{\frac{1}{\alpha}})} + o_d(1).$$

By using $\lambda_j = j^{-\alpha}$ in the indicator function, we rewrite

$$\mathcal{R}_d(t) = 1 - \frac{1}{\|\mathbf{\Lambda}\|_{\mathrm{F}}^2} \sum_{i=1}^{r_s} \lambda_j^2 \mathbb{1}\{(1 \pm o_d(1))t \geq j\}$$

We define a sequence of measures supported on $\mathbb{N}$, where $\mu_d\{j\} \propto j^{-2\alpha}$ for $j = 1, \cdots, r_s$. We observe the following:

- $\mu_d$ converges weakly weakly to a limiting probability measure $\mu$ supported on $\mathbb{N}$, such that $\mu\{j\} = \frac{j^{-2\alpha}}{\mathcal{Z}}$.

- Moreover, the risk can be expressed as

$$\mathcal{R}_d(t) = \mathbb{E}_{X \sim \mu_d}[\mathbb{1}\{(1 \pm o_d(1)t \vee r_s < X\}]$$

Since $t \notin \mathbb{N}$, we have

$$\mathbb{R}_d(t) \to \mu([t \vee r_s, \infty)).$$

By observing that $\mu([t, \infty)) \in \Theta(t^{1-2\alpha})$, the result follows for a fixed $t \in (0, (Cr_s)^{\alpha}) \setminus \{j^{\alpha} : j \in \mathbb{N}\}$. Since the limit is piecewise continuous and non increasing, it is sufficient to take a union over $t \in \{0.5, 1.5, \cdots, r_s + 0.5\}$ to extend the result for all $t > 0$. $\qquad \square$

# E  Details of the Fine-tuning Step

In this part, we describe how to efficiently solve the empirical risk minimization problem used in the fine-tuning step of Algorithm 1. Recall that this step aims to find a rotation matrix $\boldsymbol{\Omega} \in \mathbb{R}^{r_s \times r_s}$ that aligns the learned features with the teacher directions by minimizing the empirical loss over $N_{\mathrm{Ft}}$ fresh samples:

$$\boldsymbol{\Omega}_* = \underset{\boldsymbol{\Omega} \in \mathbb{R}^{r_s \times r_s}}{\arg \min} \sum_{j=1}^{N_{\mathrm{Ft}}} \mathcal{L}\big(\boldsymbol{W}_t \boldsymbol{\Omega}; (\boldsymbol{x}_{t+j}, y_{t+j})\big), \tag{E.1}$$

where each sample loss is given by

$$\mathcal{L}\big(\boldsymbol{W}_t \boldsymbol{\Omega}; (\boldsymbol{x}_{t+j}, y_{t+j})\big) = \frac{1}{16}\left(y_{t+j} - \frac{1}{\sqrt{r_s}}\mathrm{Tr}\big(\boldsymbol{\Omega}\boldsymbol{\Omega}^\top \boldsymbol{W}_t^\top (\boldsymbol{x}_{t+j}\boldsymbol{x}_{t+j}^\top - \boldsymbol{I}_d)\boldsymbol{W}_t\big)\right)^2.$$

Let us define $\boldsymbol{A}_j := \boldsymbol{W}_t^\top (\boldsymbol{x}_{t+j}\boldsymbol{x}_{t+j}^\top - \boldsymbol{I}_d)\boldsymbol{W}_t$. We observe that the loss becomes quadratic in the symmetric matrix positive semidefinite matrix $\boldsymbol{S} := \boldsymbol{\Omega}\boldsymbol{\Omega}^\top$. Then, the fine-tuning objective reduces to a standard least squares problem over the cone of symmetric matrix positive semidefinite matrices:

$$\boldsymbol{S}_* := \underset{\substack{\boldsymbol{S} \in \mathbb{R}^{r_s \times r_s} \\ \boldsymbol{S}=\boldsymbol{S}^\top, \boldsymbol{S} \succeq 0}}{\arg \min} \underbrace{\frac{1}{2N_{\mathrm{Ft}}}\sum_{j=1}^{N_{\mathrm{Ft}}}\left(\sqrt{r_s}y_{t+j} - \mathrm{Tr}(\boldsymbol{S}\boldsymbol{A}_j)\right)^2}_{:=\mathrm{Ft}(\boldsymbol{S})}. \tag{E.2}$$

For the following, we also define the global minimum of the least square objective in (E.2) as:

$$\boldsymbol{S}_{\mathrm{glob}} := \underset{\substack{\boldsymbol{S} \in \mathbb{R}^{r_s \times r_s} \\ \boldsymbol{S}=\boldsymbol{S}^\top}}{\arg \min} \mathrm{Ft}(\boldsymbol{S}). \tag{E.3}$$

## E.1  Characterizing the Minimum

Since the fine-tuning objective reduces to a least squares regression problem over symmetric matrices, we can write

$$\mathrm{Ft}(\boldsymbol{S}) = \mathrm{Ft}(\boldsymbol{S}_{\mathrm{glob}}) + \mathrm{Tr}\big((\boldsymbol{S} - \boldsymbol{S}_{\mathrm{glob}})\mathsf{L}(\boldsymbol{S} - \boldsymbol{S}_{\mathrm{glob}})\big)$$

where $\mathsf{L}$ is defined as the linear operator acting on symmetric matrices via

$$\mathsf{L}(\boldsymbol{S}) := \frac{1}{2N_{\mathrm{Ft}}}\sum_{j=1}^{N_{\mathrm{Ft}}}\mathrm{Tr}(\boldsymbol{S}\boldsymbol{A}_j)\boldsymbol{A}_j,$$

which corresponds to the empirical second moment operator associated with the covariates $\boldsymbol{A}_j$. We note that the operator $\mathsf{L}$ is self-adjoint and positive semi-definite on the space of symmetric matrices, and we can write the characterization in (E.2) equivalently

$$\boldsymbol{S}_* := \underset{\substack{\boldsymbol{S} \in \mathbb{R}^{r_s \times r_s} \\ \boldsymbol{S}=\boldsymbol{S}^\top, \boldsymbol{S} \succeq 0}}{\arg \min} \mathrm{Tr}\big((\boldsymbol{S} - \boldsymbol{S}_{\mathrm{glob}})\mathsf{L}(\boldsymbol{S} - \boldsymbol{S}_{\mathrm{glob}})\big).$$

We define the projection on the cone of symmetric positive semi-definite matrices as:

$$\Pi(\widetilde{\boldsymbol{S}}) := \underset{\substack{\boldsymbol{S} \in \mathbb{R}^{r_s \times r_s} \\ \boldsymbol{S}=\boldsymbol{S}^\top, \boldsymbol{S} \succeq 0}}{\arg \min} \|\boldsymbol{S} - \widetilde{\boldsymbol{S}}\|_F^2.$$

In the following, we will show that the operator $\mathsf{L}$ is close to the identity, and thus, $\boldsymbol{S}_*$ is close to $\Pi \circ \mathsf{L}(\boldsymbol{S}_{\mathrm{glob}})$. Before proceeding, we make the following observations:

- We observe that by the first-order optimality condition applied in (E.3), we have

$$\mathsf{L}(\boldsymbol{S}_{\mathrm{glob}}) = \frac{\sqrt{r_s}}{2N_{\mathrm{Ft}}}\sum_{j=1}^{N_{\mathrm{Ft}}} y_{t+j}\boldsymbol{A}_j. \tag{E.4}$$

- By the generalized Pythagorean theorem [Bub14, Lemma 3.1], we have

$$\|\boldsymbol{S}_* - \Pi \circ \mathsf{L}(\boldsymbol{S}_{\mathrm{glob}})\|_F^2 \leq \|\boldsymbol{S}_* - \mathsf{L}(\boldsymbol{S}_{\mathrm{glob}})\|_F^2 - \|\Pi \circ \mathsf{L}(\boldsymbol{S}_{\mathrm{glob}}) - \mathsf{L}(\boldsymbol{S}_{\mathrm{glob}})\|_F^2$$

$$= \mathrm{Ft}(\boldsymbol{S}_*) - \mathrm{Ft}\big(\Pi \circ \mathsf{L}(\boldsymbol{S}_{\mathrm{glob}})\big)$$

$$- \mathrm{Tr}\big((\boldsymbol{S}_* - \Pi \circ \mathsf{L}(\boldsymbol{S}_{\mathrm{glob}}))(\mathsf{L} - \mathsf{Id})(\boldsymbol{S}_* + \Pi \circ \mathsf{L}(\boldsymbol{S}_{\mathrm{glob}}))\big), \tag{E.5}$$

where we use $\mathsf{Id}$ to denote the identity map on symmetric matrices.

## E.2 Computing the Minimum

We define the approximate solution for (E.1) as:

$$\hat{\boldsymbol{\Omega}} := \left( \Pi \circ \mathsf{L}(\boldsymbol{S}_{\mathrm{glob}}) \right)^{\frac{1}{2}}, \tag{E.6}$$

where $\boldsymbol{S} \to \boldsymbol{S}^{1/2}$ denotes the square root operator on symmetric positive semidefinite matrices. Note that the approximation in (E.6) can be computed by taking the spectral decomposition of $\mathsf{L}(\boldsymbol{S}_{\mathrm{glob}})$ given in (E.4), which requires $\tilde{O}(dr_s^3)$ including the computation of $\mathsf{L}(\boldsymbol{S}_{\mathrm{glob}})$. This is negligible compared to the feature learning phase, whose complexity scales as $O(Tdr_s)$. The following statement shows that $\hat{\boldsymbol{\Omega}}$ is sufficiently close to the fine-tuning solution $\boldsymbol{\Omega}^*$:

**Proposition 11.** *Suppose $N_{\mathrm{Ft}} \geq r_s^2 \log^5 d$. Then, with probability at least $1 - 2d^{-3}$, the final risk incurred by $\boldsymbol{W}_t \hat{\boldsymbol{\Omega}}$ is close to that of the optimal fine-tuning solution:*

$$R(\boldsymbol{W}_t \hat{\boldsymbol{\Omega}}) \leq R(\boldsymbol{W}_t \boldsymbol{\Omega}_*) + \frac{1}{\log d} \leq 1 - \frac{\|\boldsymbol{\Lambda}^{\frac{1}{2}} \boldsymbol{G}_t \boldsymbol{\Lambda}^{\frac{1}{2}}\|_F^2}{\|\boldsymbol{\Lambda}\|_F^2} + \frac{O(1)}{\log d}.$$

### E.2.1 Proof of Proposition 11

We define the operator norm of $\mathsf{L}$ as

$$\|\mathsf{L}\|_2 = \sup_{\substack{\boldsymbol{S} \in \mathbb{R}^{r_s \times r_s} \\ \boldsymbol{S} = \boldsymbol{S}^\top}} \|\mathsf{L}(\boldsymbol{S})\|_F.$$

We consider the intersection of the following events:

- $\|\mathsf{L} - \mathsf{Id}\|_2 \leq \frac{6}{\sqrt{\log d}}$

- $\left\| \frac{1}{2N_{\mathrm{Ft}}} \sum_{j=1}^{N_{\mathrm{Ft}}} y_{t+j} \boldsymbol{A}_j - \frac{1}{\|\boldsymbol{\Lambda}\|_F} \boldsymbol{W}_t^\top \boldsymbol{\Theta} \boldsymbol{\Lambda} \boldsymbol{\Theta}^\top \boldsymbol{W}_t \right\|_F^2 \leq \frac{1}{\log d}.$

We note that for $d \geq \Omega(1)$ the first item holds with probability $1 - d^{-3}$ by Proposition 31, where we choose $C = 5$ and $u = \log d$, and the second item holds follows with probability $1 - d^{-3}$ by Proposition 32 where we choose $C = 16$. Given the events, we have

$$
\begin{aligned}
R(\boldsymbol{W}_t \hat{\boldsymbol{\Omega}}) &= \frac{1}{r_s} \left\| \Pi \circ \mathsf{L}(\boldsymbol{S}_{\mathrm{glob}}) - \frac{\sqrt{r_s}}{\|\boldsymbol{\Lambda}\|_F} \boldsymbol{W}_t^\top \boldsymbol{\Theta} \boldsymbol{\Lambda} \boldsymbol{\Theta}^\top \boldsymbol{W}_t \right\|_F^2 + \left( 1 - \frac{\|\boldsymbol{\Lambda}^{\frac{1}{2}} \boldsymbol{G}_t \boldsymbol{\Lambda}^{\frac{1}{2}}\|_F^2}{\|\boldsymbol{\Lambda}\|_F^2} \right) \\
&\overset{(a)}{\leq} \frac{1}{r_s} \left\| \mathsf{L}(\boldsymbol{S}_{\mathrm{glob}}) - \frac{\sqrt{r_s}}{\|\boldsymbol{\Lambda}\|_F} \boldsymbol{W}_t^\top \boldsymbol{\Theta} \boldsymbol{\Lambda} \boldsymbol{\Theta}^\top \boldsymbol{W}_t \right\|_F^2 + \left( 1 - \frac{\|\boldsymbol{\Lambda}^{\frac{1}{2}} \boldsymbol{G}_t \boldsymbol{\Lambda}^{\frac{1}{2}}\|_F^2}{\|\boldsymbol{\Lambda}\|_F^2} \right) \\
&\overset{(b)}{\leq} \frac{1}{\log d} + R(\boldsymbol{W}_t \boldsymbol{\Omega}_*).
\end{aligned}
$$

where we use the convexity of the cone of symmetric positive semi-definite matrices in (a) and the second event above in (b). By using (E.5), we have

$$\|\boldsymbol{S}_* - \Pi \circ \mathsf{L}(\boldsymbol{S}_{\mathrm{glob}})\|_F \leq \|\mathsf{L} - \mathsf{Id}\|_2 \|\boldsymbol{S}_* + \Pi \circ \mathsf{L}(\boldsymbol{S}_{\mathrm{glob}})\|_F.$$

Therefore,

$$\|\boldsymbol{S}_*\|_F \leq \frac{1 + \|\mathsf{L} - \mathsf{Id}\|_2}{1 - \|\mathsf{L} - \mathsf{Id}\|_2} \|\Pi \circ \mathsf{L}(\boldsymbol{S}_{\mathrm{glob}})\|_F,$$

and thus,

$$\|\boldsymbol{S}_* - \Pi \circ \mathsf{L}(\boldsymbol{S}_{\mathrm{glob}})\|_F \leq \frac{2\|\mathsf{L} - \mathsf{Id}\|_2 \|\Pi \circ \mathsf{L}(\boldsymbol{S}_{\mathrm{glob}})\|_F}{1 - \|\mathsf{L} - \mathsf{Id}\|_2} \overset{(c)}{\leq} \frac{15 r_s}{\sqrt{\log d}}$$

where we followed the reasoning in (a)-(b) to bound $\|\Pi \circ \mathsf{L}(\boldsymbol{S}_{\mathrm{glob}})\|_F$ in (c). Therefore,

$$R(\boldsymbol{W}_t \boldsymbol{\Omega}_*) = \frac{1}{r_s} \left\| \boldsymbol{S}_* - \frac{\sqrt{r_s}}{\|\boldsymbol{\Lambda}\|_F} \boldsymbol{W}_t^\top \boldsymbol{\Theta} \boldsymbol{\Lambda} \boldsymbol{\Theta}^\top \boldsymbol{W}_t \right\|_F^2 + \left( 1 - \frac{\|\boldsymbol{\Lambda}^{\frac{1}{2}} \boldsymbol{G}_t \boldsymbol{\Lambda}^{\frac{1}{2}}\|_F^2}{\|\boldsymbol{\Lambda}\|_F^2} \right)$$

$$\leq \frac{2}{r_s} \|\boldsymbol{S}_* - \Pi \circ \mathsf{L}(\boldsymbol{S}_{\mathrm{glob}})\|_F^2 + \frac{2}{r_s} \left\| \Pi \circ \mathsf{L}(\boldsymbol{S}_{\mathrm{glob}}) - \frac{\sqrt{r_s}}{\|\boldsymbol{\Lambda}\|_{\mathrm{F}}} \boldsymbol{W}_t^\top \boldsymbol{\Theta} \boldsymbol{\Lambda} \boldsymbol{\Theta}^\top \boldsymbol{W}_t \right\|_F^2$$

$$+ \left( 1 - \frac{\|\boldsymbol{\Lambda}^{\frac{1}{2}} \boldsymbol{G}_t \boldsymbol{\Lambda}^{\frac{1}{2}}\|_F^2}{\|\boldsymbol{\Lambda}\|_{\mathrm{F}}^2} \right)$$

$$\leq \frac{O(1)}{\log d} + \left( 1 - \frac{\|\boldsymbol{\Lambda}^{\frac{1}{2}} \boldsymbol{G}_t \boldsymbol{\Lambda}^{\frac{1}{2}}\|_F^2}{\|\boldsymbol{\Lambda}\|_{\mathrm{F}}^2} \right).$$

# F  Deferred Proofs for Online SGD

## F.1  Preliminaries

We consider

$$y_{t+1} = \frac{1}{\|\boldsymbol{\Lambda}\|_F} \sum_{j=1}^r \lambda_j \big( \langle \boldsymbol{\theta}_j, \boldsymbol{x}_{t+1} \rangle^2 - 1 \big) + \epsilon_{t+1} \ \text{ and } \ \hat{y}(\boldsymbol{W}_t; \boldsymbol{x}_{t+1}) = \frac{1}{\sqrt{r_s}} \sum_{j=1}^{r_s} \langle \boldsymbol{w}_{t,j}, \boldsymbol{x}_{t+1} \rangle^2 - 1,$$

where $\|\epsilon_{t+1}\|_{\psi_2} \leq \sigma$. We use $\hat{y}_{t+1} := \hat{y}(\boldsymbol{W}_t; \boldsymbol{x}_{t+1})$ and consider

- The loss function is $\mathcal{L}(\boldsymbol{W}_t; (\boldsymbol{x}_{t+1}, y_{t+1})) = \frac{1}{16} \big( y_{t+1} - \hat{y}_{t+1} \big)^2$
- The Euclidean gradient is $\nabla \mathcal{L}(\boldsymbol{W}_t) = \frac{-1}{4\sqrt{r_s}} \big( y_{t+1} - \hat{y}_{t+1} \big) \boldsymbol{x}_{t+1} \boldsymbol{x}_{t+1}^\top \boldsymbol{W}_t$. Therefore, we have

$$\nabla_{\mathrm{St}} \mathcal{L}(\boldsymbol{W}_t) = \frac{-1/4}{\sqrt{r_s}} \big( \boldsymbol{I}_d - \boldsymbol{W}_t \boldsymbol{W}_t^\top \big) \big( y_{t+1} - \hat{y}_{t+1} \big) \boldsymbol{x}_{t+1} \boldsymbol{x}_{t+1}^\top \boldsymbol{W}_t.$$

- We recall that $\boldsymbol{G}_t = \boldsymbol{\Theta}^\top \boldsymbol{W}_t \boldsymbol{W}_t^\top \boldsymbol{\Theta}$.

Then, (SGD) reads

$$\widetilde{\boldsymbol{W}}_{t+1} = \boldsymbol{W}_t + \frac{\eta/4}{\sqrt{r_s}} \underbrace{\big( \boldsymbol{I}_d - \boldsymbol{W}_t \boldsymbol{W}_t^\top \big) \big( y_{t+1} - \hat{y}_{t+1} \big) \boldsymbol{x}_{t+1} \boldsymbol{x}_{t+1}^\top \boldsymbol{W}_t}_{:= \nabla_{\mathrm{St}} \boldsymbol{L}_{t+1}}$$

$$\boldsymbol{W}_{t+1} = \widetilde{\boldsymbol{W}}_{t+1} \left( \boldsymbol{I}_{r_s} + \frac{\eta^2/16}{r_s} \underbrace{\nabla_{\mathrm{St}} \boldsymbol{L}_{t+1}^\top \nabla_{\mathrm{St}} \boldsymbol{L}_{t+1}}_{:= \boldsymbol{\mathcal{P}}_{t+1}} \right)^{-1/2}. \tag{SGD}$$

We observe that

$$\frac{\eta^2/16}{r_s} \boldsymbol{\mathcal{P}}_{t+1} = \frac{\eta^2/16}{r_s} \big( y_{t+1} - \hat{y}_{t+1} \big)^2 \boldsymbol{W}_t^\top \boldsymbol{x}_{t+1} \boldsymbol{x}_{t+1}^\top \big( \boldsymbol{I}_d - \boldsymbol{W}_t \boldsymbol{W}_t^\top \big) \boldsymbol{x}_{t+1} \boldsymbol{x}_{t+1}^\top \boldsymbol{W}_t$$

$$= \frac{\eta^2/16}{r_s} \big( y_{t+1} - \hat{y}_{t+1} \big)^2 \| \big( \boldsymbol{I}_d - \boldsymbol{W}_t \boldsymbol{W}_t^\top \big) \boldsymbol{x}_{t+1} \|_2^2 \boldsymbol{W}_t^\top \boldsymbol{x}_{t+1} \boldsymbol{x}_{t+1}^\top \boldsymbol{W}_t.$$

Let

$$c_{t+1}^2 := \frac{\eta^2/16}{r_s} \|\boldsymbol{\mathcal{P}}_{t+1}\|_2 = \frac{\eta^2/16}{r_s} \big( y_{t+1} - \hat{y}_{t+1} \big)^2 \| \big( \boldsymbol{I}_d - \boldsymbol{W}_t \boldsymbol{W}_t^\top \big) \boldsymbol{x}_{t+1} \|_2^2 \| \boldsymbol{W}_t^\top \boldsymbol{x}_{t+1} \|_2^2.$$

We define $\boldsymbol{P}_{t+1} := \left( \boldsymbol{I}_{r_s} + \frac{\eta^2/16}{r_s} \boldsymbol{\mathcal{P}}_{t+1} \right)^{-1/2}$ and since $\boldsymbol{\mathcal{P}}_{t+1}$ is 1-rank, we have

$$\boldsymbol{P}_{t+1}^2 = \boldsymbol{I}_{r_s} - \frac{\eta^2/16}{r_s} \frac{\boldsymbol{\mathcal{P}}_{t+1}}{1 + c_{t+1}^2}.$$

We let

$$\boldsymbol{M}_t := \boldsymbol{\Theta}^\top \boldsymbol{W}_t \ \text{ and } \ \hat{\boldsymbol{M}}_{t+1} := \boldsymbol{\Theta}^\top \hat{\boldsymbol{W}}_{t+1}.$$

We have

$$\hat{M}_{t+1} = M_t + \frac{\eta/4}{\sqrt{r_s}} \Theta^\top \nabla_{\text{St}} L_{t+1}.$$

By recalling that $G_t = M_t M_t^\top$, we have

$$\begin{aligned}
G_{t+1} &= \hat{M}_{t+1} \hat{M}_{t+1}^\top + \hat{M}_{t+1} (P_{t+1}^2 - I_{r_s}) \hat{M}_{t+1}^\top \\
&= G_t + \frac{\eta/4}{\sqrt{r_s}} M_t \nabla_{\text{St}} L_{t+1}^\top \Theta + \frac{\eta/4}{\sqrt{r_s}} \Theta^\top \nabla_{\text{St}} L_{t+1} M_t^\top + \frac{\eta^2}{16 r_s} \Theta^\top \nabla_{\text{St}} L_{t+1} \nabla_{\text{St}} L_{t+1}^\top \Theta \\
&\quad - \frac{\eta^2}{16 r_s} \frac{\hat{M}_{t+1} P_{t+1} \hat{M}_{t+1}^\top}{1 + c_{t+1}^2}.
\end{aligned}$$

We have

$$\nabla_{\text{St}} L_{t+1} = \frac{2}{\|\Lambda\|_{\text{F}}} \left( I_d - W_t W_t^\top \right) \Theta \Lambda \Theta^\top W_t + \left( \nabla_{\text{St}} L_{t+1} - \mathbb{E}_t \left[ \nabla_{\text{St}} L_{t+1} \right] \right),$$

Therefore,

$$\begin{aligned}
\Theta^\top \nabla_{\text{St}} L_{t+1} M_t^\top &= \frac{2}{\|\Lambda\|_{\text{F}}} \Theta^\top \left( I_d - W_t W_t^\top \right) \Theta \Lambda \Theta^\top W_t W_t^\top \Theta \\
&\quad + \Theta^\top \left( \nabla_{\text{St}} L_{t+1} - \mathbb{E}_t \left[ \nabla_{\text{St}} L_{t+1} \right] \right) M_t^\top \\
&= \frac{2}{\|\Lambda\|_{\text{F}}} \left( I_r - G_t \right) \Lambda G_t + \Theta^\top \left( \nabla_{\text{St}} L_{t+1} - \mathbb{E}_t \left[ \nabla_{\text{St}} L_{t+1} \right] \right) M_t^\top.
\end{aligned}$$

Hence, we have

$$\begin{aligned}
G_{t+1} &= G_t + \frac{\eta/2}{\|\Lambda\|_{\text{F}} \sqrt{r_s}} \left( \Lambda G_t + G_t \Lambda - 2 G_t \Lambda G_t \right) \\
&\quad + \frac{\eta/2}{\sqrt{r_s}} \text{Sym} \left( \Theta^\top \left( \nabla_{\text{St}} L_{t+1} - \mathbb{E}_t \left[ \nabla_{\text{St}} L_{t+1} \right] \right) M_t^\top \right) \\
&\quad + \frac{\eta^2}{16 r_s} \Theta^\top \nabla_{\text{St}} L_{t+1} \nabla_{\text{St}} L_{t+1}^\top \Theta - \frac{\eta^2}{16 r_s} \frac{\hat{M}_{t+1} P_{t+1} \hat{M}_{t+1}^\top}{1 + c_{t+1}^2}
\end{aligned}$$

On the other hand,

$$\begin{aligned}
\hat{M}_{t+1} P_{t+1} \hat{M}_{t+1}^\top &= \left( M_t + \frac{\eta/4}{\sqrt{r_s}} \Theta^\top \nabla_{\text{St}} L_{t+1} \right) P_{t+1} \left( M_t + \frac{\eta/4}{\sqrt{r_s}} \Theta^\top \nabla_{\text{St}} L_{t+1} \right)^\top \\
&= M_t P_{t+1} M_t^\top + \frac{\eta/2}{\sqrt{r_s}} \text{Sym} \left( \Theta^\top \nabla_{\text{St}} L_{t+1} P_{t+1} M_t^\top \right) + \frac{\eta^2}{16 r_s} \Theta^\top \nabla_{\text{St}} L_{t+1} P_{t+1} \nabla_{\text{St}} L_{t+1}^\top \Theta.
\end{aligned}$$

We collect the higher order terms in a single term defined as follows:

$$\begin{aligned}
R_{\text{so}}[G_t] &:= \frac{\eta^2}{16 r_s} \Theta^\top \mathbb{E}_t \left[ \nabla_{\text{St}} L_{t+1} \nabla_{\text{St}} L_{t+1}^\top \right] \Theta - \frac{\eta^2}{16 r_s} M_t \mathbb{E}_t \left[ \frac{P_{t+1}}{1 + c_{t+1}^2} \right] M_t^\top \\
&\quad - \frac{\eta^3}{32 r_s^{3/2}} \text{Sym} \left( \Theta^\top \mathbb{E}_t \left[ \frac{\nabla_{\text{St}} L_{t+1} P_{t+1}}{1 + c_{t+1}^2} \right] M_t^\top \right) \\
&\quad - \frac{\eta^4}{256 r_s^2} \Theta^\top \mathbb{E}_t \left[ \frac{\nabla_{\text{St}} L_{t+1} P_{t+1} \nabla_{\text{St}} L_{t+1}^\top}{1 + c_{t+1}^2} \right] \Theta.
\end{aligned}$$

We collect the noise terms in a single term defined as follows:

$$\begin{aligned}
\frac{\eta/2}{\sqrt{r_s}} \nu_{t+1} &:= \frac{\eta/2}{\sqrt{r_s}} \text{Sym} \left( \Theta^\top \left( \nabla_{\text{St}} L_{t+1} - \mathbb{E}_t \left[ \nabla_{\text{St}} L_{t+1} \right] \right) M_t^\top \right) \\
&\quad - \frac{\eta^2}{16 r_s} M_t \left( \frac{P_{t+1}}{1 + c_{t+1}^2} - \mathbb{E}_t \left[ \frac{P_{t+1}}{1 + c_{t+1}^2} \right] \right) M_t^\top
\end{aligned}$$

$$+ \frac{\eta^2}{16 r_s} \boldsymbol{\Theta}^\top \left( \nabla_{\mathrm{St}} \boldsymbol{L}_{t+1} \nabla_{\mathrm{St}} \boldsymbol{L}_{t+1}^\top - \mathbb{E}_t \left[ \nabla_{\mathrm{St}} \boldsymbol{L}_{t+1} \nabla_{\mathrm{St}} \boldsymbol{L}_{t+1}^\top \right] \right) \boldsymbol{\Theta}$$

$$- \frac{\eta^3}{32 r_s^{3/2}} \mathrm{Sym} \left( \boldsymbol{\Theta}^\top \left( \frac{\nabla_{\mathrm{St}} \boldsymbol{L}_{t+1} \boldsymbol{\mathcal{P}}_{t+1}}{1 + c_{t+1}^2} - \mathbb{E}_t \left[ \frac{\nabla_{\mathrm{St}} \boldsymbol{L}_{t+1} \boldsymbol{\mathcal{P}}_{t+1}}{1 + c_{t+1}^2} \right] \right) \boldsymbol{M}_t^\top \right)$$

$$- \frac{\eta^4}{256 r_s^2} \boldsymbol{\Theta}^\top \left( \frac{\nabla_{\mathrm{St}} \boldsymbol{L}_{t+1} \boldsymbol{\mathcal{P}}_{t+1} \nabla_{\mathrm{St}} \boldsymbol{L}_{t+1}^\top}{1 + c_{t+1}^2} - \mathbb{E}_t \left[ \frac{\nabla_{\mathrm{St}} \boldsymbol{L}_{t+1} \boldsymbol{\mathcal{P}}_{t+1} \nabla_{\mathrm{St}} \boldsymbol{L}_{t+1}^\top}{1 + c_{t+1}^2} \right] \right) \boldsymbol{\Theta}.$$

With these definitions in hand, we have

$$\boldsymbol{G}_{t+1} = \boldsymbol{G}_t + \frac{\eta/2}{\|\boldsymbol{\Lambda}\|_{\mathrm{F}} \sqrt{r_s}} \left( \boldsymbol{\Lambda} \boldsymbol{G}_t + \boldsymbol{G}_t \boldsymbol{\Lambda} - 2 \boldsymbol{G}_t \boldsymbol{\Lambda} \boldsymbol{G}_t \right) + R_{\mathrm{so}}[\boldsymbol{G}_t] + \frac{\eta/2}{\sqrt{r_s}} \boldsymbol{\nu}_{t+1}.$$

## F.2 Including second-order terms and monotone bounds

For $C > 1$, we define

$$\boldsymbol{\Lambda}_{\ell_1} := \boldsymbol{\Lambda} - C\|\boldsymbol{\Lambda}\|_{\mathrm{F}} \frac{\eta d}{\sqrt{r_s}} \boldsymbol{I}_r \text{ and } \boldsymbol{\Lambda}_{u_1} := \boldsymbol{\Lambda} + C\|\boldsymbol{\Lambda}\|_{\mathrm{F}} \frac{\eta d}{\sqrt{r_s}} \boldsymbol{I}_r. \tag{F.2}$$

We recall the definition of effective learning rate $\eta = \frac{\eta/2}{\|\boldsymbol{\Lambda}\|_{\mathrm{F}} \sqrt{r_s}}$. By Proposition 17, we have

$$\boldsymbol{G}_{t+1} \succeq \boldsymbol{G}_t + \eta \left( \boldsymbol{\Lambda}_{\ell_1} \boldsymbol{G}_t + \boldsymbol{G}_t \boldsymbol{\Lambda}_{\ell_1} - 2 \boldsymbol{G}_t \boldsymbol{\Lambda} \boldsymbol{G}_t \right) - \frac{C}{2} \eta^2 \|\boldsymbol{\Lambda}\|_{\mathrm{F}}^2 r_s \boldsymbol{I}_r + \frac{\eta/2}{\sqrt{r_s}} \boldsymbol{\nu}_{t+1}, \tag{F.3}$$

$$\boldsymbol{G}_{t+1} \preceq \boldsymbol{G}_t + \eta \left( \boldsymbol{\Lambda}_{u_1} \boldsymbol{G}_t + \boldsymbol{G}_t \boldsymbol{\Lambda}_{u_1} - 2 \boldsymbol{G}_t \boldsymbol{\Lambda} \boldsymbol{G}_t \right) + \frac{C}{2} \eta^2 \|\boldsymbol{\Lambda}\|_{\mathrm{F}}^2 r_s \boldsymbol{I}_r + \frac{\eta/2}{\sqrt{r_s}} \boldsymbol{\nu}_{t+1}. \tag{F.4}$$

### F.2.1 Heavy tailed case - $\alpha \in [0, 0.5)$

**Proposition 12.** *We consider* $\alpha \in [0, 0.5)$, $\frac{r_s}{r} \to (0, \infty]$ *and* $\eta \ll \frac{1}{d \log^4 d} \sqrt{\frac{r_s}{r}}$. *We define*

$$\boldsymbol{V}_t^- := 2 \boldsymbol{\Lambda}^{\frac{1}{2}} \boldsymbol{G}_t \boldsymbol{\Lambda}^{\frac{1}{2}} - \boldsymbol{\Lambda}_{\ell_1} \text{ and } \boldsymbol{V}_t^+ := 2 \boldsymbol{\Lambda}^{\frac{1}{2}} \boldsymbol{G}_t \boldsymbol{\Lambda}^{\frac{1}{2}} - \boldsymbol{\Lambda}_{u_1}.$$

*For* $d \geq \Omega(1)$, *we have*

$$\boldsymbol{\Lambda} + \frac{0.1 r^{-\alpha}}{\log^4 d} \boldsymbol{I}_r \succ \boldsymbol{\Lambda}_{u_1} \succ \boldsymbol{\Lambda} \succ \boldsymbol{\Lambda}_{\ell_1} \succ \boldsymbol{\Lambda} - \frac{0.1 r^{-\alpha}}{\log^4 d} \boldsymbol{I}_r \tag{F.5}$$

*and*

$$\boldsymbol{V}_{t+1}^- \succeq \boldsymbol{V}_t^- \left( \boldsymbol{I}_r + \frac{\eta}{1 - 1.1\eta} \boldsymbol{V}_t^- \right)^{-1} + \eta \boldsymbol{\Lambda}_{\ell_1}^2 - C\eta^2 \|\boldsymbol{\Lambda}\|_{\mathrm{F}}^2 r_s \boldsymbol{\Lambda} + \frac{\eta}{\sqrt{r_s}} \boldsymbol{\Lambda}^{\frac{1}{2}} \boldsymbol{\nu}_{t+1} \boldsymbol{\Lambda}^{\frac{1}{2}}$$

$$\boldsymbol{V}_{t+1}^+ \preceq \boldsymbol{V}_t^+ \left( \boldsymbol{I}_r + \frac{\eta}{1 + 1.1\eta} \boldsymbol{V}_t^+ \right)^{-1} + \eta \boldsymbol{\Lambda}_{u_1}^2 + C\eta^2 \|\boldsymbol{\Lambda}\|_{\mathrm{F}}^2 r_s \boldsymbol{\Lambda} + \frac{\eta}{\sqrt{r_s}} \boldsymbol{\Lambda}^{\frac{1}{2}} \boldsymbol{\nu}_{t+1} \boldsymbol{\Lambda}^{\frac{1}{2}} \tag{F.6}$$

*where the bounding iterations are monotone in the sense defined in Proposition 4.*

*Proof.* We first note that since $\|\boldsymbol{\Lambda}\|_{\mathrm{F}} \asymp r^{\frac{1}{2} - \alpha}$ for $\alpha \in [0, 0.5)$, we have

$$\|\boldsymbol{\Lambda}\|_{\mathrm{F}} \frac{\eta d}{\sqrt{r_s}} \ll \frac{r^{-\alpha}}{\log^4 d}.$$

Therefore, (F.5) holds for $d \geq \Omega(1)$, which implies

$$\|\boldsymbol{V}_t^-\|_2 \vee \|\boldsymbol{V}_t^+\|_2 \leq 1 + \frac{0.1 r^{-\alpha}}{\log^4 d}, \text{ for all } t \in \mathbb{N}. \tag{F.7}$$

Therefore, the monotonicity follows from Proposition 4.

For the remaining part, we introduce the following notation, $\boldsymbol{K}_t := \boldsymbol{\Lambda}^{\frac{1}{2}} \boldsymbol{G}_t \boldsymbol{\Lambda}^{\frac{1}{2}}$. For the lower bound, by (F.3), we have

$$\boldsymbol{K}_{t+1} \succeq \boldsymbol{K}_t + \frac{\eta}{2} \left( \boldsymbol{\Lambda}_{\ell_1}^2 - (2\boldsymbol{K}_t - \boldsymbol{\Lambda}_{\ell_1})^2 \right) - \frac{C}{2} \eta^2 \|\boldsymbol{\Lambda}\|_{\mathrm{F}}^2 r_s \boldsymbol{\Lambda} + \frac{\eta/2}{\sqrt{r_s}} \boldsymbol{\Lambda}^{\frac{1}{2}} \boldsymbol{\nu}_{t+1} \boldsymbol{\Lambda}^{\frac{1}{2}}.$$

By multiplying both sides with 2 and subtracting $\boldsymbol{\Lambda}_{\ell_1}$ from both sides, we have

$$\boldsymbol{V}_{t+1}^{-} \succeq \boldsymbol{V}_t^{-} - \eta(\boldsymbol{V}_t^{-})^2 + \eta\boldsymbol{\Lambda}_{\ell_1}^2 - C\eta^2\|\boldsymbol{\Lambda}\|_{\mathrm{F}}^2 r_s\boldsymbol{\Lambda} + \frac{\eta}{\sqrt{r_s}}\boldsymbol{\Lambda}^{\frac{1}{2}}\boldsymbol{\nu}_{t+1}\boldsymbol{\Lambda}^{\frac{1}{2}}$$

$$\overset{(a)}{\succeq} \boldsymbol{V}_t^{-} - \frac{\eta}{1-1.1\eta}(\boldsymbol{V}_t^{-})^2\left(\boldsymbol{I}_r + \frac{\eta}{1-1.1\eta}\boldsymbol{V}_t^{-}\right)^{-1} + \eta\boldsymbol{\Lambda}_{\ell_1}^2$$

$$- C\eta^2\|\boldsymbol{\Lambda}\|_{\mathrm{F}}^2 r_s\boldsymbol{\Lambda} + \frac{\eta}{\sqrt{r_s}}\boldsymbol{\Lambda}^{\frac{1}{2}}\boldsymbol{\nu}_{t+1}\boldsymbol{\Lambda}^{\frac{1}{2}}$$

$$= \boldsymbol{V}_t^{-}\left(\boldsymbol{I}_r + \frac{\eta}{1-1.1\eta}\boldsymbol{V}_t^{-}\right)^{-1} + \eta\boldsymbol{\Lambda}_{\ell_1}^2 - C\eta^2\|\boldsymbol{\Lambda}\|_{\mathrm{F}}^2 r_s\boldsymbol{\Lambda} + \frac{\eta}{\sqrt{r_s}}\boldsymbol{\Lambda}^{\frac{1}{2}}\boldsymbol{\nu}_{t+1}\boldsymbol{\Lambda}^{\frac{1}{2}},$$

where we used (F.7) for (a).

For the upper bound, by (F.4), we have

$$\boldsymbol{K}_{t+1} \preceq \boldsymbol{K}_t + \frac{\eta}{2}\left(\boldsymbol{\Lambda}_{u_1}^2 - (2\boldsymbol{K}_t - \boldsymbol{\Lambda}_{u_1})^2\right) + \frac{C}{2}\eta^2\|\boldsymbol{\Lambda}\|_{\mathrm{F}}^2 r_s\boldsymbol{\Lambda} + \frac{\eta/2}{\sqrt{r_s}}\boldsymbol{\Lambda}^{\frac{1}{2}}\boldsymbol{\nu}_{t+1}\boldsymbol{\Lambda}^{\frac{1}{2}}.$$

By multiplying both sides with 2 and subtracting $\boldsymbol{\Lambda}_{u_1}$ from both sides, we get

$$\boldsymbol{V}_{t+1}^{+} \preceq \boldsymbol{V}_t^{+} - \eta(\boldsymbol{V}_t^{+})^2 + \eta\boldsymbol{\Lambda}_{u_1}^2 + C\eta^2\|\boldsymbol{\Lambda}\|_{\mathrm{F}}^2 r_s\boldsymbol{\Lambda} + \frac{\eta}{\sqrt{r_s}}\boldsymbol{\Lambda}^{\frac{1}{2}}\boldsymbol{\nu}_{t+1}\boldsymbol{\Lambda}^{\frac{1}{2}}$$

$$\overset{(b)}{\preceq} \boldsymbol{V}_t^{+} - \frac{\eta}{1+1.1\eta}(\boldsymbol{V}_t^{+})^2\left(\boldsymbol{I}_r + \frac{\eta}{1+1.1\eta}\boldsymbol{V}_t^{+}\right)^{-1} + \eta\boldsymbol{\Lambda}_{u_1}^2$$

$$+ C\eta^2\|\boldsymbol{\Lambda}\|_{\mathrm{F}}^2 r_s\boldsymbol{\Lambda} + \frac{\eta}{\sqrt{r_s}}\boldsymbol{\Lambda}^{\frac{1}{2}}\boldsymbol{\nu}_{t+1}\boldsymbol{\Lambda}^{\frac{1}{2}}$$

$$= \boldsymbol{V}_t^{+}\left(\boldsymbol{I}_r + \frac{\eta}{1+1.1\eta}\boldsymbol{V}_t^{+}\right)^{-1} + \eta\boldsymbol{\Lambda}_{u_1}^2 + C\eta^2\|\boldsymbol{\Lambda}\|_{\mathrm{F}}^2 r_s\boldsymbol{\Lambda} + \frac{\eta}{\sqrt{r_s}}\boldsymbol{\Lambda}^{\frac{\dot{u}_2}{2}}\boldsymbol{\nu}_{t+1}\boldsymbol{\Lambda}^{\frac{\dot{u}_2}{2}},$$

where we used (F.7) for (b). $\qquad\qquad\square$

### F.2.2  Light tailed case - $\alpha > 0.5$

We introduce the submatrix notation

$$\boldsymbol{G}_t =: \begin{bmatrix} \boldsymbol{G}_{t,11} & \boldsymbol{G}_{t,12} \\ \boldsymbol{G}_{t,12}^{\top} & \boldsymbol{G}_{t,22} \end{bmatrix} \quad \boldsymbol{\nu}_t =: \begin{bmatrix} \boldsymbol{\nu}_{t,11} & \boldsymbol{\nu}_{t,12} \\ \boldsymbol{\nu}_{t,12}^{\top} & \boldsymbol{\nu}_{t,22} \end{bmatrix} \quad \boldsymbol{\Lambda} =: \begin{bmatrix} \boldsymbol{\Lambda}_{11} & 0 \\ 0 & \boldsymbol{\Lambda}_{22} \end{bmatrix} \quad \boldsymbol{\Lambda}_{\ell_1} =: \begin{bmatrix} \boldsymbol{\Lambda}_{\ell_1,11} & 0 \\ 0 & \boldsymbol{\Lambda}_{\ell_1,22} \end{bmatrix},$$

where $\boldsymbol{G}_{t,11}, \boldsymbol{\nu}_{t,11}, \boldsymbol{\Lambda}_{11}, \boldsymbol{\Lambda}_{\ell_1,11} \in \mathbb{R}^{r_u \times r_u}$ for $r_u < r$. Similarly, we define the block matrices of $\boldsymbol{\Lambda}_{u_1}$ as $\boldsymbol{\Lambda}_{u_1,11} \in \mathbb{R}^{r_u \times r_u}$ and $\boldsymbol{\Lambda}_{u_1,22}$. We can write iterations (F.3) and (F.4) for the left-top submatrix as:

$$\boldsymbol{G}_{t+1,11} \succeq \boldsymbol{G}_{t,11} + \eta\left(\boldsymbol{\Lambda}_{\ell_1,11}\boldsymbol{G}_{t,11} + \boldsymbol{G}_{t,11}\boldsymbol{\Lambda}_{\ell_1,11} - 2\boldsymbol{G}_{t,11}\boldsymbol{\Lambda}_{11}\boldsymbol{G}_{t,11} - 2\boldsymbol{G}_{t,12}\boldsymbol{\Lambda}_{22}\boldsymbol{G}_{t,12}^{\top}\right)$$

$$- \frac{C}{2}\eta^2\|\boldsymbol{\Lambda}\|_{\mathrm{F}}^2 r_s\boldsymbol{I}_{r_u} + \frac{\eta/2}{\sqrt{r_s}}\boldsymbol{\nu}_{t+1,11} \tag{F.8}$$

$$\boldsymbol{G}_{t+1,11} \preceq \boldsymbol{G}_{t,11} + \eta\left(\boldsymbol{\Lambda}_{u_1,11}\boldsymbol{G}_{t,11} + \boldsymbol{G}_{t,11}\boldsymbol{\Lambda}_{u_1,11} - 2\boldsymbol{G}_{t,11}\boldsymbol{\Lambda}_{11}\boldsymbol{G}_{t,11} - 2\boldsymbol{G}_{t,12}\boldsymbol{\Lambda}_{22}\boldsymbol{G}_{t,12}^{\top}\right)$$

$$+ \frac{C}{2}\eta^2\|\boldsymbol{\Lambda}\|_{\mathrm{F}}^2 r_s\boldsymbol{I}_{r_u} + \frac{\eta/2}{\sqrt{r_s}}\boldsymbol{\nu}_{t+1,11}. \tag{F.9}$$

The following statement is analogous to Proposition 12 in the case $\alpha > 0.5$.

**Proposition 13.** *We consider $\alpha > 0.5$, $r_s \asymp 1$, and*

$$\eta \ll \frac{1}{d\log^3 d}\frac{1}{r_u^{2+\alpha}} \quad \text{and} \quad r_u = \lceil\log^{2.5} d\rceil.$$

*We define $\boldsymbol{V}_t^{+} := 2\boldsymbol{\Lambda}_{11}^{\frac{1}{2}}\boldsymbol{G}_{t,11}\boldsymbol{\Lambda}_{11}^{\frac{1}{2}} - \boldsymbol{\Lambda}_{u_1,11}$ and*

$$\boldsymbol{V}_t^{-} := 2\left(\boldsymbol{\Lambda}_{11} - \frac{1}{(r_u+1)^{\alpha}}\boldsymbol{I}_{r_u}\right)^{\frac{1}{2}}\boldsymbol{G}_{t,11}\left(\boldsymbol{\Lambda}_{11} - \frac{1}{(r_u+1)^{\alpha}}\boldsymbol{I}_{r_u}\right)^{\frac{1}{2}} - \left(\boldsymbol{\Lambda}_{\ell_1,11} - \frac{1}{(r_u+1)^{\alpha}}\boldsymbol{I}_{r_u}\right).$$

*For $d \geq \Omega(1)$, we have*

$$\mathbf{\Lambda}_{11} + \tfrac{0.1}{r_u^{2+\alpha}\log^3 d}\boldsymbol{I}_{r_u} \succ \mathbf{\Lambda}_{u_1,11} \succ \mathbf{\Lambda}_{11} \succ \mathbf{\Lambda}_{\ell_1,11} \succ \mathbf{\Lambda}_{11} - \tfrac{0.1}{r_u^{2+\alpha}\log^3 d}\boldsymbol{I}_{r_u} \qquad (\text{F.10})$$

*and*

$$\boldsymbol{V}_{t+1}^{-} \succeq \boldsymbol{V}_t^{-}\Big(\boldsymbol{I}_{r_u} + \tfrac{\eta}{1-1.1\eta}\boldsymbol{V}_t^{-}\Big)^{-1} + \eta\Big(\mathbf{\Lambda}_{\ell_1,11} - \tfrac{1}{(r_u+1)^\alpha}\boldsymbol{I}_{r_u}\Big)^2 - C\eta^2\|\mathbf{\Lambda}\|_{\mathrm{F}}^2 r_s\Big(\mathbf{\Lambda}_{11} - \tfrac{1}{(r_u+1)^\alpha}\boldsymbol{I}_{r_u}\Big)$$

$$+ \frac{\eta}{\sqrt{r_s}}\Big(\mathbf{\Lambda}_{11} - \tfrac{1}{(r_u+1)^\alpha}\boldsymbol{I}_{r_u}\Big)^{\frac{1}{2}}\boldsymbol{\nu}_{t+1,11}\Big(\mathbf{\Lambda}_{11} - \tfrac{1}{(r_u+1)^\alpha}\boldsymbol{I}_{r_u}\Big)^{\frac{1}{2}}$$

$$\boldsymbol{V}_{t+1}^{+} \preceq \boldsymbol{V}_t^{+}\Big(\boldsymbol{I}_{r_u} + \tfrac{\eta}{1+1.1\eta}\boldsymbol{V}_t^{+}\Big)^{-1} + \eta\mathbf{\Lambda}_{u_1,11}^2 + C\eta^2\|\mathbf{\Lambda}\|_{\mathrm{F}}^2 r_s\mathbf{\Lambda}_{11} + \frac{\eta}{\sqrt{r_s}}\mathbf{\Lambda}_{11}^{\frac{1}{2}}\boldsymbol{\nu}_{t+1,11}\mathbf{\Lambda}_{11}^{\frac{1}{2}}.$$

*where the bounding iterations are monotone in the sense defined in Proposition 4.*

**Proof.** We first note that since $r_s \asymp 1$ and $\|\mathbf{\Lambda}\|_{\mathrm{F}} \asymp 1$ for $\alpha > 0.5$, we have

$$\|\mathbf{\Lambda}\|_{\mathrm{F}}\frac{\eta d}{\sqrt{r_s}} \ll \frac{1}{r_u^{2+\alpha}\log^3 d}.$$

Therefore, (F.10) holds for $d \geq \Omega(1)$, which implies

$$\|\boldsymbol{V}_t^{-}\|_2 \vee \|\boldsymbol{V}_t^{+}\|_2 \leq 1 + \frac{0.1}{r_u^{2+\alpha}\log^3 d} + \frac{1}{(r_u+1)^\alpha}, \quad \text{for all } t \in \mathbb{N}. \qquad (\text{F.11})$$

For the remaining part, we introduce the following notation,

$$\boldsymbol{K}_t^{-} := \Big(\mathbf{\Lambda}_{11} - \tfrac{1}{(r_u+1)^\alpha}\boldsymbol{I}_{r_u}\Big)^{\frac{1}{2}}\boldsymbol{G}_{t,11}\Big(\mathbf{\Lambda}_{11} - \tfrac{1}{(r_u+1)^\alpha}\boldsymbol{I}_{r_u}\Big)^{\frac{1}{2}} \quad \text{and} \quad \boldsymbol{K}_t^{+} := \mathbf{\Lambda}_{11}^{\frac{1}{2}}\boldsymbol{G}_{t,11}\mathbf{\Lambda}_{11}^{\frac{1}{2}}.$$

For the upper bound, since $\mathbf{\Lambda}_{22} \succ 0$, by (F.9) we have

$$\boldsymbol{K}_{t+1}^{+} \preceq \boldsymbol{K}_t^{+} + \frac{\eta}{2}\big(\mathbf{\Lambda}_{u_1,11}^2 - (2\boldsymbol{K}_t^{+} - \mathbf{\Lambda}_{u_1,11})^2\big) + \frac{C}{2}\eta^2\|\mathbf{\Lambda}\|_{\mathrm{F}}^2 r_s\mathbf{\Lambda}_{11} + \frac{\eta/2}{\sqrt{r_s}}\mathbf{\Lambda}_{11}^{\frac{1}{2}}\boldsymbol{\nu}_{t+1,11}\mathbf{\Lambda}_{11}^{\frac{1}{2}}.$$

By multiplying both sides with 2 and subtracting $\mathbf{\Lambda}_{u_1,11}$ from both sides, we get

$$\boldsymbol{V}_{t+1}^{+} \preceq \boldsymbol{V}_t^{+} - \eta(\boldsymbol{V}_t^{+})^2 + \eta\mathbf{\Lambda}_{u_1,11}^2 + C\eta^2\|\mathbf{\Lambda}\|_{\mathrm{F}}^2 r_s\mathbf{\Lambda}_{11} + \frac{\eta}{\sqrt{r_s}}\mathbf{\Lambda}_{11}^{\frac{1}{2}}\boldsymbol{\nu}_{t+1,11}\mathbf{\Lambda}_{11}^{\frac{1}{2}}$$

$$\overset{(a)}{\preceq} \boldsymbol{V}_t^{+} - \frac{\eta}{1+1.1\eta}(\boldsymbol{V}_t^{+})^2\Big(\boldsymbol{I}_{r_u} + \frac{\eta}{1+1.1\eta}\boldsymbol{V}_t^{+}\Big)^{-1} + \eta\mathbf{\Lambda}_{u_1,11}^2$$

$$+ C\eta^2\|\mathbf{\Lambda}\|_{\mathrm{F}}^2 r_s\mathbf{\Lambda}_{11} + \frac{\eta}{\sqrt{r_s}}\mathbf{\Lambda}_{11}^{\frac{1}{2}}\boldsymbol{\nu}_{t+1,11}\mathbf{\Lambda}_{11}^{\frac{1}{2}}$$

$$= \boldsymbol{V}_t^{+}\Big(\boldsymbol{I}_{r_u} + \frac{\eta}{1+1.1\eta}\boldsymbol{V}_t^{+}\Big)^{-1} + \eta\mathbf{\Lambda}_{u_1,11}^2 + C\eta^2\|\mathbf{\Lambda}\|_{\mathrm{F}}^2 r_s\mathbf{\Lambda}_{11} + \frac{\eta}{\sqrt{r_s}}\mathbf{\Lambda}_{11}^{\frac{1}{2}}\boldsymbol{\nu}_{t+1,11}\mathbf{\Lambda}_{11}^{\frac{1}{2}},$$

where we used (F.11) for (a).

For the lower bound, we first observe that $\boldsymbol{G}_{t,11}(\boldsymbol{I}_{r_u} - \boldsymbol{G}_{t,11}) - \boldsymbol{G}_{t,12}\boldsymbol{G}_{t,12}^{\top} \succeq 0$ since it corresponds to the left-top submatrix of $\boldsymbol{G}_t(\boldsymbol{I}_r - \boldsymbol{G}_t)$. Therefore, by (F.8)

$$\boldsymbol{G}_{t+1,11} \succeq \boldsymbol{G}_{t,11} + \eta\Big(\mathbf{\Lambda}_{\ell_1,11}\boldsymbol{G}_{t,11} + \boldsymbol{G}_{t,11}\mathbf{\Lambda}_{\ell_1,11} - 2\boldsymbol{G}_{t,11}\mathbf{\Lambda}_{11}\boldsymbol{G}_{t,11} - 2\boldsymbol{G}_{t,12}\mathbf{\Lambda}_{22}\boldsymbol{G}_{t,12}^{\top}\Big)$$

$$- \frac{C}{2}\eta^2\|\mathbf{\Lambda}\|_{\mathrm{F}}^2 r_s\boldsymbol{I}_{r_u} + \frac{\eta/2}{\sqrt{r_s}}\boldsymbol{\nu}_{t+1,11} - \frac{2\eta}{(r_u+1)^\alpha}\big(\boldsymbol{G}_{t,11}(\boldsymbol{I}_{r_u} - \boldsymbol{G}_{t,11}) - \boldsymbol{G}_{t,12}\boldsymbol{G}_{t,12}^{\top}\big)$$

$$\overset{(b)}{\succeq} \boldsymbol{G}_{t,11} + \eta\Big(\big(\mathbf{\Lambda}_{\ell_1,11} - \tfrac{1}{(r_u+1)^\alpha}\boldsymbol{I}_{r_u}\big)\boldsymbol{G}_{t,11} + \boldsymbol{G}_{t,11}\big(\mathbf{\Lambda}_{\ell_1,11} - \tfrac{1}{(r_u+1)^\alpha}\boldsymbol{I}_{r_u}\big)\Big)$$

$$- 2\eta\boldsymbol{G}_{t,11}\big(\mathbf{\Lambda}_{11} - \tfrac{1}{(r_u+1)^\alpha}\boldsymbol{I}_{r_u}\big)\boldsymbol{G}_{t,11} - \frac{C}{2}\eta^2\|\mathbf{\Lambda}\|_{\mathrm{F}}^2 r_s\boldsymbol{I}_{r_u} + \frac{\eta/2}{\sqrt{r_s}}\boldsymbol{\nu}_{t+1,11},$$

where (b) follows by $\boldsymbol{\Lambda}_{22} \preceq \frac{1}{(r_u+1)^\alpha} \boldsymbol{I}_{r-r_u}$. Therefore, we have

$$\boldsymbol{K}_{t+1}^- \succeq \boldsymbol{K}_t^- + \frac{\eta}{2}\left(\left(\boldsymbol{\Lambda}_{\ell_1,11} - \frac{1}{(r_u+1)^\alpha}\boldsymbol{I}_{r_u}\right)^2 - \left(2\boldsymbol{K}_t^- - \left(\boldsymbol{\Lambda}_{\ell_1,11} - \frac{1}{(r_u+1)^\alpha}\boldsymbol{I}_{r_u}\right)\right)^2\right)$$

$$-\frac{C}{2}\eta^2\|\boldsymbol{\Lambda}\|_{\mathrm{F}}^2 r_s\left(\boldsymbol{\Lambda}_{11} - \frac{1}{(r_u+1)^\alpha}\boldsymbol{I}_{r_u}\right)$$

$$+\frac{\eta/2}{\sqrt{r_s}}\left(\boldsymbol{\Lambda}_{11} - \frac{1}{(r_u+1)^\alpha}\boldsymbol{I}_{r_u}\right)^{\frac{1}{2}}\boldsymbol{\nu}_{t+1,11}\left(\boldsymbol{\Lambda}_{11} - \frac{1}{(r_u+1)^\alpha}\boldsymbol{I}_{r_u}\right)^{\frac{1}{2}}.$$

By multiplying both sides with 2 and subtracting $\boldsymbol{\Lambda}_{\ell_1,11}$ from both sides, we get

$$\boldsymbol{V}_{t+1}^- \succeq \boldsymbol{V}_t^- - \eta(\boldsymbol{V}_t^-)^2 + \eta\left(\boldsymbol{\Lambda}_{\ell_1,11} - \frac{1}{(r_u+1)^\alpha}\boldsymbol{I}_{r_u}\right)^2 - C\eta^2\|\boldsymbol{\Lambda}\|_{\mathrm{F}}^2 r_s\left(\boldsymbol{\Lambda}_{11} - \frac{1}{(r_u+1)^\alpha}\boldsymbol{I}_{r_u}\right)$$

$$+\frac{\eta}{\sqrt{r_s}}\left(\boldsymbol{\Lambda}_{11} - \frac{1}{(r_u+1)^\alpha}\boldsymbol{I}_{r_u}\right)^{\frac{1}{2}}\boldsymbol{\nu}_{t+1,11}\left(\boldsymbol{\Lambda}_{11} - \frac{1}{(r_u+1)^\alpha}\boldsymbol{I}_{r_u}\right)^{\frac{1}{2}}$$

$$\overset{(c)}{\succeq} \boldsymbol{V}_t^- - \frac{\eta}{1-1.1\eta}(\boldsymbol{V}_t^-)^2\left(\boldsymbol{I}_{r_u} + \frac{\eta}{1-1.1\eta}\boldsymbol{V}_t^-\right)^{-1} + \eta\left(\boldsymbol{\Lambda}_{\ell_1,11} - \frac{1}{(r_u+1)^\alpha}\boldsymbol{I}_{r_u}\right)^2$$

$$- C\eta^2\|\boldsymbol{\Lambda}\|_{\mathrm{F}}^2 r_s\left(\boldsymbol{\Lambda}_{11} - \frac{1}{(r_u+1)^\alpha}\boldsymbol{I}_{r_u}\right)$$

$$+\frac{\eta}{\sqrt{r_s}}\left(\boldsymbol{\Lambda}_{11} - \frac{1}{(r_u+1)^\alpha}\boldsymbol{I}_{r_u}\right)^{\frac{1}{2}}\boldsymbol{\nu}_{t+1,11}\left(\boldsymbol{\Lambda}_{11} - \frac{1}{(r_u+1)^\alpha}\boldsymbol{I}_{r_u}\right)^{\frac{1}{2}}$$

$$= \boldsymbol{V}_t^-\left(\boldsymbol{I}_{r_u} + \frac{\eta}{1-1.1\eta}\boldsymbol{V}_t^-\right)^{-1} + \eta\left(\boldsymbol{\Lambda}_{\ell_1,11} - \frac{1}{(r_u+1)^\alpha}\boldsymbol{I}_{r_u}\right)^2$$

$$- C\eta^2\|\boldsymbol{\Lambda}\|_{\mathrm{F}}^2 r_s\left(\boldsymbol{\Lambda}_{11} - \frac{1}{(r_u+1)^\alpha}\boldsymbol{I}_{r_u}\right)$$

$$+\frac{\eta}{\sqrt{r_s}}\left(\boldsymbol{\Lambda}_{11} - \frac{1}{(r_u+1)^\alpha}\boldsymbol{I}_{r_u}\right)^{\frac{1}{2}}\boldsymbol{\nu}_{t+1,11}\left(\boldsymbol{\Lambda}_{11} - \frac{1}{(r_u+1)^\alpha}\boldsymbol{I}_{r_u}\right)^{\frac{1}{2}},$$

where we used (F.11) for (c). The monotonicity of the update follows the same argument in the heavy-tailed case. $\qquad\square$

### F.3 Definitions and bounding systems

To avoid repetition in the derivations, we introduce the following unified notation:

$$\mathsf{rk} \in \{r, r_u\}, \quad \mathsf{G}_t \in \{\boldsymbol{G}_t, \boldsymbol{G}_{t,11}\}, \quad \mathsf{v}_t \in \{\boldsymbol{\nu}_t, \boldsymbol{\nu}_{t,11}\}.$$

where each variable will take its first value in the heavy-tailed case and its second value in the light-tailed case. To avoid repetition in the following sections, we make the following simplifications by slight abuse of notation:

$$\boldsymbol{\Lambda}_{\ell_1} \leftarrow \begin{cases} \boldsymbol{\Lambda}_{\ell_1}, & \alpha \in [0, 0.5) \\ \boldsymbol{\Lambda}_{\ell_1,11} - \frac{1}{(r_u+1)^\alpha}\boldsymbol{I}_{r_u}, & \alpha > 0.5 \end{cases} \quad \text{and} \quad \boldsymbol{\Lambda}_{\ell_2} \leftarrow \begin{cases} \boldsymbol{\Lambda}, & \alpha \in [0, 0.5) \\ \boldsymbol{\Lambda}_{11} - \frac{1}{(r_u+1)^\alpha}\boldsymbol{I}_{r_u}, & \alpha > 0.5 \end{cases}$$

and

$$\boldsymbol{\Lambda}_{u_1} \leftarrow \begin{cases} \boldsymbol{\Lambda}_{u_1}, & \alpha \in [0, 0.5) \\ \boldsymbol{\Lambda}_{u_1,11}, & \alpha > 0.5 \end{cases} \quad \text{and} \quad \boldsymbol{\Lambda}_{u_2} \leftarrow \begin{cases} \boldsymbol{\Lambda}, & \alpha \in [0, 0.5) \\ \boldsymbol{\Lambda}_{11}, & \alpha > 0.5. \end{cases}$$

The dimension of each block is $r_u < r$ for $\alpha > 0.5$ and $r$ for $\alpha \in [0, 0.5)$, from which readers can distinguish the light tailed case from the heavy tailed case. Throughout the proof, we will also use constants $\kappa_d \in o_d(1)$ and $\tilde{C} \in O(1)$ that will be specified later. Moreover, we make the following definitions:

- **Noise sequence.** For $\underline{\boldsymbol{\nu}}_0 = 0$, we define the noise sequence $\underline{\boldsymbol{\nu}}_{t+1} := \underline{\boldsymbol{\nu}}_t + \frac{\eta/2}{\sqrt{r_s}}\mathsf{v}_{t+1}$.

- **Reference sequence.** For $\boldsymbol{T}_0 = \frac{\kappa_d r_s}{d}\boldsymbol{I}_{\mathsf{rk}}$, we define the reference sequence

$$\boldsymbol{T}_{t+1} = \boldsymbol{T}_t + 2(1 - 2\kappa_d)\eta\left(\boldsymbol{\Lambda}_{\ell_1}\boldsymbol{T}_t - \frac{3\kappa_d + 1}{\kappa_d(1 - 2\kappa_d)}\boldsymbol{\Lambda}_{u_2}\boldsymbol{T}_t^2\right).$$

- **Bounding systems.** We define the lower and upper bounding recursions as

$$\underline{V}_{t+1} = \underline{V}_t \left( I_{\mathrm{rk}} + \frac{\eta(1+2\kappa_d)}{1-1.2\eta} \underline{V}_t \right)^{-1} + \frac{\eta(1+2\kappa_d)}{1-1.2\eta}\left( \frac{\Lambda_{\ell_1}^2}{(1+2\kappa_d)^2} - \tilde{C}\eta\|\Lambda\|_{\mathrm{F}}^2 r_s \Lambda_{\ell_1} \right) \quad \text{(F.12)}$$

$$\bar{V}_{t+1} = \bar{V}_t \left( I_{\mathrm{rk}} + \frac{\eta(1-2\kappa_d)}{1+1.2\eta} \bar{V}_t \right)^{-1} + \frac{\eta(1-2\kappa_d)}{1+1.2\eta}\left( \frac{\Lambda_{u_1}^2}{(1-2\kappa_d)^2} + \tilde{C}\eta\|\Lambda\|_{\mathrm{F}}^2 r_s \Lambda_{u_1} \right). \quad \text{(F.13)}$$

where the iterates $\{\underline{V}_t\}_{t\in\mathbb{N}}$ and $\{\bar{V}_t\}_{t\in\mathbb{N}}$ are functions of the bounding sequences $\{\underline{G}_t\}_{t\in\mathbb{N}}$ and $\{\bar{G}_t\}_{t\in\mathbb{N}}$ as following:

$$\underline{K}_t := \Lambda_{\ell_2}^{\frac{1}{2}} \underline{G}_t \Lambda_{\ell_2}^{\frac{1}{2}} \quad \text{and} \quad \underline{V}_t = 2\underline{K}_t - \frac{\Lambda_{\ell_1}}{1+2\kappa_d} \quad \text{and} \quad \underline{G}_0 \preceq \mathsf{G}_0 - T_0,$$

$$\bar{K}_t := \Lambda_{u_2}^{\frac{1}{2}} \bar{G}_t \Lambda_{u_2}^{\frac{1}{2}} \quad \text{and} \quad \bar{V}_t = 2\bar{K}_t - \frac{\Lambda_{u_1}}{1-2\kappa_d} \quad \text{and} \quad \bar{G}_0 \succeq \mathsf{G}_0 + T_0.$$

- **Stopping times.** We define a sequence of events $\{\mathcal{E}_t\}_{t\geq 0}$

$$\mathcal{E}_t := \begin{cases} \left\{ -\kappa_d r^{\frac{-\alpha}{2}} T_t \preceq \nu_t \preceq \kappa_d r^{\frac{-\alpha}{2}} T_t \right\} \cap \left\{ -\kappa_d^2 r^{-\alpha} T_t \preceq \Lambda^{\frac{1}{2}} \nu_t \Lambda^{\frac{1}{2}} \preceq \kappa_d^2 r^{-\alpha} T_t \right\}, & \alpha \in [0, 0.5) \\ \left\{ -\kappa_d r_u^{\frac{-\alpha}{2}} T_t \preceq \nu_t \preceq \kappa_d r_u^{\frac{-\alpha}{2}} T_t \right\} \cap \left\{ \frac{-\kappa_d^2}{4} r_u^{-\alpha} T_t \preceq \Lambda_{11}^{\frac{1}{2}} \nu_t \Lambda_{11}^{\frac{1}{2}} \preceq \frac{\kappa_d^2}{4} r_u^{-\alpha} T_t \right\}, & \alpha > 0.5. \end{cases}$$

We define the stopping times

$$\mathcal{T}_{\mathrm{noise}}(\omega) := \inf\left\{ t \geq 0 \mid \omega \notin \mathcal{E}_t \right\} \wedge d^3 \quad \text{and} \quad \mathcal{T}_{\mathrm{bounded}} := \inf\left\{ t \geq 0 \mid \|\underline{G}_t\|_2 \vee \|\bar{G}_t\|_2 \geq 1.5 \right\},$$

and

$$\mathcal{T}_{\mathrm{bad}} := \mathcal{T}_{\mathrm{noise}} \wedge \mathcal{T}_{\mathrm{bounded}} \wedge \{t \geq 0 : \|T_t\|_2 > 1.2\kappa_d\}. \quad \text{(F.14)}$$

The main result of this section is the following:

**Proposition 14.** *Let $\kappa_d$ satisfy (F.15) and consider $d$ large enough so that $\kappa_d \leq \frac{1}{50}$. Under the learning rate conditions considered in Propositions 12 and 13, we have for $\tilde{C} > 1 \vee \Omega(1)$*

$$\underline{G}_{t \wedge \mathcal{T}_{bad}} + T_{t \wedge \mathcal{T}_{bad}} + \underline{\nu}_{t \wedge \mathcal{T}_{bad}} \preceq \mathsf{G}_{t \wedge \mathcal{T}_{bad}} \preceq \bar{G}_{t \wedge \mathcal{T}_{bad}} - T_{t \wedge \mathcal{T}_{bad}} + \bar{\nu}_{t \wedge \mathcal{T}_{bad}}.$$

Before starting the proof, we provide an auxiliary statement.

**Lemma 4.** *We consider the learning rate conditions considered in Propositions 12 and 13 with*

$$\kappa_d \ll \begin{cases} \frac{1}{\log d}, & \alpha \in [0, 0.5) \\ \frac{r_u^{-1}}{\log d}, & \alpha > 0.5. \end{cases} \quad \text{(F.15)}$$

*The event $\mathcal{E}_t$ implies for $d \geq \Omega(1)$ and $t \leq \mathcal{T}_{bounded} \wedge \{t : \|T_t\|_2 > 1.2\kappa_d\}$ that*

1. $-3\kappa_d \Lambda_{\ell_1}^{\frac{1}{2}} T_t \Lambda_{\ell_1}^{\frac{1}{2}} \preceq \Lambda_{\ell_1} \underline{\nu}_t + \underline{\nu}_t \Lambda_{\ell_1}$

2. $\Lambda_{u_1} \bar{\nu}_t + \bar{\nu}_t \Lambda_{u_1} \preceq 3\kappa_d \Lambda_{u_1}^{\frac{1}{2}} T_t \Lambda_{u_1}^{\frac{1}{2}}$

3. $\left( \Lambda_{\ell_2}^{\frac{1}{2}} \underline{\nu}_t \Lambda_{\ell_2}^{\frac{1}{2}} \right)^2 \preceq \frac{\kappa_d^2}{4} \Lambda_{\ell_1} T_t \Lambda_{\ell_1}$

4. $\left( \Lambda_{u_2}^{\frac{1}{2}} \bar{\nu}_t \Lambda_{u_2}^{\frac{1}{2}} \right)^2 \preceq \frac{\kappa_d^2}{4} \Lambda_{u_1} T_t \Lambda_{u_1}$

*Proof.* For notational convenience, we define $\widetilde{\nu}_t := T_t^{\frac{-1}{2}} \nu_t T_t^{\frac{-1}{2}}$. We initially observe that $\mathcal{E}_t$ implies for $\alpha \in [0.0.5)$ that

$$\Lambda \widetilde{\nu}_t + \widetilde{\nu}_t \Lambda \preceq \kappa_d^2 \Lambda + \frac{1}{\kappa_d^2} \Lambda^{\frac{-1}{2}} \left( \Lambda^{\frac{1}{2}} \widetilde{\nu}_t \Lambda^{\frac{1}{2}} \right)^2 \Lambda^{\frac{-1}{2}} \preceq \kappa_d^2 \Lambda + \kappa_d^2 r^{-\alpha} I_r$$

$$\Lambda \widetilde{\nu}_t + \widetilde{\nu}_t \Lambda \succeq -\kappa_d^2 \Lambda - \frac{1}{\kappa_d^2} \Lambda^{\frac{-1}{2}} \left( \Lambda^{\frac{1}{2}} \widetilde{\nu}_t \Lambda^{\frac{1}{2}} \right)^2 \Lambda^{\frac{-1}{2}} \succeq -\kappa_d^2 \Lambda - \kappa_d^2 r^{-\alpha} I_r.$$

For $\alpha > 0.5$, these bounds become

$$\mathbf{\Lambda}_{11}\widetilde{\boldsymbol{\nu}}_t + \widetilde{\boldsymbol{\nu}}_t\mathbf{\Lambda}_{11} \preceq \frac{\kappa_d^2}{4}\mathbf{\Lambda}_{11} + \frac{4}{\kappa_d^2}\mathbf{\Lambda}_{11}^{\frac{-1}{2}}\left(\mathbf{\Lambda}_{11}^{\frac{1}{2}}\widetilde{\boldsymbol{\nu}}_t\mathbf{\Lambda}_{11}^{\frac{1}{2}}\right)^2\mathbf{\Lambda}_{11}^{\frac{-1}{2}} \preceq \frac{\kappa_d^2}{4}\mathbf{\Lambda}_{11} + \frac{\kappa_d^2}{4}r_u^{-\alpha}\boldsymbol{I}_{r_u}$$

$$\mathbf{\Lambda}_{11}\widetilde{\boldsymbol{\nu}}_t + \widetilde{\boldsymbol{\nu}}_t\mathbf{\Lambda}_{11} \succeq -\frac{\kappa_d^2}{4}r_u^{-1}\mathbf{\Lambda}_{11} - \frac{4}{\kappa_d^2}\mathbf{\Lambda}_{11}^{\frac{-1}{2}}\left(\mathbf{\Lambda}_{11}^{\frac{1}{2}}\widetilde{\boldsymbol{\nu}}_t\mathbf{\Lambda}_{11}^{\frac{1}{2}}\right)^2\mathbf{\Lambda}_{11}^{\frac{-1}{2}} \succeq -\frac{\kappa_d^2}{4}r_u^{-1}\mathbf{\Lambda}_{11} - \frac{\kappa_d^2}{4}r_u^{-\alpha}\boldsymbol{I}_{r_u}.$$

In the following, we will use these bounds. For the first item, we have

$$\mathbf{\Lambda}_{\ell_1}\widetilde{\boldsymbol{\nu}}_t + \widetilde{\boldsymbol{\nu}}_t\mathbf{\Lambda}_{\ell_1} \succeq \begin{cases} -\kappa_d\left(\kappa_d\mathbf{\Lambda} + \kappa_d r^{-\alpha}\boldsymbol{I}_r + r^{\frac{-\alpha}{2}}C\|\mathbf{\Lambda}\|_{\mathrm{F}}\frac{\eta d}{\sqrt{r_s}}\boldsymbol{I}_r\right), & \alpha \in [0, 0.5) \\ -\kappa_d\left(\frac{\kappa_d}{4}\mathbf{\Lambda}_{11} + \frac{\kappa_d}{4}r_u^{-\alpha}\boldsymbol{I}_{r_u} + r_u^{\frac{-\alpha}{2}}C\|\mathbf{\Lambda}\|_{\mathrm{F}}\frac{\eta d}{\sqrt{r_s}}\boldsymbol{I}_{r_u}\right), & \alpha > 0.5 \end{cases}$$

$$\succeq -3\kappa_d\mathbf{\Lambda}_{\ell_1}.$$

For the second item, we have

$$\mathbf{\Lambda}_{u_1}\widetilde{\boldsymbol{\nu}}_t + \widetilde{\boldsymbol{\nu}}_t\mathbf{\Lambda}_{u_1} \preceq \begin{cases} \kappa_d\left(\kappa_d\mathbf{\Lambda} + \kappa_d r^{-\alpha}\boldsymbol{I}_r + r^{\frac{-\alpha}{2}}C\|\mathbf{\Lambda}\|_{\mathrm{F}}\frac{\eta d}{\sqrt{r_s}}\boldsymbol{I}_r\right), & \alpha \in [0, 0.5) \\ \kappa_d\left(\frac{\kappa_d}{4}\mathbf{\Lambda}_{11} + \frac{\kappa_d}{4}r_u^{-\alpha}\boldsymbol{I}_{r_u} + r_u^{\frac{-\alpha}{2}}C\|\mathbf{\Lambda}\|_{\mathrm{F}}\frac{\eta d}{\sqrt{r_s}}\boldsymbol{I}_{r_u}\right), & \alpha > 0.5 \end{cases}$$

$$\preceq 3\kappa_d\mathbf{\Lambda}_{u_1}.$$

For the third item, we immediately observe that $\left(\mathbf{\Lambda}_{\ell_2}^{\frac{1}{2}}\boldsymbol{\nu}_t\mathbf{\Lambda}_{\ell_2}^{\frac{1}{2}}\right)^2 \preceq \mathbf{\Lambda}_{\ell_2}^{\frac{1}{2}}\boldsymbol{\nu}_t^2\mathbf{\Lambda}_{\ell_2}^{\frac{1}{2}}$. Therefore,

$$\boldsymbol{T}_t^{\frac{-1}{2}}\mathbf{\Lambda}_{\ell_2}^{\frac{1}{2}}\boldsymbol{\nu}_t^2\mathbf{\Lambda}_{\ell_2}^{\frac{1}{2}}\boldsymbol{T}_t^{\frac{-1}{2}} \preceq 1.2\kappa_d\mathbf{\Lambda}_{\ell_2}^{\frac{1}{2}}\widetilde{\boldsymbol{\nu}}_t^2\mathbf{\Lambda}_{\ell_2}^{\frac{1}{2}} \preceq 1.2\kappa_d^3\mathsf{rk}^{-\alpha}\mathbf{\Lambda}_{\ell_2} \preceq \frac{\kappa_d^2}{4}\mathbf{\Lambda}_{\ell_1}.$$

For the fourth item, we observe that

$$\left(\mathbf{\Lambda}_{u_2}^{\frac{1}{2}}\boldsymbol{\nu}_t\mathbf{\Lambda}_{u_2}^{\frac{1}{2}}\right)^2 = \mathbf{\Lambda}_{u_2}^{\frac{1}{2}}\boldsymbol{\nu}_t\mathbf{\Lambda}_{u_2}\boldsymbol{\nu}_t\mathbf{\Lambda}_{u_2}^{\frac{1}{2}} \preceq \left(1 + \frac{0.1\mathsf{rk}^{-\alpha}}{\log^3 d}\right)\mathbf{\Lambda}_{u_2}^{\frac{1}{2}}\boldsymbol{\nu}_t^2\mathbf{\Lambda}_{u_2}^{\frac{1}{2}}.$$

Therefore

$$\left(1 + \frac{0.1\mathsf{rk}^{-\alpha}}{\log^3 d}\right)\boldsymbol{T}_t^{\frac{-1}{2}}\mathbf{\Lambda}_{u_2}^{\frac{1}{2}}\boldsymbol{\nu}_t^2\mathbf{\Lambda}_{u_2}^{\frac{1}{2}}\boldsymbol{T}_t^{\frac{-1}{2}} \preceq 1.25\kappa_d\mathbf{\Lambda}_{u_2}^{\frac{1}{2}}\widetilde{\boldsymbol{\nu}}_t^2\mathbf{\Lambda}_{u_2}^{\frac{1}{2}} \preceq 1.25\kappa_d^3\mathsf{rk}^{-\alpha}\mathbf{\Lambda}_{u_2} \preceq \frac{\kappa_d^2}{4}\mathbf{\Lambda}_{u_1}.$$

□

### F.3.1 Proof of Proposition 14

*Proof.* For the proof, we introduce the following notations

$$\underline{\boldsymbol{\zeta}}_t := 2\mathbf{\Lambda}_{\ell_2}^{\frac{1}{2}}\boldsymbol{\nu}_t\mathbf{\Lambda}_{\ell_2}^{\frac{1}{2}} \text{ and } \bar{\boldsymbol{\zeta}}_t := 2\mathbf{\Lambda}_{u_2}^{\frac{1}{2}}\boldsymbol{\nu}_t\mathbf{\Lambda}_{u_2}^{\frac{1}{2}} \text{ and } \underline{\boldsymbol{B}}_t := 2\mathbf{\Lambda}_{\ell_2}^{\frac{1}{2}}\boldsymbol{T}_t\mathbf{\Lambda}_{\ell_2}^{\frac{1}{2}} \text{ and } \bar{\boldsymbol{B}}_t := 2\mathbf{\Lambda}_{u_2}^{\frac{1}{2}}\boldsymbol{T}_t\mathbf{\Lambda}_{u_2}^{\frac{1}{2}}.$$

Using this notation, we obtain:

$$\underline{\boldsymbol{B}}_{t+1} \preceq \underline{\boldsymbol{B}}_t + \frac{2(1 - 2\kappa_d)\eta}{1 - 1.1\eta}\left(\mathbf{\Lambda}_{\ell_1}\underline{\boldsymbol{B}}_t - \frac{1.5\kappa_d + 0.5}{\kappa_d(1 - 2\kappa_d)}\underline{\boldsymbol{B}}_t^2\right) + 10\eta^2\kappa_d\mathbf{\Lambda}_{\ell_1} \quad \text{(F.16)}$$

$$\bar{\boldsymbol{B}}_{t+1} \preceq \bar{\boldsymbol{B}}_t + \frac{2(1 - 2\kappa_d)\eta}{1 + 1.1\eta}\left(\mathbf{\Lambda}_{u_1}\bar{\boldsymbol{B}}_t - \frac{1.5\kappa_d + 0.5}{\kappa_d(1 - 2\kappa_d)}\bar{\boldsymbol{B}}_t^2\right) + 10\eta^2\kappa_d\mathbf{\Lambda}_{u_1} \quad \text{(F.17)}$$

Before proceeding with the proof, we observe that the following inequalities hold:

$$\|\mathbf{\Lambda}_{\ell_2}^{-1}\mathbf{\Lambda}_{\ell_1}\|_2 \le 1, \ \|\mathbf{\Lambda}_{\ell_1}^{-1}\mathbf{\Lambda}_{\ell_2}\|_2 \le \frac{1}{1 - \frac{0.1}{\log^4 d}} \text{ and } \|\mathbf{\Lambda}_{u_2}^{-1}\mathbf{\Lambda}_{u_1}\|_2 \le 1, \ \|\mathbf{\Lambda}_{u_1}^{-1}\mathbf{\Lambda}_{u_2}\|_2 \le \frac{1}{1 - \frac{0.1}{\log^4 d}}.$$

These bounds will be used in the following whenever we apply Propositions 29 and 30, without explicitly restating them each time. We will establish the upper and lower bounds simultaneously for $\mathsf{rk} \in \{r, r_u\}$.

**Upper bound proof:** We will use proof by induction. Specifically, we will show that for $t < \mathcal{T}_{\text{bad}}$,

$$\boldsymbol{V}_t^+ \preceq \bar{\boldsymbol{V}}_t + \bar{\boldsymbol{\zeta}}_t - \bar{\boldsymbol{B}}_t + \frac{2\kappa_d \boldsymbol{\Lambda}_{u_1}}{1 - 2\kappa_d} \quad \Rightarrow \quad \boldsymbol{V}_{t+1}^+ \preceq \bar{\boldsymbol{V}}_{t+1} + \bar{\boldsymbol{\zeta}}_{t+1} - \bar{\boldsymbol{B}}_{t+1} + \frac{2\kappa_d \boldsymbol{\Lambda}_{u_1}}{1 - 2\kappa_d}. \quad \text{(F.18)}$$

Since the base case holds at $t = 0$ and $\mathcal{T}_{\text{bad}} > 0$, it remains to prove (F.18). By (F.6), we have

$$\boldsymbol{V}_{t+1}^+ \preceq \left( \bar{\boldsymbol{V}}_t + \bar{\boldsymbol{\zeta}}_t - \bar{\boldsymbol{B}}_t + \frac{2\kappa_d \boldsymbol{\Lambda}_{u_1}}{1 - 2\kappa_d} \right) \left( \boldsymbol{I}_{\text{rk}} + \frac{\eta}{1 + 1.1\eta}(\bar{\boldsymbol{V}}_t + \bar{\boldsymbol{\zeta}}_t - \bar{\boldsymbol{B}}_t + \frac{2\kappa_d \boldsymbol{\Lambda}_{u_1}}{1 - 2\kappa_d}) \right)^{-1}$$
$$+ \eta \boldsymbol{\Lambda}_{u_1}^2 + C\eta^2 \|\boldsymbol{\Lambda}\|_{\text{F}}^2 r_s \boldsymbol{\Lambda}_{u_2} + \frac{\eta}{\sqrt{r_s}} \boldsymbol{\Lambda}_{u_2}^{\frac{1}{2}} \boldsymbol{\nu}_{t+1} \boldsymbol{\Lambda}_{u_2}^{\frac{1}{2}}$$

$$= \bar{\boldsymbol{V}}_t \left( \boldsymbol{I}_{\text{rk}} + \frac{\eta}{1 + 1.1\eta} \bar{\boldsymbol{V}}_t \right)^{-1} + \eta \boldsymbol{\Lambda}_{u_1}^2 + C\eta^2 \|\boldsymbol{\Lambda}\|_{\text{F}}^2 r_s \boldsymbol{\Lambda}_{u_2} + \frac{\eta}{\sqrt{r_s}} \boldsymbol{\Lambda}_{u_2}^{\frac{1}{2}} \boldsymbol{\nu}_{t+1} \boldsymbol{\Lambda}_{u_2}^{\frac{1}{2}}$$

$$+ \left( \boldsymbol{I}_{\text{rk}} + \frac{\eta}{1 + 1.1\eta} \bar{\boldsymbol{V}}_t \right)^{-1} (\bar{\boldsymbol{\zeta}}_t - \bar{\boldsymbol{B}}_t + \frac{2\kappa_d \boldsymbol{\Lambda}_{u_1}}{1 - 2\kappa_d}) \left( \boldsymbol{I}_{\text{rk}} + \frac{\eta}{1 + 1.1\eta}(\bar{\boldsymbol{V}}_t + \bar{\boldsymbol{\zeta}}_t - \bar{\boldsymbol{B}}_t + \frac{2\kappa_d \boldsymbol{\Lambda}_{u_1}}{1 - 2\kappa_d}) \right)^{-1}.$$
$$\text{(F.19)}$$

By using Proposition 29, we have for $t < \mathcal{T}_{\text{bad}}$

$$\left( \boldsymbol{I}_{\text{rk}} + \frac{\eta}{1 + 1.1\eta} \bar{\boldsymbol{V}}_t \right)^{-1} (\bar{\boldsymbol{\zeta}}_t - \bar{\boldsymbol{B}}_t + \frac{2\kappa_d \boldsymbol{\Lambda}_{u_1}}{1 - 2\kappa_d}) \left( \boldsymbol{I}_{\text{rk}} + \frac{\eta}{1 + 1.1\eta}(\bar{\boldsymbol{V}}_t + \bar{\boldsymbol{\zeta}}_t - \bar{\boldsymbol{B}}_t + \frac{2\kappa_d \boldsymbol{\Lambda}_{u_1}}{1 - 2\kappa_d}) \right)^{-1}$$

$$\preceq \left( \bar{\boldsymbol{\zeta}}_t - \bar{\boldsymbol{B}}_t + \frac{2\kappa_d \boldsymbol{\Lambda}_{u_1}}{1 - 2\kappa_d} \right) - \frac{\eta}{1 + 1.1\eta} \bar{\boldsymbol{V}}_t \left( \bar{\boldsymbol{\zeta}}_t - \bar{\boldsymbol{B}}_t + \frac{2\kappa_d \boldsymbol{\Lambda}_{u_1}}{1 - 2\kappa_d} \right)$$

$$- \frac{\eta}{1 + 1.1\eta} \left( \bar{\boldsymbol{\zeta}}_t - \bar{\boldsymbol{B}}_t + \frac{2\kappa_d \boldsymbol{\Lambda}_{u_1}}{1 - 2\kappa_d} \right) \bar{\boldsymbol{V}}_t - \frac{\eta}{1 + 1.1\eta} \left( \bar{\boldsymbol{\zeta}}_t - \bar{\boldsymbol{B}}_t + \frac{2\kappa_d \boldsymbol{\Lambda}_{u_1}}{1 - 2\kappa_d} \right)^2 + \frac{\eta^2 \kappa_d^2}{(1 + 1.1\eta)^2} \bar{\boldsymbol{V}}_t^4$$

$$+ \frac{2\eta^2 / \kappa_d^2}{(1 + 1.1\eta)^2} \left( \bar{\boldsymbol{\zeta}}_t - \bar{\boldsymbol{B}}_t + \frac{2\kappa_d \boldsymbol{\Lambda}_{u_1}}{1 - 2\kappa_d} \right)^2 + \frac{\eta^2 \kappa_d^2}{(1 + 1.1\eta)^2} \bar{\boldsymbol{V}}_t \left( \bar{\boldsymbol{\zeta}}_t - \bar{\boldsymbol{B}}_t + \frac{2\kappa_d \boldsymbol{\Lambda}_{u_1}}{1 - 2\kappa_d} \right)^2 \bar{\boldsymbol{V}}_t$$

$$+ \frac{\eta^2}{(1 + 1.1\eta)^2} \left( \bar{\boldsymbol{\zeta}}_t - \bar{\boldsymbol{B}}_t + \frac{2\kappa_d \boldsymbol{\Lambda}_{u_1}}{1 - 2\kappa_d} \right) \bar{\boldsymbol{V}}_t \left( \bar{\boldsymbol{\zeta}}_t - \bar{\boldsymbol{B}}_t + \frac{2\kappa_d \boldsymbol{\Lambda}_{u_1}}{1 - 2\kappa_d} \right) + \eta^3 \tilde{C}_1 \boldsymbol{\Lambda}_{u_1}$$

$$+ \frac{\eta^2}{(1 + 1.1\eta)^2} \left( \bar{\boldsymbol{\zeta}}_t - \bar{\boldsymbol{B}}_t + \frac{2\kappa_d \boldsymbol{\Lambda}_{u_1}}{1 - 2c_d} \right)^3 + \frac{\eta^2}{(1 + 1.1\eta)^2} \bar{\boldsymbol{V}}_t \left( \bar{\boldsymbol{\zeta}}_t - \bar{\boldsymbol{B}}_t + \frac{2\kappa_d \boldsymbol{\Lambda}_{u_1}}{1 - 2\kappa_d} \right) \bar{\boldsymbol{V}}_t$$

for some $\tilde{C}_1 = O(1)$ . We have the following: First:

$$\kappa_d^2 \bar{\boldsymbol{V}}_t^4 + \kappa_d^2 \bar{\boldsymbol{V}}_t \left( \bar{\boldsymbol{\zeta}}_t - \bar{\boldsymbol{B}}_t + \frac{2\kappa_d \boldsymbol{\Lambda}_{u_1}}{1 - 2\kappa_d} \right)^2 \bar{\boldsymbol{V}}_t + \bar{\boldsymbol{V}}_t \left( \bar{\boldsymbol{\zeta}}_t - \bar{\boldsymbol{B}}_t + \frac{2\kappa_d \boldsymbol{\Lambda}_{u_1}}{1 - 2\kappa_d} \right) \bar{\boldsymbol{V}}_t \overset{(a)}{\preceq} 4\kappa_d \bar{\boldsymbol{V}}_t^2$$

$$\overset{(b)}{\preceq} \frac{1}{15} \frac{1 + 1.1\eta}{1 + 1.2\eta} \bar{\boldsymbol{V}}_t^2,$$

where (a) follows by $\|\bar{\boldsymbol{V}}_t\|_2 \le 5$ and $\left\| \bar{\boldsymbol{\zeta}}_t - \bar{\boldsymbol{B}}_t + \frac{2\kappa_d \boldsymbol{\Lambda}_{u_1}}{1 - 2\kappa_d} \right\|_2 \le 2.5\kappa_d$, and (b) follows by $\kappa_d \le \frac{1}{50}$.
Second,

$$\frac{2}{\kappa_d^2} \left( \bar{\boldsymbol{\zeta}}_t - \bar{\boldsymbol{B}}_t + \frac{2\kappa_d \boldsymbol{\Lambda}_{u_1}}{1 - 2\kappa_d} \right)^2 + \left( \bar{\boldsymbol{\zeta}}_t - \bar{\boldsymbol{B}}_t + \frac{2\kappa_d \boldsymbol{\Lambda}_{u_1}}{1 - 2\kappa_d} \right) \bar{\boldsymbol{V}}_t \left( \bar{\boldsymbol{\zeta}}_t - \bar{\boldsymbol{B}}_t + \frac{2\kappa_d \boldsymbol{\Lambda}_{u_1}}{1 - 2\kappa_d} \right)$$

$$+ \left( \bar{\boldsymbol{\zeta}}_t + \bar{\boldsymbol{B}}_t + \frac{2\kappa_d \boldsymbol{\Lambda}_{u_1}}{1 - 2c_d} \right)^3 \overset{(c)}{\preceq} \frac{3}{\kappa_d^2} \left( \bar{\boldsymbol{\zeta}}_t - \bar{\boldsymbol{B}}_t + \frac{2\kappa_d \boldsymbol{\Lambda}_{u_1}}{1 - 2\kappa_d} \right)^2 \preceq \frac{9}{\kappa_d^2} \left( \bar{\boldsymbol{\zeta}}_t^2 + \bar{\boldsymbol{B}}_t^2 \right) + \frac{36}{(1 - 2\kappa_d)^2} \boldsymbol{\Lambda}_{u_1}^2,$$

where (c) follows by $\|\bar{\boldsymbol{V}}_t\|_2 \le 5$ and $\left\| \bar{\boldsymbol{\zeta}}_t - \bar{\boldsymbol{B}}_t + \frac{2\kappa_d \boldsymbol{\Lambda}_{u_1}}{1 - 2\kappa_d} \right\|_2 \le 2.5\kappa_d$. Third:

$$- \bar{\boldsymbol{V}}_t \left( \bar{\boldsymbol{\zeta}}_t - \bar{\boldsymbol{B}}_t + \frac{2\kappa_d \boldsymbol{\Lambda}_{u_1}}{1 - 2\kappa_d} \right) - \left( \bar{\boldsymbol{\zeta}}_t - \bar{\boldsymbol{B}}_t + \frac{2\kappa_d \boldsymbol{\Lambda}_{u_1}}{1 - 2\kappa_d} \right) \bar{\boldsymbol{V}}_t - \left( \bar{\boldsymbol{\zeta}}_t - \bar{\boldsymbol{B}}_t + \frac{2\kappa_d \boldsymbol{\Lambda}_{u_1}}{1 - 2\kappa_d} \right)^2$$

$$+ \frac{9\eta / \kappa_d^2}{1 + 1.1\eta} \left( \bar{\boldsymbol{\zeta}}_t^2 + \bar{\boldsymbol{B}}_t^2 \right)$$

$$\preceq -2(\bar{K}_t\bar{\zeta}_t + \bar{\zeta}_t\bar{K}_t) + 2(\bar{K}_t\bar{B}_t + \bar{B}_t\bar{K}_t) - \frac{4\kappa_d}{1-2\kappa_d}(\bar{K}_t\Lambda_{u_1} + \Lambda_{u_1}\bar{K}_t) + 3\left(\bar{\zeta}_t^2 + \bar{B}_t^2\right)$$

$$+ (\Lambda_{u_1}\bar{\zeta}_t + \bar{\zeta}_t\Lambda_{u_1}) - (\Lambda_{u_1}\bar{B}_t + \bar{B}_t\Lambda_{u_1}) + \frac{4\kappa_d(1-\kappa_d)}{(1-2\kappa_d)^2}\Lambda_{u_1}^2$$

$$\overset{(d)}{\preceq} 8\kappa_d\bar{K}_t^2 - \frac{4\kappa_d}{1-2\kappa_d}(\bar{K}_t\Lambda_{u_1} + \Lambda_{u_1}\bar{K}_t) + \frac{4\kappa_d(1-\kappa_d)}{(1-2\kappa_d)^2}\Lambda_{u_1}^2$$

$$- (2-4\kappa_d)\Lambda_{u_1}\bar{B}_t + \left(3 + \frac{1}{\kappa_d}\right)\bar{B}_t^2$$

$$= 2\kappa_d\bar{V}_t^2 + \frac{2\kappa_d}{1-2\kappa_d}\Lambda_{u_1}^2 - (2-4\kappa_d)\Lambda_{u_1}\bar{B}_t + \left(3 + \frac{1}{\kappa_d}\right)\bar{B}_t^2,$$

where we used Proposition 22, and the second and fourth items in Lemma 4 in (d). Therefore, we have

(F.19) $$\preceq \bar{V}_t\left(\boldsymbol{I}_{\mathrm{rk}} + \frac{\eta}{1+1.1\eta}\bar{V}_t\right)^{-1} + \frac{2\kappa_d\eta}{1+1.1\eta}\bar{V}_t^2 + \frac{1}{10}\frac{(1-2\kappa_d)\eta^2}{(1+1.1\eta)(1+1.2\eta)}\bar{V}_t^2 + \frac{2\kappa_d\Lambda_{u_1}}{1-2\kappa_d}$$

$$+ \frac{\eta}{1+1.1\eta}\left(\frac{\Lambda_{u_1}^2}{1-2\kappa_d} + \tilde{C}\eta\|\Lambda\|_{\mathrm{F}}^2 r_s\Lambda_{u_2}\right) + \bar{\zeta}_{t+1} - \bar{B}_t$$

$$- \frac{2(1-2\kappa_d)\eta}{1+1.1\eta}\left(\Lambda_{u_1}\bar{B}_t - \frac{1.5\kappa_d + 0.5}{\kappa_d(1-2\kappa_d)}\bar{B}_t^2\right)$$

$$\overset{(e)}{\preceq} \bar{V}_t\left(\boldsymbol{I}_{\mathrm{rk}} + \frac{\eta(1-2\kappa_d)}{1+1.2\eta}\bar{V}_t\right)^{-1} + \frac{\eta}{1+1.2\eta}\left(\frac{\Lambda_{u_1}^2}{1-2\kappa_d} + \tilde{C}\eta\|\Lambda\|_{\mathrm{F}}^2 r_s\Lambda_{u_1}\right)$$

$$+ \bar{\zeta}_{t+1} + \frac{2\kappa_d\Lambda_{u_1}}{1-2\kappa_d} - \bar{B}_{t+1}$$

$$\preceq \bar{V}_{t+1} + \bar{\zeta}_{t+1} + \frac{2\kappa_d\Lambda_{u_1}}{1-2\kappa_d} - \bar{B}_{t+1},$$

where we used Proposition 30 and (F.17) in (e).

**Lower bound proof:** Similar to the upper bound proof, here we will show that for $t < \mathcal{T}_{\mathrm{bad}}$,

$$\underline{V}_t + \underline{\zeta}_t + \underline{B}_t - \frac{2\kappa_d\Lambda_{\ell_1}}{1+2\kappa_d} \preceq V_t^- \implies \underline{V}_{t+1} + \underline{\zeta}_{t+1} + \underline{B}_{t+1} - \frac{2\kappa_d}{1+2\kappa_d}\Lambda_{\ell_1} \preceq V_{t+1}^-. \text{(F.20)}$$

Since the base case holds at $t = 0$ and $\mathcal{T}_{\mathrm{bad}} > 0$, it remains to prove (F.20). By (F.6), we have

$$V_{t+1}^- \succeq \left(\underline{V}_t + \underline{\zeta}_t + \underline{B}_t - \frac{2\kappa_d\Lambda_{\ell_1}}{1+2\kappa_d}\right)\left(\boldsymbol{I}_{\mathrm{rk}} + \frac{\eta}{1-1.1\eta}(\underline{V}_t + \underline{\zeta}_t + \underline{B}_t - \frac{2\kappa_d\Lambda_{\ell_1}}{1+2\kappa_d})\right)^{-1} + \eta\Lambda_{\ell_1}^2$$

$$- C\eta^2\|\Lambda\|_{\mathrm{F}}^2 r_s\Lambda_{\ell_2} + \frac{\eta}{\sqrt{r_s}}\Lambda_{\ell_2}^{\frac{1}{2}}\nu_{t+1}\Lambda_{\ell_2}^{\frac{1}{2}}$$

$$= \underline{V}_t\left(\boldsymbol{I}_{\mathrm{rk}} + \frac{\eta}{1-1.1\eta}\underline{V}_t\right)^{-1} + \eta\Lambda_{\ell_1}^2 - C\eta^2\|\Lambda\|_{\mathrm{F}}^2 r_s\Lambda_{\ell_2} + \frac{\eta}{\sqrt{r_s}}\Lambda_{\ell_2}^{\frac{1}{2}}\nu_{t+1}\Lambda_{\ell_2}^{\frac{1}{2}}$$

$$+ \left(\boldsymbol{I}_{\mathrm{rk}} + \frac{\eta}{1-1.1\eta}\underline{V}_t\right)^{-1}(\underline{\zeta}_t + \underline{B}_t - \frac{2\kappa_d\Lambda_{\ell_1}}{1+2\kappa_d})\left(\boldsymbol{I}_{\mathrm{rk}} + \frac{\eta}{1-1.1\eta}(\underline{V}_t + \underline{\zeta}_t + \underline{B}_t - \frac{2\kappa_d\Lambda_{\ell_1}}{1+2\kappa_d})\right)^{-1}.$$
(F.21)

By using Proposition 29, we have for $t < \mathcal{T}_{\mathrm{bad}}$

$$\left(\boldsymbol{I}_{\mathrm{rk}} + \frac{\eta}{1-1.1\eta}\underline{V}_t\right)^{-1}\left(\underline{\zeta}_t + \underline{B}_t - \frac{2\kappa_d\Lambda_{\ell_1}}{1+2\kappa_d}\right)\left(\boldsymbol{I}_{\mathrm{rk}} + \frac{\eta}{1-1.1\eta}(\underline{V}_t + \underline{\zeta}_t + \underline{B}_t - \frac{2\kappa_d\Lambda_{\ell_1}}{1+2\kappa_d})\right)^{-1}$$

$$\succeq \left(\underline{\zeta}_t + \underline{B}_t - \frac{2\kappa_d\Lambda_{\ell_1}}{1+2\kappa_d}\right) - \frac{\eta}{1-1.1\eta}\underline{V}_t\left(\underline{\zeta}_t + \underline{B}_t - \frac{2\kappa_d\Lambda_{\ell_1}}{1+2\kappa_d}\right) - \frac{\eta^2\kappa_d^2}{(1-1.1\eta)^2}\underline{V}_t^4$$

$$- \frac{\eta}{1-1.1\eta}\left(\underline{\zeta}_t + \underline{B}_t - \frac{2\kappa_d\Lambda_{\ell_1}}{1+2\kappa_d}\right)\underline{V}_t - \frac{\eta}{1-1.1\eta}\left(\underline{\zeta}_t + \underline{B}_t - \frac{2\kappa_d\Lambda_{\ell_1}}{1+2\kappa_d}\right)^2$$

$$
-\frac{2\eta^2/\kappa_d^2}{(1-1.1\eta)^2}\left(\boldsymbol{\zeta}_t+\boldsymbol{B}_t-\frac{2\kappa_d\boldsymbol{\Lambda}_{\ell_1}}{1+2\kappa_d}\right)^2-\frac{\eta^2\kappa_d^2}{(1-1.1\eta)^2}\boldsymbol{V}_t\left(\boldsymbol{\zeta}_t+\boldsymbol{B}_t-\frac{2\kappa_d\boldsymbol{\Lambda}_{\ell_1}}{1+2\kappa_d}\right)^2\boldsymbol{V}_t
$$

$$
+\frac{\eta^2}{(1-1.1\eta)^2}\left(\boldsymbol{\zeta}_t+\boldsymbol{B}_t-\frac{2\kappa_d\boldsymbol{\Lambda}_{\ell_1}}{1+2\kappa_d}\right)\boldsymbol{V}_t\left(\boldsymbol{\zeta}_t+\boldsymbol{B}_t-\frac{2\kappa_d\boldsymbol{\Lambda}_{\ell_1}}{1+2\kappa_d}\right)-\eta^3\tilde{C}_2\boldsymbol{\Lambda}_{\ell_1}
$$

$$
+\frac{\eta^2}{(1-1.1\eta)^2}\left(\boldsymbol{\zeta}_t+\boldsymbol{B}_t-\frac{2\kappa_d\boldsymbol{\Lambda}_{\ell_1}}{1+2\kappa_d}\right)^3+\frac{\eta^2}{(1-1.1\eta)^2}\boldsymbol{V}_t\left(\boldsymbol{\zeta}_t+\boldsymbol{B}_t-\frac{2\kappa_d\boldsymbol{\Lambda}_{\ell_1}}{1+2\kappa_d}\right)\boldsymbol{V}_t
$$

for some $\tilde{C}_2=O(1)$. We have the following: First:

$$
-\kappa_d^2\boldsymbol{V}_t^4-\kappa_d^2\boldsymbol{V}_t\left(\boldsymbol{\zeta}_t+\boldsymbol{B}_t-\frac{2\kappa_d\boldsymbol{\Lambda}_{\ell_1}}{1+2\kappa_d}\right)^2\boldsymbol{V}_t+\boldsymbol{V}_t\left(\boldsymbol{\zeta}_t+\boldsymbol{B}_t-\frac{2\kappa_d\boldsymbol{\Lambda}_{\ell_1}}{1+2\kappa_d}\right)\boldsymbol{V}_t\overset{(f)}{\succeq}-3.2\kappa_d\boldsymbol{V}_t^2
$$

$$
\overset{(g)}{\succeq}-\frac{1}{15}\frac{1-1.1\eta}{1-1.2\eta}\boldsymbol{V}_t^2.
$$

where (f) follows by $\|\boldsymbol{V}_t\|_2\le 5$ and $\left\|\boldsymbol{\zeta}_t+\boldsymbol{B}_t-\frac{2\kappa_d\boldsymbol{\Lambda}_{\ell_1}}{1+2\kappa_d}\right\|_2\le 2.5\kappa_d$, and (g) follows by $\kappa_d\le\frac{1}{50}$.
Second:

$$
-\frac{2}{\kappa_d^2}\left(\boldsymbol{\zeta}_t+\boldsymbol{B}_t-\frac{2\kappa_d\boldsymbol{\Lambda}_{\ell_1}}{1+2\kappa_d}\right)^2+\left(\boldsymbol{\zeta}_t+\boldsymbol{B}_t-\frac{2c_d\boldsymbol{\Lambda}_{\ell_1}}{1+2\kappa_d}\right)\boldsymbol{V}_t\left(\boldsymbol{\zeta}_t+\boldsymbol{B}_t-\frac{2\kappa_d\boldsymbol{\Lambda}_{\ell_1}}{1+2\kappa_d}\right)
$$

$$
+\left(\boldsymbol{\zeta}_t+\boldsymbol{B}_t-\frac{2\kappa_d\boldsymbol{\Lambda}_{\ell_1}}{1+2\kappa_d}\right)^3\overset{(h)}{\succeq}\frac{-3}{\kappa_d^2}\left(\boldsymbol{\zeta}_t+\boldsymbol{B}_t-\frac{2\kappa_d\boldsymbol{\Lambda}_{\ell_1}}{1+2\kappa_d}\right)^2\succeq\frac{-9}{\kappa_d^2}\left(\boldsymbol{\zeta}_t^2+\boldsymbol{B}_t^2\right)-36\boldsymbol{\Lambda}_{\ell_1}^2,
$$

where (h) follows by $\|\boldsymbol{V}_t\|_2\le 5$ and $\left\|\boldsymbol{\zeta}_t+\boldsymbol{B}_t-\frac{2\kappa_d\boldsymbol{\Lambda}_{\ell_1}}{1+2\kappa_d}\right\|_2\le 2.5\kappa_d$. Third:

$$
-\boldsymbol{V}_t\left(\boldsymbol{\zeta}_t+\boldsymbol{B}_t-\frac{2\kappa_d\boldsymbol{\Lambda}_{\ell_1}}{1+2\kappa_d}\right)-\left(\boldsymbol{\zeta}_t+\boldsymbol{B}_t-\frac{2\kappa_d\boldsymbol{\Lambda}_{\ell_1}}{1+2\kappa_d}\right)\boldsymbol{V}_t-\left(\boldsymbol{\zeta}_t+\boldsymbol{B}_t-\frac{2\kappa_d\boldsymbol{\Lambda}_{\ell_1}}{1+2\kappa_d}\right)^2
$$

$$
-\frac{9\eta/\kappa_d^2}{1-1.1\eta}\left(\boldsymbol{\zeta}_t^2+\boldsymbol{B}_t^2\right)
$$

$$
\succeq-2\left(\boldsymbol{K}_t\boldsymbol{\zeta}_t+\boldsymbol{\zeta}_t\boldsymbol{K}_t\right)-2\left(\boldsymbol{K}_t\boldsymbol{B}_t+\boldsymbol{B}_t\boldsymbol{K}_t\right)+\frac{4\kappa_d}{1+2\kappa_d}\left(\boldsymbol{K}_t\boldsymbol{\Lambda}_{\ell_1}+\boldsymbol{\Lambda}_{\ell_1}\boldsymbol{K}_t\right)-3(\boldsymbol{\zeta}_t^2+\boldsymbol{B}_t^2)
$$

$$
+\left(\boldsymbol{\Lambda}_{\ell_1}\boldsymbol{\zeta}_t+\boldsymbol{\zeta}_t\boldsymbol{\Lambda}_{\ell_1}\right)+\left(\boldsymbol{\Lambda}_{\ell_1}\boldsymbol{B}_t+\boldsymbol{B}_t\boldsymbol{\Lambda}_{\ell_1}\right)-\frac{4\kappa_d(1+\kappa_d)\boldsymbol{\Lambda}_{\ell_1}^2}{(1+2\kappa_d)^2}
$$

$$
\overset{(i)}{\succeq}-8c_d\boldsymbol{K}_t^2+\frac{4\kappa_d}{1+2\kappa_d}\left(\boldsymbol{K}_t\boldsymbol{\Lambda}_{\ell_1}+\boldsymbol{\Lambda}_{\ell_1}\boldsymbol{K}_t\right)-\frac{4\kappa_d(1+\kappa_d)\boldsymbol{\Lambda}_{\ell_1}^2}{(1+2\kappa_d)^2}
$$

$$
+(2-4\kappa_d)\boldsymbol{\Lambda}_{\ell_1}\boldsymbol{B}_t-\left(3+\frac{1}{\kappa_d}\right)\boldsymbol{B}_t^2
$$

$$
=-2\kappa_d\boldsymbol{V}_t^2-\frac{2\kappa_d\boldsymbol{\Lambda}_{\ell_1}^2}{(1+2\kappa_d)}+(2-4\kappa_d)\boldsymbol{B}_t\boldsymbol{\Lambda}_{\ell_1}-\left(3+\frac{1}{\kappa_d}\right)\boldsymbol{B}_t^2,
$$

where we used Proposition 22, and the first and third items in Lemma 4 in (i). Therefore, we have

$$
\text{(F.21)}\succeq\boldsymbol{V}_t\left(\boldsymbol{I}_r+\frac{\eta}{1-1.1\eta}\boldsymbol{V}_t\right)^{-1}-\frac{2\kappa_d\eta}{1-1.1\eta}\boldsymbol{V}_t^2-\frac{(1+2\kappa_d)\eta^2/15}{(1-1.1\eta)(1-1.2\eta)}\boldsymbol{V}_t^2-\frac{2\kappa_d}{1+2\kappa_d}\boldsymbol{\Lambda}_{\ell_1}
$$

$$
+\frac{\eta}{1-1.1\eta}\left(\frac{\boldsymbol{\Lambda}_{\ell_1}^2}{1+2\kappa_d}-\tilde{C}\eta\|\boldsymbol{\Lambda}\|_{\mathrm{F}}^2 r_s\boldsymbol{\Lambda}_{\ell_2}\right)+\boldsymbol{\zeta}_{t+1}+\boldsymbol{B}_t
$$

$$
+\frac{2(1-2\kappa_d)\eta}{1-1.1\eta}\left(\boldsymbol{\Lambda}_{\ell_1}\boldsymbol{B}_t-\frac{1.5\kappa_d+0.5}{\kappa_d(1-2\kappa_d)}\boldsymbol{B}_t^2\right)
$$

$$
\overset{(j)}{\succeq}\boldsymbol{V}_t\left(\boldsymbol{I}_r+\frac{\eta(1+2\kappa_d)}{1-1.2\eta}\boldsymbol{V}_t\right)^{-1}+\frac{\eta}{1-1.1\eta}\left(\frac{\boldsymbol{\Lambda}_{\ell_1}^2}{1+2\kappa_d}-\tilde{C}\eta\|\boldsymbol{\Lambda}\|_{\mathrm{F}}^2 r_s\boldsymbol{\Lambda}_{\ell_1}\right)+\boldsymbol{\zeta}_{t+1}
$$

$$
+\boldsymbol{B}_{t+1}-\frac{2\kappa_d}{1+2\kappa_d}\boldsymbol{\Lambda}_{\ell_1}
$$

$$= \underline{\boldsymbol{V}}_{t+1} + \underline{\boldsymbol{\zeta}}_{t+1} + \underline{\boldsymbol{B}}_{t+1} - \frac{2\kappa_d}{1 + 2\kappa_d}\boldsymbol{\Lambda}_{\ell_1},$$

where we used Proposition 30 and (F.16) in (j). □

### F.4 Analysis of the bounding systems

#### F.4.1 Lower bounding system

In this section, we consider (F.12). For notational convenience, we multiply both sides by the factor $(1 + 2\kappa_d)$ and use a generic learning rate $\eta$, i.e.,

$$\underline{\boldsymbol{V}}_{t+1} = \underline{\boldsymbol{V}}_t \left(\boldsymbol{I}_{\mathsf{rk}} + \eta\underline{\boldsymbol{V}}_t\right)^{-1} + \eta \left(\boldsymbol{\Lambda}_{\ell_1}^2 - \tilde{C}\eta\|\boldsymbol{\Lambda}\|_{\mathrm{F}}^2 r_s \boldsymbol{\Lambda}_{\ell_1}\right), \quad \text{where } \underline{\boldsymbol{V}}_t = 2\boldsymbol{\Lambda}_{\ell_2}^{\frac{1}{2}}\underline{\boldsymbol{G}}_t\boldsymbol{\Lambda}_{\ell_2}^{\frac{1}{2}} - \boldsymbol{\Lambda}_{\ell_1}.$$

The main result of this section is stated in Proposition 15. To establish it, we first prove an auxiliary result, Lemma 5. For the following, we define

$$\hat{\boldsymbol{\Lambda}} := \sqrt{\boldsymbol{\Lambda}_{\ell_1}^2 - \tilde{C}\eta\|\boldsymbol{\Lambda}\|_{\mathrm{F}}^2 r_s \boldsymbol{\Lambda}_{\ell_1}} = \mathrm{diag}(\{\hat{\lambda}_i\}_{i=1}^r), \quad \boldsymbol{D}_t := \frac{\boldsymbol{\Lambda}_{\ell_2}^{-1}\hat{\boldsymbol{\Lambda}}\left(\frac{\boldsymbol{A}_{t,11}}{\boldsymbol{A}_{t,12}} - \boldsymbol{I}_{\mathsf{rk}}\right)}{2} - \frac{1.1\kappa_d r_s}{d}\boldsymbol{I}_{\mathsf{rk}}.$$

By Corollary 3, we have

$$\underline{\boldsymbol{G}}_t = \frac{1}{2}\left(\frac{\boldsymbol{\Lambda}_{\ell_1}}{\boldsymbol{\Lambda}_{\ell_2}} + \frac{\boldsymbol{A}_{t,22}}{\boldsymbol{A}_{t,12}}\frac{\hat{\boldsymbol{\Lambda}}}{\boldsymbol{\Lambda}_{\ell_2}}\right) - \frac{1}{4}\frac{\boldsymbol{A}_{t,12}^{-1}\hat{\boldsymbol{\Lambda}}}{\boldsymbol{\Lambda}_{\ell_2}}\left(\frac{\frac{\hat{\boldsymbol{\Lambda}}}{\boldsymbol{\Lambda}_{\ell_2}}\left(\frac{\boldsymbol{A}_{t,11}}{\boldsymbol{A}_{t,12}} - \boldsymbol{I}_{\mathsf{rk}}\right)}{2} + \frac{\left(\hat{\boldsymbol{\Lambda}} - \boldsymbol{\Lambda}_{\ell_1}\right)}{2\boldsymbol{\Lambda}_{\ell_2}} + \underline{\boldsymbol{G}}_0\right)^{-1}\frac{\hat{\boldsymbol{\Lambda}}\boldsymbol{A}_{t,12}^{-1}}{\boldsymbol{\Lambda}_{\ell_2}},$$

$$(\text{F.22})$$

where $\boldsymbol{A}_{t,11}$, $\boldsymbol{A}_{t,12}$, and $\boldsymbol{A}_{t,22}$ are defined as in (R.1) with $\hat{\boldsymbol{\Lambda}}$. *For $\alpha = 0$, we will consider $\{\underline{\boldsymbol{G}}_t\}_{t\in\mathbb{N}}$ in the basis of $\underline{\boldsymbol{G}}_0$ without writing explicitly, which will imply that $\{\underline{\boldsymbol{G}}_t\}_{t\in\mathbb{N}}$ is diagonal due to the rotational symmetry for $\alpha = 0$.*

We further decompose $\{\underline{\boldsymbol{G}}_t\}_{t\in N}$ and related matrices to isolate their top-left submatrices of dimension $\mathsf{rk}_\star \in \{r_\star, r_{u_\star}\}$, where $r_\star < r$ and $r_{u_\star} < r_u$ which we will denote as $\mathsf{rk}_\star < \mathsf{rk}$. The decompositions are as follows:

$$\underline{\boldsymbol{G}}_t := \begin{bmatrix} \underline{\boldsymbol{G}}_{t,11} & \underline{\boldsymbol{G}}_{t,12} \\ \underline{\boldsymbol{G}}_{t,12}^\top & \underline{\boldsymbol{G}}_{t,22} \end{bmatrix}, \quad \hat{\boldsymbol{\Lambda}} := \begin{bmatrix} \hat{\boldsymbol{\Lambda}}_{11} & 0 \\ 0 & \hat{\boldsymbol{\Lambda}}_{22} \end{bmatrix}, \quad \boldsymbol{D}_t := \begin{bmatrix} \boldsymbol{D}_{t,1} & 0 \\ 0 & \boldsymbol{D}_{t,2} \end{bmatrix}, \quad \boldsymbol{Z}_{1:\mathsf{rk}} := \begin{bmatrix} \boldsymbol{Z}_{1:\mathsf{rk}_\star} \\ \boldsymbol{Z}_2 \end{bmatrix},$$

where $\underline{\boldsymbol{G}}_{t,11}, \boldsymbol{D}_{t,1}, \hat{\boldsymbol{\Lambda}}_{11} \in \mathbb{R}^{\mathsf{rk}_\star \times \mathsf{rk}_\star}$. We define

$$\boldsymbol{\Gamma}_t := \boldsymbol{D}_t + \frac{1}{1.05}\boldsymbol{Z}_{1:\mathsf{rk}}\boldsymbol{Z}_{1:\mathsf{rk}}^\top = \begin{bmatrix} \frac{1}{1.05}\boldsymbol{Z}_{1:\mathsf{rk}_\star}\boldsymbol{Z}_{1:\mathsf{rk}_\star}^\top + \boldsymbol{D}_{t,1} & \frac{1}{1.05d}\boldsymbol{Z}_{1:\mathsf{rk}_\star}\boldsymbol{Z}_2^\top \\ \frac{1}{1.05}\boldsymbol{Z}_2\boldsymbol{Z}_{1:\mathsf{rk}_\star}^\top & \frac{1}{1.05}\boldsymbol{Z}_2\boldsymbol{Z}_2^\top + \boldsymbol{D}_{t,2} \end{bmatrix}$$

and

$$\boldsymbol{\Gamma}_t^{-1} := \begin{bmatrix} (\boldsymbol{\Gamma}_t^{-1})_{11} & (\boldsymbol{\Gamma}_t^{-1})_{12} \\ (\boldsymbol{\Gamma}_t^{-\top})_{12} & (\boldsymbol{\Gamma}_t^{-1})_{22} \end{bmatrix}$$

whenever $\boldsymbol{\Gamma}_t$ is invertible. Lemma 5 is stated as follows:

**Lemma 5.** *We consider the following setting:*

$$\alpha \in [0, 0.5) : \quad \frac{r_s}{r} \to \varphi \in (0, \infty), \quad \eta \ll \frac{1}{d\,r^{1-\alpha}\log^4 d}, \quad \kappa_d = \frac{1}{\log^{3.5} d},$$

$$\alpha > 0.5 : \quad r_s \asymp 1, \quad \eta \ll \frac{1}{d\,r_u^{2+\alpha}\log^3 d}, \quad \kappa_d = \frac{1}{r_u \log^{2.5} d}.$$

$\mathcal{G}_{init}$ *implies the following:*

- *For $\alpha \geq 0$ and $K \leq \mathsf{rk}_\star \leq \mathsf{rk}$, we have for $\eta t \leq \frac{1}{2}(K+1)^\alpha \log\left(\frac{d\log^{1.5} d}{r_s}\right)$,*

$$\boldsymbol{D}_t \succeq \frac{\boldsymbol{\Lambda}_{\ell_2}^{-1}\hat{\boldsymbol{\Lambda}}\left(\frac{\boldsymbol{A}_{t,22}}{\boldsymbol{A}_{t,12}} - \boldsymbol{I}_{\mathsf{rk}}\right)}{2} - \frac{1.2\kappa_d r_s}{d}\boldsymbol{I}_{\mathsf{rk}}, \quad \boldsymbol{D}_{t,2} \succeq \frac{\log^3 d - 1}{\log^3 d}\left(\frac{0.5 r_s}{d\log^{1.5} d}\right)^{\left(\frac{K+1}{\mathsf{rk}_\star + 1}\right)^\alpha}\boldsymbol{I}_{\mathsf{rk}-\mathsf{rk}_\star}.$$

- *For $r_\star = \lfloor r_s(1 - \log^{\frac{-1}{2}} d) \wedge r \rfloor$ and $r_{u_\star} = r_s$, and $\eta t \leq \frac{1}{2}(\mathsf{rk}_\star + 1)^\alpha \log\left(\frac{d \log^{1.5} d}{r_s}\right)$, we have*

$$\boldsymbol{\Gamma}_t \succeq \frac{\boldsymbol{\Lambda}_{\ell_2}^{-1} \hat{\boldsymbol{\Lambda}}\left(\frac{\boldsymbol{A}_{t,22}}{\boldsymbol{A}_{t,12}} - \boldsymbol{I}_{\mathsf{rk}}\right)}{2} + \begin{bmatrix} \frac{C_1 r_s}{d \log^{4.5} d} \boldsymbol{I}_{\mathsf{rk}_\star} & 0 \\ 0 & -\frac{C_2 r_s}{d \log^2 d} \boldsymbol{I}_{\mathsf{rk}-\mathsf{rk}_\star} \end{bmatrix} \succ 0, \qquad \text{(F.23)}$$

*where*

$$C_1 = \begin{cases} \frac{1}{10}, & \alpha \in [0, 0.5) \\ \left(\frac{1}{1.1 r_s^6} - \frac{1.3}{\sqrt{\log d}}\right), & \alpha > 0.5 \end{cases} \quad \text{and} \quad C_2 = \begin{cases} 2.1\left(1 + \frac{1}{\sqrt{\varphi}}\right)^2, & \alpha \in [0, 0.5) \\ 2, & \alpha > 0.5. \end{cases}$$

*Within the same time interval, we have*

$$\boldsymbol{\Gamma}_t^{-1} \preceq \left( \frac{\boldsymbol{\Lambda}_{\ell_2}^{-1} \hat{\boldsymbol{\Lambda}}\left(\frac{\boldsymbol{A}_{t,22}}{\boldsymbol{A}_{t,12}} - \boldsymbol{I}_{\mathsf{rk}}\right)}{2} + \begin{bmatrix} \frac{C_1 r_s}{d \log^{4.5} d} \boldsymbol{I}_{\mathsf{rk}_\star} & 0 \\ 0 & \frac{-C_2 r_s}{d \log^2 d} \boldsymbol{I}_{\mathsf{rk}-\mathsf{rk}_\star} \end{bmatrix} \right)^{-1}. \qquad \text{(F.24)}$$

- *For $\alpha > 0$, we have*

$$(\boldsymbol{\Gamma}_t^{-1})_{11} \preceq \left( \boldsymbol{D}_{t,1} + \frac{1}{2} \boldsymbol{Z}_{1:\mathsf{rk}_\star} \boldsymbol{Z}_{1:\mathsf{rk}_\star}^\top \right)^{-1},$$

*for $0.001 > \delta \geq \log^{\frac{-1}{4}} d$*

$$\begin{cases} r_\star = \lfloor r_s(1 - \delta) \wedge r \rfloor \text{ and } \eta t \leq \frac{1}{2}\left(r_s(1 - \sqrt{\delta}) \wedge r\right)^\alpha \log\left(\frac{d \log^{1.5} d}{r_s}\right), & \alpha \in (0, 0.5) \\ r_{u_\star} = r_s \text{ and } \eta t \leq \frac{1}{2} r_s^\alpha \log\left(\frac{d \log^{1.5} d}{r_s}\right), & \alpha > 0.5 \end{cases}$$

*provided that*

$$\begin{cases} d \geq \Omega_\beta(1) \vee \exp\left(2.5 \alpha^{-8}\right), & \alpha \in (0, 0.5) \\ d \geq \Omega_{r_s}(1), & \alpha > 0.5. \end{cases}$$

*Proof.* For the first part of the first item, by (R.2), we have

$$\boldsymbol{D}_t = \frac{\boldsymbol{\Lambda}_{\ell_2}^{-1} \hat{\boldsymbol{\Lambda}}\left(\frac{\boldsymbol{A}_{t,22}}{\boldsymbol{A}_{t,12}} - \boldsymbol{I}_{\mathsf{rk}}\right)}{2} - \frac{\eta}{2} \boldsymbol{\Lambda}_{\ell_2}^{-1} \hat{\boldsymbol{\Lambda}}^2 - \frac{1.1 \kappa_d r_s}{d} \boldsymbol{I}_{\mathsf{rk}} \overset{(a)}{\succeq} \frac{\boldsymbol{\Lambda}_{\ell_2}^{-1} \hat{\boldsymbol{\Lambda}}\left(\frac{\boldsymbol{A}_{t,22}}{\boldsymbol{A}_{t,12}} - \boldsymbol{I}_{\mathsf{rk}}\right)}{2} - \frac{1.2 \kappa_d r_s}{d} \boldsymbol{I}_{\mathsf{rk}},$$

where (a) follows $\eta \ll \frac{\kappa_d r_s}{d}$. Moreover, since $\boldsymbol{\Lambda}_{\ell_2}^{-1} \hat{\boldsymbol{\Lambda}} \succeq (1 - \frac{1}{\log^3 d}) \boldsymbol{I}_{\mathsf{rk}}$, by (R.4), we have

$$\boldsymbol{D}_{t,2} \succeq \left(1 - \frac{1}{\log^3 d}\right) \left[ \frac{\left(\boldsymbol{I}_{\mathsf{rk}-\mathsf{rk}_\star} - \eta \hat{\boldsymbol{\Lambda}}_{22}\right)^t}{\left(\boldsymbol{I}_{\mathsf{rk}-\mathsf{rk}_\star} + \eta \hat{\boldsymbol{\Lambda}}_{22}\right)^t - \left(\boldsymbol{I}_{\mathsf{rk}-\mathsf{rk}_\star} - \eta \hat{\boldsymbol{\Lambda}}_{22}\right)^t} - \frac{1.3 \kappa_d r_s}{d} \boldsymbol{I}_{\mathsf{rk}-\mathsf{rk}_\star} \right] \quad \text{(F.25)}$$

We observe that

$$\frac{\left(\boldsymbol{I}_{\mathsf{rk}-\mathsf{rk}_\star} - \eta \hat{\boldsymbol{\Lambda}}_{22}\right)^t}{\left(\boldsymbol{I}_{\mathsf{rk}-\mathsf{rk}_\star} + \eta \hat{\boldsymbol{\Lambda}}_{22}\right)^t - \left(\boldsymbol{I}_{\mathsf{rk}-\mathsf{rk}_\star} - \eta \hat{\boldsymbol{\Lambda}}_{22}\right)^t} \succeq \frac{\boldsymbol{I}_{\mathsf{rk}-\mathsf{rk}_\star}}{\exp\left(\frac{2t \eta \hat{\boldsymbol{\Lambda}}_{22}}{1 - \eta \hat{\boldsymbol{\Lambda}}_{22}}\right) - \boldsymbol{I}_{\mathsf{rk}-\mathsf{rk}_\star}}$$

$$\overset{(b)}{\succeq} \left(\frac{r_s}{d \log^{1.5} d}\right)^{(1 + 2\eta \mathsf{rk}_\star^{-\alpha})\left(\frac{K+1}{\mathsf{rk}_\star+1}\right)^\alpha} \boldsymbol{I}_{\mathsf{rk}-\mathsf{rk}_\star}$$

$$\overset{(c)}{\succeq} \left(\frac{0.9 r_s}{d \log^{1.5} d}\right)^{\left(\frac{K+1}{\mathsf{rk}_\star+1}\right)^\alpha} \boldsymbol{I}_{\mathsf{rk}-\mathsf{rk}_\star},$$

where we use $\eta t \leq \frac{1}{2}(K + 1)^\alpha \log\left(\frac{d \log^{1.5} d}{r_s}\right)$ in (b) and $d \geq \Omega(1)$ in (c). By (F.25), the first item follows.

For the second item, by Proposition 24 with $\varepsilon = \frac{1}{\log^2 d}$ for $\alpha \in [0, 0.5)$ and $\varepsilon = \frac{1}{r_u \log^2 d}$ for $\alpha > 0.5$, we have

$$\mathbf{\Gamma}_t \succeq \frac{\mathbf{\Lambda}_{\ell_2}^{-1} \hat{\mathbf{\Lambda}} \left( \frac{\mathbf{A}_{t,22}}{\mathbf{A}_{t,12}} - \mathbf{I}_{\mathsf{rk}} \right)}{2} - \frac{1.2\kappa_d r_s}{d} \mathbf{I}_{\mathsf{rk}} + \frac{1}{1.05} \begin{bmatrix} \varepsilon \mathbf{Z}_{1:\mathsf{rk}_\star} \mathbf{Z}_{1:\mathsf{rk}_\star}^\top & 0 \\ 0 & \frac{-\varepsilon}{1-\varepsilon} \mathbf{Z}_2 \mathbf{Z}_2^\top \end{bmatrix}.$$

For $\alpha \in [0, 0.5)$, since $\kappa_d = \frac{1}{\log^{3.5} d}$, by (H.1), we have

$$\frac{\varepsilon}{1.05} \mathbf{Z}_{1:r_\star} \mathbf{Z}_{1:r_\star}^\top - \frac{1.2\kappa_d r_s}{d} \mathbf{I}_{r_\star} \succeq \frac{r_s}{d \log^3 d} \frac{1}{6.25} \mathbf{I}_{r_\star} - \frac{1.2\kappa_d r_s}{d} \mathbf{I}_{r_\star} \succ \frac{1}{10} \frac{r_s}{d \log^3 d} \mathbf{I}_{r_\star}.$$

Similarly by (H.3), we have

$$\frac{-\varepsilon}{1-\varepsilon} \frac{1}{1.05} \mathbf{Z}_2 \mathbf{Z}_2^\top - \frac{1.2\kappa_d r_s}{d} \mathbf{I}_{r-r_\star} \succeq -\left(1 + \frac{1}{\sqrt{\varphi}}\right)^2 \frac{2.1 r_s}{d \log^2 d} \mathbf{I}_{r-r_\star}.$$

For $\alpha > 0.5$, since $\kappa_d = \frac{1}{r_u \log^{2.5} d}$ and $r_u = \lceil \log^{2.5} d \rceil$, by (L.1), we have

$$\frac{\varepsilon}{1.05} \mathbf{Z}_{1:r_{u\star}} \mathbf{Z}_{1:r_{u\star}}^\top - \frac{1.2\kappa_d r_s}{d} \mathbf{I}_{r_{u\star}} \succ \frac{r_s}{d \log^{4.5} d} \left( \frac{1}{1.1 r_s^6} - \frac{1.3}{\sqrt{\log d}} \right) \mathbf{I}_{r_{u\star}}.$$

Similarly by (L.2),

$$\frac{-\varepsilon}{1-\varepsilon} \frac{1}{1.05 d} \mathbf{Z}_2 \mathbf{Z}_2^\top - \frac{1.2\kappa_d r_s}{d} \mathbf{I}_{r_u - r_{u\star}} \succeq \frac{-2 r_s}{d \log^2 d} \mathbf{I}_{r_u - r_{u\star}}.$$

Therefore, we have (F.23). By Proposition 25, we have (F.24).

For the last item, we have

$$(\mathbf{\Gamma}_t^{-1})_{11} = \left( \mathbf{D}_{t,1} + \frac{1}{1.05} \mathbf{Z}_{1:\mathsf{rk}_\star} \left( \mathbf{I}_{r_s} + \frac{1}{1.05} \mathbf{Z}_2^\top \mathbf{D}_{t,2}^{-1} \mathbf{Z}_2 \right)^{-1} \mathbf{Z}_{1:\mathsf{rk}_\star}^\top \right)^{-1}. \tag{F.26}$$

For $\alpha \in (0, 0.5)$, if $r_\star = r$, the statement follows. If not by the first item, for $K = \lfloor (1 - \sqrt{\delta}) r_s \rfloor$ and $r_\star = \lfloor r_s (1 - \delta) \rfloor$, we have

$$\frac{K+1}{r_\star + 1} \leq \frac{1 - \sqrt{\delta}}{1 - \delta} + \frac{2}{r_s} \leq 1 - 0.9\sqrt{\delta} \;\Rightarrow\; \left( \frac{K+1}{r_\star + 1} \right)^\alpha \leq 1 - \alpha 0.9 \sqrt{\delta}.$$

Therefore,

$$\mathbf{D}_{t,2} \succeq \left( \frac{0.5 r_s}{d \log^{1.5} d} \right)^{1 - \alpha 0.9\sqrt{\delta}} \mathbf{I}_{r - r_\star} \overset{(d)}{\succeq} \frac{0.5 r_s}{d \log^{1.5} d} \left( \frac{d}{r_s} \right)^{\log^{-1/4} d} \mathbf{I}_{r - r_\star} \overset{(e)}{\succeq} \frac{r_s \log d}{d} \mathbf{I}_{r - r_\star},$$

where we used $d \geq \Omega(1) \vee \exp(2.5 \alpha^{-8})$ in (d) and $d \geq \Omega_\beta(1)$ in (e). By (F.26) and (H.3), we have the statement for $\alpha \in (0, 0.5)$. For $\alpha > 0.5$, $K = r_\star = r_s$, we have

$$\left( \frac{K+1}{r_\star + 1} \right)^\alpha \leq \left( 1 + \frac{1}{r_s + 1} \right)^{0.5} \leq 1 - \frac{1}{2(r_s + 1)}.$$

Therefore,

$$\mathbf{D}_{t,2} \succeq \left( \frac{0.5 r_s}{d \log^{1.5} d} \right)^{1 - \frac{1}{2(r_s + 1)}} \mathbf{I}_{r_u - r_{u\star}} \succeq \frac{r_s \log^8 d}{d} \mathbf{I}_{r - r_\star}$$

for $d \geq \Omega_{r_s}(1)$. By (F.26) and (L.2), we have the statement for $\alpha > 0.5$. $\qquad\square$

**Proposition 15.** *Let*

$$\underline{\mathbf{G}}_0 = (1 + 2\kappa_d) \left( \mathsf{G}_0 - \frac{\kappa_d r_s}{d} \mathbf{I}_{\mathsf{rk}} \right),$$

*Under the parameter choice in Lemma 5, $\mathcal{G}_{init}$ guarantees that:*

- *We have* $\Omega\big(-\log^{\frac{-1}{2}}d\big)\boldsymbol{I}_{\mathsf{rk}} \preceq \underline{\boldsymbol{G}}_t$ *whenever*

$$\eta t \le \begin{cases} \frac{1}{2}\Big(r_s\big(1-\log^{\frac{-1}{2}}d\big)\wedge r\Big)^\alpha \log\Big(\frac{d\log^{1.5}d}{r_s}\Big), & \alpha \in [0,0.5) \\ \frac{1}{2}r_s^\alpha \log\Big(\frac{d\log^{1.5}d}{r_s}\Big), & \alpha > 0.5 \end{cases}.$$

- *Let* $\boldsymbol{\Lambda}_{11}$ *be the* $\mathsf{rk}_\star \times \mathsf{rk}_\star$ *dimensional top-left sub-matrix of* $\boldsymbol{\Lambda}$. *Given* $0.001 \ge \delta \ge \log^{\frac{-1}{4}}d$ *and* $\mathsf{rk}_\star = \Big\{r_\star = \lfloor r_s(1-\delta)\wedge r\rfloor, r_{u_\star} = r_s\Big\}$, *we have*

$$\underline{\boldsymbol{G}}_{t,11} \succeq \frac{1 - \frac{10}{\log^3 d}}{\frac{1.2}{C_{lb}}\frac{d}{r_s}\exp\big(-2\eta t\boldsymbol{\Lambda}_{11}\big)+1}$$

*and*

$$\|\hat{\boldsymbol{\Lambda}}\|_F^2 - \|\hat{\boldsymbol{\Lambda}}_1^{\frac{1}{2}}\underline{\boldsymbol{G}}_{t,11}\hat{\boldsymbol{\Lambda}}_1^{\frac{1}{2}}\|_F^2 \le \sum_{i=(\mathsf{rk}_\star\wedge r)+1}^{r}\hat{\lambda}_i^2 + \sum_{i=1}^{\mathsf{rk}_\star}\hat{\lambda}_i^2\Bigg(1 - \frac{1 - \frac{10}{\log^3 d}}{\frac{1.2}{C_{lb}}\frac{d}{r_s}\exp\big(-2\eta t\lambda_i\big)+1}\Bigg)^2,$$

*for*

$$C_{lb} = \frac{1}{15}\begin{cases}\delta^2, & \alpha\in[0,0.5) \\ \frac{1}{r_s^6}, & \alpha>0.5\end{cases} \quad \text{and} \quad \eta t \le \begin{cases}\frac{1}{2}\Big(r_s\big(1-\sqrt{\delta}\big)\wedge r\Big)^\alpha\log\Big(\frac{d\log^{1.5}d}{r_s}\Big), & \alpha\in[0,0.5) \\ \frac{1}{2}r_s^\alpha\log\Big(\frac{d\log^{1.5}d}{r_s}\Big), & \alpha>0.5.\end{cases}$$

- *For* $\delta = \log^{\frac{-1}{4}}d$, *we define*

$$\mathcal{T}_{lb} := \inf\Bigg\{n\ge 0 \ \Bigg|\ \|\hat{\boldsymbol{\Lambda}}\|_F^2 - \|\hat{\boldsymbol{\Lambda}}_1^{\frac{1}{2}}\underline{\boldsymbol{G}}_{t,11}\hat{\boldsymbol{\Lambda}}_1^{\frac{1}{2}}\|_F^2 \le \sum_{j=(r_s\wedge r)+1}^{r}\lambda_j^2 + \frac{3\|\hat{\boldsymbol{\Lambda}}\|_F^2}{\log^{\frac{1}{8}}d}\Bigg\}.$$

*Then,*

$$\mathcal{T}_{lb} \le \begin{cases}\frac{1}{2\eta}\Big(r_s\big(1-\log^{\frac{-1}{8}}d\big)\wedge r\Big)^\alpha\log\Big(\frac{20d\log^{\frac{3}{4}}(1+d/r_s)}{r_s}\Big), & \alpha\in[0,0.5) \\ \frac{1}{2\eta}r_s^\alpha\log\Big(\frac{20d\log^{\frac{3}{4}}d}{r_s}\Big), & \alpha>0.5.\end{cases}$$

*Proof.* For $\alpha > 0.5$, we assume that $d$ is large enough to guarantee that $\Big(\frac{1}{1.1r_s^6} - \frac{1.3}{\sqrt{\log d}}\Big) > 0$. We observe that

$$\frac{\boldsymbol{\Lambda}_{\ell_2}^{-1}\hat{\boldsymbol{\Lambda}}\Big(\frac{\boldsymbol{A}_{t,11}}{\boldsymbol{A}_{t,12}} - \boldsymbol{I}_{\mathsf{rk}}\Big)}{2} + \frac{\boldsymbol{\Lambda}_{\ell_2}^{-1}\Big(\hat{\boldsymbol{\Lambda}} - \boldsymbol{\Lambda}_{\ell_1}\Big)}{2} + \underline{\boldsymbol{G}}_0 \succeq \boldsymbol{\Gamma}_t,$$

where we used (E.1), $\boldsymbol{\Lambda}_{\ell_2}\succeq\boldsymbol{\Lambda}_{\ell_1}$ and $\eta\|\boldsymbol{\Lambda}\|_F^2 r_s\boldsymbol{I}_{\mathsf{rk}} \ll \frac{\kappa_d r_s}{d}\boldsymbol{\Lambda}_{\ell_1}$.

For the first item, by using $\mathsf{rk}_\star = \Big\{r_\star = \lfloor r_s(1-\log^{\frac{-1}{2}}d)\wedge r\rfloor, r_{u_\star} = r_s\Big\}$, we define

$$\boldsymbol{D}_{lb} := \begin{bmatrix} \frac{C_1 r_s}{d\log^{4.5}d}\boldsymbol{I}_{\mathsf{rk}_\star} & 0 \\ 0 & \frac{-C_2 r_s}{d\log^2 d}\boldsymbol{I}_{\mathsf{rk}-\mathsf{rk}_\star}\end{bmatrix} \quad \text{and} \quad \tilde{\boldsymbol{D}}_{lb} := \frac{\boldsymbol{\Lambda}_{\ell_2}}{\hat{\boldsymbol{\Lambda}}}\boldsymbol{D}_{lb}.$$

We introduce submatrix notation for block-diagonal matrices. Specifically, we write

$$\tilde{\boldsymbol{D}}_{lb} = \begin{bmatrix}\tilde{\boldsymbol{D}}_{lb,1} & 0 \\ 0 & \tilde{\boldsymbol{D}}_{lb,2}\end{bmatrix} \quad \text{and} \quad \frac{\boldsymbol{A}_{t,22}}{\boldsymbol{A}_{t,12}}\pm\boldsymbol{I}_{\mathsf{rk}} = \begin{bmatrix}\Big(\frac{\boldsymbol{A}_{t,22}}{\boldsymbol{A}_{t,12}}\pm\boldsymbol{I}_{\mathsf{rk}}\Big)_{11} & 0 \\ 0 & \Big(\frac{\boldsymbol{A}_{t,22}}{\boldsymbol{A}_{t,12}}\pm\boldsymbol{I}_{\mathsf{rk}}\Big)_{22}\end{bmatrix},$$

where the block dimensions of each submatrix match those of $\boldsymbol{D}_{lb}$. We start with proving the lower bound part. By the second item in Lemma 5, we have

$$\underline{\boldsymbol{G}}_t \succeq \frac{1}{2}\sqrt{\frac{\hat{\boldsymbol{\Lambda}}}{\boldsymbol{\Lambda}_{\ell_2}}}\Bigg(\Big(\frac{\boldsymbol{A}_{t,22}}{\boldsymbol{A}_{t,12}}+\boldsymbol{I}_{\mathsf{rk}}\Big) - \boldsymbol{A}_{t,12}^{-1}\Big(\Big(\frac{\boldsymbol{A}_{t,22}}{\boldsymbol{A}_{t,12}}-\boldsymbol{I}_{\mathsf{rk}}\Big) + 2\tilde{\boldsymbol{D}}_{lb}\Big)^{-1}\boldsymbol{A}_{t,12}^{-1}\Bigg)\sqrt{\frac{\hat{\boldsymbol{\Lambda}}}{\boldsymbol{\Lambda}_{\ell_2}}}$$

$$
+\frac{\frac{\boldsymbol{\Lambda}_{\ell_1}}{\boldsymbol{\Lambda}_{\ell_2}}-\frac{\hat{\boldsymbol{\Lambda}}}{\boldsymbol{\Lambda}_{\ell_2}}}{2}
$$

$$
\succeq\frac{1}{2}\sqrt{\frac{\hat{\boldsymbol{\Lambda}}}{\boldsymbol{\Lambda}_{\ell_2}}}\left(\left(\frac{\boldsymbol{A}_{t,22}}{\boldsymbol{A}_{t,12}}+\boldsymbol{I}_{\mathsf{rk}}\right)-\boldsymbol{A}_{t,12}^{-1}\left(\left(\frac{\boldsymbol{A}_{t,22}}{\boldsymbol{A}_{t,12}}-\boldsymbol{I}_{\mathsf{rk}}\right)+2\tilde{\boldsymbol{D}}_{lb}\right)^{-1}\boldsymbol{A}_{t,12}^{-1}\right)\sqrt{\frac{\hat{\boldsymbol{\Lambda}}}{\boldsymbol{\Lambda}_{\ell_2}}}\tag{F.27}
$$

where we used $\boldsymbol{\Lambda}_{\ell_1}\succ\hat{\boldsymbol{\Lambda}}$ in the second line. We have

$$
\left(\frac{\boldsymbol{A}_{t,22}}{\boldsymbol{A}_{t,12}}+\boldsymbol{I}_{\mathsf{rk}}\right)-\boldsymbol{A}_{t,12}^{-1}\left(\left(\frac{\boldsymbol{A}_{t,22}}{\boldsymbol{A}_{t,12}}-\boldsymbol{I}_{\mathsf{rk}}\right)+2\tilde{\boldsymbol{D}}_{lb}\right)^{-1}\boldsymbol{A}_{t,12}^{-1}
$$

$$
=\frac{(\boldsymbol{A}_{t,22}+\boldsymbol{A}_{t,12})(\boldsymbol{A}_{t,22}-\boldsymbol{A}_{t,12}+2\tilde{\boldsymbol{D}}_{lb}\boldsymbol{A}_{t,12})-\boldsymbol{I}_{\mathsf{rk}}}{\boldsymbol{A}_{t,12}(\boldsymbol{A}_{t,22}-\boldsymbol{A}_{t,12}+2\tilde{\boldsymbol{D}}_{lb}\boldsymbol{A}_{t,12})}
$$

$$
\overset{(a)}{\succeq}\frac{2\tilde{\boldsymbol{D}}_{lb}\left(\frac{\boldsymbol{A}_{t,22}}{\boldsymbol{A}_{t,12}}+\boldsymbol{I}_{\mathsf{rk}}\right)}{\frac{\boldsymbol{A}_{t,22}}{\boldsymbol{A}_{t,12}}-\boldsymbol{I}_{\mathsf{rk}}+2\tilde{\boldsymbol{D}}_{lb}}
$$

$$
=\begin{bmatrix}\dfrac{2\tilde{\boldsymbol{D}}_{lb,1}\left(\frac{\boldsymbol{A}_{t,22}}{\boldsymbol{A}_{t,12}}+\boldsymbol{I}_{\mathsf{rk}}\right)_{11}}{\left(\frac{\boldsymbol{A}_{t,22}}{\boldsymbol{A}_{t,12}}-\boldsymbol{I}_{\mathsf{rk}}\right)_{11}+2\tilde{\boldsymbol{D}}_{lb,1}}&0\\[4ex]0&\dfrac{2\tilde{\boldsymbol{D}}_{lb,2}\left(\frac{\boldsymbol{A}_{t,22}}{\boldsymbol{A}_{t,12}}+\boldsymbol{I}_{\mathsf{rk}}\right)_{22}}{\left(\frac{\boldsymbol{A}_{t,22}}{\boldsymbol{A}_{t,12}}-\boldsymbol{I}_{\mathsf{rk}}\right)_{22}+2\tilde{\boldsymbol{D}}_{lb,2}}\end{bmatrix},\tag{F.28}
$$

where we used $\boldsymbol{A}_{t,22}^2-\boldsymbol{A}_{t,12}^2\succ\boldsymbol{I}_{\mathsf{rk}}$ (by (R.2)) and $\boldsymbol{A}_{t,22}-\boldsymbol{A}_{t,12}+2\tilde{\boldsymbol{D}}_{lb}\boldsymbol{A}_{t,12}\succ 0$ (by (F.23)) in (a). Since $\frac{\boldsymbol{A}_{t,22}}{\boldsymbol{A}_{t,12}}-\boldsymbol{I}_{\mathsf{rk}}\succ 0$ and $\tilde{\boldsymbol{D}}_{lb,1}\succ 0$, it is enough to look at the bottom-right submatrix in (F.28) for the lower bound part. We have

$$
\frac{2\tilde{\boldsymbol{D}}_{lb,2}\left(\frac{\boldsymbol{A}_{t,22}}{\boldsymbol{A}_{t,12}}+\boldsymbol{I}_{\mathsf{rk}}\right)_{22}}{\left(\frac{\boldsymbol{A}_{t,22}}{\boldsymbol{A}_{t,12}}-\boldsymbol{I}_{\mathsf{rk}}\right)_{22}+2\tilde{\boldsymbol{D}}_{lb,2}}=\frac{2\tilde{\boldsymbol{D}}_{lb,2}\left(\frac{\boldsymbol{A}_{t,22}}{\boldsymbol{A}_{t,12}}+\boldsymbol{I}_{\mathsf{rk}}\right)_{22}}{\left(\frac{\boldsymbol{A}_{t,22}}{\boldsymbol{A}_{t,12}}+\boldsymbol{I}_{\mathsf{rk}}\right)_{22}-2\boldsymbol{I}_{\mathsf{rk}-\mathsf{rk}_\star}+2\tilde{\boldsymbol{D}}_{lb,2}}.\tag{F.29}
$$

Note that by (R.4),

$$
\left(\frac{\boldsymbol{A}_{t,22}}{\boldsymbol{A}_{t,12}}+\boldsymbol{I}_{\mathsf{rk}}\right)_2\succeq\frac{2(\boldsymbol{I}_{\mathsf{rk}-\mathsf{rk}_\star}+\eta\hat{\boldsymbol{\Lambda}}_2)^t}{(\boldsymbol{I}_{\mathsf{rk}-\mathsf{rk}_\star}+\eta\hat{\boldsymbol{\Lambda}}_2)^t-(\boldsymbol{I}_{\mathsf{rk}-\mathsf{rk}_\star}-\eta\hat{\boldsymbol{\Lambda}}_2)^t}
$$

$$
\succeq 2\boldsymbol{I}_{\mathsf{rk}-\mathsf{rk}_\star}+\frac{2(\boldsymbol{I}_{\mathsf{rk}-\mathsf{rk}_\star}-\eta^2\hat{\boldsymbol{\Lambda}}_2^2)^t\exp\left(-2t\eta\hat{\boldsymbol{\Lambda}}_2\right)}{\boldsymbol{I}_{\mathsf{rk}-\mathsf{rk}_\star}-(\boldsymbol{I}_{\mathsf{rk}-\mathsf{rk}_\star}-\eta^2\hat{\boldsymbol{\Lambda}}_2^2)^t\exp\left(-2t\eta\hat{\boldsymbol{\Lambda}}_2\right)}
$$

$$
\overset{(b)}{\succeq}\left(2+\frac{0.9r_s}{d\log^{1.5}d}\right)\boldsymbol{I}_{\mathsf{rk}-\mathsf{rk}_\star},
$$

where we use $\eta t\leq\frac{1}{2}(\mathsf{rk}_\star+1)^\alpha\log\left(\frac{d\log^{1.5}d}{r_s}\right)$ in (b). Hence, for $d\geq\Omega(1)$

$$
\text{(F.29)}\succeq\frac{2\left(2+\frac{0.9r_s}{d\log^{1.5}d}\right)\tilde{\boldsymbol{D}}_{lb,2}}{\frac{0.9r_s}{d\log^{1.5}d}\boldsymbol{I}_{\mathsf{rk}-\mathsf{rk}_\star}+\tilde{\boldsymbol{D}}_{lb,2}}\overset{(c)}{\succeq}\frac{12\tilde{\boldsymbol{D}}_{lb,2}}{\frac{r_s}{d\log^{1.5}d}\boldsymbol{I}_{\mathsf{rk}-\mathsf{rk}_\star}}\overset{(d)}{\succeq}\frac{-15C_2}{\log^{0.5}d}\boldsymbol{I}_{\mathsf{rk}-\mathsf{rk}_\star},\tag{F.30}
$$

where we used $\tilde{\boldsymbol{D}}_{lb,2}\succeq\frac{-1.1C_2r_s}{d\log^2d}\boldsymbol{I}_{\mathsf{rk}}$ in (c) and (d). The first item follows from (F.30).

For the second and third items, let $\left(\frac{\boldsymbol{A}_{t,22}}{\boldsymbol{A}_{t,12}}\pm\boldsymbol{I}_{\mathsf{rk}}\right)_{11}$ denote the $\mathsf{rk}_\star\times\mathsf{rk}_\star$ dimensional top-left submatrices with $\mathsf{rk}_\star=\left\{r_\star=\lfloor r_s(1-\delta)\wedge r\rfloor,r_{u_\star}=r_s\right\}$. By using the third item in Lemma 5, we immediately observe that for $\alpha>0$, $\underline{\boldsymbol{G}}_{t,11}\succeq 0$ and

$$
\underline{\boldsymbol{G}}_{t,11}\overset{(e)}{\succeq}\left(1-\frac{10}{\log^3d}\right)\frac{1}{2}\frac{\frac{2C_{\mathrm{lb}}r_s}{d}\left(\frac{\boldsymbol{A}_{t,22}}{\boldsymbol{A}_{t,12}}+\boldsymbol{I}_{\mathsf{rk}}\right)_{11}}{\left(\frac{\boldsymbol{A}_{t,11}}{\boldsymbol{A}_{t,12}}-\boldsymbol{I}_{\mathsf{rk}}\right)_{11}+\frac{2C_{\mathrm{lb}}r_s}{d}\boldsymbol{I}_{\mathsf{rk}_\star}},\tag{F.31}
$$

for

$$C_{\text{lb}} = \frac{1}{15} \begin{cases} \delta^2, & \alpha \in (0, 0.5) \\ \frac{1}{r_s^6}, & \alpha > 0.5 \end{cases} \quad \text{and } \eta t \leq \begin{cases} \frac{1}{2}\left(r_s(1 - \sqrt{\delta}) \wedge r\right)^\alpha \log\left(\frac{d\log^{1.5} d}{r_s}\right), & \alpha \in (0, 0.5) \\ \frac{1}{2} r_s^\alpha \log\left(\frac{d\log^{1.5} d}{r_s}\right), & \alpha > 0.5, \end{cases}$$

where we used $\boldsymbol{\Lambda}_{\ell_2} \succeq \hat{\boldsymbol{\Lambda}} \succeq \left(1 - \frac{0.5}{\log^4 d}\right)\boldsymbol{\Lambda}_{\ell_2}$, and followed the steps in (F.27)- (F.28) with (H.1) and (L.1) to obtain (e). Then, by (R.4), we have

$$\frac{1}{2} \frac{\frac{2C_{\text{lb}} r_s}{d}\left(\frac{\boldsymbol{A}_{t,22}}{\boldsymbol{A}_{t,12}} + \boldsymbol{I}_{\text{rk}}\right)_{11}}{\left(\frac{\boldsymbol{A}_{t,11}}{\boldsymbol{A}_{t,12}} - \boldsymbol{I}_{\text{rk}}\right)_{11} + \frac{2C_{\text{lb}} r_s}{d}\boldsymbol{I}_{\text{rk}_\star}} \succeq \frac{\boldsymbol{I}_{\text{rk}_\star}}{\left(\frac{1}{C_{\text{lb}}}\frac{d}{r_s} - 1\right)\frac{(\boldsymbol{I}_{\text{rk}_\star} - \eta\hat{\boldsymbol{\Lambda}}_1)^t}{(\boldsymbol{I}_{\text{rk}_\star} + \eta\hat{\boldsymbol{\Lambda}}_1)^t} + \boldsymbol{I}_{\text{rk}_\star}}$$

$$\succeq \frac{\boldsymbol{I}_{\text{rk}_\star}}{\frac{1.1}{C_{\text{lb}}}\frac{d}{r_s}\exp\left(-2\eta t\hat{\boldsymbol{\Lambda}}_1\right) + \boldsymbol{I}_{\text{rk}_\star}}.$$

Consequently, by observing $\hat{\boldsymbol{\Lambda}} \succeq \boldsymbol{\Lambda}_{\ell_1} - \tilde{C}\eta\boldsymbol{I}_{\text{rk}}$ and using the lower bounds for $\boldsymbol{\Lambda}_{\ell_1}$ in Propositions 12 and 13, we have

$$\underline{\boldsymbol{G}}_{t,11} \succeq \frac{1 - \frac{10}{\log^3 d}}{\frac{1.2}{C_{\text{lb}}}\frac{d}{r_s}\exp\left(-2\eta t\boldsymbol{\Lambda}_{11}\right) + 1},$$

where $\boldsymbol{\Lambda}_{11}$ denotes the $\text{rk}_\star \times \text{rk}_\star$ dimensional top-left sub-matrix of $\boldsymbol{\Lambda}$. Therefore,

$$\|\hat{\boldsymbol{\Lambda}}\|_F^2 - \|\hat{\boldsymbol{\Lambda}}^{\frac{1}{2}}\underline{\boldsymbol{G}}_{t,11}\hat{\boldsymbol{\Lambda}}^{\frac{1}{2}}\|_F^2 \leq \sum_{i=(\text{rk}_\star \wedge r)+1}^{r}\hat{\lambda}_i^2 + \sum_{i=1}^{\text{rk}_\star}\hat{\lambda}_i^2\left(1 - \frac{1 - \frac{10}{\log^3 d}}{\frac{1.2}{C_{\text{lb}}}\frac{d}{r_s}\exp\left(-2\eta t\lambda_i\right) + 1}\right)^2, \quad \text{(F.32)}$$

which proves the second item for $\alpha > 0$. Moreover, since (F.22) is in the eigenbasis of $\underline{\boldsymbol{G}}_0$, the arguments in (F.27)-(F.28) and the condition in (H.2) extend (F.31) to $\alpha = 0$ in the eigenbasis of $\underline{\boldsymbol{G}}_0$ for $d \geq \Omega(1)$. Given (F.31), we can extend (F.32) to $\alpha = 0$ as the Frobenious norm is basis independent.

For the third item, for $\alpha > 0.5$ and $t \geq \frac{1}{2\eta}r_s^\alpha\log\left(\frac{20d\log^{\frac{3}{4}} d}{r_s}\right)$, we have

$$\text{(F.32)} \leq \sum_{i=(r_s \wedge r)+1}^{r}\hat{\lambda}_i^2 + \frac{\|\hat{\boldsymbol{\Lambda}}\|_F^2}{\log^{\frac{1}{2}} d} \leq \sum_{i=(r_s \wedge r)+1}^{r}\lambda_i^2 + \frac{\|\hat{\boldsymbol{\Lambda}}\|_F^2}{\log^{\frac{1}{2}} d},$$

which gives us the corresponding bound for $\mathcal{T}_{lb}$.

For $\alpha \in [0, 0.5)$ and $t \geq \frac{1}{2\eta}\left(r_s(1 - \log^{\frac{-1}{8}} d) \wedge r\right)^\alpha\log\left(\frac{20d\log^{\frac{3}{4}} d(1 + d/r_s)}{r_s}\right)$, we have

$$\text{(F.32)} \leq \sum_{i=(r_s \wedge r)+1}^{r}\hat{\lambda}_i^2 + \sum_{i=\lfloor r_s\left(1 - \log^{\frac{-1}{8}} d\right)\wedge r\rfloor+1}^{r_s \wedge r}\hat{\lambda}_i^2$$

$$+ \sum_{i=1}^{\lfloor r_s\left(1 - \log^{\frac{-1}{8}} d\right)\wedge r\rfloor}\hat{\lambda}_i^2\left(1 - \frac{1 - \frac{10}{\log^3 d}}{\frac{1.2}{C_{\text{lb}}}\frac{d}{r_s}\exp\left(-2\eta t\lambda_i\right) + 1}\right)^2$$

$$\leq \sum_{i=(r_s \wedge r)+1}^{r}\lambda_i^2 + \frac{3\|\hat{\boldsymbol{\Lambda}}\|_F^2}{\log^{\frac{1}{8}} d},$$

which gives us its bound for $\mathcal{T}_{lb}$. $\qquad\square$

### F.4.2 Upper bounding system

In this section, we consider (F.13). For notational convenience, we multiply both sides by the factor $(1 - 2\kappa_d)$ and use a generic learning rate $\eta$, i.e.,

$$\bar{\boldsymbol{V}}_{t+1} = \bar{\boldsymbol{V}}_t(\boldsymbol{I}_{\text{rk}} + \eta\bar{\boldsymbol{V}}_t)^{-1} + \eta\left(\boldsymbol{\Lambda}_{u_1}^2 + \tilde{C}\eta\|\boldsymbol{\Lambda}\|_F^2 r_s\boldsymbol{\Lambda}_{u_1}\right), \quad \text{where } \bar{\boldsymbol{V}}_t = 2\boldsymbol{\Lambda}_{u_2}^{\frac{1}{2}}\bar{\boldsymbol{G}}_t\boldsymbol{\Lambda}_{u_2}^{\frac{1}{2}} - \boldsymbol{\Lambda}_{u_1}.$$

The main result of this section is stated in Proposition 16. To establish it, we first prove an auxiliary result:

**Lemma 6.** *The following statement holds:*

- *The reference sequence satisfies $T_t \succeq \frac{\kappa_d r_s}{d} I_{\mathsf{rk}}$ and $\{t \geq 0 : \|T_t\|_2 > 1.2\kappa_d\} = \infty$.*

- *For $r_{u_\star} = 2r_s$, we have*

$$
\begin{cases}
\bar{G}_0 = \frac{2.2\left(1+\frac{1}{\sqrt{\varphi}}\right)^2 r_s}{d} I_r \succeq (1 - 2\kappa_d)\left(G_0 + \frac{\kappa_d r_s}{d} I_r\right), & \alpha \in [0, 0.5) \\[2mm]
\bar{G}_0 = \frac{5.5}{d}\begin{bmatrix} 2r_s I_{r_{u_\star}} & 0 \\ 0 & r_u I_{r-r_{u_\star}} \end{bmatrix} \succeq (1 - 2\kappa_d)\left(G_{0,11} + \frac{\kappa_d r_s}{d} I_{r_u}\right), & \alpha > 0.5
\end{cases}
\tag{F.33}
$$

*provided that $\mathcal{G}_{init}$ holds.*

For the following, we introduce $\hat{T}_t := \frac{\Lambda_{u_2}}{\Lambda_{\ell_1}}\frac{(3\kappa_d+1)}{\kappa_d(1-\kappa_d)}T_t$. Note that for $d \geq \Omega(1)$, we have

$$
\hat{T}_{t+1} = \hat{T}_t + 2(1-2\kappa_d)\eta\Lambda_{\ell_1}\hat{T}_t\left(I_{\mathsf{rk}} - \hat{T}_t\right) \quad \text{and} \quad \frac{\kappa_d}{1.1}\frac{\Lambda_{\ell_1}}{\Lambda_{u_2}} \preceq \frac{T_t}{\hat{T}_t} \preceq \kappa_d I_{\mathsf{rk}}. \tag{F.34}
$$

By Proposition 34, we have

$$
1.1 \wedge \hat{T}_{0,ii} \exp\left(2\eta t\lambda_i\right) \geq \hat{T}_{t,ii}
$$

$$
\geq \frac{1}{2}
\begin{cases}
1 \wedge \hat{T}_{0,ii} \exp\left(\frac{(1-2\kappa_d)2\eta t\left(\lambda_i - \frac{0.1 r^{-\alpha}}{\log^4 d}\right)}{1+2(1-2\kappa_d)\eta\lambda_i}\right), & \alpha \in [0, 0.5) \\[3mm]
1 \wedge \hat{T}_{0,ii} \exp\left(\frac{(1-2\kappa_d)2\eta t\left(\lambda_i - \frac{1}{(r_u+1)^\alpha} - \frac{0.1}{r_u^{2+\alpha}\log^4 d}\right)}{1+2(1-2\kappa_d)\eta\lambda_i}\right), & \alpha > 0.5.
\end{cases}
\tag{F.35}
$$

*Proof of Lemma 6.* For the first item, by Proposition 34, we have

$$
\hat{T}_t \succeq \hat{T}_0 \overset{(a)}{\Rightarrow} T_t \succeq T_0 = \frac{\kappa_d r_s}{d} I_{\mathsf{rk}},
$$

where we multiplied each side with $\frac{\kappa_d(1-\kappa_d)}{3\kappa_d+1}\frac{\Lambda_{\ell_1}}{\Lambda_{u_2}}$ for (a). Moreover, by (F.34)-(F.35), we have

$$
T_t \preceq \kappa_d\hat{T}_t \preceq 1.1 I_{\mathsf{rk}} \Rightarrow \{t \geq 0 : \|T_t\|_2 > 1.2\kappa_d\} = \infty.
$$

The second item follows (E.2) and (H.3) (for $\alpha \in [0, 0.5)$) and (L.2) (for $\alpha > 0.5$). $\qquad\square$

**Proposition 16.** *We consider $\mathsf{rk} \in \{r, r_u\}$, where $r_u = \lceil \log^{2.5} d \rceil$, and*

$$
\alpha \in [0, 0.5): \quad \frac{r_s}{r} \to \varphi \in (0, \infty), \quad \eta \ll \frac{1}{d\,r^{1-\alpha}\log^4 d}, \quad \kappa_d = \frac{1}{\log^{3.5} d},
$$

$$
\alpha > 0.5: \qquad r_s \asymp 1, \qquad\qquad \eta \ll \frac{1}{d\,r_u^{2+\alpha}\log^3 d}, \quad \kappa_d = \frac{1}{r_u\log^{2.5} d}.
$$

*If $\bar{G}_0$ are taken as in (F.33), we have the following:*

- *$\{\bar{G}_t\}_{n\in\mathbb{N}}$ is diagonal and satisfies*

$$
\frac{r_s}{d} I_{\mathsf{rk}} \preceq \bar{G}_{t+1} \preceq \bar{G}_t + \eta\left((1+\kappa_d)\Lambda_{u_1}\bar{G}_t + (1+\kappa_d)\bar{G}_t\Lambda_{u_1} - 2\bar{G}_t\Lambda_{u_2}\bar{G}_t\right) \preceq 1.1 I_{\mathsf{rk}}.
$$

- *For $\alpha \in [0, 0.5)$ and $d \geq \Omega(1)$, we have for $t \leq \frac{1}{2\eta}r^\alpha \log\left(\frac{d\log^{1.5} d}{r_s}\right)$ :*

  - $T_t^{-\frac{1}{2}}\bar{G}_j T_t^{-\frac{1}{2}} \preceq \frac{11\left(1+\frac{1}{\sqrt{\varphi}}\right)^2}{\kappa_d} I_r$ for $0 \leq j \leq t$.
  - $T_t^{-\frac{1}{2}}\left(\eta\sum_{j=1}^t \bar{G}_{j-1}\right)T_t^{-\frac{1}{2}} \preceq \frac{5.5\left(1+\frac{1}{\sqrt{\varphi}}\right)^2}{\kappa_d}(2\eta t \vee r^\alpha)I_r$.
  - $\bar{G}_t \preceq \left(1.1\bar{G}_0 \exp\left(2\eta t\Lambda\right) \wedge I_r\right) + o_d(1)$

- $-\|\mathbf{\Lambda}\|_{\mathrm{F}}^2 - Tr(\mathbf{\Lambda}\bar{\mathbf{G}}_t\mathbf{\Lambda}) \geq \sum_{i=1}^{r}\lambda_i^2\Big(1 - \frac{2.5\left(1+\frac{1}{\sqrt{\varphi}}\right)^2 r_s}{d}\exp\left(2\eta t\lambda_i\right)\Big)_+ - o_d(1)$.

- *For $\alpha > 0.5$ and $d \geq \Omega_{r_s}(1)$, we have for $t \leq \frac{1}{2\eta}r_s^\alpha\log\left(\frac{d\log^{1.5}d}{r_s}\right)$ :*

  - $\mathbf{T}_t^{-\frac{1}{2}}\bar{\mathbf{G}}_j\mathbf{T}_t^{-\frac{1}{2}} \preceq \frac{26.4 r_u}{\kappa_d}\mathbf{I}_{r_u}$ *for $0 \leq j \leq t$.*
  - $\mathbf{T}_t^{-\frac{1}{2}}\left(\eta\sum_{j=1}^{t}\bar{\mathbf{G}}_{j-1}\right)\mathbf{T}_n^{-\frac{1}{2}} \preceq \frac{15 r_u}{\kappa_d}(2r_s)^\alpha\log d\mathbf{I}_{r_u}$.
  - $\bar{\mathbf{G}}_t \preceq \left(1.1\bar{\mathbf{G}}_0\exp\left(2\eta t\mathbf{\Lambda}_{11}\right)\wedge\mathbf{I}_{r_u}\right) + o_d(1)$.
  - $\|\mathbf{\Lambda}_{11}\|_F^2 - Tr(\mathbf{\Lambda}_{11}\bar{\mathbf{G}}_{t,11}\mathbf{\Lambda}_{11}) \geq \sum_{i=1}^{r_s}\lambda_i^2\Big(1 - \frac{12.1 r_s}{d}\exp\left(2\eta t\lambda_i\right)\Big)_+ + \sum_{i=r_s+1}^{r_u}\lambda_i^2 - o_d(1)$.

*Proof.* Given that $\frac{r_s}{d}\mathbf{I}_{\mathrm{rk}} \preceq \bar{\mathbf{G}}_t \preceq 1.1\mathbf{I}_{\mathrm{rk}}$, we have

$$\bar{\mathbf{G}}_{t+1} \overset{(a)}{\preceq} \bar{\mathbf{G}}_t + \eta\left(\mathbf{\Lambda}_{u_1}\bar{\mathbf{G}}_t + \bar{\mathbf{G}}_t\mathbf{\Lambda}_{u_1} - 2\bar{\mathbf{G}}_t\mathbf{\Lambda}_{u_2}\bar{\mathbf{G}}_t\right) + 1.1\tilde{C}\eta^2\|\mathbf{\Lambda}\|_{\mathrm{F}}^2 r_s\mathbf{I}_{\mathrm{rk}}$$

$$\overset{(b)}{\preceq} \bar{\mathbf{G}}_t + \eta\left((1+\kappa_d)\mathbf{\Lambda}_{u_1}\bar{\mathbf{G}}_t + (1+\kappa_d)\bar{\mathbf{G}}_t\mathbf{\Lambda}_{u_1} - 2\bar{\mathbf{G}}_t\mathbf{\Lambda}_{u_2}\bar{\mathbf{G}}_t\right),$$

$$\bar{\mathbf{G}}_{t+1} \overset{(c)}{\succeq} \bar{\mathbf{G}}_t + \eta\left(\mathbf{\Lambda}_{u_1}\bar{\mathbf{G}}_t + \bar{\mathbf{G}}_t\mathbf{\Lambda}_{u_1} - 2\bar{\mathbf{G}}_t\mathbf{\Lambda}_{u_2}\bar{\mathbf{G}}_t\right) - 1.1\tilde{C}\|\mathbf{\Lambda}\|_{\mathrm{F}}^2 r_s\eta^2\mathbf{I}_{\mathrm{rk}}$$

$$\overset{(d)}{\succeq} \bar{\mathbf{G}}_t + \eta\left((1-\kappa_d)\mathbf{\Lambda}_{u_1}\bar{\mathbf{G}}_t + (1-\kappa_d)\bar{\mathbf{G}}_t\mathbf{\Lambda}_{u_1} - 2\bar{\mathbf{G}}_t\mathbf{\Lambda}_{u_2}\bar{\mathbf{G}}_t\right)$$

where we use $-2\mathbf{\Lambda}_{u_2} \preceq \bar{\mathbf{V}}_t^3\left(\mathbf{I}_r + \eta\bar{\mathbf{V}}_t\right)^{-1} \preceq 2\mathbf{\Lambda}_{u_2}$ in (a) and (c), and we use $\eta\|\mathbf{\Lambda}\|_{\mathrm{F}}^2 r_s \ll \kappa_d\mathrm{rk}^{-\alpha}\frac{r_s}{d}$ in (b) and (d). By Proposition 34, we have $\frac{r_s}{d}\mathbf{I}_{\mathrm{rk}} \preceq \bar{\mathbf{G}}_{t+1} \preceq 1.1\mathbf{I}_{\mathrm{rk}}$, hence, the induction hypothesis holds. Therefore, we have the first item.

By using the first item and Proposition 34, we can write for the given time horizons in second and third items that

$$\bar{\mathbf{G}}_t \preceq \left(\bar{\mathbf{G}}_0\exp\left(2(1+\kappa_d)\eta t\mathbf{\Lambda}_{u_1}\right)\wedge\left(1 + (1+\kappa_d)^2\eta^2\mathbf{\Lambda}_{u_1}^2\right)\mathbf{I}_{\mathrm{rk}}\right)$$

$$\preceq \begin{cases} \left(1.1\bar{\mathbf{G}}_0\exp\left(2\eta t\mathbf{\Lambda}\right)\wedge\mathbf{I}_{\mathrm{rk}}\right) + 2\eta^2\mathbf{I}_{\mathrm{rk}} \\ 1.2\left(\bar{\mathbf{G}}_0\exp\left(2\eta t\mathbf{\Lambda}\right)\wedge\mathbf{I}_{\mathrm{rk}}\right), \end{cases} \tag{F.36}$$

where both upper bounds in (F.36) are valid and will be used in different parts of the proof. The third sub-items immediately follow from the first bound.

For $\alpha \in [0, 0.5)$, we have

$$\hat{\mathbf{T}}_{t,ii} \geq \frac{1}{3}\left(1\wedge\hat{\mathbf{T}}_{0,ii}\exp\left(2\eta t\lambda_i\right)\right) \Rightarrow \mathbf{T}_{t,ii} \geq \frac{1}{4}\left(\kappa_d\wedge\mathbf{T}_{0,ii}\exp\left(2\eta t\lambda_i\right)\right)$$

$$\overset{(e)}{\Rightarrow} \mathbf{T}_{t,ii} \geq \frac{\kappa_d}{4}\left(1\wedge\frac{r_s}{d}\exp\left(2\eta t\lambda_i\right)\right),$$

where we used $\mathbf{T}_0 = \frac{\kappa_d r_s}{d}\mathbf{I}_{\mathrm{rk}}$ in (e). Therefore by (F.33) and the second bound in (F.36), we have for $j \leq t$

$$\frac{\bar{\mathbf{G}}_{j,ii}}{\mathbf{T}_{t,ii}} \leq \frac{1.2\left(1\wedge\left(1+\frac{1}{\sqrt{\varphi}}\right)^2\frac{2.2 r_s}{d}\exp\left(2\eta t\lambda_i\right)\right)}{0.25\kappa_d\left(1\wedge\frac{r_s}{d}\exp\left(2\eta t\lambda_i\right)\right)} \leq \frac{11}{\kappa_d}\left(1+\frac{1}{\sqrt{\varphi}}\right)^2.$$

On the other hand, by using the second bound in (F.36),

$$\frac{\eta\sum_{j=0}^{t-1}\bar{\mathbf{G}}_{j,ii}}{\mathbf{T}_{t,ii}} \leq \frac{11\left(1+\frac{1}{\sqrt{\varphi}}\right)^2}{\kappa_d}\frac{\eta\left(t\wedge\frac{r_s}{d}\sum_{j=0}^{t-1}\exp(2\eta j\lambda_i)\right)}{\left(1\wedge\frac{r_s}{d}\exp\left(2\eta t\lambda_i\right)\right)}$$

$$\leq \frac{5.5\left(1+\frac{1}{\sqrt{\varphi}}\right)^2}{\kappa_d}\begin{cases} \frac{1}{\lambda_i}, & 2\eta t \leq \frac{\log\frac{d}{r_s}}{\lambda_i} \\ 2\eta t, & 2\eta t > \frac{\log\frac{d}{r_s}}{\lambda_i} \end{cases}$$

$$\leq \frac{5.5\left(1+\frac{1}{\sqrt{\varphi}}\right)^2}{\kappa_d}(2\eta t \vee r^\alpha).$$

Lastly, by using the first bound in (F.36), we get

$$\|\mathbf{\Lambda}\|_F^2 - \mathrm{Tr}(\mathbf{\Lambda}\bar{\mathbf{G}}_t\mathbf{\Lambda}) \geq \sum_{i=1}^r \lambda_i^2\left(1 - \frac{2.5\left(1+\frac{1}{\sqrt{\varphi}}\right)^2 r_s}{d}\exp\left(2\eta t\lambda_i\right)\right)_+ - 2\eta^2\|\mathbf{\Lambda}\|_F^2.$$

For $\alpha > 0.5$, we have for $i \leq 2r_s\log^{\frac{1}{\alpha}}d$ and $d \geq \Omega_{r_s}(1)$,

$$\hat{\mathbf{T}}_{t,ii} \geq \frac{1}{3}\left(1 \wedge \hat{\mathbf{T}}_{0,ii}\exp\left(2\eta t\lambda_i\right)\right) \Rightarrow \mathbf{T}_{t,ii} \geq \frac{1}{4}\left(\kappa_d \wedge \mathbf{T}_{0,ii}\exp\left(2\eta t\lambda_i\right)\right).$$

Therefore, we have

$$\mathbf{T}_{t,ii} \geq \kappa_d \begin{cases} 0.25\left(1 \wedge \frac{r_s}{d}\exp\left(2\eta t\lambda_i\right)\right), & i \leq 2r_s\log^{\frac{1}{\alpha}}d \\ \frac{r_s}{d}, & i > 2r_s\log^{\frac{1}{\alpha}}d. \end{cases}$$

On the other hand, for $\eta t \leq \frac{1}{2}r_s^\alpha\log(\frac{d\log^{1.5}d}{r_s})$ and $i > 2r_s\log^{\frac{1}{\alpha}}d$, we have for $d \geq \Omega(1)$.

$$\bar{\mathbf{G}}_{t,ii} \leq 1.2\left(\bar{\mathbf{G}}_{0,ii}\exp\left(2\eta t\lambda_i\right) \wedge 1\right) \leq 1.2\left(\bar{\mathbf{G}}_{0,ii}\exp\left(\frac{r_s^\alpha\log(\frac{d\log d}{r_s})}{2^\alpha r_s^\alpha\log d}\right) \wedge 1\right) \leq 1.5\bar{\mathbf{G}}_{0,ii}.$$

Therefore, for $\eta t \leq \frac{1}{2}r_s^\alpha\log(\frac{d\log^{1.5}d}{r_s})$,

$$\frac{\bar{\mathbf{G}}_{j,ii}}{\mathbf{T}_{t,ii}} \leq \begin{cases} \dfrac{1.2\left(1 \wedge \frac{5.5r_ur_s}{d}\exp\left(2\eta t\lambda_i\right)\right)}{0.25\kappa_d\left(1 \wedge \frac{r_s}{d}\exp\left(2\eta t\lambda_i\right)\right)}, & i \leq 2r_s\log^{\frac{1}{\alpha}}d \\ \dfrac{d}{\kappa_d r_s}\dfrac{8.25r_ur_s}{d}, & i > 2r_s\log^{\frac{1}{\alpha}}d \end{cases}$$

$$\leq \frac{26.4r_u}{\kappa_d}.$$

Moreover, for $\eta t \leq \frac{1}{2}r_s^\alpha\log(\frac{d\log^{1.5}d}{r_s})$,

$$\frac{\eta\sum_{j=0}^{t-1}\bar{\mathbf{G}}_{j,ii}}{\mathbf{T}_{t,ii}} \leq \begin{cases} \dfrac{26.4r_u}{\kappa_d}\dfrac{\eta\left(t \wedge \frac{r_s}{d}\sum_{j=0}^{t-1}\exp(2\eta j\lambda_i)\right)}{\left(1 \wedge \frac{r_s}{d}\exp\left(2\eta t\lambda_i\right)\right)}, & i \leq 2r_s\log^{\frac{1}{\alpha}}d \\ \dfrac{d}{\kappa_d r_s}\dfrac{8.25r_ur_s}{d}\eta t, & i > 2r_s\log^{\frac{1}{\alpha}}d \end{cases}$$

$$\leq \frac{13.2r_u}{\kappa_d}\begin{cases} \frac{1}{\lambda_i}, & 2\eta t \leq \frac{\log\frac{d}{r_s}}{\lambda_i} \text{ and } i \leq 2r_s\log^{\frac{1}{\alpha}}d \\ 2\eta t, & \text{otherwise} \end{cases}$$

$$\leq \frac{13.2r_u}{\kappa_d}(2\eta t \vee (2r_s)^\alpha\log d) \leq \frac{15r_u}{\kappa_d}(2r_s)^\alpha\log d.$$

Finally for $\eta t \leq \frac{1}{2}r_s^\alpha\log(\frac{d\log^{1.5}d}{r_s})$ and $d \geq \Omega_{r_s}(1)$, by using the first bound in (F.36),

$$\|\mathbf{\Lambda}_{11}\|_F^2 - \mathrm{Tr}(\mathbf{\Lambda}_{11}\bar{\mathbf{G}}_{t,11}\mathbf{\Lambda}_{11}) \geq \sum_{i=1}^{r_s}\lambda_i^2\left(1 - \frac{12.1r_s}{d}\exp\left(2\eta t\lambda_i\right)\right)_+$$

$$+ \sum_{i=r_s+1}^{2r_s}\lambda_i^2\left(1 - \frac{12.1r_s}{d}\exp\left(2\eta t\lambda_i\right)\right)_+ + \sum_{i=2r_s+1}^{r_u}\lambda_i^2\left(1 - \frac{6.05r_ur_s}{d}\exp\left(2\eta t\lambda_i\right)\right)_+ - 2\eta^2\|\mathbf{\Lambda}\|_F^2$$

$$\overset{(f)}{\geq} \sum_{i=1}^{r_s}\lambda_i^2\left(1 - \frac{12.1r_s}{d}\exp\left(2\eta t\lambda_i\right)\right)_+ + \left(1 - \frac{1}{\log d}\right)\sum_{i=r_s+1}^{2r_s}\lambda_i^2$$

$$+ \left(1 - 6.05 r_u \log^{\frac{1.5}{\sqrt{2}}} d \left(\frac{r_s}{d}\right)^{1-\frac{1}{\sqrt{2}}}\right) \sum_{i=2r_s+1}^{r_u} \lambda_i^2 - 2\eta^2 \|\mathbf{\Lambda}\|_{\mathrm{F}}^2$$

$$\geq \sum_{i=1}^{r_s} \lambda_i^2 \left(1 - \frac{12.1 r_s}{d} \exp\left(2\eta t \lambda_i\right)\right)_+ + \left(1 - 6.05 r_u \log^{\frac{1.5}{\sqrt{2}}} d \left(\frac{r_s}{d}\right)^{1-\frac{1}{\sqrt{2}}}\right) \sum_{i=r_s+1}^{r_u} \lambda_i^2$$

$$- \frac{(r_s+1)^{1+2\alpha}}{\log d} - 2\eta^2 \|\mathbf{\Lambda}\|_{\mathrm{F}}^2,$$

where we used the bounds for $t, d$ in (f). $\qquad\qquad\square$

### F.5 Bounds for the second-order terms

We recall

$$R_{\mathrm{so}}[\mathbf{G}_t] = \frac{\eta^2}{16 r_s} \mathbf{\Theta}^\top \mathbb{E}_t \left[\nabla_{\mathrm{St}} \mathbf{L}_{t+1} \nabla_{\mathrm{St}} \mathbf{L}_{t+1}^\top\right] \mathbf{\Theta}$$

$$- \frac{\eta^2}{16 r_s} \mathbf{M}_t \mathbb{E}_t \left[\frac{\mathbf{\mathcal{P}}_{t+1}}{1 + c_{t+1}^2}\right] \mathbf{M}_t^\top - \frac{\eta^3}{32 r_s^{3/2}} \mathrm{Sym}\left(\mathbf{\Theta}^\top \mathbb{E}_t \left[\frac{\nabla_{\mathrm{St}} \mathbf{L}_{t+1} \mathbf{\mathcal{P}}_{t+1}}{1 + c_{t+1}^2}\right] \mathbf{M}_t^\top\right)$$

$$- \frac{\eta^4}{256 r_s^2} \mathbf{\Theta}^\top \mathbb{E}_t \left[\frac{\nabla_{\mathrm{St}} \mathbf{L}_{t+1} \mathbf{\mathcal{P}}_{t+1} \nabla_{\mathrm{St}} \mathbf{L}_{t+1}^\top}{1 + c_{t+1}^2}\right] \mathbf{\Theta}. \tag{F.37}$$

**Proposition 17.** *For $\eta \ll d^{-1/2}$, there exists a universal constant $C > 0$ such that*

$$-C\left(\frac{\eta^2 d}{r_s} \mathbf{G}_t + \eta^2 \mathbf{I}_r\right) \preceq R_{so}[\mathbf{G}_t] \preceq C\left(\frac{\eta^2 d}{r_s} \mathbf{G}_t + \eta^2 \mathbf{I}_r\right).$$

*Proof.* We bound each term in (F.37). In the following, $\mathbf{v}$ denotes a generic unit norm vector with proper dimensionality. For the first term,

$$\mathbf{\Theta}^\top \mathbb{E}_t \left[\nabla_{\mathrm{St}} \mathbf{L}_{t+1} \nabla_{\mathrm{St}} \mathbf{L}_{t+1}^\top\right] \mathbf{\Theta}$$
$$= \mathbf{\Theta}^\top \left(\mathbf{I}_d - \mathbf{W}_t \mathbf{W}_t^\top\right) \mathbb{E}_t \left[(y_{t+1} - \hat{y}_{t+1})^2 \|\mathbf{W}_t^\top \mathbf{x}_{t+1}\|_2^2 \mathbf{x}_{t+1} \mathbf{x}_{t+1}^\top\right] \left(\mathbf{I}_d - \mathbf{W}_t \mathbf{W}_t^\top\right) \mathbf{\Theta}.$$

We have

$$\mathbb{E}_t \left[(y_{t+1} - \hat{y}_{t+1})^2 \|\mathbf{W}_t^\top \mathbf{x}_{t+1}\|_2^2 \langle \mathbf{v}, \mathbf{x}_{t+1}\rangle^2\right] \leq C r_s.$$

Therefore,

$$0 \preceq \frac{\eta^2}{16 r_s} \mathbf{\Theta}^\top \mathbb{E}_t \left[\nabla_{\mathrm{St}} \mathbf{L}_{t+1} \nabla_{\mathrm{St}} \mathbf{L}_{t+1}^\top\right] \mathbf{\Theta} \preceq C\eta^2 (\mathbf{I}_r - \mathbf{G}_t).$$

For the second term,

$$\mathbf{M}_t \mathbb{E}_t \left[\frac{\mathbf{\mathcal{P}}_{t+1}}{1 + c_{t+1}^2}\right] \mathbf{M}_t^\top$$
$$= \mathbf{M}_t \mathbb{E}_t \left[\frac{(y_{t+1} - \hat{y}_{t+1})^2 \|(\mathbf{I}_d - \mathbf{W}_t \mathbf{W}_t^\top) \mathbf{x}_{t+1}\|_2^2 \mathbf{W}_t^\top \mathbf{x}_{t+1} \mathbf{x}_{t+1}^\top \mathbf{W}_t}{1 + c_{t+1}^2}\right] \mathbf{M}_t^\top.$$

We have

$$\mathbb{E}_t \left[\frac{(y_{t+1} - \hat{y}_{t+1})^2 \|(\mathbf{I}_d - \mathbf{W}_t \mathbf{W}_t^\top) \mathbf{x}_{t+1}\|_2^2 \langle \mathbf{v}, \mathbf{W}_t^\top \mathbf{x}_{t+1}\rangle^2}{1 + c_{t+1}^2}\right] \leq Cd.$$

Therefore,

$$0 \preceq \frac{\eta^2}{16 r_s} \mathbf{M}_t \mathbb{E}_t \left[\frac{\mathbf{\mathcal{P}}_{t+1}}{1 + c_{t+1}^2}\right] \mathbf{M}_t^\top \preceq C\frac{\eta^2 d}{r_s} \mathbf{G}_t.$$

For the third term by using Proposition 22,

$$\frac{\eta^3}{32r_s^{3/2}}\mathrm{Sym}\left(\mathbf{\Theta}^\top\mathbb{E}_t\left[\frac{\nabla_{\mathrm{St}}\mathbf{L}_{t+1}\mathcal{P}_{t+1}}{1+c_{t+1}^2}\right]\mathbf{M}_t^\top\right)$$
$$\preceq C\left(\frac{\eta^4}{r_s^2 d}\mathbf{\Theta}^\top\mathbb{E}_t\left[\frac{\nabla_{\mathrm{St}}\mathbf{L}_{t+1}\mathcal{P}_{t+1}}{1+c_{t+1}^2}\right]\mathbb{E}_t\left[\frac{\mathcal{P}_{t+1}\nabla_{\mathrm{St}}\mathbf{L}_{t+1}^\top}{1+c_{t+1}^2}\right]\mathbf{\Theta}+\frac{\eta^2 d}{r_s}\mathbf{G}_t\right)$$

We have

$$\mathbf{\Theta}^\top\mathbb{E}_t\left[\frac{\nabla_{\mathrm{St}}\mathbf{L}_{t+1}\mathcal{P}_{t+1}}{1+c_{t+1}^2}\right]$$
$$=\mathbf{\Theta}^\top(\mathbf{I}_d-\mathbf{W}_t\mathbf{W}_t^\top)$$
$$\times\mathbb{E}_t\left[\frac{(y_{t+1}-\hat{y}_{t+1})^3}{1+c_{t+1}^2}\|(\mathbf{I}_d-\mathbf{W}_t\mathbf{W}_t^\top)\,\mathbf{x}_{t+1}\|_2^2\|\mathbf{W}_t^\top\mathbf{x}_{t+1}\|_2^2\mathbf{x}_{t+1}\mathbf{x}_{t+1}^\top\mathbf{W}_t\right].$$

Then, by using Cauchy-Schwartz inequality, we can show that

$$\left\|\mathbb{E}_t\left[\frac{(y_{t+1}-\hat{y}_{t+1})^3}{1+c_{t+1}^2}\|(\mathbf{I}_d-\mathbf{W}_t\mathbf{W}_t^\top)\,\mathbf{x}_{t+1}\|_2^2\|\mathbf{W}_t^\top\mathbf{x}_{t+1}\|_2^2\mathbf{x}_{t+1}\mathbf{x}_{t+1}^\top\mathbf{W}_t\right]\right\|_2\leq Cdr_s.$$

Therefore,

$$\frac{\eta^4}{r_s^2 d}\mathbf{\Theta}^\top\mathbb{E}_t\left[\frac{\nabla_{\mathrm{St}}\mathbf{L}_{t+1}\mathcal{P}_{t+1}}{1+c_{t+1}^2}\right]\mathbb{E}_t\left[\frac{\mathcal{P}_{t+1}\nabla_{\mathrm{St}}\mathbf{L}_{t+1}^\top}{1+c_{t+1}^2}\right]\mathbf{\Theta}\leq C\eta^4 d(\mathbf{I}_r-\mathbf{G}_t).$$

We get

$$\frac{\eta^3}{32r_s^{3/2}}\mathrm{Sym}\left(\mathbf{\Theta}^\top\mathbb{E}_t\left[\frac{\nabla_{\mathrm{St}}\mathbf{L}_{t+1}\mathcal{P}_{t+1}}{1+c_{t+1}^2}\right]\mathbf{M}_t^\top\right)\preceq C\left(\eta^4 d(\mathbf{I}_r-\mathbf{G}_t)+\frac{\eta^2 d}{r_s}\mathbf{G}_t\right).$$

By repeating the argument with the lower bound in Proposition 22, we can also show

$$\frac{\eta^3}{32r_s^{3/2}}\mathrm{Sym}\left(\mathbf{\Theta}^\top\mathbb{E}_t\left[\frac{\nabla_{\mathrm{St}}\mathbf{L}_{t+1}\mathcal{P}_{t+1}}{1+c_{t+1}^2}\right]\mathbf{M}_t^\top\right)\succeq -C\left(\eta^4 d(\mathbf{I}_r-\mathbf{G}_t)+\frac{\eta^2 d}{r_s}\mathbf{G}_t\right).$$

For the last term, we write

$$\mathbf{\Theta}^\top\mathbb{E}_t\left[\frac{\nabla_{\mathrm{St}}\mathbf{L}_{t+1}\mathcal{P}_{t+1}\nabla_{\mathrm{St}}\mathbf{L}_{t+1}^\top}{1+c_{t+1}^2}\right]\mathbf{\Theta}$$
$$=\mathbf{\Theta}^\top(\mathbf{I}_d-\mathbf{W}_t\mathbf{W}_t^\top)$$
$$\times\mathbb{E}_t\left[\frac{(y_{t+1}-\hat{y}_{t+1})^4}{1+c_{t+1}^2}\|\mathbf{W}_t^\top\mathbf{x}_{t+1}\|_2^4\|(\mathbf{I}_d-\mathbf{W}_t\mathbf{W}_t^\top)\,\mathbf{x}_{t+1}\|_2^2\mathbf{x}_{t+1}\mathbf{x}_{t+1}^\top\right](\mathbf{I}_d-\mathbf{W}_t\mathbf{W}_t^\top)\,\mathbf{\Theta}.$$

We have

$$\mathbb{E}_t\left[\frac{(y_{t+1}-\hat{y}_{t+1})^4}{1+c_{t+1}^2}\|\mathbf{W}_t^\top\mathbf{x}_{t+1}\|_2^4\|(\mathbf{I}_d-\mathbf{W}_t\mathbf{W}_t^\top)\,\mathbf{x}_{t+1}\|_2^2\langle\mathbf{v},\mathbf{x}_{t+1}\rangle^2\right]\leq Cdr_s^2.$$

Therefore,

$$0\preceq\frac{\eta^4}{r_s^2}\mathbf{\Theta}^\top\mathbb{E}_t\left[\frac{\nabla_{\mathrm{St}}\mathbf{L}_{t+1}\mathcal{P}_{t+1}\nabla_{\mathrm{St}}\mathbf{L}_{t+1}^\top}{1+c_{t+1}^2}\right]\mathbf{\Theta}\preceq C\eta^4 d(\mathbf{I}_r-\mathbf{G}_t).$$

By using $\mathbf{G}_t\succeq 0$ and $\eta\ll d^{-1/2}$, the result follows. $\qquad\square$

### F.6 Noise characterization

To prove the noise concentration bound for both the heavy-tailed and light-tailed cases simultaneously, we introduce some new notation. Specifically, we define the submatrix notation:

$$\boldsymbol{\Theta} =: \begin{bmatrix} \boldsymbol{\Theta}_1 & \boldsymbol{\Theta}_2 \end{bmatrix} \quad \text{and} \quad \boldsymbol{M}_t =: \begin{bmatrix} \boldsymbol{M}_{t,1} \\ \boldsymbol{M}_{t,2} \end{bmatrix} = \begin{bmatrix} \boldsymbol{\Theta}_1^\top \boldsymbol{W}_t \\ \boldsymbol{\Theta}_2^\top \boldsymbol{W}_t \end{bmatrix},$$

where $\boldsymbol{\Theta}_1 \in \mathbb{R}^{d \times r_u}$ and $\boldsymbol{M}_{t,1} \in \mathbb{R}^{r_u \times r_s}$. We note that $\boldsymbol{G}_{t,11} = \boldsymbol{M}_{t,1}\boldsymbol{M}_{t,1}^\top$. To unify the treatment of the heavy-tailed and light-tailed cases, we use the following notation to represent both cases:

$$\Theta := \{\boldsymbol{\Theta}, \boldsymbol{\Theta}_1\} \quad \mathsf{M}_t := \{\boldsymbol{M}_t, \boldsymbol{M}_{t,1}\}.$$

With the new notation, we have

$$
\begin{aligned}
\frac{\eta/2}{\sqrt{r_s}}\mathsf{v}_{t+1} =\ & \frac{\eta/2}{\sqrt{r_s}}\mathrm{Sym}\left(\Theta^\top\left(\nabla_{\mathrm{St}}\boldsymbol{L}_{t+1} - \mathbb{E}_t\left[\nabla_{\mathrm{St}}\boldsymbol{L}_{t+1}\right]\right)\mathsf{M}_t^\top\right) \\
& -\frac{\eta^2}{16r_s}\mathsf{M}_t\left(\frac{\boldsymbol{\mathcal{P}}_{t+1}}{1+c_{t+1}^2} - \mathbb{E}_t\left[\frac{\boldsymbol{\mathcal{P}}_{t+1}}{1+c_{t+1}^2}\right]\right)\mathsf{M}_t^\top \\
& +\frac{\eta^2}{16r_s}\Theta^\top\left(\nabla_{\mathrm{St}}\boldsymbol{L}_{t+1}\nabla_{\mathrm{St}}\boldsymbol{L}_{t+1}^\top - \mathbb{E}_t\left[\nabla_{\mathrm{St}}\boldsymbol{L}_{t+1}\nabla_{\mathrm{St}}\boldsymbol{L}_{t+1}^\top\right]\right)\Theta \\
& -\frac{\eta^3}{32r_s^{3/2}}\mathrm{Sym}\left(\Theta^\top\left(\frac{\nabla_{\mathrm{St}}\boldsymbol{L}_{t+1}\boldsymbol{\mathcal{P}}_{t+1}}{1+c_{t+1}^2} - \mathbb{E}_t\left[\frac{\nabla_{\mathrm{St}}\boldsymbol{L}_{t+1}\boldsymbol{\mathcal{P}}_{t+1}}{1+c_{t+1}^2}\right]\right)\mathsf{M}_t^\top\right) \\
& -\frac{\eta^4}{256r_s^2}\Theta^\top\left(\frac{\nabla_{\mathrm{St}}\boldsymbol{L}_{t+1}\boldsymbol{\mathcal{P}}_{t+1}\nabla_{\mathrm{St}}\boldsymbol{L}_{t+1}^\top}{1+c_{t+1}^2} - \mathbb{E}_t\left[\frac{\nabla_{\mathrm{St}}\boldsymbol{L}_{t+1}\boldsymbol{\mathcal{P}}_{t+1}\nabla_{\mathrm{St}}\boldsymbol{L}_{t+1}^\top}{1+c_{t+1}^2}\right]\right)\Theta.
\end{aligned}
$$

For $\mathsf{rk} \in \{r, r_u\}$ and $\boldsymbol{T}_1, \boldsymbol{T}_2 \in \mathbb{R}^{\mathsf{rk} \times \mathsf{rk}}$ be a deterministic symmetric positive definite matrices, we define

$$
\mathcal{A}_{t+1}\left(\boldsymbol{T}_1, \boldsymbol{T}_2\right) \equiv \left\{\left\|\boldsymbol{T}_1\Theta^\top\nabla_{\mathrm{St}}\boldsymbol{L}_{t+1}\mathsf{M}_t^\top\boldsymbol{T}_2\right\|_2 \leq \frac{L^2}{2}\sqrt{\mathrm{Tr}\big(\boldsymbol{T}_1^2(\boldsymbol{I}_{\mathsf{rk}} - \mathsf{G}_t)\big)\mathrm{Tr}\big(\boldsymbol{T}_2^2\mathsf{G}_t\big)}\right\}
$$

$$
\mathcal{B}_{t+1}\left(\boldsymbol{T}_1, \boldsymbol{T}_2\right) \equiv \left\{\left\|\boldsymbol{T}_1\mathsf{M}_t\boldsymbol{\mathcal{P}}_{t+1}\mathsf{M}_t^\top\boldsymbol{T}_2\right\|_2 \leq \frac{L^4 d}{2}\sqrt{\mathrm{Tr}\big(\boldsymbol{T}_1^2\mathsf{G}_t\big)\mathrm{Tr}\big(\boldsymbol{T}_2^2\mathsf{G}_t\big)}\right\}
$$

$$
\mathcal{C}_{t+1}\left(\boldsymbol{T}_1, \boldsymbol{T}_2\right) \equiv \left\{\left\|\boldsymbol{T}_1\Theta^\top\nabla_{\mathrm{St}}\boldsymbol{L}_{t+1}\nabla_{\mathrm{St}}\boldsymbol{L}_{t+1}^\top\Theta\boldsymbol{T}_2\right\|_2 \leq \frac{L^4 r_s}{2}\sqrt{\mathrm{Tr}\big(\boldsymbol{T}_1^2(\boldsymbol{I}_{\mathsf{rk}} - \mathsf{G}_t)\big)\mathrm{Tr}\big(\boldsymbol{T}_2^2(\boldsymbol{I}_{\mathsf{rk}} - \mathsf{G}_t)\big)}\right\}
$$

$$
\mathcal{D}_{t+1}\left(\boldsymbol{T}_1, \boldsymbol{T}_2\right) \equiv \left\{\left\|\boldsymbol{T}_1\Theta^\top\nabla_{\mathrm{St}}\boldsymbol{L}_{t+1}\boldsymbol{\mathcal{P}}_{t+1}\mathsf{M}_t^\top\boldsymbol{T}_2\right\|_2 \leq \frac{L^6 d r_s}{2}\sqrt{\mathrm{Tr}\big(\boldsymbol{T}_1^2(\boldsymbol{I}_{\mathsf{rk}} - \mathsf{G}_t)\big)\mathrm{Tr}\big(\boldsymbol{T}_2^2\mathsf{G}_t\big)}\right\}
$$

$$
\mathcal{F}_{t+1}\left(\boldsymbol{T}_1, \boldsymbol{T}_2\right) \equiv \left\{\left\|\boldsymbol{T}_1\Theta^\top\nabla_{\mathrm{St}}\boldsymbol{L}_{t+1}\boldsymbol{\mathcal{P}}_{t+1}\nabla_{\mathrm{St}}\boldsymbol{L}_{t+1}^\top\Theta\boldsymbol{T}_2\right\|_2 \leq \frac{L^8 d r_s^2}{2}\sqrt{\mathrm{Tr}\big(\boldsymbol{T}_1^2(\boldsymbol{I}_{\mathsf{rk}} - \mathsf{G}_t)\big)\mathrm{Tr}\big(\boldsymbol{T}_2^2(\boldsymbol{I}_{\mathsf{rk}} - \mathsf{G}_t)\big)}\right\}.
$$

We start with the following statement:

**Proposition 18.** *Let $e_{t+1} := (y_{t+1} - \hat{y}_{t+1})$. There exists a universal constant $C > 0$ such that for $L \geq 2e(\sqrt{8} + \sigma)$, the following statements hold:*

*1. We have $\mathbb{P}_t\left[\mathcal{A}_{t+1}\left(\boldsymbol{T}_1, \boldsymbol{T}_2\right) \cap \{|e_{t+1}| \leq L\}\right] \geq 1 - 2e^{\frac{-L/e}{(\sqrt{8}+\sigma)}}$. Moreover,*

$$
\begin{aligned}
\mathbb{E}_t\left[\left(\mathrm{Sym}\left(\boldsymbol{T}_1\Theta^\top\nabla_{St}\boldsymbol{L}_{t+1}\mathsf{M}_t^\top\boldsymbol{T}_2\right)\right)^2\right] \\
\preceq C\left(\mathrm{Tr}(\boldsymbol{T}_2^2\mathsf{G}_t)\boldsymbol{T}_1(\boldsymbol{I}_{\mathsf{rk}} - \mathsf{G}_t)\boldsymbol{T}_1 + \mathrm{Tr}\big(\boldsymbol{T}_1^2(\boldsymbol{I}_{\mathsf{rk}} - \mathsf{G}_t)\big)\boldsymbol{T}_2\mathsf{G}_t\boldsymbol{T}_2\right).
\end{aligned}
$$

*2. We have $\mathbb{P}_t\left[\mathcal{B}_{t+1}\left(\boldsymbol{T}_1, \boldsymbol{T}_2\right) \cap \{|e_{t+1}| \leq L\}\right] \geq 1 - 2e^{\frac{-L/e}{(\sqrt{8}+\sigma)}}$. Moreover,*

$$
\begin{aligned}
\mathbb{E}_t\left[\left(\mathrm{Sym}\left(\boldsymbol{T}_1\mathsf{M}_t\boldsymbol{\mathcal{P}}_{t+1}\mathsf{M}_t^\top\boldsymbol{T}_2\right)\right)^2\right] \\
\preceq Cd^2\left(\mathrm{Tr}(\boldsymbol{T}_2^2\mathsf{G}_t)\boldsymbol{T}_1\mathsf{G}_t\boldsymbol{T}_1 + \mathrm{Tr}(\boldsymbol{T}_1^2\mathsf{G}_t)\boldsymbol{T}_2\mathsf{G}_t\boldsymbol{T}_2\right).
\end{aligned}
$$

3. We have $\mathbb{P}_t\left[\mathcal{C}_{t+1}\left(\boldsymbol{T}_1, \boldsymbol{T}_2\right) \cap \{|e_{t+1}| \leq L\}\right] \geq 1 - 2e^{\frac{-L/e}{(\sqrt{8}+\sigma)}}$. *Moreover,*

$$\mathbb{E}_t\left[\left(\operatorname{Sym}\left(\boldsymbol{T}_1 \Theta^\top \nabla_{St} \boldsymbol{L}_{t+1} \nabla_{St} \boldsymbol{L}_{t+1}^\top \boldsymbol{T}_2\right)\right)^2\right]$$
$$\preceq C r_s^2 \Big( Tr\big(\boldsymbol{T}_2^2(\boldsymbol{I}_{\mathsf{rk}} - \mathsf{G}_t)\big) \boldsymbol{T}_1(\boldsymbol{I}_{\mathsf{rk}} - \mathsf{G}_t)\boldsymbol{T}_1 + Tr\big(\boldsymbol{T}_1^2(\boldsymbol{I}_{\mathsf{rk}} - \mathsf{G}_t)\big)\boldsymbol{T}_2(\boldsymbol{I}_{\mathsf{rk}} - \mathsf{G}_t)\boldsymbol{T}_2 \Big).$$

4. We have $\mathbb{P}_t\left[\mathcal{D}_{t+1}\left(\boldsymbol{T}_1, \boldsymbol{T}_2\right) \cap \{|e_{t+1}| \leq L\}\right] \geq 1 - 2e^{\frac{-L/e}{(7/2+\sigma)}}$. *Moreover,*

$$\mathbb{E}_t\left[\left(\operatorname{Sym}\left(\boldsymbol{T}_1 \Theta^\top \nabla_{St} \boldsymbol{L}_{t+1} \boldsymbol{\mathcal{P}}_{t+1} \mathsf{M}_t^\top \boldsymbol{T}_2\right)\right)^2\right]$$
$$\preceq C d^2 r_s^2 \Big( Tr(\boldsymbol{T}_2^2 \mathsf{G}_t) \boldsymbol{T}_1(\boldsymbol{I}_{\mathsf{rk}} - \mathsf{G}_t)\boldsymbol{T}_1 + Tr\big(\boldsymbol{T}_1^2(\boldsymbol{I}_{\mathsf{rk}} - \mathsf{G}_t)\big)\boldsymbol{T}_2 \mathsf{G}_t \boldsymbol{T}_2 \Big).$$

5. We have $\mathbb{P}_t\left[\mathcal{F}_{t+1}\left(\boldsymbol{T}_1, \boldsymbol{T}_2\right) \cap \{|e_{t+1}| \leq L\}\right] \geq 1 - 2e^{\frac{-L/e}{(4\sqrt{2}+\sigma)}}$. *Moreover,*

$$\mathbb{E}_t\left[\left(\operatorname{Sym}\left(\boldsymbol{T}_1 \Theta^\top \nabla_{St} \boldsymbol{L}_{t+1} \boldsymbol{\mathcal{P}}_{t+1} \nabla_{St} \boldsymbol{L}_{t+1}^\top \Theta \boldsymbol{T}_2\right)\right)^2\right]$$
$$\preceq C d^2 r_s^4 \Big( Tr\big(\boldsymbol{T}_2^2(\boldsymbol{I}_{\mathsf{rk}} - \mathsf{G}_t)\big) \boldsymbol{T}_1(\boldsymbol{I}_{\mathsf{rk}} - \mathsf{G}_t)\boldsymbol{T}_1 + Tr\big(\boldsymbol{T}_1^2(\boldsymbol{I}_{\mathsf{rk}} - \mathsf{G}_t)\big)\boldsymbol{T}_2(\boldsymbol{I}_{\mathsf{rk}} - \mathsf{G}_t)\boldsymbol{T}_2 \Big).$$

*Proof.* First, we derive a concentration bound for $|e_{t+1}|$. By Corollary 6 and Proposition 33 we have

$$\mathbb{E}_t\left[|e_{t+1}|^p\right] \leq \left(\mathbb{E}_t\left[|y_{t+1}|^p\right]^{\frac{1}{p}} + \mathbb{E}_t\left[|\hat{y}_{t+1}|^p\right]^{\frac{1}{p}}\right)^p \leq (\sqrt{8}+\sigma)^p p^p \text{ for } p \geq 2,$$

which implies $\mathbb{P}_t\left[|e_{t+1}| \geq u\right] \leq e^{\frac{-u/e}{(\sqrt{8}+\sigma)}}$ for $u \geq 2e(\sqrt{8}+\sigma)$. In the following, we prove each item separately.

**First item.** We define

$$\boldsymbol{T}_1 \Theta^\top \nabla_{St} \boldsymbol{L}_{t+1} \mathsf{M}_t^\top \boldsymbol{T}_2 = \underbrace{\boldsymbol{T}_1 \Theta^\top (\boldsymbol{I}_d - \boldsymbol{W}_t \boldsymbol{W}_t^\top) e_{t+1} \boldsymbol{x}_{t+1}}_{:=\boldsymbol{u}_{t+1}} \underbrace{\boldsymbol{x}_{t+1}^\top \boldsymbol{W}_t \boldsymbol{W}_t^\top \Theta \boldsymbol{T}_2}_{:=\boldsymbol{v}_{t+1}^\top}.$$

For $u, L > 0$

$$\mathbb{P}_t\left[\left\|\boldsymbol{u}_{t+1} \boldsymbol{v}_{t+1}^\top\right\|_2 \geq uL\sqrt{Tr\big(\boldsymbol{T}_1^2(\boldsymbol{I}_{\mathsf{rk}} - \mathsf{G}_t)\big)Tr(\boldsymbol{T}_2^2 \mathsf{G}_t)} \text{ or } |e_{t+1}| \geq L\right]$$
$$\leq \mathbb{P}_t\left[\left\|\boldsymbol{u}_{t+1} \boldsymbol{v}_{t+1}^\top\right\|_2 \geq uL\sqrt{Tr\big(\boldsymbol{T}_1^2(\boldsymbol{I}_{\mathsf{rk}} - \mathsf{G}_t)\big)Tr(\boldsymbol{T}_2^2 \mathsf{G}_t)} \text{ and } |e_{t+1}| \leq L\right] + \mathbb{P}_t\left[|e_{t+1}| \geq L\right]$$
$$\leq \mathbb{P}_t\left[\left\|\mathbb{1}_{|e_{t+1}| \leq L} \boldsymbol{u}_{t+1} \boldsymbol{v}_{t+1}^\top\right\|_2 \geq uL\sqrt{Tr\big(\boldsymbol{T}_1^2(\boldsymbol{I}_{\mathsf{rk}} - \mathsf{G}_t)\big)Tr(\boldsymbol{T}_2^2 \mathsf{G}_t)}\right] + \mathbb{P}_t\left[|e_{t+1}| \geq L\right].$$

We have for $p \geq 2$

$$\mathbb{E}_t\left[\left\|\mathbb{1}_{|e_{t+1}| \leq L} \boldsymbol{u}_{t+1} \boldsymbol{v}_{t+1}^\top\right\|_2^p\right]$$
$$\leq L^p \mathbb{E}_t\left[\|\boldsymbol{T}_1 \Theta^\top (\boldsymbol{I}_d - \boldsymbol{W}_t \boldsymbol{W}_t^\top)\boldsymbol{x}_{t+1}\|_2^p\right] \mathbb{E}_t\left[\|\boldsymbol{T}_2 \Theta^\top \boldsymbol{W}_t \boldsymbol{W}_t^\top \boldsymbol{x}_{t+1}\|_2^p\right]$$
$$\overset{(a)}{\leq} L^p \left(\frac{p}{2}\right)^p \left(3 Tr\big(\boldsymbol{T}_1^2(\boldsymbol{I}_{\mathsf{rk}} - \mathsf{G}_t)\big)Tr(\boldsymbol{T}_2^2 \mathsf{G}_t)\right)^{\frac{p}{2}},$$

where we used Corollary 7 in (a). By Proposition 33, we have for $u \geq 2e$

$$\mathbb{P}_t\left[\left\|\mathbb{1}_{|e_{t+1}| \leq t} \boldsymbol{u}_{t+1} \boldsymbol{v}_{t+1}^\top\right\|_2 \geq uL\sqrt{Tr\big(\boldsymbol{T}_1^2(\boldsymbol{I}_{\mathsf{rk}} - \mathsf{G}_t)\big)Tr(\boldsymbol{T}_2^2 \mathsf{G}_t)}\right] \leq e^{-\frac{u}{e}}.$$

By choosing $u = \frac{L}{2}$, we have the probability bound.

For the variance bound, we have

$$\mathbb{E}_t\left[\operatorname{Sym}\left(\boldsymbol{T}_1 \Theta^\top \nabla_{St} \boldsymbol{L}_{t+1} \mathsf{M}_t^\top \boldsymbol{T}_2\right)^2\right] = \mathbb{E}_t\left[\operatorname{Sym}\left(\boldsymbol{u}_{t+1} \boldsymbol{v}_{t+1}^\top\right)^2\right].$$

By using Proposition 22, we have

$$\mathbb{E}_t\left[\operatorname{Sym}\left(\boldsymbol{u}_{t+1} \boldsymbol{v}_{t+1}^\top\right)^2\right]$$

$$\preceq \boldsymbol{T}_1\Theta^\top(\boldsymbol{I}_d - \boldsymbol{W}_t\boldsymbol{W}_t^\top)\mathbb{E}_t\left[e_{t+1}^2\|\boldsymbol{T}_2\Theta^\top\boldsymbol{W}_t\boldsymbol{W}_t^\top\boldsymbol{x}_{t+1}\|_2^2\boldsymbol{x}_{t+1}\boldsymbol{x}_{t+1}^\top\right](\boldsymbol{I}_d - \boldsymbol{W}_t\boldsymbol{W}_t^\top)\Theta\boldsymbol{T}_1$$
$$+ \boldsymbol{T}_2\Theta^\top\boldsymbol{W}_t\boldsymbol{W}_t^\top\mathbb{E}_t\left[e_{t+1}^2\|\boldsymbol{T}_1\Theta^\top(\boldsymbol{I}_d - \boldsymbol{W}_t\boldsymbol{W}_t^\top)\boldsymbol{x}_{t+1}\|_2^2\boldsymbol{x}_{t+1}\boldsymbol{x}_{t+1}^\top\right]\boldsymbol{W}_t\boldsymbol{W}_t^\top\Theta\boldsymbol{T}_2$$
$$\overset{(b)}{\preceq} C\Big(\text{Tr}(\boldsymbol{T}_2^2\mathsf{G}_t)\boldsymbol{T}_1(\boldsymbol{I}_{\mathsf{rk}} - \mathsf{G}_t)\boldsymbol{T}_1 + \text{Tr}\Big(\boldsymbol{T}_1^2(\boldsymbol{I}_{\mathsf{rk}} - \mathsf{G}_t)\Big)\boldsymbol{T}_2\mathsf{G}_t\boldsymbol{T}_2\Big),$$

where we used the Cauchy-Schwartz inequality in (b).

**Second item.** We define

$$\boldsymbol{T}_1\mathsf{M}_t\boldsymbol{\mathcal{P}}_{t+1}\mathsf{M}_t^\top\boldsymbol{T}_2 = \underbrace{e_{t+1}^2\|\left(\boldsymbol{I}_d - \boldsymbol{W}_t\boldsymbol{W}_t^\top\right)\boldsymbol{x}_{t+1}\|_2^2\boldsymbol{T}_1\Theta^\top\boldsymbol{W}_t\boldsymbol{W}_t^\top\boldsymbol{x}_{t+1}}_{:=\boldsymbol{u}_{t+1}}\underbrace{\boldsymbol{x}_{t+1}^\top\boldsymbol{W}_t\boldsymbol{W}_t^\top\Theta\boldsymbol{T}_2}_{:=\boldsymbol{v}_{t+1}^\top}.$$

We have for $p \geq 2$

$$\mathbb{E}_t\left[\left\|\mathbb{1}_{|e_{t+1}|\leq L}\boldsymbol{u}_{t+1}\boldsymbol{v}_{t+1}^\top\right\|_2^p\right]$$
$$\leq L^{2p}\mathbb{E}_t\left[\|\left(\boldsymbol{I}_d - \boldsymbol{W}_t\boldsymbol{W}_t^\top\right)\boldsymbol{x}_{t+1}\|_2^{2p}\right]\mathbb{E}_t\left[\|\boldsymbol{T}_1\Theta^\top\boldsymbol{W}_t\boldsymbol{W}_t^\top\boldsymbol{x}_{t+1}\|_2^{2p}\right]^{\frac{1}{2}}$$
$$\times \mathbb{E}_t\left[\|\boldsymbol{T}_2\Theta^\top\boldsymbol{W}_t\boldsymbol{W}_t^\top\boldsymbol{x}_{t+1}\|_2^{2p}\right]^{\frac{1}{2}}$$
$$\overset{(c)}{\leq} L^{2p}p^{2p}\left(3d\sqrt{\text{Tr}(\boldsymbol{T}_1^2\mathsf{G}_t)\text{Tr}(\boldsymbol{T}_2^2\mathsf{G}_t)}\right)^p,$$

where we use Corollary 7 in (c). By Proposition 33, we have for $u \geq (2e)^2$

$$\mathbb{P}_t\left[\left\|\mathbb{1}_{|e_{t+1}|\leq L}\boldsymbol{u}_{t+1}\boldsymbol{v}_{t+1}^\top\right\|_2 \geq uL^23d\sqrt{\text{Tr}(\boldsymbol{T}_1^2\mathsf{G}_t)\text{Tr}(\boldsymbol{T}_2^2\mathsf{G}_t)}\right] \leq e^{-\frac{u^{1/2}}{e}}.$$

By choosing $u = \frac{L^2}{6}$, we have the probability bound. For the variance bound, we have

$$\mathbb{E}_t\left[\left(\text{Sym}\left(\boldsymbol{T}_1\mathsf{M}_t\boldsymbol{\mathcal{P}}_{t+1}\mathsf{M}_t^\top\boldsymbol{T}_2\right)\right)^2\right] = \mathbb{E}_t\left[\text{Sym}\left(\boldsymbol{u}_{t+1}\boldsymbol{v}_{t+1}^\top\right)^2\right].$$

By using Proposition 22, we have

$$\mathbb{E}_t\left[\text{Sym}\left(\boldsymbol{u}_{t+1}\boldsymbol{v}_{t+1}^\top\right)^2\right]$$
$$\preceq \boldsymbol{T}_1\Theta^\top\boldsymbol{W}_t\boldsymbol{W}_t^\top$$
$$\times \mathbb{E}_t\left[e_{t+1}^4\|\left(\boldsymbol{I}_d - \boldsymbol{W}_t\boldsymbol{W}_t^\top\right)\boldsymbol{x}_{t+1}\|_2^4\|\boldsymbol{T}_2\Theta^\top\boldsymbol{W}_t\boldsymbol{W}_t^\top\boldsymbol{x}_{t+1}\|_2^2\boldsymbol{x}_{t+1}\boldsymbol{x}_{t+1}^\top\right]\boldsymbol{W}_t\boldsymbol{W}_t^\top\Theta\boldsymbol{T}_1$$
$$+ \boldsymbol{T}_2\Theta^\top\boldsymbol{W}_t\boldsymbol{W}_t^\top$$
$$\times \mathbb{E}_t\left[e_{t+1}^4\|\left(\boldsymbol{I}_d - \boldsymbol{W}_t\boldsymbol{W}_t^\top\right)\boldsymbol{x}_{t+1}\|_2^4\|\boldsymbol{T}_1\Theta^\top\boldsymbol{W}_t\boldsymbol{W}_t^\top\boldsymbol{x}_{t+1}\|_2^2\boldsymbol{x}_{t+1}\boldsymbol{x}_{t+1}^\top\right]\boldsymbol{W}_t\boldsymbol{W}_t^\top\Theta\boldsymbol{T}_2$$
$$\overset{(d)}{\preceq} Cd^2\left(\text{Tr}(\boldsymbol{T}_2^2\mathsf{G}_t)\boldsymbol{T}_1\mathsf{G}_t\boldsymbol{T}_1 + \text{Tr}(\boldsymbol{T}_1^2\mathsf{G}_t)\boldsymbol{T}_2\mathsf{G}_t\boldsymbol{T}_2\right),$$

where we use the Cauchy-Schwartz inequality in (d).

**Third item.** We define

$$\boldsymbol{T}_1\Theta^\top\nabla_{\text{St}}\boldsymbol{L}_{t+1}\nabla_{\text{St}}\boldsymbol{L}_{t+1}^\top\Theta\boldsymbol{T}_2$$
$$= \underbrace{e_{t+1}^2\|\boldsymbol{W}_t^\top\boldsymbol{x}_{t+1}\|_2^2\boldsymbol{T}_1\Theta^\top(\boldsymbol{I}_d - \boldsymbol{W}_t\boldsymbol{W}_t^\top)\boldsymbol{x}_{t+1}}_{:=\boldsymbol{u}_{t+1}}\underbrace{\boldsymbol{x}_{t+1}^\top(\boldsymbol{I}_d - \boldsymbol{W}_t\boldsymbol{W}_t^\top)\Theta\boldsymbol{T}_2}_{:=\boldsymbol{v}_{t+1}^\top}.$$

We have for $p \geq 2$

$$\mathbb{E}_t\left[\left\|\mathbb{1}_{|e_{t+1}|\leq L}\boldsymbol{u}_{t+1}\boldsymbol{v}_{t+1}^\top\right\|_2^p\right]$$
$$\leq L^{2p}\mathbb{E}_t\left[\|\boldsymbol{W}_t^\top\boldsymbol{x}_{t+1}\|_2^{2p}\right]\mathbb{E}_t\left[\|\boldsymbol{T}_1\Theta^\top(\boldsymbol{I}_d - \boldsymbol{W}_t\boldsymbol{W}_t^\top)\boldsymbol{x}_{t+1}\|_2^{2p}\right]^{\frac{1}{2}}$$
$$\times \mathbb{E}_t\left[\|\boldsymbol{T}_2\Theta^\top(\boldsymbol{I}_d - \boldsymbol{W}_t\boldsymbol{W}_t^\top)\boldsymbol{x}_{t+1}\|_2^{2p}\right]^{\frac{1}{2}}$$

$$\overset{(e)}{\leq} L^{2p}p^{2p}\left(3r_s\sqrt{\operatorname{Tr}\left(T_1^2(I_{\mathrm{rk}}-G_t)\right)\operatorname{Tr}\left(T_2^2(I_{\mathrm{rk}}-G_t)\right)}\right)^p,$$

where we use Corollary 7 in (e). By Proposition 33, we have for $u \geq (2e)^2$

$$\mathbb{P}_t\left[\left\|\mathbb{1}_{|e_{t+1}|\leq L}u_{t+1}v_{t+1}^\top\right\|_2 \geq uL^2 3r_s\sqrt{\operatorname{Tr}\left(T_1^2(I_{\mathrm{rk}}-G_t)\right)\operatorname{Tr}\left(T_2^2(I_{\mathrm{rk}}-G_t)\right)}\right] \leq e^{-\frac{u^{1/2}}{e}}.$$

By choosing $u = \frac{L^2}{6}$, we have the probability bound. For the variance bound, we have

$$\mathbb{E}_t\left[\left(\operatorname{Sym}\left(T_1\Theta^\top\nabla_{\mathrm{St}}L_{t+1}\nabla_{\mathrm{St}}L_{t+1}^\top\Theta T_2\right)\right)^2\right] = \mathbb{E}_t\left[\operatorname{Sym}\left(u_{t+1}v_{t+1}^\top\right)^2\right].$$

By using Proposition 22, we have

$$\mathbb{E}_t\left[\operatorname{Sym}\left(u_{t+1}v_{t+1}^\top\right)^2\right]$$
$$\preceq T_1\Theta^\top(I_d-W_tW_t^\top)$$
$$\quad\times \mathbb{E}_t[e_{t+1}^4\|W_t^\top x_{t+1}\|_2^4\|T_2\Theta^\top(I_d-W_tW_t^\top)x_{t+1}\|_2^2 x_{t+1}x_{t+1}^\top]\left(I_d-W_tW_t^\top\right)\Theta T_1$$
$$\quad+T_2\Theta^\top(I_d-W_tW_t^\top)$$
$$\quad\times \mathbb{E}_t[e_{t+1}^4\|W_t^\top x_{t+1}\|_2^4\|T_1\Theta^\top(I_d-W_tW_t^\top)x_{t+1}\|_2^2 x_{t+1}x_{t+1}^\top]\left(I_d-W_tW_t^\top\right)\Theta T_2$$
$$\overset{(f)}{\preceq} Cr_s^2\left(\operatorname{Tr}\left(T_2^2(I_{\mathrm{rk}}-G_t)\right)T_1(I_{\mathrm{rk}}-G_t)T_1 + \operatorname{Tr}\left(T_1^2(I_{\mathrm{rk}}-G_t)\right)T_2(I_{\mathrm{rk}}-G_t)T_2\right),$$

where we used the Cauchy-Schwartz inequality in (f).

**Fourth item.** We define

$$T_1\Theta^\top\nabla_{\mathrm{St}}L_{t+1}\mathcal{P}_{t+1}M_t^\top T_2$$
$$= \underbrace{e_{t+1}^3\|(I_d-W_tW_t^\top)x_{t+1}\|_2^2\|W_t^\top x_{t+1}\|_2^2 T_1\Theta^\top(I_d-W_tW_t^\top)x_{t+1}}_{:=u_{t+1}}\underbrace{x_{t+1}^\top W_tW_t^\top\Theta T_2}_{:=v_{t+1}^\top}.$$

We have for $p \geq 2$

$$\mathbb{E}_t\left[\left\|\mathbb{1}_{|e_{t+1}|\leq L}u_{t+1}v_{t+1}^\top\right\|_2^p\right]$$
$$\leq L^{3p}\mathbb{E}_t\left[\left\|(I_d-W_tW_t^\top)x_{t+1}\right\|_2^{2p}\|T_1\Theta^\top(I_d-W_tW_t^\top)x_{t+1}\|_2^p\right]$$
$$\quad\times \mathbb{E}_t\left[\|W_t^\top x_{t+1}\|_2^{2p}\|T_2\Theta^\top W_tW_t^\top x_{t+1}\|_2^p\right]$$
$$\leq L^{3p}(2p)^p(\sqrt{3}d)^p p^{\frac{p}{2}}\left(\sqrt{3}\operatorname{Tr}\left(T_1^2(I_{\mathrm{rk}}-G_t)\right)\right)^{\frac{p}{2}}(2p)^p(\sqrt{3}r_s)^p p^{\frac{p}{2}}\left(\sqrt{3}\operatorname{Tr}\left(T_2^2 G_t\right)\right)^{\frac{p}{2}}$$
$$= L^{3p}(12\sqrt{3})^p p^{3p}\left(dr_s\sqrt{\operatorname{Tr}\left(T_1^2(I_{\mathrm{rk}}-G_t)\right)\operatorname{Tr}\left(T_2^2 G_t\right)}\right)^p.$$

By Proposition 33, we have for $u \geq (2e)^3$

$$\mathbb{P}_t\left[\left\|\mathbb{1}_{|e_{t+1}|\leq L}u_{t+1}v_{t+1}^\top\right\|_2 \geq uL^3 12\sqrt{3}dr_s\sqrt{\operatorname{Tr}\left(T_1^2(I_{\mathrm{rk}}-G_t)\right)\operatorname{Tr}\left(T_2^2 G_t\right)}\right] \leq e^{-\frac{u^{1/3}}{e}}.$$

By choosing $u = \frac{L^3}{24\sqrt{3}}$, we have the probability bound. For the variance bound, we have

$$\mathbb{E}_t\left[\left(\operatorname{Sym}\left(T_1\Theta^\top\nabla_{\mathrm{St}}L_{t+1}\mathcal{P}_{t+1}M_t^\top T_2\right)\right)^2\right] = \mathbb{E}_t\left[\operatorname{Sym}\left(u_{t+1}v_{t+1}^\top\right)^2\right].$$

By using Proposition 22, we have

$$\mathbb{E}_t\left[\operatorname{Sym}\left(u_{t+1}v_{t+1}^\top\right)^2\right]$$
$$\preceq T_1\Theta^\top(I_d-W_tW_t^\top)$$
$$\quad\times \mathbb{E}_t\left[e_{t+1}^6\|(I_d-W_tW_t^\top)x_{t+1}\|_2^4\|W_t^\top x_{t+1}\|_2^4\|T_2\Theta^\top W_tW_t^\top x_{t+1}\|_2^2 x_{t+1}x_{t+1}^\top\right]$$
$$\quad\times (I_d-W_tW_t^\top)\Theta T_1$$

$$
\begin{aligned}
&+ \boldsymbol{T}_2\Theta^\top \boldsymbol{W}_t \boldsymbol{W}_t^\top \\
&\quad \times \mathbb{E}_t\left[e_{t+1}^6\|(\boldsymbol{I}_d - \boldsymbol{W}_t\boldsymbol{W}_t^\top)\boldsymbol{x}_{t+1}\|_2^4\|\boldsymbol{W}_t^\top\boldsymbol{x}_{t+1}\|_2^4\|\boldsymbol{T}_1\Theta^\top(\boldsymbol{I}_d - \boldsymbol{W}_t\boldsymbol{W}_t^\top)\boldsymbol{x}_{t+1}\|_2^2\boldsymbol{x}_{t+1}\boldsymbol{x}_{t+1}^\top\right] \\
&\quad \times \boldsymbol{W}_t\boldsymbol{W}_t^\top\Theta\boldsymbol{T}_2 \\
&\preceq Cd^2 r_s^2\left(\mathrm{Tr}(\boldsymbol{T}_2^2\mathsf{G}_t)\boldsymbol{T}_1(\boldsymbol{I}_{\mathrm{rk}} - \mathsf{G}_t)\boldsymbol{T}_1 + \mathrm{Tr}\big(\boldsymbol{T}_1^2(\boldsymbol{I}_{\mathrm{rk}} - \mathsf{G}_t)\big)\boldsymbol{T}_2\mathsf{G}_t\boldsymbol{T}_2\right).
\end{aligned}
$$

**Fifth item.** We define

$$
\begin{aligned}
\boldsymbol{T}_1\Theta^\top&\nabla_{\mathrm{St}}\boldsymbol{L}_{t+1}\boldsymbol{\mathcal{P}}_{t+1}\nabla_{\mathrm{St}}\boldsymbol{L}_{t+1}^\top\Theta\boldsymbol{T}_2 \\
&= \underbrace{e_{t+1}^4\|(\boldsymbol{I}_d - \boldsymbol{W}_t\boldsymbol{W}_t^\top)\boldsymbol{x}_{t+1}\|_2^2\|\boldsymbol{W}_t^\top\boldsymbol{x}_{t+1}\|_2^4\boldsymbol{T}_1\Theta^\top(\boldsymbol{I}_d - \boldsymbol{W}_t\boldsymbol{W}_t^\top)\boldsymbol{x}_{t+1}}_{:=\boldsymbol{u}_{t+1}} \\
&\quad \times \underbrace{\boldsymbol{x}_{t+1}^\top(\boldsymbol{I}_d - \boldsymbol{W}_t\boldsymbol{W}_t^\top)\Theta\boldsymbol{T}_2}_{:=\boldsymbol{v}_{t+1}^\top}.
\end{aligned}
$$

We have for $p \geq 2$

$$
\begin{aligned}
\mathbb{E}_t&\left[\left\|\mathbb{1}_{|e_{t+1}|\leq L}\boldsymbol{u}_{t+1}\boldsymbol{v}_{t+1}^\top\right\|_2^p\right] \\
&\leq L^{4p}\mathbb{E}_t\Big[\left\|(\boldsymbol{I}_d - \boldsymbol{W}_t\boldsymbol{W}_t^\top)\,\boldsymbol{x}_{t+1}\right\|_2^{2p}\|\boldsymbol{W}_t^\top\boldsymbol{x}_{t+1}\|_2^{4p} \\
&\qquad\qquad \times \|\boldsymbol{T}_1\Theta^\top(\boldsymbol{I}_d - \boldsymbol{W}_t\boldsymbol{W}_t^\top)\boldsymbol{x}_{t+1}\|_2^p\|\boldsymbol{T}_1\Theta^\top(\boldsymbol{I}_d - \boldsymbol{W}_t\boldsymbol{W}_t^\top)\boldsymbol{x}_{t+1}\|_2^p\Big] \\
&\leq L^{4p}\mathbb{E}_t\Big[\left\|(\boldsymbol{I}_d - \boldsymbol{W}_t\boldsymbol{W}_t^\top)\,\boldsymbol{x}_{t+1}\right\|_2^{4p}\Big]^{\frac{1}{2}}\mathbb{E}_t\Big[\|\boldsymbol{T}_1\Theta^\top(\boldsymbol{I}_d - \boldsymbol{W}_t\boldsymbol{W}_t^\top)\boldsymbol{x}_{t+1}\|_2^{4p}\Big]^{\frac{1}{4}} \\
&\qquad \times \mathbb{E}_t\Big[\|\boldsymbol{T}_2\Theta^\top(\boldsymbol{I}_d - \boldsymbol{W}_t\boldsymbol{W}_t^\top)\boldsymbol{x}_{t+1}\|_2^{4p}\Big]^{\frac{1}{4}}\mathbb{E}_t\Big[\|\boldsymbol{W}_t^\top\boldsymbol{x}_{t+1}\|_2^{4p}\Big] \\
&\leq L^{4p}(2p)^p(\sqrt{3}d)^p(2p)^p\left(3\mathrm{Tr}\big(\boldsymbol{T}_1^2(\boldsymbol{I}_{\mathrm{rk}} - \mathsf{G}_t)\big)\mathrm{Tr}\big(\boldsymbol{T}_2^2(\boldsymbol{I}_{\mathrm{rk}} - \mathsf{G}_t)\big)\right)^{\frac{p}{2}}(2p)^{2p}(\sqrt{3}r_s)^{2p} \\
&= L^{4p}(2\sqrt{3})^{4p}p^{4p}\left(dr_s^2\sqrt{\mathrm{Tr}\big(\boldsymbol{T}_1^2(\boldsymbol{I}_{\mathrm{rk}} - \mathsf{G}_t)\big)\mathrm{Tr}\big(\boldsymbol{T}_2^2(\boldsymbol{I}_{\mathrm{rk}} - \mathsf{G}_t)\big)}\right)^p.
\end{aligned}
$$

By Proposition 33, we have for $u \geq (2e)^4$

$$
\mathbb{P}_t\left[\left\|\mathbb{1}_{|e_{t+1}|\leq L}\boldsymbol{u}_{t+1}\boldsymbol{v}_{t+1}^\top\right\|_2 \geq uL^4(2\sqrt{3})^4dr_s^2\sqrt{\mathrm{Tr}\big(\boldsymbol{T}_1^2(\boldsymbol{I}_{\mathrm{rk}} - \mathsf{G}_t)\big)\mathrm{Tr}\big(\boldsymbol{T}_2^2(\boldsymbol{I}_{\mathrm{rk}} - \mathsf{G}_t)\big)}\right] \leq 2e^{-\frac{u^{1/4}}{e}}.
$$

By choosing $u = \frac{L^4}{2(2\sqrt{3})^4}$, we have the probability bound. For the variance bound, we have

$$
\mathbb{E}_t\left[\left(\mathrm{Sym}\left(\boldsymbol{T}_1\Theta^\top\nabla_{\mathrm{St}}\boldsymbol{L}_{t+1}\boldsymbol{\mathcal{P}}_{t+1}\nabla_{\mathrm{St}}\boldsymbol{L}_{t+1}^\top\Theta\boldsymbol{T}_2\right)\right)^2\right] = \mathbb{E}_t\left[\mathrm{Sym}\left(\boldsymbol{u}_{t+1}\boldsymbol{v}_{t+1}^\top\right)^2\right].
$$

By using Proposition 22, we have

$$
\begin{aligned}
\mathbb{E}_t&\left[\mathrm{Sym}\left(\boldsymbol{u}_{t+1}\boldsymbol{v}_{t+1}^\top\right)^2\right] \\
&\preceq \boldsymbol{T}_1\Theta^\top(\boldsymbol{I}_d - \boldsymbol{W}_t\boldsymbol{W}_t^\top) \\
&\quad \times \mathbb{E}_t\left[e_{t+1}^8\|(\boldsymbol{I}_d - \boldsymbol{W}_t\boldsymbol{W}_t^\top)\boldsymbol{x}_{t+1}\|_2^4\|\boldsymbol{W}_t^\top\boldsymbol{x}_{t+1}\|_2^8\|\boldsymbol{T}_2\Theta^\top(\boldsymbol{I}_d - \boldsymbol{W}_t\boldsymbol{W}_t^\top)\boldsymbol{x}_{t+1}\|_2^2\boldsymbol{x}_{t+1}\boldsymbol{x}_{t+1}^\top\right] \\
&\quad \times (\boldsymbol{I}_d - \boldsymbol{W}_t\boldsymbol{W}_t^\top)\Theta\boldsymbol{T}_1 \\
&+ \boldsymbol{T}_2\Theta^\top(\boldsymbol{I}_d - \boldsymbol{W}_t\boldsymbol{W}_t^\top) \\
&\quad \times \mathbb{E}_t\left[e_{t+1}^8\|(\boldsymbol{I}_d - \boldsymbol{W}_t\boldsymbol{W}_t^\top)\boldsymbol{x}_{t+1}\|_2^4\|\boldsymbol{W}_t^\top\boldsymbol{x}_{t+1}\|_2^8\|\boldsymbol{T}_1\Theta^\top(\boldsymbol{I}_d - \boldsymbol{W}_t\boldsymbol{W}_t^\top)\boldsymbol{x}_{t+1}\|_2^2\boldsymbol{x}_{t+1}\boldsymbol{x}_{t+1}^\top\right] \\
&\quad \times (\boldsymbol{I}_d - \boldsymbol{W}_t\boldsymbol{W}_t^\top)\Theta\boldsymbol{T}_2 \\
&\preceq Cd^2 r_s^4\left(\mathrm{Tr}\big(\boldsymbol{T}_2^2(\boldsymbol{I}_{\mathrm{rk}} - \mathsf{G}_t)\big)\boldsymbol{T}_1(\boldsymbol{I}_{\mathrm{rk}} - \mathsf{G}_t)\boldsymbol{T}_1 + \mathrm{Tr}\big(\boldsymbol{T}_1^2(\boldsymbol{I}_{\mathrm{rk}} - \mathsf{G}_t)\big)\boldsymbol{T}_2(\boldsymbol{I}_{\mathrm{rk}} - \mathsf{G}_t)\boldsymbol{T}_2\right).
\end{aligned}
$$

$\square$

By recalling the definitions $\{T_t\}_{t\in\mathbb{N}}$, $\Lambda$, $\Lambda_{11}$ in Sections F.2 and F.3, we define the event:

$$\mathcal{A}_{t+1} := \begin{cases} \mathcal{A}_{t+1}\left(T_t^{\frac{-1}{2}}, T_t^{\frac{-1}{2}}\right) \cap \mathcal{A}_{t+1}\left(\Lambda^{\frac{1}{2}}T_t^{\frac{-1}{2}}, T_t^{\frac{-1}{2}}\Lambda^{\frac{1}{2}}\right) \cap \{e_{t+1}\le L\}, & \alpha\in[0,0.5) \\ \mathcal{A}_{t+1}\left(T_t^{\frac{-1}{2}}, T_t^{\frac{-1}{2}}\right) \cap \mathcal{A}_{t+1}\left(\Lambda_{11}^{\frac{1}{2}}T_t^{\frac{-1}{2}}, T_t^{\frac{-1}{2}}\Lambda_{11}^{\frac{1}{2}}\right) \cap \{e_{t+1}\le L\}, & \alpha > 0.5. \end{cases}$$

We define the events $\mathcal{B}_{t+1}, \mathcal{C}_{t+1}, \mathcal{D}_{t+1}$, and $\mathcal{F}_{t+1}$ in the same way. Based on these events, we define the clipped versions of the noise matrices:

$$\mathsf{A}_{t+1} := \mathrm{Sym}\left(\Theta^\top \nabla_{\mathrm{St}} L_{t+1} \mathsf{M}_t^\top \mathbb{1}_{\mathcal{A}_{t+1}} - \mathbb{E}_t\left[\Theta^\top \nabla_{\mathrm{St}} L_{t+1} \mathsf{M}_t^\top \mathbb{1}_{\mathcal{A}_{t+1}}\right]\right)$$

$$\mathsf{B}_{t+1} := \frac{\mathsf{M}_t \mathcal{P}_{t+1}\mathsf{M}_t^\top \mathbb{1}_{\mathcal{B}_{t+1}}}{1+c_{t+1}^2} - \mathbb{E}_t\left[\frac{\mathsf{M}_t \mathcal{P}_{t+1}\mathsf{M}_t^\top \mathbb{1}_{\mathcal{B}_{t+1}}}{1+c_{t+1}^2}\right]$$

$$\mathsf{C}_{t+1} := \Theta^\top \nabla_{\mathrm{St}} L_{t+1} \nabla_{\mathrm{St}} L_{t+1}^\top \Theta \mathbb{1}_{\mathcal{C}_{t+1}} - \mathbb{E}_t\left[\Theta^\top \nabla_{\mathrm{St}} L_{t+1} \nabla_{\mathrm{St}} L_{t+1}^\top \Theta \mathbb{1}_{\mathcal{C}_{t+1}}\right]$$

$$\mathsf{D}_{t+1} := \mathrm{Sym}\left(\frac{\Theta^\top \nabla_{\mathrm{St}} L_{t+1} \mathcal{P}_{t+1}\mathsf{M}_t^\top \mathbb{1}_{\mathcal{D}_{t+1}}}{1+c_{t+1}^2} - \mathbb{E}_t\left[\frac{\Theta^\top \nabla_{\mathrm{St}} L_{t+1} \mathcal{P}_{t+1}\mathsf{M}_t^\top \mathbb{1}_{\mathcal{D}_{t+1}}}{1+c_{t+1}^2}\right]\right)$$

$$\mathsf{F}_{t+1} := \frac{\Theta^\top \nabla_{\mathrm{St}} L_{t+1} \mathcal{P}_{t+1}\nabla_{\mathrm{St}} L_{t+1}^\top \Theta \mathbb{1}_{\mathcal{F}_{t+1}}}{1+c_{t+1}^2} - \mathbb{E}_t\left[\frac{\Theta^\top \nabla_{\mathrm{St}} L_{t+1} \mathcal{P}_{t+1}\nabla_{\mathrm{St}} L_{t+1}^\top \Theta \mathbb{1}_{\mathcal{F}_{t+1}}}{1+c_{t+1}^2}\right] \tag{F.38}$$

Let $\mathsf{X} \in \left\{\frac{\eta/2}{\sqrt{r_s}}\mathsf{A}, \frac{\eta^2/16}{r_s}\mathsf{B}, \frac{\eta^2/16}{r_s}\mathsf{C}, \frac{\eta^3/32}{r_s^{3/2}}\mathsf{D}, \frac{\eta^4/256}{r_s^2}\mathsf{F}\right\}$ and

$$\Gamma_1 := \begin{cases} \mathbf{I}_r, & \alpha\in[0,0.5) \\ \mathbf{I}_{r_u} & \alpha > 0.5 \end{cases} \qquad \Gamma_2 := \begin{cases} \Lambda^{\frac{1}{2}}, & \alpha\in[0,0.5) \\ \Lambda_{11}^{\frac{1}{2}}, & \alpha > 0.5. \end{cases}$$

For $\ell\in\{1,2\}$, we define:

$$\mathrm{Quad}_{k,t}^{(\ell)}(\mathsf{X}) := \sum_{j=1}^k \mathbb{E}_{j-1}\left[\left(\Gamma_\ell T_t^{\frac{-1}{2}} \mathsf{X}_j T_t^{\frac{-1}{2}} \Gamma_\ell\right)^2\right].$$

We have the following corollary.

**Corollary 4.** *Let* $\mathrm{rk}\in\{r, r_u\}$, $\mathrm{rk}_\star\in\{r, r_s\}$ *and*

$$\mathsf{S}_t := \eta\sum_{j=1}^t \mathsf{G}_{j-1} \ \text{and}\ \eta = \frac{\eta}{\sqrt{r_s}\|\Lambda\|_{\mathrm{F}}} \ \text{and}\ r_u = \lceil\log^{2.5} d\rceil.$$

*Assume the following conditions hold:*

- $T_t \succeq \frac{\kappa_d r_s}{d}\mathbf{I}_{\mathrm{rk}}$,

- $T_t^{\frac{-1}{2}}\mathsf{S}_t T_t^{\frac{-1}{2}} \preceq \frac{C_{ub}}{\kappa_d}\begin{cases}(2\eta t\vee r^\alpha)\mathbf{I}_r, & \alpha\in[0,0.5) \\ r_s^\alpha \log d\,\mathbf{I}_{r_u}, & \alpha > 0.5,\end{cases}$

- $T_t^{\frac{-1}{2}}\mathsf{G}_j T_t^{\frac{-1}{2}} \preceq \frac{C_{ub}}{\kappa_d}\mathbf{I}_{\mathrm{rk}}$ *for* $j\le t-1$.

*Let*

$$\begin{cases} \mathsf{p}_1 = 1 \ \text{and}\ \mathsf{p}_2 = 1-\alpha, & \alpha\in[0,1) \\ \mathsf{p}_1 = 1 \ \text{and}\ \mathsf{p}_2 = \frac{2\log\log r_u}{\log r_u} & \alpha = 1 \\ \mathsf{p}_1 = 1 \ \text{and}\ \mathsf{p}_2 = 0 & \alpha > 1. \end{cases}$$

*For* $\eta t \le \frac{1}{2}\mathrm{rk}_\star^\alpha \log\left(\frac{d\log^{1.5} d}{r_s}\right)$, *the following results hold:*

**(a) Quadratic variation bounds.** *We have:*

$$\|Quad_{t,t}^{(\ell)}(\mathsf{X})\|_2 \le \frac{C\log d}{\kappa_d^2}\frac{\|\mathbf{\Lambda}\|_{\mathrm{F}}\mathsf{rk}^{\mathsf{p}_\ell}\mathsf{rk}_\star^\alpha}{r_s^{3/2}}\begin{cases} C_{ub}\eta d & \mathsf{X} = \frac{\eta/2}{\sqrt{r_s}}\mathsf{A} \\ C_{ub}^2\eta^3 d^2, & \mathsf{X} = \frac{\eta^2/16}{r_s}\mathsf{B} \\ \eta^3 d^2, & \mathsf{X} = \frac{\eta^2/16}{r_s}\mathsf{C} \\ C_{ub}\eta^5 d^3, & \mathsf{X} = \frac{\eta^3/32}{r_s^{3/2}}\mathsf{D} \\ \eta^7 d^2, & \mathsf{X} = \frac{\eta^4/256}{r_s^2}\mathsf{F}. \end{cases}$$

**(b) Operator norm bounds.** *For $L \ge 8\sqrt{2}e$, there exists $C > 0$ such that*

$$\left\|\Gamma_\ell\boldsymbol{T}_t^{\frac{-1}{2}}\mathsf{X}_j\boldsymbol{T}_t^{\frac{-1}{2}}\Gamma_\ell\right\|_2 \le r_{j,t}^{(\ell)}(\mathsf{X}) := \begin{cases} \frac{\eta L^2}{2\sqrt{r_s}}\sqrt{Tr(\Gamma_\ell^2\boldsymbol{T}_t^{-1})Tr(\Gamma_\ell^2\boldsymbol{T}_t^{\frac{-1}{2}}\mathsf{G}_{j-1}\boldsymbol{T}_t^{\frac{-1}{2}})}, & \mathsf{X} = \frac{\eta/2}{\sqrt{r_s}}\mathsf{A} \\ \frac{\eta^2 L^4}{16r_s}dTr(\Gamma_\ell^2\boldsymbol{T}_t^{\frac{-1}{2}}\mathsf{G}_{j-1}\boldsymbol{T}_t^{\frac{-1}{2}}), & \mathsf{X} = \frac{\eta^2/16}{r_s}\mathsf{B} \\ \frac{\eta^2 L^4}{16}Tr(\Gamma_\ell^2\boldsymbol{T}_t^{-1}), & \mathsf{X} = \frac{\eta^2/16}{r_s}\mathsf{C} \\ \frac{\eta^3 L^6}{32\sqrt{r_s}}d\sqrt{Tr(\Gamma_\ell^2\boldsymbol{T}_t^{-1})Tr(\Gamma_\ell^2\boldsymbol{T}_t^{\frac{-1}{2}}\mathsf{G}_{j-1}\boldsymbol{T}_t^{\frac{-1}{2}})}, & \mathsf{X} = \frac{\eta^3/32}{r_s^{3/2}}\mathsf{D} \\ \frac{\eta^4 L^8}{256}dTr(\Gamma_\ell^2\boldsymbol{T}_t^{-1}), & \mathsf{X} = \frac{\eta^4/256}{r_s^2}\mathsf{F} \end{cases}$$

$$\le \frac{C}{\kappa_d}\frac{\mathsf{rk}^{\mathsf{p}_\ell}}{r_s}\begin{cases} L^2\sqrt{C_{ub}}\eta\sqrt{d}, & \mathsf{X} = \frac{\eta/2}{\sqrt{r_s}}\mathsf{A} \\ L^4 C_{ub}\eta^2 d, & \mathsf{X} = \frac{\eta^2/16}{r_s}\mathsf{B} \\ L^4\eta^2 d, & \mathsf{X} = \frac{\eta^2/16}{r_s}\mathsf{C} \\ L^6\sqrt{C_{ub}}\eta^3 d^{3/2}, & \mathsf{X} = \frac{\eta^3/32}{r_s^{3/2}}\mathsf{D} \\ L^8\eta^4 d^2, & \mathsf{X} = \frac{\eta^4/256}{r_s^2}\mathsf{F}. \end{cases}$$

*Proof. Quadratic variation bounds.* We will use the variance bounds given in Proposition 18. For $\mathsf{X} = \frac{\eta/2}{\sqrt{r_s}}\mathsf{A}$, we have

$$Quad_{t,t}^{(\ell)}(\tfrac{\eta/2}{\sqrt{r_s}}\mathsf{A}) \preceq \frac{C\eta\|\mathbf{\Lambda}\|_{\mathrm{F}}}{\sqrt{r_s}}\left(Tr\left(\Gamma_\ell^2\boldsymbol{T}_t^{\frac{-1}{2}}\mathsf{S}_t\boldsymbol{T}_t^{\frac{-1}{2}}\right)\Gamma_\ell\boldsymbol{T}_t^{-1}\Gamma_\ell + Tr\left(\Gamma_\ell^2\boldsymbol{T}_t^{-1}\right)\Gamma_\ell\boldsymbol{T}_t^{\frac{-1}{2}}\mathsf{S}_t\boldsymbol{T}_t^{\frac{-1}{2}}\Gamma_\ell\right)$$

$$\preceq \frac{C\eta\|\mathbf{\Lambda}\|_{\mathrm{F}}}{\sqrt{r_s}}\frac{C_{ub}\mathsf{rk}^{\mathsf{p}_\ell}\mathsf{rk}_\star^\alpha\log d}{\kappa_d^2}\frac{d}{r_s}\boldsymbol{I}_{\mathsf{rk}}.$$

For $\mathsf{X} = \frac{\eta^2/16}{r_s}\mathsf{B}$, we have

$$Quad_{t,t}^{(\ell)}(\tfrac{\eta^2/16}{r_s}\mathsf{B}) \preceq \frac{CC_{ub}\eta^3 d^2\|\mathbf{\Lambda}\|_{\mathrm{F}}}{r_s^{3/2}}\sup_{j\le t}\left(Tr\left(\Gamma_\ell^2\boldsymbol{T}_t^{\frac{-1}{2}}\mathsf{G}_{j-1}\boldsymbol{T}_t^{\frac{-1}{2}}\right)\right)\Gamma_\ell\boldsymbol{T}_t^{\frac{-1}{2}}\mathsf{S}_t\boldsymbol{T}_t^{\frac{-1}{2}}\Gamma_\ell$$

$$\preceq \frac{C\eta^3 d^2\|\mathbf{\Lambda}\|_{\mathrm{F}}}{r_s^{3/2}}\frac{C_{ub}^2\mathsf{rk}^{\mathsf{p}_\ell}\mathsf{rk}_\star^\alpha\log d}{\kappa_d^2}\boldsymbol{I}_{\mathsf{rk}}.$$

For $\mathsf{X} = \frac{\eta^2/16}{r_s}\mathsf{C}$, we have

$$Quad_{t,t}^{(\ell)}(\tfrac{\eta^2/16}{r_s}\mathsf{C}) \preceq C\eta^4 tTr(\Gamma_\ell^2\boldsymbol{T}_t^{-1})\Gamma_\ell\boldsymbol{T}_t^{-1}\Gamma_\ell \preceq \frac{C\eta^3 d^2\|\mathbf{\Lambda}\|_{\mathrm{F}}}{r_s^{3/2}}\frac{\mathsf{rk}^{\mathsf{p}_\ell}\mathsf{rk}_\star^\alpha\log d}{\kappa_d^2}\boldsymbol{I}_{\mathsf{rk}}.$$

For $\mathsf{X} = \frac{\eta^3/32}{r_s^{3/2}}\mathsf{D}$, we have

$$Quad_{t,t}^{(\ell)}(\tfrac{\eta^3/32}{r_s^{3/2}}\mathsf{D}) \preceq \frac{C\eta^5 d^2\|\mathbf{\Lambda}\|_{\mathrm{F}}}{\sqrt{r_s}}\left(Tr(\Gamma_\ell^2\boldsymbol{T}_t^{\frac{-1}{2}}\mathsf{S}_t\boldsymbol{T}_t^{\frac{-1}{2}})\Gamma_\ell\boldsymbol{T}_t^{-1}\Gamma_\ell + Tr(\Gamma_\ell^2\boldsymbol{T}_t^{-1})\Gamma_\ell\boldsymbol{T}_t^{\frac{-1}{2}}\mathsf{S}_t\boldsymbol{T}_t^{\frac{-1}{2}}\Gamma_\ell\right)$$

$$\preceq \frac{C\eta^5 d^2\|\mathbf{\Lambda}\|_{\mathrm{F}}}{\sqrt{r_s}}\frac{C_{ub}\mathsf{rk}^{\mathsf{p}_\ell}\mathsf{rk}_\star^\alpha\log d}{\kappa_d^2}\frac{d}{r_s}\boldsymbol{I}_{\mathsf{rk}}.$$

For $X = \frac{\eta^4/256}{r_s^2}F$, we have

$$\mathrm{Quad}_{t,t}^{(\ell)}(\tfrac{\eta^4/256}{r_s^2}F) \preceq C\eta^8 d^2 t \mathrm{Tr}(\Gamma_\ell^2 \boldsymbol{T}_t^{-1})\Gamma_\ell \boldsymbol{T}_t^{-1}\Gamma_\ell \preceq \frac{C\eta^7 d^2 \|\boldsymbol{\Lambda}\|_{\mathrm{F}}}{r_s^{3/2}} \frac{\mathsf{rk}^{\mathsf{p}_\ell}\mathsf{rk}_\star^\alpha \log d}{\kappa_d^2} \boldsymbol{I}_{\mathrm{rk}}.$$

*Operator Norm Bounds.* We will use the events defined in Proposition 18. For $X = \frac{\eta/2}{\sqrt{r_s}}A$, we have

$$r_{j,t}^{(\ell)}(\tfrac{\eta/2}{\sqrt{r_s}}A) = \frac{\eta/2}{\sqrt{r_s}}L^2 \sqrt{\mathrm{Tr}(\Gamma_\ell^2 \boldsymbol{T}_t^{-1})\mathrm{Tr}(\Gamma_\ell^2 \boldsymbol{T}_t^{\frac{-1}{2}}\mathsf{G}_{j-1}\boldsymbol{T}_t^{\frac{-1}{2}})} \leq \frac{CL^2}{\kappa_d}\frac{\sqrt{C_{\mathrm{ub}}}\eta\sqrt{d}\mathsf{rk}^{\mathsf{p}_\ell}}{r_s}.$$

For $X = \frac{\eta^2/16}{r_s}B$, we have

$$r_{j,t}^{(\ell)}(\tfrac{\eta^2/16}{r_s}B) = \frac{\eta^2}{16r_s}L^4 d\,\mathrm{Tr}(\Gamma_\ell^2 \boldsymbol{T}_t^{\frac{-1}{2}}\mathsf{G}_{j-1}\boldsymbol{T}_t^{\frac{-1}{2}}) \leq \frac{CL^4}{\kappa_d}\frac{C_{\mathrm{ub}}\eta^2 d\mathsf{rk}^{\mathsf{p}_\ell}}{r_s}.$$

For $X = \frac{\eta^2/16}{r_s}C$, we have

$$r_{j,t}^{(\ell)}(\tfrac{\eta^2/16}{r_s}C) = \frac{\eta^2}{16r_s}L^4\mathrm{Tr}(\Gamma_\ell^2 \boldsymbol{T}_t^{-1}) \leq \frac{CL^4}{\kappa_d}\frac{\eta^2 d\mathsf{rk}^{\mathsf{p}_\ell}}{r_s}.$$

For $X = \frac{\eta^3/32}{r_s^{3/2}}D$, we have

$$r_{j,t}^{(\ell)}(\tfrac{\eta^3/32}{r_s^{3/2}}D) = \frac{\eta^3}{32\sqrt{r_s}}L^6 d\sqrt{\mathrm{Tr}(\Gamma_\ell^2 \boldsymbol{T}_t^{\frac{-1}{2}}\mathsf{G}_{j-1}\boldsymbol{T}_t^{\frac{-1}{2}})\mathrm{Tr}(\Gamma_\ell^2 \boldsymbol{T}_t^{-1})} \leq \frac{CL^6}{\kappa_d}\frac{\sqrt{C_{\mathrm{ub}}}\eta^3 d^{3/2}\mathsf{rk}^{\mathsf{p}_\ell}}{r_s}.$$

For $X = \frac{\eta^4/256}{r_s^2}F$, we have

$$r_{j,t}^{(\ell)}(\tfrac{\eta^4/256}{r_s^2}F) = \frac{\eta^4}{256}L^8 d\mathrm{Tr}(\Gamma_\ell^2 \boldsymbol{T}_t^{-1}) \leq \frac{CL^8}{\kappa_d}\frac{\eta^4 d^2\mathsf{rk}^{\mathsf{p}_\ell}}{r_s}.$$

$\square$

**Proposition 19.** *Let $\{\mathsf{Y}_t,\ t=1,2\cdots\}$ be a symmetric-matrix martingale with difference sequence $\{\mathsf{X}_t := \mathsf{Y}_{t+1} - \mathsf{Y}_t,\ t=1,2\cdots\}$, whose values are symmetric matrices with dimension $r \leq d$. Let $\{\boldsymbol{T}_t,\ t=1,2,\cdots\}$ be a deterministic sequence, whose values are positive semi-definite matrices with the same dimensionality. Assume that the difference sequence is uniformly bounded in the sense that for a predictable triangular sequence $\{r_{j,t}\}_{j\leq t}$, we have*

$$\lambda_{max}(\boldsymbol{T}_t^{\frac{-1}{2}}\mathsf{X}_j\boldsymbol{T}_t^{\frac{-1}{2}}) \leq r_{j,t}\ \text{for}\ j=1,2,\cdots,t.$$

*Define the predictable uniform bound and quadratic variation process of the martingale:*

$$R_{k,t} := \max_{j\leq k} r_{j,t}\ \text{and}\ Quad_{k,t}(\mathsf{X}) := \sum_{j=1}^k \mathbb{E}_{j-1}\left[\left(\boldsymbol{T}_t^{\frac{-1}{2}}\mathsf{X}_j\boldsymbol{T}_t^{\frac{-1}{2}}\right)^2\right]\ \text{for}\ k\leq t=1,2,\cdots.$$

*Let $\mathcal{T} \leq d^3$ be a bounded stopping time. Then, for any deterministic $\sigma^2, \widetilde{L} > 0$*

$$\mathbb{P}\left[\exists t\leq\mathcal{T}, \mathsf{Y}_t \npreceq u\boldsymbol{T}_t\ \text{and}\ \max_{t\leq\mathcal{T}}\|Quad_{t,t}(\mathsf{X})\|_2 \leq \sigma^2\ \text{and}\ \max_{t\leq\mathcal{T}} R_{t,t} \leq \widetilde{L}\right]$$

$$\leq d^4 \cdot \exp\left(\frac{-u^2/2}{\sigma^2 + \widetilde{L}u/3}\right).$$

*Proof.* We have

$$\mathcal{E}_{\mathrm{target}} \equiv \left\{\exists t\leq\mathcal{T}, \mathsf{Y}_t \npreceq u\boldsymbol{T}_t\ \text{and}\ \max_{t\leq\mathcal{T}}\|\mathrm{Quad}_{t,t}(\mathsf{X})\|_2 \leq \sigma^2\ \text{and}\ \max_{t\leq\mathcal{T}} R_{t,t} \leq \widetilde{L}\right\}$$

$$\subseteq \bigcup_{t=0}^{\mathcal{T}}\left\{\exists k\leq t, \mathsf{Y}_k \npreceq u\boldsymbol{T}_t\ \text{and}\ \|\mathrm{Quad}_{t,t}(\mathsf{X})\|_2 \leq \sigma^2\ \text{and}\ R_{t,t} \leq \widetilde{L}\right\}.$$

Therefore, we have

$$\mathbb{P}\big[\mathcal{E}_{\text{target}}\big] \leq \sum_{n=1}^{d^3} \mathbb{P}\left[\exists k \leq t, \ \mathsf{Y}_k \npreceq u\boldsymbol{T}_t \ \text{and} \ \|\text{Quad}_{t,t}(\mathsf{X})\|_2 \leq \sigma^2 \ \text{and} \ R_{t,t} \leq \widetilde{L}\right]. \quad \text{(F.39)}$$

In the following, we will bound the each term in the right hands-side of (F.39). By [Tro10, Lemma 6.7], we have for $k = 1, \cdots, t$ and $\theta > 0$,

$$\mathbb{1}_{R_{k,t} \leq \widetilde{L}} \mathbb{E}_{k-1}\left[e^{\frac{\theta}{L}\boldsymbol{T}_t^{\frac{-1}{2}}\mathsf{X}_k\boldsymbol{T}_t^{\frac{-1}{2}}}\right] \preceq \mathbb{1}_{R_{k,t} \leq \widetilde{L}} \exp\left(\frac{e^\theta - \theta - 1}{\widetilde{L}^2}\mathbb{E}_{k-1}\left[(\boldsymbol{T}_t^{\frac{-1}{2}}\mathsf{X}_k\boldsymbol{T}_t^{\frac{-1}{2}})^2\right]\right). \quad \text{(F.40)}$$

For notational convenience call $g(\theta) := e^\theta - \theta - 1$. We define a super martingale such that for $0 < k \leq t$,

$$S_k := \text{Tr}\left(\exp\left(\frac{\theta}{L}\boldsymbol{T}_t^{\frac{-1}{2}}\mathsf{Y}_k\boldsymbol{T}_t^{\frac{-1}{2}} - \frac{g(\theta)}{\widetilde{L}^2}\text{Quad}_{k,t}(\mathsf{X})\right)\right)\mathbb{1}_{R_{k,t} \leq \widetilde{L}},$$

with initial values $R_{0,t} = 0$, $\mathsf{Y}_0 = \text{Quad}_{0,t} = 0$, and thus, $S_0 = r$. Note that by (F.40) and [Tro10, Corollary 3.3], we can show that $\mathbb{E}_{k-1}S_k \leq S_{k-1}$. We define a stopping time and an event

$$\mathcal{T}_{\text{hit}} := \{k \geq 0 \mid \lambda_{\max}(\boldsymbol{T}_t^{\frac{-1}{2}}\mathsf{Y}_k\boldsymbol{T}_t^{\frac{-1}{2}}) \geq u\} \wedge t,$$

$$\mathcal{E}_{\text{hit}} := \{\mathcal{T}_{\text{hit}} \leq t\} \cap \{\|\text{Quad}_{t,t}(\mathsf{X})\|_2 \leq \sigma^2 \ \text{and} \ R_{t,t} \leq \widetilde{L}\}.$$

We have

$$\mathbb{1}_{\mathcal{E}_{\text{hit}}}S_{\mathcal{T}_{\text{hit}}} \geq \mathbb{1}_{\mathcal{E}_{\text{hit}}} \exp\left(\frac{\theta}{\widetilde{L}}u - \frac{g(\theta)}{\widetilde{L}^2}\sigma^2\right) \overset{(a)}{\Rightarrow} r \geq \mathbb{P}\left[\mathcal{E}_{\text{hit}}\right]\exp\left(\frac{\theta}{\widetilde{L}}u - \frac{g(\theta)}{\widetilde{L}^2}\sigma^2\right)$$

$$\Rightarrow r\inf_{\theta > 0}\exp\left(-\theta\frac{u}{\widetilde{L}} + g(\theta)\frac{\sigma^2}{\widetilde{L}^2}\right) \geq \mathbb{P}\left[\mathcal{E}_{\text{hit}}\right],$$

where we use Doob's optional sampling theorem in (a). Since the infimum is attained at $\theta > 0$ and the convex conjugate of $g(\theta)$ is $g^\star(\theta) = (\theta + 1)\log(\theta + 1) - \theta$, we have

$$\mathbb{P}\left[\mathcal{E}_{\text{hit}}\right] \leq r \cdot \exp\left(-\frac{\sigma^2}{\widetilde{L}^2}g^\star\left(\frac{u\widetilde{L}}{\sigma^2}\right)\right) \leq r \cdot \exp\left(\frac{-u^2/2}{\sigma^2 + \widetilde{L}u/3}\right),$$

where we used $g^\star(\theta) \geq \frac{u^2/2}{1+u/3}$ in the last step. By $r \leq d$ and (F.39), we have the statement. $\qquad\square$

**Proposition 20.** *Let $\mathbb{P}_0$ denote the conditional probability conditioned on $\boldsymbol{W}_0$. We consider $r_u = \lceil\log^{2.5}d\rceil$, and*

$$\alpha \in [0, 0.5] : \quad \frac{r_s}{r} \to \varphi, \quad \eta \asymp \frac{1}{dr^\alpha \log^{20}(1+d/r_s)}, \quad \kappa_d = \frac{1}{\log^{3.5}d}, \qquad C_{ub} = 12\left(1 + \frac{1}{\sqrt{\varphi}}\right)^2$$

$$\alpha > 0.5 : \quad r_s \asymp 1, \quad \eta \asymp \frac{1}{dr_u^{4\alpha+3}\log^{18}d}, \qquad \kappa_d = \frac{1}{r_u\log^{2.5}d}, \quad C_{ub} = 2^\alpha 30r_u.$$

*For $\alpha \in [0, 0.5)$, we define $\mathcal{T} := \mathcal{T}_{bad} \wedge \frac{1}{2\eta}\left(r_s\big(1 - \log^{\frac{-1}{2}}d\big) \wedge r\right)^\alpha \log\left(\frac{d\log^{1.5}d}{r_s}\right)$. We have for $d \geq \Omega_{\alpha,\varphi,\beta}(1)$*

$$\mathbb{P}_0\left[\sup_{t \leq \mathcal{T}}\|\boldsymbol{T}_t^{\frac{-1}{2}}\boldsymbol{\nu}_t\boldsymbol{T}_t^{\frac{-1}{2}}\|_2 \vee r^{\frac{\alpha}{2}}\|\boldsymbol{\Lambda}^{\frac{1}{2}}\boldsymbol{T}_t^{\frac{-1}{2}}\boldsymbol{\nu}_t\boldsymbol{T}_t^{\frac{-1}{2}}\boldsymbol{\Lambda}^{\frac{1}{2}}\|_2 \geq \kappa_d r^{\frac{-\alpha}{2}} \ \text{and} \ \mathcal{G}_{init}\right]$$

$$\leq 20d^4\exp(-\log^2 d).$$

*For $\alpha > 0.5$, we set $\mathcal{T} := \mathcal{T}_{bad} \wedge \frac{1}{2\eta}r_s^\alpha \log\left(\frac{d\log^{1.5}d}{r_s}\right)$. We have for $d \geq \Omega_{\alpha,r_s}(1)$*

$$\mathbb{P}_0\left[\sup_{t \leq \mathcal{T}}\|\boldsymbol{T}_t^{\frac{-1}{2}}\boldsymbol{\nu}_t\boldsymbol{T}_t^{\frac{-1}{2}}\|_2 \vee r_u^{\frac{\alpha}{2}}\|\boldsymbol{\Lambda}_{11}^{\frac{1}{2}}\boldsymbol{T}_t^{\frac{-1}{2}}\boldsymbol{\nu}_t\boldsymbol{T}_t^{\frac{-1}{2}}\boldsymbol{\Lambda}_{11}^{\frac{1}{2}}\|_2 \geq \kappa_d r_u^{\frac{-\alpha}{2}} \ \text{and} \ \mathcal{G}_{init}\right]$$

$$\leq 20d^4\exp(-\log^2 d).$$

*Proof.* For notational convenience, we introduce $\mathcal{X} := \left\{ \frac{\eta/2}{\sqrt{r_s}}\mathsf{A}, \frac{\eta^2/16}{r_s}\mathsf{B}, \frac{\eta^2/16}{r_s}\mathsf{C}, \frac{\eta^3/32}{r_s^{3/2}}\mathsf{D}, \frac{\eta^4/256}{r_s^2}\mathsf{F} \right\}$.

For both cases, we will set the clip threshold to $L = \log^2 d$. We introduce the notation $R_t^{(\ell)} := \max_{\mathsf{X}\in\mathcal{X}} \max_{j\leq t} r_{j,t}^{(\ell)}(\mathsf{X})$ and $\|\mathsf{Quad}_t^{(\ell)}\|_2 := \max_{\mathsf{X}\in\mathcal{X}}\|\mathsf{Quad}_{t,t}^{(\ell)}(\mathsf{X})\|_2$ for $\ell = 1,2$.

For $\alpha \in [0, 0.5)$, we can write for all $\mathsf{X} \in \mathcal{X}$,

$$\mathbb{P}_0\left[ \sup_{t\leq\mathcal{T}} \|\boldsymbol{T}_t^{\frac{-1}{2}}\boldsymbol{\nu}_t\boldsymbol{T}_t^{\frac{-1}{2}}\|_2 \vee r^{\frac{\alpha}{2}}\|\boldsymbol{\Lambda}^{\frac{1}{2}}\boldsymbol{T}_t^{\frac{-1}{2}}\boldsymbol{\nu}_t\boldsymbol{T}_t^{\frac{-1}{2}}\boldsymbol{\Lambda}^{\frac{1}{2}}\|_2 \geq \kappa_d r^{\frac{-\alpha}{2}} \text{ and } \mathcal{G}_{\text{init}} \right]$$

$$\leq \sum_{\mathsf{X}\in\mathcal{X}} \mathbb{P}_0\left[ \sup_{t\leq\mathcal{T}} \left\|\boldsymbol{T}_t^{\frac{-1}{2}}\Big(\sum_{j\leq t}\mathsf{X}_j\Big)\boldsymbol{T}_t^{\frac{-1}{2}}\right\|_2 \geq \frac{\kappa_d r^{\frac{-\alpha}{2}}}{10} \text{ and } \mathcal{G}_{\text{init}} \right]$$

$$+ \sum_{\mathsf{X}\in\mathcal{X}} \mathbb{P}_0\left[ \sup_{t\leq\mathcal{T}} \left\|\boldsymbol{\Lambda}^{\frac{1}{2}}\boldsymbol{T}_t^{\frac{-1}{2}}\Big(\sum_{j\leq t}\mathsf{X}_j\Big)\boldsymbol{T}_t^{\frac{-1}{2}}\boldsymbol{\Lambda}^{\frac{1}{2}}\right\|_2 \geq \frac{\kappa_d r^{-\alpha}}{10} \text{ and } \mathcal{G}_{\text{init}} \right].$$

By Propositions 14 and 16 and Corollary 4, $\mathcal{G}_{\text{init}}$ implies the events

$$\mathcal{E}_{\text{ht},1} \equiv \left\{ \max_{t\leq\mathcal{T}}\|\mathsf{Quad}_t^{(1)}\|_2 \leq \frac{O_{\alpha,\beta,\varphi}\left(r^{-\alpha}\right)}{\log^{12} d} \text{ and } \max_{t\leq\mathcal{T}} R_t^{(1)} \leq \frac{O_{\alpha,\beta,\varphi}\left(r^{-\alpha}\right)}{\sqrt{d}\log^{12.5} d} \right\}$$

$$\mathcal{E}_{\text{ht},2} \equiv \left\{ \max_{t\leq\mathcal{T}}\|\mathsf{Quad}_t^{(2)}\|_2 \leq \frac{O_{\alpha,\beta,\varphi}\left(r^{-2\alpha}\right)}{\log^{12} d} \text{ and } \max_{t\leq\mathcal{T}} R_t^{(2)} \leq \frac{O_{\alpha,\beta,\varphi}\left(r^{-2\alpha}\right)}{\sqrt{d}\log^{12.5} d} \right\}.$$

Therefore, by using Proposition 19

$$\mathbb{P}_0\left[ \sup_{t\leq\mathcal{T}} \left\|\boldsymbol{T}_t^{\frac{-1}{2}}\Big(\sum_{j\leq t}\mathsf{X}_j\Big)\boldsymbol{T}_t^{\frac{-1}{2}}\right\|_2 \geq \frac{\kappa_d r^{\frac{-\alpha}{2}}}{10} \text{ and } \mathcal{G}_{\text{init}} \right]$$

$$\leq \mathbb{P}_0\left[ \sup_{t\leq\mathcal{T}} \left\|\boldsymbol{T}_t^{\frac{-1}{2}}\Big(\sum_{j\leq t}\mathsf{X}_j\Big)\boldsymbol{T}_t^{\frac{-1}{2}}\right\|_2 \geq \frac{\kappa_d r^{\frac{-\alpha}{2}}}{10} \text{ and } \mathcal{E}_{\text{ht},1} \right]$$

$$\leq 2d^4 \exp\left(-\log^2 d\right).$$

Similarly,

$$\mathbb{P}_0\left[ \sup_{t\leq\mathcal{T}} \left\|\boldsymbol{\Lambda}^{\frac{1}{2}}\boldsymbol{T}_t^{\frac{-1}{2}}\Big(\sum_{j\leq t}\mathsf{X}_j\Big)\boldsymbol{T}_t^{\frac{-1}{2}}\boldsymbol{\Lambda}^{\frac{1}{2}}\right\|_2 \geq \frac{\kappa_d r^{-\alpha}}{10} \text{ and } \mathcal{G}_{\text{init}} \right]$$

$$\leq \mathbb{P}_0\left[ \sup_{t\leq\mathcal{T}} \left\|\boldsymbol{\Lambda}^{\frac{1}{2}}\boldsymbol{T}_t^{\frac{-1}{2}}\Big(\sum_{j\leq t}\mathsf{X}_j\Big)\boldsymbol{T}_t^{\frac{-1}{2}}\boldsymbol{\Lambda}^{\frac{1}{2}}\right\|_2 \geq \frac{\kappa_d r^{-\alpha}}{10} \text{ and } \mathcal{E}_{\text{ht},2} \right]$$

$$\leq 2d^4 \exp\left(-\log^2 d\right).$$

For $\alpha > 0.5$, we can write for all $\mathsf{X} \in \mathcal{X}$,

$$\mathbb{P}_0\left[ \sup_{t\leq\mathcal{T}} \|\boldsymbol{T}_t^{\frac{-1}{2}}\boldsymbol{\nu}_t\boldsymbol{T}_t^{\frac{-1}{2}}\|_2 \vee r_u^{\frac{\alpha}{2}}\|\boldsymbol{\Lambda}_{11}^{\frac{1}{2}}\boldsymbol{T}_t^{\frac{-1}{2}}\boldsymbol{\nu}_t\boldsymbol{T}_t^{\frac{-1}{2}}\boldsymbol{\Lambda}_{11}^{\frac{1}{2}}\|_2 \geq \kappa_d r_u^{\frac{-\alpha}{2}} \text{ and } \mathcal{G}_{\text{init}} \right]$$

$$\leq \sum_{\mathsf{X}\in\mathcal{X}} \mathbb{P}_0\left[ \sup_{t\leq\mathcal{T}} \left\|\boldsymbol{T}_t^{\frac{-1}{2}}\Big(\sum_{j\leq t}\mathsf{X}_j\Big)\boldsymbol{T}_t^{\frac{-1}{2}}\right\|_2 \geq \frac{\kappa_d r_u^{\frac{-\alpha}{2}}}{10} \text{ and } \mathcal{G}_{\text{init}} \right]$$

$$+ \sum_{\mathsf{X}\in\mathcal{X}} \mathbb{P}_0\left[ \sup_{t\leq\mathcal{T}} \left\|\boldsymbol{\Lambda}_{11}^{\frac{1}{2}}\boldsymbol{T}_t^{\frac{-1}{2}}\Big(\sum_{j\leq t}\mathsf{X}_j\Big)\boldsymbol{T}_t^{\frac{-1}{2}}\boldsymbol{\Lambda}_{11}^{\frac{1}{2}}\right\|_2 \geq \frac{\kappa_d r_u^{-\alpha}}{10} \text{ and } \mathcal{G}_{\text{init}} \right]$$

By Propositions 14 and 16 and Corollary 4, $\mathcal{G}_{\text{init}}$ implies the events

$$\mathcal{E}_{\text{lt},1} \equiv \left\{ \max_{t\leq\mathcal{T}}\|\mathsf{Quad}_t^{(1)}\|_2 \leq \frac{O_{\alpha,r_s}(r_u^{-4\alpha})}{\log^{12} d} \text{ and } \max_{t\leq\mathcal{T}} R_t^{(1)} \leq \frac{O_{r_s}(r_u^{-4\alpha-1})}{\sqrt{d}\log^{11.5} d} \right\}$$

$$\mathcal{E}_{\text{lt},2} \equiv \left\{ \max_{t\leq\mathcal{T}}\|\mathsf{Quad}_t^{(2)}\|_2 \leq \frac{O_{\alpha,r_s}(r_u^{-4\alpha-(\alpha\wedge 1)})}{\log^{12} d \log^{-2} r_u} \text{ and } \max_{t\leq\mathcal{T}} R_t^{(2)} \leq \frac{O_{r_s}(r_u^{-4\alpha-1}r_u^{-(\alpha\wedge 1)})}{\sqrt{d}\log^{11.5} d \log^{-2} r_u} \right\}.$$

Therefore, by using Proposition 19

$$\mathbb{P}_0\Big[\sup_{t\leq\mathcal{T}}\Big\|\boldsymbol{T}_t^{\frac{-1}{2}}\big(\sum_{j\leq t}\mathsf{X}_j\big)\boldsymbol{T}_t^{\frac{-1}{2}}\Big\|_2\geq\frac{\mathsf{K}_d r_u^{\frac{-\alpha}{2}}}{10}\ \text{ and }\ \mathcal{G}_{\mathrm{init}}\Big]$$

$$\leq\mathbb{P}_0\Big[\sup_{n\leq\mathcal{T}}\Big\|\boldsymbol{T}_t^{\frac{-1}{2}}\big(\sum_{j\leq t}\mathsf{X}_j\big)\boldsymbol{T}_t^{\frac{-1}{2}}\Big\|_2\geq\frac{\mathsf{K}_d r_u^{\frac{-\alpha}{2}}}{10}\ \text{ and }\ \mathcal{E}_{\mathrm{lt},1}\Big]$$

$$\leq 2d^4\exp\big(-\log^2 d\big).$$

Similarly,

$$\mathbb{P}_0\Big[\sup_{t\leq\mathcal{T}}\Big\|\boldsymbol{\Lambda}_{11}^{\frac{1}{2}}\boldsymbol{T}_t^{\frac{-1}{2}}\big(\sum_{j\leq t}\mathsf{X}_j\big)\boldsymbol{T}_t^{\frac{-1}{2}}\boldsymbol{\Lambda}_{11}^{\frac{1}{2}}\Big\|_2\geq\frac{\mathsf{K}_d r_u^{-\alpha}}{10}\ \text{ and }\ \mathcal{G}_{\mathrm{init}}\Big]$$

$$\leq\mathbb{P}_0\Big[\sup_{t\leq\mathcal{T}}\Big\|\boldsymbol{\Lambda}_{11}^{\frac{1}{2}}\boldsymbol{T}_t^{\frac{-1}{2}}\big(\sum_{j\leq t}\mathsf{X}_j\big)\boldsymbol{T}_t^{\frac{-1}{2}}\boldsymbol{\Lambda}_{11}^{\frac{1}{2}}\Big\|_2\geq\frac{\mathsf{K}_d r_u^{-\alpha}}{10}\ \text{ and }\ \mathcal{E}_{\mathrm{lt},2}\Big]$$

$$\leq 2d^4\exp\big(-\log^2 d\big).$$

$\square$

**Corollary 5.** *Consider* $\mathsf{rk}_\star=\{r_\star=\lfloor r_s\big(1-\log^{-1/2} d\big)\wedge r\rfloor, r_{u_\star}=r_s\}$ *and the parameters in Proposition* 20. *We have*

$$\mathbb{P}_0\Big[\mathcal{T}_{bad}\geq\tfrac{1}{2\eta}\mathsf{rk}_\star\log\Big(\tfrac{d\log^{1.5} d}{r_s}\Big)\ \text{and}\ \mathcal{G}_{init}\Big]\geq 1-20d^4\exp(-\log^2 d).$$

*Proof.* By using the first items in Proposition 15 and 16 ,and Lemma 6, $\mathcal{G}_{\mathrm{init}}$ implies that

$$\mathcal{T}_{\mathrm{bad}}\geq\mathcal{T}_{\mathrm{noise}}\wedge\tfrac{1}{2\eta}\mathsf{rk}_\star\log\Big(\tfrac{d\log^{1.5} d}{r_s}\Big).$$

On the other hand, within the (negation) of the events given in Proposition 20, we have

$$\mathcal{T}_{\mathrm{noise}}>\mathcal{T}_{\mathrm{bad}}\wedge\tfrac{1}{2\eta}\mathsf{rk}_\star\log\Big(\tfrac{d\log^{1.5} d}{r_s}\Big).$$

Therefore, the statement follows. $\square$

### F.7 Stability near minima

In this section, we will establish that given (SGD) is near global minimum it will stay near global minimum. For the statement, we (re)introduce the block matrix notation: $\mathsf{rk}_\star=\{r_\star=\lfloor r_s\big(1-\log^{-1/8} d\big)\wedge r\rfloor, r_{u_\star}=r_s\}$, we have

$$\boldsymbol{G}_t=\begin{bmatrix}\boldsymbol{G}_{t,11} & \boldsymbol{G}_{t,12}\\ \boldsymbol{G}_{t,12}^\top & \boldsymbol{G}_{t,22}\end{bmatrix}\quad \boldsymbol{\nu}_t=\begin{bmatrix}\boldsymbol{\nu}_{t,11} & \boldsymbol{\nu}_{t,12}\\ \boldsymbol{\nu}_{t,12}^\top & \boldsymbol{\nu}_{t,22}\end{bmatrix}\quad \boldsymbol{\Lambda}=\begin{bmatrix}\boldsymbol{\Lambda}_{11} & 0\\ 0 & \boldsymbol{\Lambda}_{22}\end{bmatrix},\ \boldsymbol{\Lambda}_{\ell_j}=\begin{bmatrix}\boldsymbol{\Lambda}_{\ell_j,11} & 0\\ 0 & \boldsymbol{\Lambda}_{\ell_j,22}\end{bmatrix},$$

where $\boldsymbol{G}_{t,11},\boldsymbol{\nu}_{t,11},\boldsymbol{\Lambda}_{11},\boldsymbol{\Lambda}_{\ell_j,11}\in\mathbb{R}^{\mathsf{rk}_\star\times\mathsf{rk}_\star}$ and $\boldsymbol{\Lambda}_{\ell_j}$ is introduced (F.2). We define the following iterations:

- Given $\underline{\boldsymbol{G}}_0=\boldsymbol{I}_{\mathsf{rk}_\star}-\frac{1}{\log d}$ diagonal and $\underline{\boldsymbol{V}}_t=2\boldsymbol{\Lambda}_{\ell_2,11}^{\frac{1}{2}}\underline{\boldsymbol{G}}_t\boldsymbol{\Lambda}_{\ell_2,11}^{\frac{1}{2}}-\boldsymbol{\Lambda}_{\ell_1,11}$, we define

$$\underline{\boldsymbol{V}}_{t+1}=\underline{\boldsymbol{V}}_t\left(\boldsymbol{I}_{\mathsf{rk}_\star}+\frac{\eta}{1-1.1\eta}\underline{\boldsymbol{V}}_t\right)^{-1}$$

$$+\frac{\eta}{1-1.1\eta}\left(\boldsymbol{\Lambda}_{\ell_1,11}^2-\frac{8.1}{\mathsf{rk}_\star^\alpha\log d}\boldsymbol{\Lambda}_{\ell_2,11}-\frac{O(1)}{\log^2 d}\boldsymbol{\Lambda}_{\ell_2,11}^2\right).$$

- For $\underline{\boldsymbol{\nu}}_0=0, \underline{\boldsymbol{\nu}}_{t+1}=\underline{\boldsymbol{\nu}}_t+\frac{\eta/2}{\sqrt{r_s}}\boldsymbol{\nu}_{t+1,11}.$

- We define a sequence of events $\{\mathcal{E}_t\}_{t \geq 0}$

$$\mathcal{E}_t := \left\{ \frac{-\mathsf{rk}_\star^{-\frac{\alpha}{2}}}{\log^2 d} \boldsymbol{I}_{\mathsf{rk}_\star} \preceq \underline{\boldsymbol{\nu}}_t \preceq \frac{\mathsf{rk}_\star^{-\frac{\alpha}{2}}}{\log^2 d} \boldsymbol{I}_{\mathsf{rk}_\star} \right\} \cap \left\{ \frac{-\mathsf{rk}_\star^{-\alpha}}{\log^4 d} \boldsymbol{I}_{\mathsf{rk}_\star} \preceq \boldsymbol{\Lambda}_{11}^{\frac{1}{2}} \underline{\boldsymbol{\nu}}_t \boldsymbol{\Lambda}_{11}^{\frac{1}{2}} \preceq \frac{\mathsf{rk}_\star^{-\alpha}}{\log^4 d} \boldsymbol{I}_{\mathsf{rk}_\star} \right\},$$

We define the stopping times

$$\mathcal{T}_{\text{noise}}(\omega) := \inf\left\{ t \geq 0 \mid \omega \notin \mathcal{E}_t \right\} \wedge d^3, \quad \mathcal{T}_{\text{bounded}} := \inf\left\{ t \geq 0 \mid \boldsymbol{G}_t \npreceq \boldsymbol{I}_{\mathsf{rk}_\star} - \frac{2}{\log d} \boldsymbol{I}_{\mathsf{rk}_\star} \right\}.$$

and $\mathcal{T}_{\text{stable}} := \mathcal{T}_{\text{noise}} \wedge \mathcal{T}_{\text{bounded}}$.

We have the following statement:

**Proposition 21.** *Consider the parameters in Proposition 20.* (SGD) *guarantees that if* $\boldsymbol{G}_{0,11} \succeq \boldsymbol{I}_{\mathsf{rk}_\star} - \frac{1}{\log d}$, *we have* $\boldsymbol{G}_{t,11} \succeq \boldsymbol{I}_{\mathsf{rk}_\star} - \frac{2}{\log d}$ *for* $t \leq \frac{\mathsf{rk}_\star^\alpha \log^2 d}{\eta}$ *with probability* $1 - d^4 \exp(-\log^2 d)$.

*Proof.* We define $\underline{\boldsymbol{\zeta}}_t := 2\boldsymbol{\Lambda}_{\ell_2,11}^{\frac{1}{2}} \underline{\boldsymbol{\nu}}_t \boldsymbol{\Lambda}_{\ell_2,11}^{\frac{1}{2}}$. We make the following observations observations:

- Since $\boldsymbol{G}_{t,11}^2 + \boldsymbol{G}_{t,12}\boldsymbol{G}_{t,12}^\top \preceq \boldsymbol{I}_{\mathsf{rk}_\star}$ for $t \leq \mathcal{T}_{\text{bounded}}$, we have

$$\boldsymbol{G}_{t,12}\boldsymbol{G}_{t,12}^\top \preceq \frac{4}{\log d} \boldsymbol{I}_{\mathsf{rk}_\star}$$

Therefore, by using (F.8), we have for $t \leq \mathcal{T}_{\text{bounded}}$

$$\boldsymbol{G}_{t+1,11} \succeq \boldsymbol{G}_{t,11} + \eta\left( \boldsymbol{\Lambda}_{\ell_1,11}\boldsymbol{G}_{t,11} + \boldsymbol{G}_{t,11}\boldsymbol{\Lambda}_{\ell_1,11} - 2\boldsymbol{G}_{t,11}\boldsymbol{\Lambda}_{\ell_2,11}\boldsymbol{G}_{t,11} \right) - \frac{4\eta}{\mathsf{rk}_\star^\alpha \log d} \boldsymbol{I}_{\mathsf{rk}_\star}$$

$$- C\eta^2 \|\boldsymbol{\Lambda}\|_{\mathrm{F}}^2 r_s \boldsymbol{I}_{\mathsf{rk}_\star} + \frac{\eta/2}{\sqrt{r_s}} \boldsymbol{\nu}_{t+1,11}$$

Then, if we define $\boldsymbol{V}_t^- := 2\boldsymbol{\Lambda}_{\ell_2,11}^{\frac{1}{2}} \boldsymbol{G}_{t,11} \boldsymbol{\Lambda}_{\ell_2,11}^{\frac{1}{2}} - \boldsymbol{\Lambda}_{\ell_1,11}$, we have

$$\boldsymbol{V}_{t+1}^- \succeq \boldsymbol{V}_t^- - \eta(\boldsymbol{V}_{t+1}^-)^2 + \eta\boldsymbol{\Lambda}_{\ell_1,11}^2 - \left( \frac{8\eta}{\mathsf{rk}_\star^\alpha \log d} + C\eta^2 \|\boldsymbol{\Lambda}\|_{\mathrm{F}}^2 r_s \right)\boldsymbol{\Lambda}_{\ell_2,11}$$

$$+ \frac{\eta}{\sqrt{r_s}} \boldsymbol{\Lambda}_{\ell_2,11}^{\frac{1}{2}} \boldsymbol{\nu}_{t+1,11} \boldsymbol{\Lambda}_{\ell_2,11}^{\frac{1}{2}}$$

$$\succeq \boldsymbol{V}_t^- \left( \boldsymbol{I}_r + \frac{\eta}{1 - 1.1\eta} \boldsymbol{V}_t^- \right)^{-1} + \eta\left( \boldsymbol{\Lambda}_{\ell_1,11}^2 - \frac{8.1}{\mathsf{rk}_\star^\alpha \log d} \boldsymbol{\Lambda}_{\ell_2,11} \right) + \boldsymbol{\Lambda}_{\ell_2,11}^{\frac{1}{2}} \boldsymbol{\nu}_{t+1,11} \boldsymbol{\Lambda}_{\ell_2,11}^{\frac{1}{2}}$$

- To derive an upper-bound for $\underline{\boldsymbol{G}}_t$, assuming $\underline{\boldsymbol{G}}_t \preceq 1.1\boldsymbol{I}_{\mathsf{rk}_\star}$, we have

$$\underline{\boldsymbol{V}}_{t+1} = \underline{\boldsymbol{V}}_t - \frac{\eta}{1 - 1.1\eta} \boldsymbol{V}_t^2 + \frac{\eta^2}{(1 - 1.1\eta)^2} \boldsymbol{V}_t^3 \left( \boldsymbol{I}_{\mathsf{rk}_\star} + \frac{\eta}{1 - 1.1\eta} \boldsymbol{V}_t \right)^{-1}$$

$$+ \frac{\eta}{1 - 1.1\eta} \left( \boldsymbol{\Lambda}_{\ell_1,11}^2 - \frac{8.1}{\mathsf{rk}_\star^\alpha \log d} \boldsymbol{\Lambda}_{\ell_2,11} - \frac{O(1)}{\log^2 d} \boldsymbol{\Lambda}_{\ell_2,11}^2 \right)$$

$$\preceq \underline{\boldsymbol{V}}_t - \frac{\eta}{1 - 1.1\eta} \boldsymbol{V}_t^2 + \frac{\eta}{1 - 1.1\eta} \boldsymbol{\Lambda}_{\ell_1,11}^2.$$

Then, by Proposition 34, we have $\underline{\boldsymbol{G}}_{t+1} \preceq 1.1\boldsymbol{I}_{\mathsf{rk}_\star}$. Since the bound holds for $t = 0$, it holds for all $t \in \mathbb{N}$.

- To derive a lower-bound, we first observe that by monotonicity $\underline{\boldsymbol{V}}_0 \succ 0$. Therefore, assuming $\underline{\boldsymbol{V}}_t \succ \underline{\boldsymbol{V}}_0$

$$\underline{\boldsymbol{V}}_{t+1} \succeq \underline{\boldsymbol{V}}_t - \frac{\eta/2}{1 - 1.1\eta} (2\boldsymbol{\Lambda}_{\ell_2,11}^{\frac{1}{2}} \underline{\boldsymbol{G}}_t \boldsymbol{\Lambda}_{\ell_2,11}^{\frac{1}{2}} - \boldsymbol{\Lambda}_{\ell_1,11})^2$$

$$+ \frac{\eta/2}{1 - 1.1\eta} \left( \boldsymbol{\Lambda}_{\ell_1,11}^2 - \frac{8.1}{\mathsf{rk}_\star^\alpha \log d} \boldsymbol{\Lambda}_{\ell_2,11} - \frac{O(1)}{\log^2 d} \boldsymbol{\Lambda}_{\ell_2,11}^2 \right)$$

$$\succeq \underline{\boldsymbol{V}}_t - \frac{\eta/2}{1 - 1.1\eta}\left(2\boldsymbol{\Lambda}_{\ell_2,11}^{\frac{1}{2}}\underline{\boldsymbol{G}}_t\boldsymbol{\Lambda}_{\ell_2,11}^{\frac{1}{2}} - \sqrt{\left(\boldsymbol{\Lambda}_{\ell_1,11}^2 - \frac{8.1}{\mathsf{rk}_\star^\alpha \log d}\boldsymbol{\Lambda}_{\ell_2,11} - \frac{O(1)}{\log^2 d}\boldsymbol{\Lambda}_{\ell_2,11}^2\right)}\right)^2$$

$$+ \frac{\eta/2}{1 - 1.1\eta}\left(\boldsymbol{\Lambda}_{\ell_1,11}^2 - \frac{8.1}{\mathsf{rk}_\star^\alpha \log d}\boldsymbol{\Lambda}_{\ell_2,11} - \frac{O(1)}{\log^2 d}\boldsymbol{\Lambda}_{\ell_2,11}^2\right).$$

Then, by Proposition 34, we have $\underline{\boldsymbol{V}}_t \succeq \underline{\boldsymbol{V}}_0$.

We start our proof by showing that $\boldsymbol{V}_t^- \succeq \underline{\boldsymbol{V}}_t + \underline{\boldsymbol{\zeta}}_t$ for $t \le \mathcal{T}_{\text{stable}}$. Assuming the statement holds for $t \in \mathbb{N}$, we have

$$\boldsymbol{V}_{t+1}^- \succeq (\underline{\boldsymbol{V}}_t + \underline{\boldsymbol{\zeta}}_t)\left(\boldsymbol{I}_r + \frac{\eta}{1 - 1.1\eta}(\underline{\boldsymbol{V}}_t + \underline{\boldsymbol{\zeta}}_t)\right)^{-1} + \eta\left(\boldsymbol{\Lambda}_{\ell_1,11}^2 - \frac{8.1}{\mathsf{rk}_\star^\alpha \log d}\boldsymbol{\Lambda}_{\ell_2,11}\right)$$

$$+ \boldsymbol{\Lambda}_{\ell_2,11}^{\frac{1}{2}}\boldsymbol{\nu}_{t+1,11}\boldsymbol{\Lambda}_{\ell_2,11}^{\frac{1}{2}}$$

$$= \underline{\boldsymbol{V}}_t\left(\boldsymbol{I}_r + \frac{\eta}{1 - 1.1\eta}\underline{\boldsymbol{V}}_t\right)^{-1} + \eta\left(\boldsymbol{\Lambda}_{\ell_1,11}^2 - \frac{8.1}{\mathsf{rk}_\star^\alpha \log d}\boldsymbol{\Lambda}_{\ell_2,11}\right)$$

$$+ \left(\boldsymbol{I}_r + \frac{\eta}{1 - 1.1\eta}\underline{\boldsymbol{V}}_t\right)^{-1}\underline{\boldsymbol{\zeta}}_t\left(\boldsymbol{I}_r + \frac{\eta}{1 - 1.1\eta}(\underline{\boldsymbol{V}}_t + \underline{\boldsymbol{\zeta}}_t)\right)^{-1} + \boldsymbol{\Lambda}_{\ell_2,11}^{\frac{1}{2}}\boldsymbol{\nu}_{t+1,11}\boldsymbol{\Lambda}_{\ell_2,11}^{\frac{1}{2}}$$

We have for $t \le \mathcal{T}_{\text{stable}}$

$$\left(\boldsymbol{I}_r + \frac{\eta}{1 - 1.1\eta}\underline{\boldsymbol{V}}_t\right)^{-1}\underline{\boldsymbol{\zeta}}_t\left(\boldsymbol{I}_r + \frac{\eta}{1 - 1.1\eta}(\underline{\boldsymbol{V}}_t + \underline{\boldsymbol{\zeta}}_t)\right)^{-1}$$

$$= \underline{\boldsymbol{\zeta}}_t - \frac{\eta}{1 - 1.1\eta}\underline{\boldsymbol{V}}_t\underline{\boldsymbol{\zeta}}_t - \frac{\eta}{1 - 1.1\eta}\underline{\boldsymbol{\zeta}}_t\underline{\boldsymbol{V}}_t - \frac{\eta}{1 - 1.1\eta}\underline{\boldsymbol{\zeta}}_t^2$$

$$- \frac{\eta^2}{(1 - 1.1\eta)^2}\underline{\boldsymbol{V}}_t\left(\boldsymbol{I}_r + \frac{\eta}{1 - 1.1\eta}\underline{\boldsymbol{V}}_t\right)^{-1}\underline{\boldsymbol{\zeta}}_t\left(\boldsymbol{I}_r + \frac{\eta}{1 - 1.1\eta}(\underline{\boldsymbol{V}}_t + \underline{\boldsymbol{\zeta}}_t)\right)^{-1}(\underline{\boldsymbol{V}}_t + \underline{\boldsymbol{\zeta}}_t)$$

$$\succeq \underline{\boldsymbol{\zeta}}_t - \frac{\eta}{1 - 1.1\eta}\frac{1}{\log^2 d}\underline{\boldsymbol{V}}_t^2 - \frac{\eta}{1 - 1.1\eta}(1 + \log^2 d)\underline{\boldsymbol{\zeta}}_t^2 - \frac{\eta^2}{(1 - 1.1\eta)^2}\frac{O(\mathsf{rk}_\star^{-\alpha})}{\log^4 d}\boldsymbol{I}_{\mathsf{rk}_\star}$$

$$\succeq \underline{\boldsymbol{\zeta}}_t - \frac{\eta}{1 - 1.1\eta}\frac{O(1)}{\log^2 d}\boldsymbol{\Lambda}_{\ell_2,11}^2.$$

Since $\boldsymbol{V}_0^- = \underline{\boldsymbol{V}}_0 + \underline{\boldsymbol{\zeta}}_0$, the claim follows. Then, by the third item above, we have for $t \le \mathcal{T}_{\text{stable}}$

$$\boldsymbol{G}_t \succeq \underline{\boldsymbol{G}}_0 + \boldsymbol{\nu}_t \succeq \boldsymbol{I}_{\mathsf{rk}_\star} - \frac{1}{\log d}\boldsymbol{I}_{\mathsf{rk}_\star} - \frac{\mathsf{rk}_\star^{\frac{-\alpha}{2}}}{\log^2 d}\boldsymbol{I}_{\mathsf{rk}_\star} \succeq \boldsymbol{I}_{\mathsf{rk}_\star} - \frac{2}{\log d}\boldsymbol{I}_{\mathsf{rk}_\star} \Rightarrow \mathcal{T}_{\text{noise}} \le \mathcal{T}_{\text{bounded}}.$$

In the following, we will bound $\mathcal{T}_{\text{noise}}$. We have

$$\mathbb{E}_t\left[\boldsymbol{\nu}_{t+1}^2\right] \overset{(a)}{\preceq} \boldsymbol{\nu}_t^2 + O(\eta^2)\boldsymbol{I}_{\mathsf{rk}_\star} \Rightarrow \mathbb{E}\left[\boldsymbol{\nu}_t^2\right] \preceq O(\eta^2 t)\boldsymbol{I}_{\mathsf{rk}_\star}$$

By clipping strategy we used with $L = \log^2 d$ in (F.38), and defining $\Gamma_1 := \boldsymbol{I}_{\mathsf{rk}_\star}$, $\Gamma_2 := \boldsymbol{\Lambda}_{11}^{\frac{1}{2}}$, and

$$\mathsf{Quad}_{k,t}^{(\ell)}(\mathsf{X}) := \sum_{j=1}^k \mathbb{E}_{j-1}\left[\left(\Gamma_\ell \boldsymbol{T}_t^{\frac{-1}{2}}\mathsf{X}_j\boldsymbol{T}_t^{\frac{-1}{2}}\Gamma_\ell\right)^2\right], \ \ell \in \{1, 2\},$$

we can show that the following events hold: For any $T \in \mathbb{N}$,

$$\widehat{\mathcal{E}}_{\text{ht},1} \equiv \left\{\max_{t \le T}\|\mathsf{Quad}_{t,t}^{(1)}\|_2 \le O(\eta^2 T) \quad \text{and} \quad \max_{t \le T}R_{t,t}^{(1)} \le O(\eta \mathsf{rk}_\star^{\frac{1}{2}}\log^2 d)\right\}$$

$$\widehat{\mathcal{E}}_{\text{ht},2} \equiv \left\{\max_{t \le T}\|\mathsf{Quad}_{t,t}^{(2)}\|_2 \le O_\alpha(\eta^2 T\mathsf{rk}_\star^{\mathsf{p}_2-1}) \quad \text{and} \quad \max_{t \le T}R_{t,t}^{(2)} \le O_\alpha(\eta \mathsf{rk}_\star^{\mathsf{p}_2-\frac{1}{2}}\log^2 d)\right\}.$$

where $\mathsf{p}_2$ is defined in Corollary 4. By using Proposition 19, we can show that with probability $d^4\exp(-\log^2 d)$, $\mathcal{T}_{\text{noise}} \ge \frac{\mathsf{rk}_\star^\alpha \log^2 d}{\eta}$. $\qquad\square$

# G   Auxiliary Statements

## G.1   Matrix bounds

**Proposition 22.** *For $A, B \in \mathbb{R}^{d \times r}$, we have*

$$-A^\top A - B^\top B \preceq A^\top B + B^\top A \preceq A^\top A + B^\top B.$$

*If $r = d$, then $(A + A^\top)^2 \preceq 2A^\top A + 2AA^\top$. Moreover, if $A_1, \cdots, A_k$ are symmetric matrices,*

$$\left( \sum_{i=1}^{k} A_i \right)^2 \preceq k \sum_{i=1}^{k} A_i^2.$$

*Proof.* We have

$$(A - B)^\top (A - B) \succeq 0 \Rightarrow A^\top A + B^\top B \succeq A^\top B + B^\top A.$$

By using $A \leftarrow -A$, we obtain the left inequality too. For the second inequality, we have

$$(A + A^\top)^2 = A^\top A + AA^\top + AA + A^\top A^\top$$
$$(A - A^\top)^\top (A - A^\top) = A^\top A + AA^\top - AA - A^\top A^\top$$

Therefore, $(A + A^\top)^2 \preceq 2 \left( A^\top A + AA^\top \right)$. For the last statement,

$$\left( \sum_{i=1}^{k} A_i \right)^2 = \sum_{i=1}^{k} A_i^2 + \sum_{i=1}^{k} \sum_{j=i+1}^{k} A_i A_j + \sum_{i=1}^{k} \sum_{j=i+1}^{k} A_j A_i \preceq k \sum_{i=1}^{k} A_i^2,$$

where we use the first statement in the last inequality. $\qquad\square$

**Proposition 23.** *Consider a symmetric square matrix with block partition*

$$M = \begin{bmatrix} A & B \\ B^\top & C \end{bmatrix}.$$

*If $A$ is invertible, then $M \succ 0$ if and only if $A \succ 0$ and $C - B^\top A^{-1} B \succ 0$.*

*Proof.* If $A$ is invertible, we have

$$\begin{bmatrix} A & B \\ B^\top & C \end{bmatrix} = \begin{bmatrix} I & 0 \\ B^\top A^{-1} & I \end{bmatrix} \begin{bmatrix} A & 0 \\ 0 & C - B^\top A^{-1} B \end{bmatrix} \begin{bmatrix} I & A^{-1} B \\ 0 & I \end{bmatrix}.$$

Note that

$$\begin{bmatrix} I & 0 \\ B^\top A^{-1} & I \end{bmatrix}^{-1} = \begin{bmatrix} I & 0 \\ -B^\top A^{-1} & I \end{bmatrix}.$$

Therefore, the statement follows. $\qquad\square$

**Proposition 24.** *Let $r_u < r$ and $Z \in \mathbb{R}^{r \times r_s}$ such that*

$$Z = \begin{bmatrix} Z_1 \\ Z_2 \end{bmatrix}, \quad \text{where } Z_1 \in \mathbb{R}^{r_u \times r_s}, Z_2 \in \mathbb{R}^{r - r_u \times r_s}.$$

*For ant $0 \leq \varepsilon < 1$*

$$ZZ^\top \succeq \varepsilon \begin{bmatrix} Z_1 Z_1^\top & 0 \\ 0 & 0 \end{bmatrix} + (1 - \varepsilon) \begin{bmatrix} Z_1 Z_1^\top - Z_1 Z_2^\top (Z_2 Z_2^\top)^+ Z_2 Z_1^\top & 0 \\ 0 & 0 \end{bmatrix}$$
$$- \frac{\varepsilon}{1 - \varepsilon} \begin{bmatrix} 0 & 0 \\ 0 & Z_2 Z_2^\top \end{bmatrix},$$

*where $A \to A^+$ denotes the pseudo inverse operator.*

*Proof.* We will denote $\boldsymbol{x} \in \mathbb{R}^r$ as

$$\boldsymbol{x} = \begin{bmatrix} \boldsymbol{x}_1 \\ \boldsymbol{x}_2 \end{bmatrix} \quad \text{where} \ \ \boldsymbol{x}_1 \in \mathbb{R}^{r_u}, \boldsymbol{x}_2 \in \mathbb{R}^{r-r_u}.$$

We have

$$\boldsymbol{x}^\top \boldsymbol{Z}\boldsymbol{Z}^\top \boldsymbol{x} = \left( \boldsymbol{x}_1^\top \boldsymbol{Z}_1 \boldsymbol{Z}_1^\top \boldsymbol{x}_1 + 2\boldsymbol{x}_1^\top \boldsymbol{Z}_1 \boldsymbol{Z}_2^\top \boldsymbol{x}_2 + \frac{1}{1-\varepsilon} \boldsymbol{x}_2^\top \boldsymbol{Z}_2 \boldsymbol{Z}_2^\top \boldsymbol{x}_2 \right) - \frac{\varepsilon}{1-\varepsilon} \boldsymbol{x}_2^\top \boldsymbol{Z}_2 \boldsymbol{Z}_2^\top \boldsymbol{x}_2$$

$$\overset{(a)}{\geq} \left( \boldsymbol{x}_1^\top \boldsymbol{Z}_1 \boldsymbol{Z}_1^\top \boldsymbol{x}_1 - (1-\varepsilon)\boldsymbol{x}_1^\top \boldsymbol{Z}_1 \boldsymbol{Z}_2^\top (\boldsymbol{Z}_2 \boldsymbol{Z}_2^\top)^+ \boldsymbol{Z}_2 \boldsymbol{Z}_1^\top \boldsymbol{x}_1 \right) - \frac{\varepsilon}{1-\varepsilon} \boldsymbol{x}_2^\top \boldsymbol{Z}_2 \boldsymbol{Z}_2^\top \boldsymbol{x}_2,$$

where we minimized the first term in the first line over $\boldsymbol{x}_2$ in (a). Since (a) holds for all $\boldsymbol{x}$, the statement follows, $\qquad\square$

**Proposition 25.** *Let $\boldsymbol{A} \in \mathbb{R}^{r \times r}$ be a symmetric matrix. For $\boldsymbol{S} \succ -\boldsymbol{A}$, $\boldsymbol{S} \to -(\boldsymbol{S} + \boldsymbol{A})^{-1}$ is monotone.*

*Proof.* Let $\boldsymbol{S}_1 \succ \boldsymbol{S}_2 \succ -\boldsymbol{A}$. We have

$$-(\boldsymbol{S}_1 + \boldsymbol{A})^{-1} + (\boldsymbol{S}_2 + \boldsymbol{A})^{-1} = (\boldsymbol{S}_2 + \boldsymbol{A})^{-1}((\boldsymbol{S}_1 - \boldsymbol{S}_2)^{-1} + (\boldsymbol{S}_2 + \boldsymbol{A})^{-1})^{-1}(\boldsymbol{S}_2 + \boldsymbol{A})^{-1} \succ 0. \tag{G.1}$$

For $\boldsymbol{S}_1 \succeq \boldsymbol{S}_2$, we can use $\boldsymbol{S}_1 + \varepsilon \boldsymbol{I}_r$ in (G.1) and take $\varepsilon \downarrow 0$ $\qquad\square$

### G.1.1 Additional bounds for continuous-time analysis

**Proposition 26.** *For a symmetric positive definite $\boldsymbol{D}_1$, $\boldsymbol{D}_2$, and $C > 0$, we have*

$$\boldsymbol{D}_1 \Big( \boldsymbol{D}_1 + \boldsymbol{Z}_1 \big( C\boldsymbol{I}_{r_s} + \boldsymbol{Z}_2^\top \boldsymbol{D}_2^{-1} \boldsymbol{Z}_2 \big)^{-1} \boldsymbol{Z}_1^\top \Big)^{-1} \boldsymbol{Z}_1 \boldsymbol{Z}_2^\top \big( \boldsymbol{Z}_2 \boldsymbol{Z}_2^\top + C\boldsymbol{D}_2 \big)^{-1} \boldsymbol{D}_2$$

$$= \boldsymbol{Z}_1 \big( C\boldsymbol{I}_{r_s} + \boldsymbol{Z}_2^\top \boldsymbol{D}_2^{-1} \boldsymbol{Z}_2 + \boldsymbol{Z}_1^\top \boldsymbol{D}_1^{-1} \boldsymbol{Z}_1 \big)^{-1} \boldsymbol{Z}_2^\top.$$

*Proof.* We have

$$\big( \boldsymbol{D}_1 + \boldsymbol{Z}_1 \big( C\boldsymbol{I}_{r_s} + \boldsymbol{Z}_2^\top \boldsymbol{D}_2^{-1} \boldsymbol{Z}_2 \big)^{-1} \boldsymbol{Z}_1^\top \big)^{-1}$$

$$= \boldsymbol{D}_1^{-1} - \boldsymbol{D}_1^{-1} \boldsymbol{Z}_1 \big( C\boldsymbol{I}_{r_s} + \boldsymbol{Z}_2^\top \boldsymbol{D}_2^{-1} \boldsymbol{Z}_2 + \boldsymbol{Z}_1^\top \boldsymbol{D}_1^{-1} \boldsymbol{Z}_1 \big)^{-1} \boldsymbol{Z}_1^\top \boldsymbol{D}_1^{-1}.$$

Therefore,

$$\boldsymbol{D}_1 \big( \boldsymbol{D}_1 + \boldsymbol{Z}_1 \big( \boldsymbol{I}_{r_s} + \boldsymbol{Z}_2^\top \boldsymbol{D}_2^{-1} \boldsymbol{Z}_2 \big)^{-1} \boldsymbol{Z}_1^\top \big)^{-1} \boldsymbol{Z}_1$$

$$= \boldsymbol{Z}_1 \Big( \boldsymbol{I}_{r_s} - \big( C\boldsymbol{I}_{r_s} + \boldsymbol{Z}_2^\top \boldsymbol{D}_2^{-1} \boldsymbol{Z}_2 + \boldsymbol{Z}_1^\top \boldsymbol{D}_1^{-1} \boldsymbol{Z}_1 \big)^{-1} \boldsymbol{Z}_1^\top \boldsymbol{D}_1^{-1} \boldsymbol{Z}_1 \Big)$$

$$= \boldsymbol{Z}_1 \big( C\boldsymbol{I}_{r_s} + \boldsymbol{Z}_2^\top \boldsymbol{D}_2^{-1} \boldsymbol{Z}_2 + \boldsymbol{Z}_1^\top \boldsymbol{D}_1^{-1} \boldsymbol{Z}_1 \big)^{-1} \big( C\boldsymbol{I}_{r_s} + \boldsymbol{Z}_2^\top \boldsymbol{D}_2^{-1} \boldsymbol{Z}_2 \big)$$

Then,

$$\boldsymbol{Z}_1 \big( C\boldsymbol{I}_{r_s} + \boldsymbol{Z}_2^\top \boldsymbol{D}_2^{-1} \boldsymbol{Z}_2 + \boldsymbol{Z}_1^\top \boldsymbol{D}_1^{-1} \boldsymbol{Z}_1 \big)^{-1} \big( C\boldsymbol{I}_{r_s} + \boldsymbol{Z}_2^\top \boldsymbol{D}_2^{-1} \boldsymbol{Z}_2 \big) \boldsymbol{Z}_2^\top \big( \boldsymbol{Z}_2 \boldsymbol{Z}_2^\top + C\boldsymbol{D}_2 \big)^{-1} \boldsymbol{D}_2$$

$$= \boldsymbol{Z}_1 \big( C\boldsymbol{I}_{r_s} + \boldsymbol{Z}_2^\top \boldsymbol{D}_2^{-1} \boldsymbol{Z}_2 + \boldsymbol{Z}_1^\top \boldsymbol{D}_1^{-1} \boldsymbol{Z}_1 \big)^{-1} \boldsymbol{Z}_2^\top.$$

$\qquad\square$

**Proposition 27.** *For some diagonal positive definite $\boldsymbol{A} := \mathrm{diag}(\{a_j\}_{j=1}^{r_u})$ and $\boldsymbol{B} := \mathrm{diag}(\{b_j\}_{j=1}^{d-r_u})$, we let*

$$\boldsymbol{D}_1 := \frac{\boldsymbol{A} \exp(-t\boldsymbol{A})}{\boldsymbol{I}_{r_u} - \exp(-t\boldsymbol{A})}, \quad \boldsymbol{D}_2 := \frac{\boldsymbol{B} \exp(-t\boldsymbol{B})}{\boldsymbol{I}_{d-r_u} - \exp(-t\boldsymbol{B})},$$

*For some $\boldsymbol{Z}_1 \in \mathbb{R}^{r_u \times r_s}$, $\boldsymbol{Z}_2 \in \mathbb{R}^{(d-r_u) \times r_s}$, and $C > 0$, we define*

$$\boldsymbol{M} := \exp(0.5t\boldsymbol{A}) \boldsymbol{Z}_1 \big( C\boldsymbol{I}_{r_s} + \boldsymbol{Z}_2^\top \boldsymbol{D}_2^{-1} \boldsymbol{Z}_2 + \boldsymbol{Z}_1^\top \boldsymbol{D}_1^{-1} \boldsymbol{Z}_1 \big)^{-1} \boldsymbol{Z}_2^\top \exp(0.5t\boldsymbol{B}).$$

*We have*

$$\|M\|_F^2 \le \tilde{C} \sum_{i=1}^{r_u \wedge r_s} \left( \lambda_{\max}(Z_1 Z_1^\top) \exp(t(a_i + b_i)) \wedge \left( C + \frac{\lambda_{\min}(Z_2^\top Z_2)}{\lambda_{\max}(D_2)} \right) \frac{a_i \exp(t(a_i + b_i))}{\exp(ta_i) - 1} \right)$$

*where*

$$\tilde{C} = \frac{\lambda_{\max}(Z_2^\top Z_2)}{\left( C + \frac{\lambda_{\min}(Z_2^\top Z_2)}{\lambda_{\max}(D_2)} \right)^2}$$

*Proof.* For convenience, we will use

$$\tilde{D}_1 := \frac{A}{I_{r_u} - \exp(-tA)}, \quad \tilde{D}_2 = \frac{B}{I_{d-r_u} - \exp(-tB)},$$
$$\tilde{Z}_1 := \exp(0.5tA)Z_1, \qquad \tilde{Z}_2 := \exp(0.5tB)Z_2.$$

We let

$$M_1 := \tilde{Z}_1 \left( CI_{r_s} + \tilde{Z}_2^\top \tilde{D}_2^{-1} \tilde{Z}_2 + \tilde{Z}_1^\top \tilde{D}_1^{-1} \tilde{Z}_1 \right)^{\frac{-1}{2}},$$

$$M_2 := \left( CI_{r_s} + \tilde{Z}_2^\top \tilde{D}_2^{-1} \tilde{Z}_2 + \tilde{Z}_1^\top \tilde{D}_1^{-1} \tilde{Z}_1 \right)^{\frac{-1}{2}} \tilde{Z}_2^\top$$

We observe that

$$\|M\|_F^2 = \text{Tr}(M_1^\top M_1 M_2 M_2^\top) \le \sum_{i=1}^{r_u \wedge r_s} \lambda_i(M_1 M_1^\top)\lambda_i(M_2^\top M_2)$$

where we used that $\text{rank}(M_1 M_1^\top) \le r_u \wedge r_s$ and Von Neumann's trace inequality in the last part. We have

$$M_2^\top M_2 \preceq \exp(0.5tB)Z_2^\top \left( CI_{r_s} + \frac{1}{\lambda_{\max}(D_2)} Z_2^\top Z_2 \right)^{-1} Z_2 \exp(0.5tB)$$

$$\preceq \frac{\lambda_{\max}(Z_2^\top Z_2)}{C + \frac{\lambda_{\max}(Z_2^\top Z_2)}{\lambda_{\max}(D_2)}} \exp(tB)$$

On the other hand,

$$M_1 M_1^\top \preceq \tilde{Z}_1 \left( \left( C + \frac{\lambda_{\min}(Z_2^\top Z_2)}{\lambda_{\max}(D_2)} \right) I_{r_s} + \tilde{Z}_1^\top \tilde{D}_1^{-1} \tilde{Z}_1 \right)^{-1} \tilde{Z}_1^\top$$

$$= \frac{1}{C + \frac{\lambda_{\min}(Z_2^\top Z_2)}{\lambda_{\max}(D_2)}} \tilde{Z}_1 \left( I_{r_s} + \tilde{Z}_1^\top \left( \left( C + \frac{\lambda_{\min}(Z_2^\top Z_2)}{\lambda_{\max}(D_2)} \right) \tilde{D}_1 \right)^{-1} \tilde{Z}_1 \right)^{-1} \tilde{Z}_1^\top$$

$$= \frac{1}{C + \frac{\lambda_{\min}(Z_2^\top Z_2)}{\lambda_{\max}(D_2)}} \tilde{Z}_1 \tilde{Z}_1^\top \left( \left( C + \frac{\lambda_{\min}(Z_2^\top Z_2)}{\lambda_{\max}(D_2)} \right) \tilde{D}_1 + \tilde{Z}_1 \tilde{Z}_1^\top \right)^{-1} \left( \left( C + \frac{\lambda_{\min}(Z_2^\top Z_2)}{\lambda_{\max}(D_2)} \right) \tilde{D}_1 \right)$$

We have the following at the same time:

- $\tilde{Z}_1 \tilde{Z}_1^\top \left( \left( C + \frac{\lambda_{\min}(Z_2^\top Z_2)}{\lambda_{\max}(D_2)} \right) \tilde{D}_1 + \tilde{Z}_1 \tilde{Z}_1^\top \right)^{-1} \left( C + \frac{\lambda_{\min}(Z_2^\top Z_2)}{\lambda_{\max}(D_2)} \right) \tilde{D}_1 \preceq \left( C + \frac{\lambda_{\min}(Z_2^\top Z_2)}{\lambda_{\max}(D_2)} \right) \tilde{D}_1$

- $\tilde{Z}_1 \tilde{Z}_1^\top \left( \left( C + \frac{\lambda_{\min}(Z_2^\top Z_2)}{\lambda_{\max}(D_2)} \right) \tilde{D}_1 + \tilde{Z}_1 \tilde{Z}_1^\top \right)^{-1} \left( C + \frac{\lambda_{\min}(Z_2^\top Z_2)}{\lambda_{\max}(D_2)} \right) \tilde{D}_1 \preceq \lambda_{\max}(Z_1 Z_1^\top) \exp(tA)$

Therefore, for $i \le r \wedge r_s$, we have

$$\lambda_i(M_1 M_1^\top) \le \frac{1}{C + \frac{\lambda_{\min}(Z_2^\top Z_2)}{\lambda_{\max}(D_2)}} \left( \lambda_{\max}(Z_1 Z_1^\top) \exp(ta_i) \wedge \left( C + \frac{\lambda_{\min}(Z_2^\top Z_2)}{\lambda_{\max}(D_2)} \right) \frac{a_i \exp(ta_i)}{\exp(ta_i) - 1} \right)$$

Therefore,

$$\|M\|_F^2 \le \tilde{C} \sum_{i=1}^{r_u \wedge r_s} \left( \lambda_{\max}(Z_1 Z_1^\top) \exp(t(a_i + b_i)) \wedge \left( C + \frac{\lambda_{\min}(Z_2^\top Z_2)}{\lambda_{\max}(D_2)} \right) \frac{a_i \exp(t(a_i + b_i))}{\exp(ta_i) - 1} \right).$$

$\square$

### G.1.2 Additional bounds for discrete-time analysis

**Proposition 28.** *For some positive definite diagonal matrices $\boldsymbol{D}_0, \boldsymbol{D}_1 \in \mathbb{R}^{r \times r}$ and symmetric matrices $\boldsymbol{G}, \boldsymbol{\nu} \in \mathbb{R}^{r \times r}$, we let*

$$\boldsymbol{V} := 2\boldsymbol{D}_0^{\frac{1}{2}} \boldsymbol{G} \boldsymbol{D}_0^{\frac{1}{2}} - \boldsymbol{D}_1 \ \ \text{and} \ \ \boldsymbol{\zeta} := \boldsymbol{D}_0^{\frac{1}{2}} \boldsymbol{\nu} \boldsymbol{D}_0^{\frac{1}{2}} \ \ \text{and} \ \ \dot{\boldsymbol{V}} := \boldsymbol{V} + \boldsymbol{\zeta},$$

*where*

- *$\|\boldsymbol{G}\|_2 \leq L_G$ and $\|\boldsymbol{\nu}\|_2 \leq L_\nu$ and $\|\boldsymbol{D}_0\|_2 \leq L_0$.*
- *$\|\boldsymbol{D}_0^{-1}\boldsymbol{D}_1\|_2 \leq L_{1/0}$ and $\|\boldsymbol{D}_0\boldsymbol{D}_1^{-1}\|_2 \leq L_{0/1}$.*
- *For notational convenience, let $L_F := 2L_G + L_{1/0}$ and $L_{\dot{F}} := 2(L_G + L_\nu) + L_{1/0}$.*

*For $0 \leq \eta < \frac{1}{L_{\dot{F}} L_0}$, we have that $(\boldsymbol{I}_r + \eta \boldsymbol{V})$ and $(\boldsymbol{I}_r + \eta \dot{\boldsymbol{V}})$ are invertible and the following bounds holds:*

$$- C_1 \boldsymbol{D}_1 \preceq \boldsymbol{V}^2 \left(\boldsymbol{I}_r + \eta \boldsymbol{V}\right)^{-1} \boldsymbol{\zeta} \dot{\boldsymbol{V}} \left(\boldsymbol{I}_r + \eta \dot{\boldsymbol{V}}\right)^{-1} \preceq C_1 \boldsymbol{D}_1, \quad - C_2 \boldsymbol{D}_1 \preceq \boldsymbol{V} \boldsymbol{\zeta} \dot{\boldsymbol{V}}^2 \left(\boldsymbol{I}_r + \eta \dot{\boldsymbol{V}}\right)^{-1} \preceq C_2 \boldsymbol{D}_1 \tag{G.2}$$

$$- C_3 \boldsymbol{D}_1 \preceq \boldsymbol{V}^3 \left(\boldsymbol{I}_r + \eta \boldsymbol{V}\right)^{-1} \boldsymbol{\zeta} \preceq C_3 \boldsymbol{D}_1, \qquad\qquad - C_4 \boldsymbol{D}_1 \preceq \boldsymbol{\zeta} \dot{\boldsymbol{V}}^3 \left(\boldsymbol{I}_r + \eta \dot{\boldsymbol{V}}\right)^{-1} \preceq C_4 \boldsymbol{D}_1, \tag{G.3}$$

*where*

$$C_1 = \frac{L_\nu L_{0/1} L_F^2 L_{\dot{F}} L_0^3}{\left(1 - \eta L_F L_0\right)\left(1 - \eta L_{\dot{F}} L_0\right)}, \quad C_2 = \frac{L_\nu L_{0/1} L_F L_{\dot{F}}^2 L_0^3}{1 - \eta L_{\dot{F}} L_0}$$

$$C_3 = \frac{L_\nu L_{0/1} L_F^3 L_0^3}{1 - \eta L_F L_0}, \qquad\qquad C_4 = \frac{L_\nu L_{0/1} L_{\dot{F}}^3 L_0^3}{1 - \eta L_{\dot{F}} L_0}.$$

*Proof.* Note that $\|\boldsymbol{V}\|_2 \vee \|\dot{\boldsymbol{V}}\|_2 \leq L_{\dot{F}} L_0$, therefore, if $0 \leq \eta < \frac{1}{L_{\dot{F}} L_0}$, $(\boldsymbol{I}_r + \eta \boldsymbol{V})$ and $(\boldsymbol{I}_r + \eta \dot{\boldsymbol{V}})$ are invertible. For the following, we introduce the notation

$$\dot{\boldsymbol{G}} := \boldsymbol{G} + \boldsymbol{\nu} \ \ \text{and} \ \ \boldsymbol{F} = 2\boldsymbol{G} + \boldsymbol{D}_0^{-1}\boldsymbol{D}_1 \ \ \text{and} \ \ \dot{\boldsymbol{F}} = 2\dot{\boldsymbol{G}} + \boldsymbol{D}_0^{-1}\boldsymbol{D}_1.$$

Note that we have $\|\boldsymbol{F}\|_2 \leq L_F$ and $\|\dot{\boldsymbol{F}}\|_2 \leq L_{\dot{F}}$. For the left part of (G.2), we write

$$\boldsymbol{D}_0^{-\frac{1}{2}} \boldsymbol{V}^2 \left(\boldsymbol{I}_r + \eta \boldsymbol{V}\right)^{-1} \boldsymbol{\zeta} \dot{\boldsymbol{V}} \left(\boldsymbol{I}_r + \eta \dot{\boldsymbol{V}}\right)^{-1} \boldsymbol{D}_0^{-\frac{1}{2}}$$

$$= \boldsymbol{F} \boldsymbol{D}_0 \boldsymbol{F} \boldsymbol{D}_0 \left(\boldsymbol{I}_r + \eta \boldsymbol{F} \boldsymbol{D}_0\right)^{-1} \boldsymbol{\nu} \boldsymbol{D}_0 \dot{\boldsymbol{F}} \left(\boldsymbol{I}_r + \eta \boldsymbol{D}_0 \dot{\boldsymbol{F}}\right)^{-1}.$$

Therefore, we have

$$\left\| \boldsymbol{F} \boldsymbol{D}_0 \boldsymbol{F} \boldsymbol{D}_0 \left(\boldsymbol{I}_r + \eta \boldsymbol{F} \boldsymbol{D}_0\right)^{-1} \boldsymbol{\nu} \boldsymbol{D}_0 \dot{\boldsymbol{F}} \left(\boldsymbol{I}_r + \eta \boldsymbol{D}_0 \dot{\boldsymbol{F}}\right)^{-1} \right\|_2$$

$$\leq \frac{\|\boldsymbol{F}\|_2^2 \|\dot{\boldsymbol{F}}\|_2 \|\boldsymbol{D}_0\|_2^3 \|\boldsymbol{\nu}\|_2}{\left(1 - \eta \|\boldsymbol{F}\|_2 \|\boldsymbol{D}_0\|_2\right)\left(1 - \eta \|\dot{\boldsymbol{F}}\|_2 \|\boldsymbol{D}_0\|_2\right)}$$

$$\leq \frac{L_F^2 L_{\dot{F}} L_0^3 L_\nu}{\left(1 - \eta L_F L_0\right)\left(1 - \eta L_{\dot{F}} L_0\right)}.$$

Therefore, we have the bound. For the right part of (G.2), we write

$$\boldsymbol{D}_0^{-\frac{1}{2}} \boldsymbol{V} \boldsymbol{\zeta} \dot{\boldsymbol{V}}^2 \left(\boldsymbol{I}_r + \eta \dot{\boldsymbol{V}}\right)^{-1} \boldsymbol{D}_0^{-\frac{1}{2}} = \boldsymbol{F} \boldsymbol{D}_0 \boldsymbol{\nu} \boldsymbol{D}_0 \dot{\boldsymbol{F}} \boldsymbol{D}_0 \dot{\boldsymbol{F}} \left(\boldsymbol{I}_r + \eta \boldsymbol{D}_0 \dot{\boldsymbol{F}}\right)^{-1}$$

Therefore, we have

$$\left\| \boldsymbol{F} \boldsymbol{D}_0 \boldsymbol{\nu} \boldsymbol{D}_0 \dot{\boldsymbol{F}} \boldsymbol{D}_0 \dot{\boldsymbol{F}} \left(\boldsymbol{I}_r + \eta \boldsymbol{D}_0 \dot{\boldsymbol{F}}\right)^{-1} \right\|_2 \leq \frac{\|\boldsymbol{F}\|_2 \|\dot{\boldsymbol{F}}\|_2^2 \|\boldsymbol{D}_0\|_2^3 \|\boldsymbol{\nu}\|_2}{1 - \eta \|\dot{\boldsymbol{F}}\|_2 \|\boldsymbol{D}_0\|_2} \leq \frac{L_F L_{\dot{F}}^2 L_0^3 L_\nu}{1 - \eta L_{\dot{F}} L_0},$$

which gives us the bound. For the left part of (G.3), we write

$$D_0^{-\frac{1}{2}} V^3 \left(I_r + \eta V\right)^{-1} \zeta D_0^{-\frac{1}{2}} = (F D_0)^3 \left(I_r + \eta F D_0\right)^{-1} \nu$$

Therefore, we have

$$\left\| (F D_0)^3 \left(I_r + \eta F D_0\right)^{-1} \nu \right\|_2 \leq \frac{\|F\|_2^3 \|D_0\|_2^3 \|\nu\|_2}{1 - \eta \|F\|_2 \|D_0\|_2} \leq \frac{L_\nu L_F^3 L_0^3}{1 - \eta L_F L_0},$$

which gives us the bound. The the right part of (G.3) can be derived similarly. $\qquad\square$

**Proposition 29.** *Let* $V, \dot{V} \in \mathbb{R}^{r \times r}$ *be symmetric matrices such that* $\dot{V} = V + \zeta$. *We have*

$$\dot{V} \left(I_r + \eta \dot{V}\right)^{-1} - V \left(I_r + \eta V\right)^{-1} = \left(I_r + \eta V\right)^{-1} \zeta \left(I_r + \eta \dot{V}\right)^{-1}.$$

*Moreover, given that*

$$M := \zeta - \eta V \zeta - \eta \zeta V - \eta \zeta^2 + \eta^2 \zeta V \zeta + \eta^2 \zeta^3 + \eta^2 V \zeta V$$

*under the conditions of Proposition 28, we have for any* $\kappa_d > 0$,

$$-\frac{2}{\kappa_d^2} \eta^2 \zeta^2 - \eta^2 \kappa_d^2 V^4 - \eta^2 \kappa_d^2 V \zeta^2 V - C \eta^3 D_1 \preceq \left(I_r + \eta V\right)^{-1} \zeta \left(I_r + \eta \dot{V}\right)^{-1} - M$$

$$\preceq \frac{2}{\kappa_d^2} \eta^2 \zeta^2 + \eta^2 \kappa_d^2 V^4 + \eta^2 \kappa_d^2 V \zeta^2 V + C \eta^3 D_1$$

*where* $C = C_1 + C_2 + C_3 + C_4$, *i.e., the sum of the constants given in Proposition 28.*

*Proof.* We write

$$\dot{V} \left(I_r + \eta \dot{V}\right)^{-1} - \left(I_r + \eta V\right)^{-1} V$$

$$= \left(I_r + \eta V\right)^{-1} \left( \left(I_r + \eta V\right)(V + \zeta) - V \left(I_r + \eta \dot{V}\right) \right) \left(I_r + \eta \dot{V}\right)^{-1}$$

$$= \left(I_r + \eta V\right)^{-1} \zeta \left(I_r + \eta \dot{V}\right)^{-1}.$$

For the second part, we write

$$\left(I_r + \eta V\right)^{-1} \zeta \left(I_r + \eta \dot{V}\right)^{-1}$$

$$= \left(I_r - \eta V \left(I_r + \eta V\right)^{-1}\right) \zeta \left(I_r - \eta \dot{V} \left(I_r + \eta \dot{V}\right)^{-1}\right)$$

$$= \zeta - \eta V \left(I_r - \eta V + \eta^2 V^2 \left(I_r + \eta V\right)^{-1}\right) \zeta$$

$$\quad - \eta \zeta \dot{V} \left(I_r - \eta \dot{V} + \eta^2 \dot{V}^2 \left(I_r + \eta V\right)^{-1}\right) + \eta^2 V \left(I_r + \eta V\right)^{-1} \zeta \dot{V} \left(I_r + \eta \dot{V}\right)^{-1}$$

$$= \zeta - \eta V \zeta - \eta \zeta V - \eta \zeta^2 + \eta^2 V^2 \zeta + \eta^2 \zeta \dot{V}^2 - \eta^3 V^3 \left(I_r + \eta V\right)^{-1} \zeta$$

$$\quad - \eta^3 \zeta \dot{V}^3 \left(I_r + \eta \dot{V}\right)^{-1} + \eta^2 V \left(I_r + \eta V\right)^{-1} \zeta \dot{V} \left(I_r + \eta \dot{V}\right)^{-1}$$

We have

$$\eta^2 V^2 \zeta + \eta^2 \zeta \dot{V}^2 = \eta^2 V^2 \zeta + \eta^2 \zeta (V + \zeta)^2 = \underbrace{\eta^2 V^2 \zeta + \eta^2 \zeta V^2 + \eta^2 \zeta^2 V}_{:= M_1} + \eta^2 \zeta V \zeta + \eta^2 \zeta^3.$$

Moreover,

$$\eta^2 V \left(I_r + \eta V\right)^{-1} \zeta \dot{V} \left(I_r + \eta \dot{V}\right)^{-1}$$

$$= \eta^2 V \zeta \dot{V} \left(I_r + \eta \dot{V}\right)^{-1} - \eta^3 V^2 \left(I_r + \eta V\right)^{-1} \zeta \dot{V} \left(I_r + \eta \dot{V}\right)^{-1}$$

$$= \eta^2 V \zeta \dot{V} - \eta^3 V \zeta \dot{V}^2 \left(I_r + \eta \dot{V}\right)^{-1} - \eta^3 V^2 \left(I_r + \eta V\right)^{-1} \zeta \dot{V} \left(I_r + \eta \dot{V}\right)^{-1}$$

$$= \eta^2 \boldsymbol{V} \boldsymbol{\zeta} \boldsymbol{V} + \underbrace{\eta^2 \boldsymbol{V} \boldsymbol{\zeta}^2}_{:=\boldsymbol{M}_2} - \eta^3 \boldsymbol{V} \boldsymbol{\zeta} \dot{\boldsymbol{V}}^2 \left( \boldsymbol{I}_r + \eta \dot{\boldsymbol{V}} \right)^{-1} - \eta^3 \boldsymbol{V}^2 \left( \boldsymbol{I}_r + \eta \boldsymbol{V} \right)^{-1} \boldsymbol{\zeta} \dot{\boldsymbol{V}} \left( \boldsymbol{I}_r + \eta \dot{\boldsymbol{V}} \right)^{-1}$$

By Proposition 22, we have

$$-2\eta^2 \boldsymbol{\zeta}^2 - \eta^2 \boldsymbol{V}^4 - \eta^2 \boldsymbol{V} \boldsymbol{\zeta}^2 \boldsymbol{V} \preceq \boldsymbol{M}_1 + \boldsymbol{M}_2 \preceq 2\eta^2 \boldsymbol{\zeta}^2 + \eta^2 \boldsymbol{V}^4 + \eta^2 \boldsymbol{V} \boldsymbol{\zeta}^2 \boldsymbol{V}.$$

Therefore by Proposition 28, we have

$$-\frac{2}{\kappa_d^2} \eta^2 \boldsymbol{\zeta}^2 - \eta^2 \kappa_d^2 \boldsymbol{V}^4 - \eta^2 \kappa_d^2 \boldsymbol{V} \boldsymbol{\zeta}^2 \boldsymbol{V} - C\eta^3 \boldsymbol{D}_1 \preceq \left( \boldsymbol{I}_r + \eta \boldsymbol{V} \right)^{-1} \boldsymbol{\zeta} \left( \boldsymbol{I}_r + \eta \dot{\boldsymbol{V}} \right)^{-1} - \boldsymbol{M}$$

$$\preceq \frac{2}{\kappa_d^2} \eta^2 \boldsymbol{\zeta}^2 + \eta^2 \kappa_d^2 \boldsymbol{V}^4 + \eta^2 \kappa_d^2 \boldsymbol{V} \boldsymbol{\zeta}^2 \boldsymbol{V} + C\eta^3 \boldsymbol{D}_1.$$

$\square$

**Proposition 30.** *By using the notation in Proposition 28, we consider*

$$\eta < \frac{1}{L_F L_0} \quad \text{and} \quad 0 < \varepsilon < \frac{0.5/\eta}{L_F L_0} - 1$$

*Then,*

$$\boldsymbol{V} \left( \boldsymbol{I}_r + \eta \boldsymbol{V} \right)^{-1} - \varepsilon \eta \boldsymbol{V}^2 \succeq \boldsymbol{V} \left( \boldsymbol{I}_r + \eta(1 + \varepsilon) \boldsymbol{V} \right)^{-1} - 2.5\varepsilon\eta^2 C \boldsymbol{D}_1$$

$$\boldsymbol{V} \left( \boldsymbol{I}_r + \eta \boldsymbol{V} \right)^{-1} + \varepsilon \eta \boldsymbol{V}^2 \preceq \boldsymbol{V} \left( \boldsymbol{I}_r + \eta(1 - \varepsilon) \boldsymbol{V} \right)^{-1} + 1.5\varepsilon\eta^2 C \boldsymbol{D}_1,$$

*where $C = \frac{L_{0/1} L_F^3 L_0^3}{1 - \eta L_F L_0}$.*

*Proof.* For the lower bound, we have

$$\boldsymbol{V} \left( \boldsymbol{I}_r + \eta \boldsymbol{V} \right)^{-1} - \varepsilon \eta \boldsymbol{V}^2$$

$$= \boldsymbol{V} - (1 + \varepsilon)\eta \boldsymbol{V}^2 + \eta^2 \boldsymbol{V}^3 (\boldsymbol{I}_r + \eta \boldsymbol{V})^{-1}$$

$$= \boldsymbol{V} - (1 + \varepsilon)\eta \boldsymbol{V}^2 + (1 + \varepsilon)^2 \eta^2 \boldsymbol{V}^3 (\boldsymbol{I}_r + (1 + \varepsilon)\eta \boldsymbol{V})^{-1}$$

$$- (2\varepsilon + \varepsilon^2)\eta^2 \boldsymbol{V}^3 (\boldsymbol{I}_r + \eta \boldsymbol{V})^{-1} + (1 + \varepsilon)^2 \varepsilon \eta^3 \boldsymbol{V}^4 (\boldsymbol{I}_r + (1 + \varepsilon)\eta \boldsymbol{V})^{-1} (\boldsymbol{I}_r + \eta \boldsymbol{V})^{-1}$$

$$\succeq \boldsymbol{V} \left( \boldsymbol{I}_r + \eta(1 + \varepsilon) \boldsymbol{V} \right)^{-1} - 2.5\varepsilon\eta^2 C \boldsymbol{D}_1,$$

where we used $C_3$ with $L_\nu = 1$ in Proposition 28 in the last step. For the upper bound,

$$\boldsymbol{V} \left( \boldsymbol{I}_r + \eta \boldsymbol{V} \right)^{-1} + \varepsilon \eta \boldsymbol{V}^2$$

$$= \boldsymbol{V} - (1 - \varepsilon)\eta \boldsymbol{V}^2 + \eta^2 \boldsymbol{V}^3 (\boldsymbol{I}_r + \eta \boldsymbol{V})^{-1}$$

$$= \boldsymbol{V} - (1 - \varepsilon)\eta \boldsymbol{V}^2 + (1 - \varepsilon)^2 \eta^2 \boldsymbol{V}^3 (\boldsymbol{I}_r + (1 - \varepsilon)\eta \boldsymbol{V})^{-1}$$

$$+ (2\varepsilon - \varepsilon^2)\eta^2 \boldsymbol{V}^3 (\boldsymbol{I}_r + \eta \boldsymbol{V})^{-1} - (1 - \varepsilon)^2 \varepsilon \eta^3 \boldsymbol{V}^4 (\boldsymbol{I}_r + (1 - \varepsilon)\eta \boldsymbol{V})^{-1} (\boldsymbol{I}_r + \eta \boldsymbol{V})^{-1}$$

$$\preceq \boldsymbol{V} \left( \boldsymbol{I}_r + \eta(1 + \varepsilon) \boldsymbol{V} \right)^{-1} + 1.5\varepsilon\eta^2 C \boldsymbol{D}_1.$$

$\square$

**Lemma 7.** *For any $\eta \in \mathbb{R}$ and $t \in \mathbb{N}$, we have*

$$\begin{bmatrix} \boldsymbol{I}_r & \eta \boldsymbol{I}_r \\ \eta \boldsymbol{\Lambda}^2 & \boldsymbol{I}_r \end{bmatrix}^t = \begin{bmatrix} \frac{(\boldsymbol{I}_r + \eta \boldsymbol{\Lambda})^t + (\boldsymbol{I}_r - \eta \boldsymbol{\Lambda})^t}{2} & \boldsymbol{\Lambda}^{-1} \frac{(\boldsymbol{I}_r + \eta \boldsymbol{\Lambda})^t - (\boldsymbol{I}_r - \eta \boldsymbol{\Lambda})^t}{2} \\ \boldsymbol{\Lambda} \frac{(\boldsymbol{I}_r + \eta \boldsymbol{\Lambda})^t - (\boldsymbol{I}_r - \eta \boldsymbol{\Lambda})^t}{2} & \frac{(\boldsymbol{I}_r + \eta \boldsymbol{\Lambda})^t + (\boldsymbol{I}_r - \eta \boldsymbol{\Lambda})^t}{2} \end{bmatrix}. \tag{G.4}$$

*Proof.* We observe that

$$\boldsymbol{A} := \begin{bmatrix} 0 & \boldsymbol{I}_r \\ \boldsymbol{\Lambda}^2 & 0 \end{bmatrix} \implies \text{(G.4)} = \sum_{k=0}^t \binom{t}{k} \eta^k \boldsymbol{A}^k.$$

Note that

$$\boldsymbol{A}^{2k} = \begin{bmatrix} \boldsymbol{\Lambda}^{2k} & 0 \\ 0 & \boldsymbol{\Lambda}^{2k} \end{bmatrix} \quad \text{and} \quad \boldsymbol{A}^{2k+1} = \begin{bmatrix} 0 & \boldsymbol{\Lambda}^{2k} \\ \boldsymbol{\Lambda}^{2k+2} & 0 \end{bmatrix}.$$

Therefore,

$$\sum_{k=0}^{t} \binom{t}{k} \eta^k \mathbf{A}^k = \begin{bmatrix} \sum_{\substack{k=0 \\ k \text{ even}}}^{t} \binom{t}{k} \eta^k \mathbf{\Lambda}^k & \sum_{\substack{k=0 \\ k \text{ odd}}}^{t} \binom{t}{k} \eta^k \mathbf{\Lambda}^{k-1} \\ \sum_{\substack{k=0 \\ k \text{ odd}}}^{t} \binom{t}{k} \eta^k \mathbf{\Lambda}^{k+1} & \sum_{\substack{k=0 \\ k \text{ even}}}^{t} \binom{t}{k} \eta^k \mathbf{\Lambda}^k \end{bmatrix}$$

$$= \begin{bmatrix} \frac{(\mathbf{I}_r + \eta\mathbf{\Lambda})^t + (\mathbf{I}_r - \eta\mathbf{\Lambda})^t}{2} & \mathbf{\Lambda}^{-1} \frac{(\mathbf{I}_r + \eta\mathbf{\Lambda})^t - (\mathbf{I}_r - \eta\mathbf{\Lambda})^t}{2} \\ \mathbf{\Lambda} \frac{(\mathbf{I}_r + \eta\mathbf{\Lambda})^t - (\mathbf{I}_r - \eta\mathbf{\Lambda})^t}{2} & \frac{(\mathbf{I}_r + \eta\mathbf{\Lambda})^t + (\mathbf{I}_r - \eta\mathbf{\Lambda})^t}{2} \end{bmatrix}.$$

$\square$

## G.2 Some moment bounds and concentration inequalities

**Lemma 8** (Hypercontractivity). *Let $P_k : \mathbb{R}^d \to \mathbb{R}$ be a polynomial of degree-$k$ and $\boldsymbol{x} \sim \mathcal{N}(0, \boldsymbol{I}_d)$. For $q \geq 2$, we have $\mathbb{E}\left[P_k(\boldsymbol{x})^q\right]^{1/q} \leq (q-1)^{k/2} \mathbb{E}\left[P_k(\boldsymbol{x})^2\right]^{1/2}$.*

**Lemma 9.** *Let $\boldsymbol{x} \sim \mathcal{N}(0, \boldsymbol{I}_d)$ and $\boldsymbol{S} \in \mathbb{R}^{d \times d}$ be a symmetric matrix. For $u > 0$,*

$$\mathbb{P}\left[|\boldsymbol{x}^\top \boldsymbol{S} \boldsymbol{x} - Tr(\boldsymbol{S})| \geq 2\|\boldsymbol{S}\|_F u + 2\|\boldsymbol{S}\|_2 u^2\right] \leq 2e^{-u^2}.$$

*Proof.* We note that $\boldsymbol{x}^\top \boldsymbol{S} \boldsymbol{x} - \text{Tr}(\boldsymbol{S})$ has the same distribution with $\sum_{i=1}^{d} \lambda_i(\boldsymbol{S})(Z_i^2 - 1)$, where $Z_i \sim_{iid} \mathcal{N}(0, 1)$. By using the Laurent-Massart lemma [LM00], we have the result. $\square$

**Corollary 6.** *Let $y = \boldsymbol{x}^\top \boldsymbol{S} \boldsymbol{x} - Tr(\boldsymbol{S})$ and $\boldsymbol{x} \sim \mathcal{N}(0, \boldsymbol{I}_d)$. For $p \geq 2$, we have $\mathbb{E}[|y|^p]^{\frac{1}{p}} \leq (p-1)\sqrt{2}\|\boldsymbol{S}\|_F$.*

*Proof.* By observing that $\mathbb{E}[|y|^2] = 2\|\boldsymbol{S}\|_F^2$, we have the result. $\square$

**Corollary 7.** *For $\boldsymbol{A} \in \mathbb{R}^{d \times r}$, $p \geq 2$ and $\boldsymbol{x} \sim \mathcal{N}(0, \boldsymbol{I}_d)$, we have $\mathbb{E}[\|\boldsymbol{A}^\top \boldsymbol{x}\|_2^{2p}]^{\frac{1}{p}} \leq \sqrt{3}(p-1)Tr(\boldsymbol{A}^\top \boldsymbol{A})$.*

*Proof.* By Lemma 8, we have $\mathbb{E}[\|\boldsymbol{A}^\top \boldsymbol{x}\|_2^{2p}]^{\frac{1}{p}} \leq (p-1)\mathbb{E}[\|\boldsymbol{A}^\top \boldsymbol{x}\|_2^4]^{\frac{1}{2}}$. For $\boldsymbol{S} = \boldsymbol{A}\boldsymbol{A}^\top$, we have

$$\mathbb{E}[\|\boldsymbol{A}^\top \boldsymbol{x}\|_2^4] = \mathbb{E}[(\boldsymbol{x}^\top \boldsymbol{S} \boldsymbol{x})^2] = \text{Tr}(\mathbb{E}[(\boldsymbol{x}^\top \boldsymbol{S} \boldsymbol{x})\boldsymbol{x}\boldsymbol{x}^\top]\boldsymbol{S}).$$

We have

$$\mathbb{E}[(\boldsymbol{x}^\top \boldsymbol{S} \boldsymbol{x})\boldsymbol{x}\boldsymbol{x}^\top] = \text{Tr}(\boldsymbol{S})\boldsymbol{I}_d + 2\boldsymbol{S} \Rightarrow \mathbb{E}[\|\boldsymbol{A}^\top \boldsymbol{x}\|_2^4] = \text{Tr}(\boldsymbol{S})^2 + 2\|\boldsymbol{S}\|_F^2 \overset{(a)}{\leq} 3\text{Tr}(\boldsymbol{S})^2,$$

where (a) follows that $\boldsymbol{S}$ is positive semi-definite. Since $\text{Tr}(\boldsymbol{S}) = \text{Tr}(\boldsymbol{A}^\top \boldsymbol{A})$, we have the statement. $\square$

**Proposition 31.** *Let $\boldsymbol{x}_j \sim_{i.i.d} \mathcal{N}(0, \boldsymbol{I}_r)$, for $j \in [N]$. There exists a constant $c > 0$ such that for $\delta = \frac{u(r + \sqrt{Cr\log d} + C\log d)}{\sqrt{N}}$, we have*

$$\mathbb{P}\left[\sup_{\substack{\boldsymbol{S} \in \mathbb{R}^{r \times r} \\ \|\boldsymbol{S}\|_F = 1}} \left|\frac{1}{N}\sum_{j=1}^{N} \tfrac{1}{2}Tr\big(\boldsymbol{S}(\boldsymbol{x}_j\boldsymbol{x}_j^\top - \boldsymbol{I}_r)\big)^2 - 1\right| \geq \max\{2\delta, \delta^2\} + 10d^{-C/2}\right]$$

$$\leq d^2 \exp(-cu^2) + 2Nd^{-C}.$$

*Proof.* We observe that

$$\frac{1}{2}\|\boldsymbol{x}_j\boldsymbol{x}_j^\top - \boldsymbol{I}_r\|_F \leq \frac{1}{2}\left(\|\boldsymbol{x}_j\|_2^2 + r\right).$$

By using Lemma 9, we can derive

$$\mathbb{P}\big[\underbrace{\|\boldsymbol{x}_j\|_2^2 \leq r + 2\sqrt{r}\sqrt{C\log d} + 2C\log d}_{=:\mathcal{E}_j}\big] \geq 1 - 2d^{-C}.$$

We have

$$\left|\mathbb{E}\left[\tfrac{1}{2}\mathrm{Tr}\big(S(x_j x_j^\top - I_r)\big)^2 \mathbb{1}_{\mathcal{E}_j}\right] - 1\right| = \tfrac{1}{2}\mathbb{E}\big[\mathrm{Tr}\big(S(x_j x_j^\top - I_r)\big)^2 \mathbb{1}_{\mathcal{E}_j^c}\big]$$

$$\leq \tfrac{1}{2}\mathbb{E}\left[\mathrm{Tr}\big(S(x_j x_j^\top - I_r)\big)^4\right]^{1/2} \sqrt{2}d^{-C/2}$$

$$\leq 9\sqrt{2}d^{-C/2}.$$

By using [Ver10, Theorem 5.41], for $\delta = \frac{u(r + \sqrt{Cr\log d} + C\log d)}{\sqrt{N}}$, we have

$$\mathbb{P}\left[\sup_{\substack{S \in \mathbb{R}^{r\times r}\\ \|S\|_F=1}} \left|\frac{1}{N}\sum_{j=1}^N \tfrac{1}{2}\mathrm{Tr}\big(S(x_j x_j^\top - I_r)\big)^2 - 1\right| \geq \max\{2\delta, \delta^2\} + 10d^{-C/2}\right]$$

$$\leq \mathbb{P}\left[\sup_{\substack{S \in \mathbb{R}^{r\times r}\\ \|S\|_F=1}} \left|\frac{1}{N}\sum_{j=1}^N \tfrac{1}{2}\mathrm{Tr}\big(S(x_j x_j^\top - I_r)\big)^2 \mathbb{1}_{\mathcal{E}_j} - \mathbb{E}\left[\tfrac{1}{2}\mathrm{Tr}\big(S(x_j x_j^\top - I_r)\big)^2 \mathbb{1}_{\mathcal{E}_j}\right]\right| \geq \max\{2\delta, \delta^2\}\right]$$

$$+ 2Nd^{-C}$$

$$\leq d^2 \exp(-cu^2) + 2Nd^{-C}.$$

$\square$

**Proposition 32.** *Let $x_j \sim_{i.i.d} \mathcal{N}(0, I_d)$, for $j \in [N]$, and $W \in \mathbb{R}^{d\times r}$ be an orthonormal matrix. For a fixed $S \in \mathbb{R}^{d\times d}$, $C \geq 16$ and $N \geq Cr\log d$, we have*

$$\mathbb{P}\left[\left\|\frac{1}{N}\sum_{j=1}^N \frac{1}{2}Tr\big(S(x_j x_j^\top - I_d)\big)W^\top(x_j x_j^\top - I_d)W - W^\top SW\right\|_2 \geq 24e\|S\|_F\left(\sqrt{\frac{Cr}{N}} + d^{\frac{-C}{2}}\right)\right]$$

$$\leq 2e^{\frac{-Cr}{8}} + 2Nd^{-C}.$$

*Proof.* Without loss of generality, we assume $\|S\|_F = 1$. By using Lemma 9, we have

$$\mathbb{P}\Big[\underbrace{|\mathrm{Tr}\big(S(x_j x_j^\top - I_d)\big)| \leq 4\sqrt{C\log d}}_{=:\mathcal{E}_j}\Big] \geq 1 - 2d^{-C}.$$

For the following, we fix a $v \in S^{d-1}$. First, to bound the bias due to clipping, we write:

$$\left|\mathbb{E}\left[\mathrm{Tr}\big(S(x_j x_j^\top - I_d)\big)(\langle v, x\rangle^2 - 1)\mathbb{1}_{\mathcal{E}_j^c}\right]\right|$$

$$\leq \mathbb{E}\left[\mathrm{Tr}\big(S(x_j x_j^\top - I_d)\big)^4\right]^{\frac{1}{4}} \mathbb{E}\left[(\langle v, x\rangle^2 - 1)^4\right]^{\frac{1}{4}} \sqrt{2}d^{-C/2} \leq 18\sqrt{2}d^{-C/2}.$$

On the other hand, to bound the moments of the clipped random variable, we have for $p \geq 2$,

$$\mathbb{E}\left[|\mathrm{Tr}\big(S(x_j x_j^\top - I_d)\big)(\langle v, x\rangle^2 - 1)|^p \mathbb{1}_{\mathcal{E}_j}\right]$$

$$\leq (4\sqrt{C\log d})^{p-2}\mathbb{E}\left[\mathrm{Tr}\big(S(x_j x_j^\top - I_d)\big)^2|(\langle v, x\rangle^2 - 1)|^p\right] \leq (12e)^2(8\sqrt{2}e\sqrt{C\log d})^{p-2}\frac{p!}{2}.$$

By using $\varepsilon$-cover argument, we can derive

$$\mathbb{P}\left[\left\|\frac{1}{N}\sum_{j=1}^N \frac{1}{2}\mathrm{Tr}\big(S(x_j x_j^\top - I_d)\big)W^\top(x_j x_j^\top - I_d)W - W^\top SW\right\|_2 \geq 24eu + 18\sqrt{2}d^{-C/2}\right]$$

$$\leq 2\cdot 9^r \exp\left(\frac{-Nu^2/2}{1 + u\sqrt{C\log d}}\right) + 2Nd^{-C}.$$

By using $u = \sqrt{Cr/N}$, we have the result. $\square$

*Proof.* Without loss of generality, we assume $\|\boldsymbol{S}\|_F = 1$. We have

$$\Big\| \sum_{j=1}^N y_j(\boldsymbol{x}_j \boldsymbol{x}_j^\top - \boldsymbol{I}_d) - \boldsymbol{S} \Big\|_F^2 \leq \sup_{\substack{\boldsymbol{S} \in \mathbb{R}^{r \times r} \\ \|\boldsymbol{S}\|_F = 1}} \bigg| \frac{1}{N} \sum_{j=1}^N \tfrac{1}{2}\mathrm{Tr}\big(\boldsymbol{S}(\boldsymbol{x}_j \boldsymbol{x}_j^\top - \boldsymbol{I}_r)\big)^2 - 1 \bigg|^2 .$$

Hence, by considering the event in Proposition 31, we have the statement. $\qquad\square$

**Proposition 33.** *Let $X \in \mathbb{R}$ be a random variable such that for some $K, C > 0$, $\mathbb{E}[|X|^p] \leq CK^p p^{pc}$ for some $c > 0$ and $p \geq k$. Then, $\mathbb{P}\left[|X| \geq Ku\right] \leq Ce^{-\frac{u^{1/c}}{e}}$ for $u \geq (ke)^c$.*

*Proof.* Use Markov inequality with $p = \frac{u^{1/c}}{e}$. $\qquad\square$

### G.3 Miscellaneous

**Proposition 34.** *We consider $\eta \leq \frac{1}{10}$. The following statements holds:*

- *For $0.2 \geq \delta > 0$, let*

$$u_{t+1} = u_t + \eta u_t(1 - u_t), \quad 1 + \delta \geq u_0 \geq 0.$$

*We have $1 + \left(\delta \vee \frac{\eta^2}{4}\right) \geq \sup_t u_t \geq 0$. Moreover, $t^* = \inf\{t : u_t \geq 1\}$, we have $u_{t+1} \geq u_t$ for $t < t^*$ and $u_{t^*} \geq u_t \geq 1$ for $t \geq t^*$.*

- *For $0.5 > \varepsilon > 0$ and $1.1 > \overline{u}_0 \geq u_0 \geq \underline{u}_0 > 0$, let*

$$\overline{u}_{t+1} = \overline{u}_t + \eta(1 + \varepsilon)\overline{u}_t(1 - \overline{u}_t) \ \text{ and } \ \underline{u}_{t+1} = \underline{u}_t + \eta\underline{u}_t(1 - \underline{u}_t).$$

*and*

$$u_t + \eta u_t(1 - u_t) \leq u_{t+1} \leq u_t + \eta(1 + \varepsilon)u_t(1 - u_t).$$

*We have*

$$\frac{1}{2}\left(1 \wedge \underline{u}_0 e^{\frac{\eta t}{1+\eta}}\right) \leq \underline{u}_t \leq u_t \leq \overline{u}_t \leq \left(1.1 \wedge \overline{u}_0 e^{\eta(1+\varepsilon)t}\right).$$

*Proof.* If $t \geq t^*$, by monotonicity of the update, we have $1 \leq u_{t+1} \leq u_t \leq u_{t^*}$. If $t^* > 0$, then for $t < t^*$, we have $1 \geq u_t \geq 0$ and $u_t(1 - u_t) \geq 0$, and thus, we have $u_{t+1} \geq u_t \geq 0$. Next, we observe that $u_{t^*} \leq 1 + 0.25\eta$ and by monotonicity of the update for $t \geq t^*$, we have $1 \leq u_t \leq u_{t^*}$. Hence, it is sufficient to bound $u_{t^*}$ to bound $\sup_t u_t$. Note that, we have $1 \geq u_{t^*-1} \geq 1 - 0.25\eta$, and thus,

$$\frac{u_{t^*}}{u_{t^*-1}} = 1 + \eta(1 - u_{t^*-1}) \leq 1 + \eta^2 \Rightarrow u_{t^*} \leq 1 + \frac{\eta^2}{4}.$$

For the second item, by monotonicity, we have $0 < \underline{u}_t \leq u_t \leq \overline{u}_t < 1.1$. Moreover, by [AGP24, Lemma A.2], we have for $t < t_u := \inf\{t : \underline{u}_t \geq 0.5\}$

$$\frac{\underline{u}_0 e^{\frac{\eta t}{1+\eta}}}{1 + \underline{u}_0 e^{\frac{\eta t}{1+\eta}}} \leq \underline{u}_t \Rightarrow \frac{\underline{u}_0}{2} e^{\frac{\eta t}{1+\eta}} \leq \underline{u}_t.$$

For $t \geq t_u$, by the first item, we have $\underline{u}_t \geq 0.5$. Therefore, we have

$$\frac{1}{2}\left(\underline{u}_0 e^{\frac{\eta t}{1+\eta}} \wedge 1\right) \leq \underline{u}_t.$$

On the other hand, for all $t \in \mathbb{N}$, we have $\overline{u}_t \leq \overline{u}_0 e^{\eta(1+\varepsilon)t}$. By the first item, we have $\overline{u}_t \leq \overline{u}_0 e^{\eta(1+\varepsilon)t} \wedge 1.1$. $\qquad\square$

**Proposition 35.** *For $t, \lambda > 0$, we have*

$$\frac{1}{t\exp(t\lambda)} \leq \frac{\lambda}{\exp(t\lambda) - 1} \leq \frac{1}{t}.$$

*Proof.* The upper bound follows $\exp(t\lambda) - 1 \geq t\lambda$. For the lower bound,

$$\frac{1}{t} - \frac{\lambda}{\exp(t\lambda) - 1} = \frac{\exp(t\lambda) - t\lambda - 1}{t(\exp(t\lambda) - 1)}. \tag{G.5}$$

We have

$$\exp(t\lambda) - t\lambda - 1 \leq \sum_{k=2}^{\infty} \frac{(t\lambda)^k}{k!} = t\lambda \sum_{k=1}^{\infty} \frac{(t\lambda)^k}{(k+1)!} \leq t\lambda \sum_{k=1}^{\infty} \frac{(t\lambda)^k}{k!} = t\lambda(\exp(t\lambda) - 1).$$

Therefore,

$$(\text{G.5}) \leq \lambda \Rightarrow \frac{1}{t} \leq \frac{\lambda}{\exp(t\lambda) - 1} + \lambda \Rightarrow \frac{1}{t\exp(t\lambda)} \leq \frac{\lambda}{\exp(t\lambda) - 1}.$$

$\square$

**Lemma 10.** *Let $r_s \asymp d^\gamma$, $\gamma \in [0,1)$, and $\log^{-1} d \ll C_d \ll \log^{10} d$. We define $F_d, G_d, H_d$ as*

$$F_d(u) := \left(1 - \frac{1}{1 + \left(\frac{dC_d}{r_s} \frac{1}{u} - 1\right)\left(\frac{d}{r_s}\right)^{-\frac{1}{u}}}\right)^2, \qquad G_d(u) := 1 - \frac{1}{1 + \left(\frac{dC_d}{r_s} - 1\right)\left(\frac{d}{r_s}\right)^{-\frac{1}{u}}},$$

$$H_d(u) := \left(1 - C_d \left(\frac{d}{r_s}\right)^{\frac{1}{u} - 1}\right)_+,$$

*We have*

- *For any $C > 0$, $\sup_{u \leq \log^C d} |F_d(u)| \leq 1$ for $d \geq \Omega_C(1)$.*

- *$\sup_u |G_d(u)| \vee |H_d(u)| \leq 1$.*

- *For any $\delta \in (0, 0.5)$, let $\mathcal{C}_\delta := \{u \geq 0 : |u - 1| < \delta\}$. For any compact $\mathcal{K} \subset (0, \infty] \setminus \mathcal{C}_\delta$, we have $F_d(u) \xrightarrow{d \to \infty} \mathbb{1}\{u > 1\}$ uniformly on $\mathcal{K}$.*

- *For any compact $\mathcal{K} \subseteq [0, \infty] \setminus \mathcal{C}_\delta$, we have*

$$G_d(u), H_d(u) \xrightarrow{d \to \infty} \mathbb{1}\{u > 1\}, \quad G_d^2(u) \xrightarrow{d \to \infty} \mathbb{1}\{u > 1\}$$

   *all uniformly on $\mathcal{K}$.*

*Proof.* For the first item, if $u \leq \log^C d$, for $d \geq \Omega_C(1)$

$$\frac{dC_d u}{r_s} - 1 \geq \frac{d}{r_s} \frac{t}{\log^{C+1} d} - 1 > 0 .$$

Therefore, $|F_d(u)| \leq 1$. For the second item, since $\frac{dC_d}{r_s} > 1$ for $d \geq \Omega(1)$, the item follows.

For the third item, since $E := [0, \infty) \setminus \mathcal{C}_\delta$ is closed in $[0, \infty)$, it suffices to establish the result on small open intervals around each point of of $E$ within $[0, \infty)$. Fix $u_0 \in E$ and choose $\epsilon \in (0, \delta/2)$. Since $B(u_0, \epsilon) := (u_0 - \epsilon, u_0 + \epsilon) \cap [0, \infty)$ is convex it can be either in $P_> := \{u : u > 1 + \delta/2\}$ or $P_< := \{u : u < 1 - \delta/2\}$. Without loss of generality let us assume it is in $P_<$. Then,

$$\sup_{u \in B(u_0, \epsilon) \subset P_<} |F_d(u)| \leq 1 - \frac{1}{1 + \left(O_\delta(C_d) - \left(\frac{d}{r_s}\right)^{-1}\right)\left(\frac{d}{r_s}\right)^{-O_\delta(1)}} \to 0.$$

A similar step can be repeated if $B_\epsilon \subset P_<$.

For the last item, we first observe that uniform convergence of $G_d(u)$ implies the uniform convergence of $G_d^2(u)$. Therefore, we will only prove the first result. Since $E := [0, \infty] \setminus \mathcal{C}_\delta$, is compact, and thus, $P_> \cap E$ and $P_< \cap E$ are also compact, we can directly use these sets. Without loss of generality let us use $P_< \cap E$. Then,

$$\sup_{u \in P_< \cap E} |G_d(u)| \vee |H_d(u)| \leq \left(1 - C_d \frac{d}{r_s}^{O_\delta(1)}\right)_+ \to 0.$$

A similar step can be repeated if $P_> \cap E$. Therefore, the statement follows. $\square$

**Proposition 36.** *Let $r_u \leq r$ and*

$$t \in \begin{cases} (0, \infty), & \alpha \in [0, 0.5) \\ (0, \infty) \setminus \{j^\alpha : j \in \mathbb{N}\}, & \alpha > 0.5, \end{cases} \qquad \kappa_{\text{eff}} := \begin{cases} r^\alpha, & \alpha \in [0, 0.5) \\ 1, & \alpha > 0.5. \end{cases}$$

*We have*

- *For* $\mathsf{K} \in \{G, H\}$ *and* $t \neq \lim_{d \to \infty} \frac{1}{\lambda_j \kappa_{\text{eff}}}$, *we have*

$$\mathsf{K}_d(\tfrac{1}{\lambda_j t \kappa_{\text{eff}}}) - \mathbb{1}\{\tfrac{1}{\lambda_j} > t \kappa_{\text{eff}}\} = o_d(1).$$

- *For* $\mathsf{K} \in \{F, G, H\}$,

$$\frac{1}{\|\mathbf{\Lambda}\|_{\mathrm{F}}^2} \sum_{j=1}^{r_u} \lambda_j^2 \left( \mathsf{K}_d(\tfrac{1}{\lambda_j t \kappa_{\text{eff}}}) - \mathbb{1}\{\tfrac{1}{\lambda_j} > t \kappa_{\text{eff}}\} \right) = o_d(1). \tag{G.6}$$

*Proof.* The first item immediately follows Lemma 10. In the following, we will prove the second item for the heavy and light tailed cases separately.

**For** $\alpha \in [0, 0.5)$**:** We define a sequence of measures $\mu_d\{j/r\} \propto j^{-2\alpha}, j \leq [r]$. We observe that

- We have $\mu_d \to \mu$ weakly such that $\mu$ is supported on $[0,1]$ and $\mu([0,\tau]) = \tau^{1-2\alpha}$ for $\tau \in [0,1]$.
- Moreover, (G.6) $= \mathbb{E}_{X \sim \mu_d}\left[(\mathsf{K}_d(X^\alpha/t) - \mathbb{1}\{X^\alpha > t\})\mathbb{1}\{X \leq \frac{r_u}{r}\}\right]$.

By using the $\mathcal{C}_\delta$ definition in Lemma 10:

$$\left| \mathbb{E}_{X \sim \mu_d}\left[(\mathsf{K}_d(X^\alpha/t) - \mathbb{1}\{X^\alpha > t\})\mathbb{1}\{X \leq \tfrac{r_u}{r}\}\right] \right|$$

$$\leq \mathbb{E}_{X \sim \mu_d}\left[|\mathsf{K}_d(X^\alpha/t) - \mathbb{1}\{X^\alpha > t\}|\mathbb{1}\{X^\alpha \in [0,1] \setminus \mathcal{C}_\delta\}\right]$$

$$+ \mathbb{E}_{X \sim \mu_d}\left[|\mathsf{K}_d(X^\alpha/t) - \mathbb{1}\{X^\alpha > t\}|\mathbb{1}\{X^\alpha \in \mathcal{C}_\delta\}\right]$$

$$\overset{(a)}{\leq} o_d(1) + \mathbb{P}_{X \sim \mu}[X^\alpha \in C_\delta],$$

where we used the second item in Lemma 10 for (a). Since $\mathbb{P}_{X \sim \mu}[X^\alpha \in C_\delta] \xrightarrow{\delta \to 0} 0$, we have the first result.

**For** $\alpha > 0.5$**:** We define a sequence of measures $\mu_d\{j\} \propto j^{-2\alpha}, j \leq [r]$. We observe that

- We have $\mu_d \to \mu$ weakly such that $\mu\{j\} \propto j^{-2\alpha}$ for $j \in \mathbb{N}$.
- Moreover, (G.6) $= \mathbb{E}_{X \sim \mu_d}\left[(\mathsf{K}_d(X^\alpha/t) - \mathbb{1}\{X^\alpha > t\})\mathbb{1}\{X \leq r_u\}\right]$.

Let $t \in \left((j-1)^\alpha, j^\alpha\right)$ for some $j \in \mathbb{N}$. For small enough $\delta > 0$, we have

$$\left| \mathbb{E}_{X \sim \mu_d}\left[(\mathsf{K}_d(X^\alpha/t) - \mathbb{1}\{X^\alpha > t\})\mathbb{1}\{X \leq r_u\}\right] \right|$$

$$= \mathbb{E}_{X \sim \mu_d}\left[|\mathsf{K}_d(X^\alpha/t) - \mathbb{1}\{X^\alpha > t\}|\mathbb{1}\{X \in [0, r_u]\}\mathbb{1}\{X^\alpha \notin \mathcal{C}_\delta\}\right] \overset{(b)}{=} o_d(1),$$

where we used both items in Lemma 10 for (b). $\qquad\square$

**Corollary 8.** *For $1 \geq c_d \gg \log^{-5} d$, we define*

$$g_d(\lambda, t) := \frac{-\lambda \exp(-t\lambda)}{1 - \exp(-t\lambda)} + \frac{\lambda^2 \exp(-t\lambda)}{(1 - \exp(-t\lambda))^2}\left(\frac{c_d}{t}\frac{r_s}{d} + \frac{\lambda \exp(-t\lambda)}{1 - \exp(-t\lambda)}\right)^{-1}$$

*Let $r_u \leq r$ and*

$$\kappa_{\text{eff}} := \begin{cases} r^\alpha, & \alpha \in [0, 0.5) \\ 1, & \alpha > 0.5 \end{cases}, \qquad \mathsf{T}_{\text{eff}} := \kappa_{\text{eff}} \log d/r_s.$$

*We have*

$$\frac{1}{\|\mathbf{\Lambda}\|_{\mathrm{F}}^2}\left(\sum_{j=1}^{r_u} g_d^2(\lambda_j; t\mathsf{T}_{\mathrm{eff}}) - \sum_{j=1}^{r_u} \lambda_j^2 \mathbb{1}\{\tfrac{1}{\lambda_j} > t\kappa_{\mathrm{eff}}\}\right) = o_d(1)$$

*for any fixed*

$$t \in \begin{cases} (0, \infty), & \alpha \in [0, 0.5) \\ (0, \infty) \setminus \{j^\alpha : j \in \mathbb{N}\}, & \alpha > 0.5. \end{cases}$$

*Proof.* We observe that

$$g_d(\lambda; t) = \lambda\left(1 - \frac{1}{1 - \exp(-t\lambda) + \frac{d}{r_s}\frac{\lambda t}{c_d}\exp(-t\lambda)}\right)$$

Therefore, we have $g_d^2(\lambda; t\mathsf{T}_{\mathrm{eff}}) = \lambda^2 F_d(\frac{1}{\lambda_j t\kappa_{\mathrm{eff}}})$. Then, by Proposition 36

$$\frac{1}{\|\mathbf{\Lambda}\|_{\mathrm{F}}^2}\left(\sum_{j=1}^{r_u} g_d^2(\lambda_j; t\mathsf{T}_{\mathrm{eff}}) - \sum_{j=1}^{r_u} \lambda_j^2 \mathbb{1}\{\tfrac{1}{\lambda_j} \geq t\kappa_{\mathrm{eff}}\}\right)$$

$$= \frac{1}{\|\mathbf{\Lambda}\|_{\mathrm{F}}^2}\sum_{j=1}^{r_u} \lambda_j^2\left(F_d(\tfrac{1}{\lambda_j t\kappa_{\mathrm{eff}}}) - \mathbb{1}\{\tfrac{1}{\lambda_j} > t\kappa_{\mathrm{eff}}\}\right) = o_d(1).$$

$\square$

