# OpenReview forum: "Learning quadratic neural networks in high dimensions: SGD dynamics and scaling laws"
_NeurIPS.cc/2025/Conference — NeurIPS 2025 poster_

### Official Review · Reviewer_u5Uk · 2025-06-25

**Clarity:** 3
**Significance:** 3
**Originality:** 4
**Rating:** 4
**Confidence:** 3

**Summary:**

This paper explores the optimization and sample complexity of training two-layer quadratic neural networks in high dimensions, focusing on the extensive-width regime where the teacher network width scales as $ r \asymp d^\beta $ $(\beta \in (0,1)$) and second-layer coefficients exhibit power-law decay $\lambda_j \asymp j^{-\alpha}$. It derives scaling laws for prediction risk, highlighting power-law dependencies on optimization time, sample size, and model width. Using matrix Riccati ODEs and novel matrix monotonicity techniques, it analyzes SGD dynamics, establishing convergence guarantees for feature learning in population and finite-sample settings, offering insights into neural scaling laws.

**Questions:**

No

**Ethical Concerns:**

["NO or VERY MINOR ethics concerns only"]

**Limitations:**

Yes

**Quality:**

3

**Strengths And Weaknesses:**

**Strengths**: The paper delivers a robust theoretical framework, elegantly connecting empirical scaling laws to rigorous mathematics through innovative matrix monotonicity and operator norm control, effectively addressing the complex extensive-width regime. It surpasses previous sample complexity bounds, especially in isotropic settings, and validates the additive model hypothesis, deepening understanding of feature learning dynamics. Its clear problem setup, thorough proofs, and relevance to practical neural network training underscore its value.

**weakness**:
1. The paper relies on idealized assumptions—uncorrelated Gaussian inputs, perfectly orthogonal teacher weights, and a strict power-law decay for second-layer coefficients—that often don’t match real-world data. I understand the challenges of theoretical research; given the complexity of analyzing practical models, it’s nearly impossible to directly characterize scaling laws for large-scale models. This isn’t a flaw in theoretical work. However, I believe the authors should delve deeper into the challenges of extending their findings to more realistic scenarios and candidly address the limitations of their current theoretical progress.

2. It’s also uncertain whether the matrix monotonicity approach applies to other activation functions like ReLU or optimization methods like Adam. The paper doesn’t explore the method’s scalability or broader applicability, which restricts its relevance to larger neural network problems.

---

> ### Author Rebuttal · Authors · 2025-07-30
>
> We thank our reviewer for constructive evaluation of our work, and address the reviewer's concerns below:
>
> ---
>
> * **Gaussian data:**  The assumption of i.i.d  Gaussian inputs is standard in the theoretical feature learning literature  (see [BAGJ21, MHPG+22, DLS22, BBSS22, AAM23, BBPV23, DKL+23,DNGL24, LOSW24], among others), as it enables analytic tractability through orthogonal polynomial expansions and concentration inequalities. While this assumption can often be relaxed via universality arguments (e.g. [Bruna et al, 2023]), a complete characterization of universality for SGD dynamics in multi-index setting remains an open and active area of research. On the other hand, our goal was to develop a rigorous arguments to reconcile scaling-law risk curves seen in practice with  high-dimensional feature learning setting, which was not very well-understood even with Gaussian inputs. Extending our results to more general input distributions is an important direction for future work.
> [Bruna et al. 2023] *On Single Index Models beyond Gaussian Data*.
> * **Strict power law decay assumption:** We stated our results under a specific power-law decay for analytical clarity and as it yields the cleanest form of the limiting risk curve. However, our main results, Theorems 1 and 2, extend to a broader class of second-layer coefficients under mild technical conditions on the asymptotic distribution of the coefficients. Under general decay rate condition, the scaling exponents in iteration and width (as shown in Table 1) remain unchanged, while the limiting values in Corollaries 1 and 2 may change. We will clarify this generality in the revised manuscript.
> * **Extension to other activations, such as ReLU:**
> While our analysis focuses on the Hermite-2 (i.e., quadratic) activation for both student and teacher networks, we expect that our monotonicity-based argument extends to more general link functions. This expectation is supported by the following arguments:
>    * For ReLU-like activations with nonzero first and second Hermite coefficients, it is known that SGD dynamics initially drive all neurons toward a degenerate rank-1 subspace, governed by the $IE = 1$ component. Once this component is learned, the Hermite-2 term dominates the dynamics, while higher-order terms remain negligible throughout the feature learning phase.
>
>    * Since the Hermite-2 dynamics exhibit monotonicity, the training trajectory for general link functions can be tightly sandwiched between the dynamics of a purely quadratic network plus/minus some vanishing terms due to higher order terms, which allows the general dynamics to closely track the quadratic case in high dimensions.
>
>    * To validate this, we conducted experiments using teacher and student networks with ReLU and two other non-quadratic activations. The results, summarized in the table below, show that the empirical scaling exponents for these activations align closely with our theoretical predictions for the quadratic case.
>
> *   **Perfectly orthogonal teacher weights:**
> We can make the assumption of orthogonal teacher weights without loss of generality in our setting, as we consider quadratic activations. For any teacher directions $(v_j)\_{j = 1}^r \subset \mathbb{R}^d$ and positive coefficients $(c_j)\_{j=1}^r$, the label can be written as:
>
> $$
> y = \frac{1}{\sqrt{\mathrm{Var}\left( \sum_{j=1}^r c_j \langle v_j, x \rangle^2 \right)}}
> \left( \sum_{j=1}^r c_j \langle v_j, x \rangle^2 -
> \mathbb{E}\left[ \sum_{j=1}^r c_j \langle v_j, x \rangle^2 \right] \right), \quad x \sim \mathcal{N}(0, I_d).
> $$
>
> In matrix notation:
>
> $$
> y = \frac{1}{\sqrt{\mathrm{Var}(x^\top V C V^\top x)}}
> \left( x^\top VC V^\top x - \mathbb{E}[x^\top V CV^\top x] \right),
> $$
>
> where \$V \in \mathbb{R}^{d \times r}\$ has columns \$v\_j\$ and \$C = \mathrm{diag}(c\_1, \ldots, c\_r)\$.
>
> By the spectral theorem, there exists an orthonormal matrix \$\Theta \in \mathbb{R}^{d \times r}\$ and a diagonal matrix \$\Lambda = \mathrm{diag}(\lambda\_1, \ldots, \lambda\_r)\$ such that:
>
> $$
> \Theta \Lambda \Theta^\top = V C V^\top.
> $$
>
> This implies that the label distribution is equivalent to:
>
> $$
> y = \frac{1}{\sqrt{2} \lVert \Lambda \rVert_F} \sum_{j=1}^r \lambda_j ( \langle \theta_j, x \rangle^2 - 1 ),
> $$
>
> using the identity \$\mathrm{Var}(x^\top \Theta \Lambda \Theta^\top x) = 2 \lVert \Lambda \rVert_F^2\$.
>
> Thus, any label generated using arbitrary directions and weights can be rewritten using orthogonal teacher directions without loss of generality -- we will include a remark in the revision. In the manuscript, we normalize the label variance to 2 and omit the \$\sqrt{2}\$ factor for notational convenience.
> *  **Adam and Larger Neural network problems:** The theoretical understanding of feature learning is still in its early stages, and the analysis of adaptive optimization methods such as Adam, as well as deeper architectures with realistic training dynamics, remains an open challenge. Extending our findings to these settings is an interesting direction for future work. We remark that our goal is not to model all practical complexities, but to provide a rigorous and analytically tractable framework for understanding fundamental phenomena—specifically, smooth scaling-law behavior in the presence of emergent feature learning risk curves. To our knowledge, SGD dynamics in the extensive-width regime is poorly understood prior to our work, even under these idealized conditions. We believe that our analysis (going beyond the "narrow-width" regime) represents significant theoretical progress in this direction.
>
> ---
>
> ### Additional Experimental Results Demonstrating Broader Applicability
>
> To demonstrate the broader applicability of our theoretical results beyond the quadratic activation function, we conducted additional experiments using three alternative activation functions: ReLU, absolute value, and $x + H_{e_2}(x)$. In each case, we considered **population gradient descent dynamics** with the **centered version** of the activation (i.e., subtracting its mean), and plotted the corresponding **risk curve**—risk versus time on a log-log scale—following the format of Figure 1(b) in the main paper.
>
> We considered the setting $d = 5000$, $r = 2400$, $r_s \in [128, 215, 362, 608, 1024]$, and ran each experiment for $T = 3 \times 10^5$ iterations. For each activation and width, we fitted a line to the decay phase of the risk curve to estimate the corresponding time exponent, and reported the resulting range alongside the theoretical prediction of Theorem 1. The results are summarized below:
>
> | Activation Function      | $\alpha = 0.8$  | $\alpha = 1.0$  | $\alpha = 1.5$  |
> | ------------------------ | --------------- | --------------- | --------------- |
> | $H_{e_2}$ (Theoretical)  | -0.75           | -1              | -1.34           |
> | $H_{e_2}$ (Experimental) | \[-0.83, -0.81] | \[-1.06, -1.04] | \[-1.36, -1.23] |
> | ReLU                     | \[-0.82, -0.72] | \[-1.02, -0.92] | \[-1.35, -1.27] |
> | Absolute Value           | \[-0.73, -0.71] | \[-0.99, -0.94] | \[-1.27, -1.23] |
> | $x + H_{e_2}(x)$         | \[-0.87, -0.84] | \[-1.11, -1.01] | \[-1.33, -1.31] |
>
> **Observations:**
>
> * The predictions of our theory continue to hold across the activation functions considered. Since we are unable to include figures in the rebuttal, we describe the qualitative behavior of the plots:
>
>   * For activations with **information exponent (IE) = 2**—i.e., $H_{e_2}$ and absolute value—the risk curves exhibit a clean power-law decay similar to Figure 1(b) in the paper.
>
>   * For activations with **IE = 1**—i.e., ReLU and $x + H_{e_2}(x)$—we observe an initial sharp drop in the early stages of training, followed by a short plateau, and then a decay phase that closely resembles that of the IE = 2 case.
>
>   * In all cases, the quality of the linear fit during the decay phase is comparable to that in Figure 1(b). For $x + H_{e_2}(x)$ and $H_{e_2}$, the risk curve flattens toward the end, replicating the behavior observed in Figure 1(b). For ReLU and absolute value, the slope decreases and the risk continues to decrease at a slower rate.
>
>
> ---
> We would be happy to answer any follow-up questions in the discussion period.

---

> > ### Comment · Area_Chair_MCZd · 2025-08-08
> >
> > Dear reviewer,
> >
> > as the end of the discussion period is imminent, please let the authors know what is your opinion about the content of the rebuttal if possible, bearing in mind that time for an extensive reply on the authors' side is very limite.
> >
> > Many thanks.
> >
> > My best regards
> >
> > the AC

---

### Official Review · Reviewer_6WHe · 2025-06-30

**Clarity:** 4
**Significance:** 3
**Originality:** 2
**Rating:** 5
**Confidence:** 4

**Summary:**

The authors study GD of the population loss on two layer networks with $x^2-1$ activation and orthonormal weights in the first layer, learning a similar 2-layer architecture. They then proceed to use the closed form solution of the dynamic to decompose the risk into a sum of independent contributions, each given by a hidden unit. This result allows the authors to derive some neural scaling laws. These results hold both for gradient flow and discrete-step GD

**Questions:**

1. Would you be able to consider the case of arbitrary covariance using the same proof technique?
2. Would the derivation change if you were to consider an activation $x + {\rm He}_2(x)$?
3. Do you expect the power-law to change if you were allowed to use the data for more than one GD iteration?
4.Minor issue: could you explicitly define $\land$ and $\lor$ as min and max somewhere in the paper?

**Ethical Concerns:**

["NO or VERY MINOR ethics concerns only"]

**Final Justification:**

The authors addressed all of my concerns, and I raised my score accordingly.

**Limitations:**

Yes

**Paper Formatting Concerns:**

No concerns

**Quality:**

2

**Strengths And Weaknesses:**

The paper is clear and well written. The proof is elegant, although not very technically innovative. The scaling laws, which constitute in my opinion the main result of this paper, are intuitively reasonable, but I would have appreciated extensive numerical simulations to verify them empirically. While the scaling laws itself are not particularly surprising, there are not exact results on learning linear width networks, so I believe the contribution is valid. I think the paper would be stronger if it also tackled the case of generic covariance and label noise, but even as is it's in my opinion a strong paper.

---

> ### Author Rebuttal · Authors · 2025-07-30
>
> We thank our reviewer for constructive evaluation of our work, and address the reviewer's concerns below:
>
> ---
>
> * **Novelty of our proof technique:**
> We emphasize that the proof techniques developed in all prior works do not extend to our setting, due to two key challenges:
>
>    * *Extensive-width regime:* In this regime, the SGD trajectory cannot be captured by finite-dimensional  *effective dynamics* [BAGJ21]. As a result, existing techniques that control the Frobenius norm of deviations around the population flow are insufficient. Instead, we establish *operator norm* bounds on the discretization error, requiring significantly tighter control over the discrete-time dynamics than in prior work.
>
>     * *Information exponent \(k = 2\):* In our setting, the dynamics cannot be reduced to decoupled systems, unlike the higher-$k$ cases studied in [OSSW24, SBH24, Ren et. al 2025]. This prevents reduction to independent single-index problems, which are central to several recent analyses.
>
>    * To overcome these challenges, we introduce an new proof technique based on the *monotonicity of matrix Riccati ODEs*. This approach departs significantly from the standard Gronwall-type arguments used to control overlap statistics [BAGJ21]. We believe our technique is of independent interest and may serve as a foundation for future work in the extensive-width setting.
>
> *  **Label noise:** Our results remain valid in the presence of label noise, since (i) it does not affect the population gradient flow, and (ii) the SGD analysis can be extended by incorporating additional martingale terms with minor modifications (see e.g., [DNGL24]). We focused on the noiseless case for mathematical convenience. However, in light of the reviews, we will include the label noise case in the final submission.
> *  **Arbitrary Covariance:** Our proof technique does not directly extend to inputs with arbitrary covariance, as the Riccati ODE structure no longer holds when the input covariance differs from the identity. However, understanding SGD dynamics under nontrivial covariance is an active area of research. Our primary goal was to establish the emergence of smooth scaling-law behavior within a well-studied feature learning setting—something previously unclear due to the theoretical challenges outlined in point 1 even for identity covariance. Extending our results to more general input distributions is a valuable direction for future work.
> * **Dynamics for $x + \mathrm{He}_2(x)$.**  We believe our analysis can be extended to the activation $x + \mathrm{He}_2(x)$, and more generally to any activation with nonzero $\mathrm{He}_1$ and $\mathrm{He}_2$ components. However, the corresponding proofs would require more involved derivations, making an already long paper even longer—without introducing substantial new technical insights. The overall structure of the argument closely follows the incremental learning dynamics studied in the multi-index setting [Abbe et. al 2023, Bietti et. al 2023]:
>
>    * For the activation $x + \mathrm{He}_2(x)$, the student network first aligns with the teacher's $\mathrm{He}_1$ component, resulting in a rapid initial drop in the risk curve. Once this mode is learned, the $\mathrm{He}_2$ component dominates the dynamics (while higher-order terms remain negligible for general activations).
>
>    * Because the Hermite-2 dynamics exhibit monotonicity, the training trajectory for general link functions can be tightly bounded between those of a purely quadratic network, up to vanishing perturbations from higher-order terms. This suggests that in high dimensions, the learning dynamics for such activations will closely track the quadratic case.
>
>    * To support our argument, we conducted experiments using teacher and student networks with $x + \mathrm{He}_2(x)$, ReLU and  absolute value activations. The results ( summarized  below) validate our argument above, and also show that the empirical scaling exponents for these activations align closely with our theoretical predictions for the quadratic case.
>
> * **Scaling exponents for offline dynamics.** Thank you for the interesting suggestion. We study the one-pass dynamics where the optimization time can be directly connected to the statistical efficiency (i.e., the data component in the scaling law), as done in prior theoretical analysis of scaling laws [PPXP24] [LWK+24]. If data reuse is allowed, then the optimization and sample complexity needs to be discussed separately, and we expect that multi-pass SGD can improve the data scaling exponent, as previously shown in the linear setting (see e.g., [Pillaud-Vivien et al. 2018]).
> [Pillaud-Vivien et al. 2018] *Statistical Optimality of Stochastic Gradient Descent on Hard Learning Problems through Multiple Passes*.
>
> * *Minor:* We will define the symbols $\wedge$ and $\vee$ explicitly.
>
> ---
>
> ### Additional Experimental Results Demonstrating Broader Applicability
>
> To demonstrate the broader applicability of our theoretical results beyond the quadratic activation function, we conducted additional experiments using three alternative activation functions: ReLU, absolute value, and $x + H_{e_2}(x)$. In each case, we considered **population gradient descent dynamics** with the **centered version** of the activation (i.e., subtracting its mean), and plotted the corresponding **risk curve**—risk versus time on a log-log scale—following the format of Figure 1(b) in the main paper.
>
> We considered the setting $d = 5000$, $r = 2400$, $r_s \in [128, 215, 362, 608, 1024]$, and ran each experiment for $T = 3 \times 10^5$ iterations. For each activation and width, we fitted a line to the decay phase of the risk curve to estimate the corresponding time exponent, and reported the resulting range alongside the theoretical prediction of Theorem 1. The results are summarized below:
>
> | Activation Function      | $\alpha = 0.8$  | $\alpha = 1.0$  | $\alpha = 1.5$  |
> | ------------------------ | --------------- | --------------- | --------------- |
> | $H_{e_2}$ (Theoretical)  | -0.75           | -1              | -1.34           |
> | $H_{e_2}$ (Experimental) | \[-0.83, -0.81] | \[-1.06, -1.04] | \[-1.36, -1.23] |
> | ReLU                     | \[-0.82, -0.72] | \[-1.02, -0.92] | \[-1.35, -1.27] |
> | Absolute Value           | \[-0.73, -0.71] | \[-0.99, -0.94] | \[-1.27, -1.23] |
> | $x + H_{e_2}(x)$         | \[-0.87, -0.84] | \[-1.11, -1.01] | \[-1.33, -1.31] |
>
> **Observations:**
>
> * The predictions of our theory continue to hold across the activation functions considered. Since we are unable to include figures in the rebuttal, we describe the qualitative behavior of the plots:
>
>   * For activations with **information exponent (IE) = 2**—i.e., $H_{e_2}$ and absolute value—the risk curves exhibit a clean power-law decay similar to Figure 1(b) in the paper.
>
>   * For activations with **IE = 1**—i.e., ReLU and $x + H_{e_2}(x)$—we observe an initial sharp drop in the early stages of training, followed by a short plateau, and then a decay phase that closely resembles that of the IE = 2 case.
>
>   * In all cases, the quality of the linear fit during the decay phase is comparable to that in Figure 1(b). For $x + H_{e_2}(x)$ and $H_{e_2}$, the risk curve flattens toward the end, replicating the behavior observed in Figure 1(b). For ReLU and absolute value, the slope decreases and the risk continues to decrease at a slower rate.
>
> ---
> We would be happy to answer any follow-up questions in the discussion period.

---

> > ### Comment · Reviewer_6WHe · 2025-08-02
> >
> > Thanks for the detailed response. I am glad that you performed some numerical experiments. I still have some doubts.
> >
> > 1. From your experiments it seems like there is good agreement between theory and experiments for $\alpha > 1$. The same doesn't seem to hold for $\alpha < 1$. Can you justify the discrepancy?
> >
> > 2. Thanks for the extensive comparison with existing literature. I am under the impression that [MBB23] also maps the problem to a matrix Riccati ODE, which they then manipulate in different ways. Could you elaborate more on the specific novelty of manipulating the matrix Riccati ODE?
> >
> > 3. More on 2. You say that your technique paves the way for future work in extensive width settings. Do you mean while keeping the activation He2? Or do you mean that you believe you can use similar tools for other activations (and in particular activations with degree larger than 2)?

---

> ### Author Response · Authors · 2025-08-03
>
> Thank you for initiating the discussion. We address the questions below.
>
> 1. The discrepancy between the empirical and theoretical slopes is due to the finite-width truncation error of the infinite power-law sum, which causes the finite-dimensional risk curve to have a steeper estimated slope than the idealized infinite-dimensional limit (and the error diminishes as the network size increases). Such finite-width effect is known in the scaling law literature: for example, similar empirical finding is reported in the concurrent work [Ren et. al, 2025], see second bullet point in page 9. The truncation error becomes more apparent in the slope  $m(\alpha) = \frac{1 - 2 \alpha}{\alpha}$ for smaller $\alpha$ because the derivative $m^\prime(\alpha) = -\frac{1}{\alpha^2}$ is larger, therefore, the slope is more sensitive to error. In particular, for $\alpha = 0.8$ experiment, we observed $m(\hat{\alpha}) = [-0.87, -0.7]$, which corresponds to $\hat{\alpha} \in [0.77, 0.88]$, which is reasonably close to the theoretical value $\alpha = 0.8$. Note that for $\alpha = 1$ and $\alpha = 1.5$, we observe $m(\hat{\alpha}) = [-1.11, -0.92]$ and $m(\hat{\alpha}) = [-1.36, -1.31]$, which correspond  to $\hat{\alpha} \in [0.93, 1.13]$ and  $\hat{\alpha} \in [1.45, 1.56]$, which gives similar error range with that of $\alpha  = 0.8$.
>
> 2. Indeed, [MBB23] considers the population gradient flow for quadratic networks which corresponds to a similar matrix Riccati ODE. However, their analysis provides only an implicit characterization of the risk trajectory and is tractable only in certain special cases. For example, they are able to analyze convergence time only when the teacher’s output layer weights are all equal, and even then, discretization of the dynamics is not studied.
> In contrast, our analysis exploits **the monotonicity structure of the matrix Riccati equation** — we are not aware of the use of similar structure in the neural network theory literature. This allows us to:
>    * derive explicit asymptotic limits for general decay rates $\alpha \geq 0$,
>    * analyze the behavior across the full training trajectory, and
>    * extend our results to the stochastic regime by discretizing the dynamics under SGD.
>
> 3. Yes, we believe that our analysis can be extended to other activations with nonzero He2 component — including ReLU, GeLU, absolute value, and higher-order polynomials ($x^4$ or $x^3 + Cx^2$, etc.) with a more involved analysis (outlined in our first response), as also supported by our empirical findings. Intuitively speaking, SGD dynamics is governed by the *harmonic decomposition* of the target function into Hermite modes (assuming Gaussian inputs):  $\mathrm{He}_1$, $\mathrm{He}_2$, and higher. The *information exponent (IE)* refers to the lowest-degree nonzero Hermite component.
> It is well-established that SGD dynamics in the feature learning regime is governed by the lowest nonzero Hermite component of the target function for an extended period of time [BAGJ21]. When there are multiple directions to be learned, the $\mathrm{He}_1$ term gives a degenerate rank-1 subspace, whereas $\mathrm{He}_2$ (the next leading term, which is the focus of our work) is the first nontrivial harmonic that captures the multi-dimensional structure of the target function (see e.g., [DLS22]); hence for different activation functions (e.g., with higher degrees), our analysis on the quadratic term captures the most dominant low-dimensional signal after the initial loss drop from $\mathrm{He}_1$. On the other hand, until the $\mathrm{He}_2$ component is learned, higher-order Hermite modes are expected to have vanishing contribution to the SGD dynamics due to their rapidly decaying gradient magnitudes.
> We note that the $\mathrm{IE} \geq 3$ setting was recently studied in a concurrent work [Ren et. al, 2025], which adopts a *decoupling approach* parallel to [OSSW2024, SBH2024]. However, this approach fails in the $\mathrm{IE} = 2$ case, the exact characterization of which (beyond isotropic teacher weights) was an open problem until our work — see our response to reviewer s1U3 for more discussions.
>
> ---
> We would be happy to answer any follow-up questions in the discussion period.

---

> > ### Comment · Reviewer_6WHe · 2025-08-05
> >
> > Thanks for clarifying these points. I will raise my score accordingly

---

### Official Review · Reviewer_s1U3 · 2025-07-07

**Clarity:** 2
**Significance:** 2
**Originality:** 3
**Rating:** 3
**Confidence:** 3

**Summary:**

This is an interesting paper with delicate theoretical results on feature learning and risk. The set-up assumes that the teacher model is a two-layer network with a quadratic activation function and with orthogonal features and algebraically decaying coefficients. It is about the extensive-width regime when the number of student model neurons is growing in specific ways with the dimension d of the data and the sample or observation size from the teacher model.

**Questions:**

Questions:
“equivalently” in line 147 does not imply that the gradient dynamics are equivalent? The authors should clarify that this refers to an equivalent representation of the student network, not necessarily equivalent gradient dynamics.

“which is negligible relative to the feature learning cost.” in line 207 only holds for d>>r_s? This should be emphasized by the authors.

Comments:

You mention scaling laws, but you could be more explicit about how the paper attempts to provide a theoretical underpinning for the empirically observed "neural scaling laws" in large language models. This is a central motivation for the work.

Conclusion:
The paper could be very interesting for a small number of highly specialized scientists working on very specific types of theory for shallow neural networks with quadratic activation functions that learn data generated from shallow neural networks with a quadratic activation function. However, this paper does not meet the required broad impact for NeurIPS and suffers from poor presentation. We highly recommend submitting this work to a more specialized venue where people working on similar topics appreciate this technically impressive but very niche result.

**Ethical Concerns:**

["NO or VERY MINOR ethics concerns only"]

**Final Justification:**

The submission adds a possibly valuable contribution to scientific progress as it pushes the frontier of a specific sub-field (quite fine-grained understanding of simplified training dynamics under quite restrictive assumptions) one step further. While this result does not give any significant insight to practitioners now, there is a chance that other theoreticians will be able to build up on this result to get more valuable insights in the future. This hope for interesting extensions gives the paper its value.

We appreciate the added experiments for ReLU activations. These experiments increase the hope for future progress in this line of work.

The downside of this submission is that most NeurIPS attendees probably won’t be able to obtain any really helpful insights from this paper. The paper by itself doesn’t tell any really exciting story, but rather feels like a challenging intermediate step in a long sequence of steps that might lead to some new insights in the future. However, a few other scientists might be interested in trying to extend this result to more meaningful results (or maybe get inspired by this point of view).

This paper should definitely be published somewhere. I argue that these results are better suited for a more specialized venue. I understand that other reviewers might think differently about the suitability of this paper for NeurIPS. However, I still believe that NeuriPS papers in deep learning theory should demonstrate more relevance for practice for the broad attendees of NeurIPS. That said, I would respect the AC’s final decision on the paper.

**Limitations:**

This work does not discuss limitations. It would be useful to add a discussion section to the main paper.

**Paper Formatting Concerns:**

No formatting concerns

**Quality:**

3

**Strengths And Weaknesses:**

Strengths:

Non-trivial mathematical results

The feature-learning setting is indeed more interesting than the lazy regime.

Better understanding of SGD dynamics and scaling laws is important

The "matrix-monotone comparison framework" might be of independent interest. However, I don’t have the expertise to evaluate this contribution and I don’t see a convincing story-line for whom this might be of interest.

Weaknesses:

The theoretical results seem to be solid, but they are asymptotic as d tends to infinity. More clear motivations for the coefficient decay assumption are necessary. How about other decay rates?  It would be insightful to see some simulation studies about the convergence rates of these asymptotic results, and under the scenario that the teacher model is observed with noise.

A key weakness of the paper is its reliance on a series of strong and specific assumptions that, while enabling the theoretical analysis, distance the results from practical applications. These assumptions include the following: Quadratic activation function (polynomial activation functions are not very popular activations functions in practice, partially because they violate the universal approximation theorem and because their gradients can explode). Also the theory only covers shallow neural networks with only 1 hidden layer. Additionally the gradient dynamics were simplified (as commonly done in theoretical literature). Additionally infinitely large training datasets are assumed. Additionally the limit $d\to\infty$ is taken. (Additionally the result does not cover transformers or convolutional NNs.) Additionally you assume that the true data generating process and the model you are training have an extremely similar structure. Additionally you assume orthogonal features, specific coefficient decay, etc.. These are multiple very specific assumptions which make the result very niche.
There are no empirical experiments with real-world architectures using ReLU or, GeLU or any non-polynomial activation function. In practice these assumptions are (almost) never met, so the paper is mainly motivated by the hope that this theory can give insights beyond its very restricting assumptions.

The presentation needs to be improved so the main take-home messages are clearer and more crisp, especially for practitioners. What does a practitioner learn from this paper that they can use to guide their practice?  How does one empirically check that the assumed teacher model captures reality for any real data sets? The paper is hard to follow. There are multiple notation-heavy mathematical results, but not a very clear storyline.

Moreover, the connections and differences between the submission and other prior works need to be made concise and clear. What precise and insightful advances have been made in this paper relative to existing works? E.g., Emergence and scaling laws in SGD learning of shallow neural networks by Yunwei Ren, Eshaan Nichani, Denny Wu, Jason D. Lee (https://arxiv.org/abs/2504.19983) seems highly related and was not cited. This preprint was only uploaded a few days before the submission deadline, so if you did not follow oral talks of this paper multiple months ago, not citing this paper is OK. Under this very specific assumptions, you might be the first to have proven this very specific result, but the question is if you have derived any impactful new insights.  The main novelty of your work seems to be the information exponent =2. A general result for an information exponent larger than 2, seems much more interesting to me than a result which only holds under your extremely restricting sets of assumptions.

---

> ### Author Rebuttal · Authors · 2025-07-31
>
> We thank the reviewer for the detailed feedback, and address the concerns below.
>
> ---
>
> ### Overlap with [Ren et al., 2025]
> Thank you for pointing out this concurrent work, which was posted to arXiv two weeks before the NeurIPS deadline. While both works address feature learning in the extensive-width regime, our results are  **complementary**  and tackle an entirely different set of theoretical challenges. We summarize the key differences below:
>
> *  **Information exponent regime:**  [Ren et al., 2025] focuses on activation functions with information exponent $IE \geq 3$, similar to the assumptions in [OSSW24] and [SBH24]. All these works rely on a decoupling argument that explicitly fails when $IE = 2$ (see the remark on page 5 of [Ren et al.]). In contrast, our work addresses the quadratic case, which models $IE = 2$. We note that extensive width setting for $IE = 2$ has been recognized as an open problem in the literature (see [OSSW24][MTM+24]).
> * **Practical relevance of $IE = 2$:** Beyond its theoretical difficulty, the $IE = 2$ regime is arguably more aligned with practical settings  than the $IE > 2$ cases.  The Hermite-2 component is the first nontrivial harmonic that captures multi-dimensional structure in the data (see e.g., [DLS22]). As such, it represents the most dominant lower dimensional signal component and thus plays a central role in the feature learning process. Moreover, commonly used activations such as ReLU and GeLU have nonzero first and second Hermite coefficients, and it is well-known that higher-order Hermite components contribute less due to their smaller gradient magnitudes.
> * **Effective reduction for ReLU-like activations:**
> For ReLU-like activations with nonzero first and second Hermite coefficients, it is known that SGD dynamics initially drive all neurons into a degenerate rank-1 subspace, governed by the $\mathrm{He}_1$ component (i.e., $IE = 1$). Once this mode is learned, the $\mathrm{He}_2$ component becomes dominant, while higher-order terms remain negligible throughout the feature learning phase. This behavior leads to an *effective reduction* to the $IE = 2$ setting we analyze.
> As a result, we expect ReLU training dynamics to exhibit a rapid initial drop in risk (driven by $\mathrm{He}_1$), followed by a power-law decay governed by the $\mathrm{He}_2$ dynamics. To support this, we conducted experiments with ReLU-based teacher and student networks. The results, summarized below, are consistent with this two-phase description and show that the empirical scaling exponents for ReLU closely match our theoretical predictions.
>
> *  **Sharp guarantees and broader coverage:**  To resolve the challenging $IE = 2$ case (where decoupling fails), we develop a new monotonicity-based analysis that precisely characterizes the Riccati dynamics of learning. Compared to [Ren et al., 2025], our approach yields:
>   (i) *sharper sample complexity:*, nearly matching information-theoretic lower bounds in several regimes (in contrast, [Ren et al.] does not provide sharp rates in the width dependence), and
>   (ii) *stronger guarantees  over a wider range of teacher widths $\alpha \in [0, 1)$*, whereas [Ren et al.] imposes a more restrictive assumption $\alpha < 0.1$.
>
> ---
>
> ### Clarifications and Additional Responses
> * **Finite $d$ and asymptotics:** Our main results—Theorems 1 and 2—hold for finite $d$ with high probability, provided $d$ exceeds a certain finite threshold. Corollaries 1 and 2 are stated in the asymptotic regime ($d \to \infty$) to highlight the analytically clean form of the limiting risk curve.
> * **Strict power-law decay assumption:**
> We stated our results under a specific power-law decay for analytical clarity and as it yields the cleanest form of the limiting risk curve. However, our main results, Theorems 1 and 2, extend to a broader class of second-layer coefficients under mild technical conditions on the asymptotic distribution of the coefficients. Under general decay rate condition, the scaling exponents in iteration and width (as shown in Table 1) remain unchanged, while  the limiting values in Corollaries 1 and 2 may change. We will clarify this generality in the revised manuscript.
> * **Label noise:** Our results remain valid in the presence of label noise, since (i) it does not affect the population gradient flow, and (ii) the SGD analysis can be extended by incorporating additional martingale terms with minor modifications (see e.g., [DNGL24]). We focused on the noiseless case for mathematical convenience. However, in light of the reviews, we will include the label noise case in the final submission.
> * **General activations functions, e.g., ReLU, GeLU:** See above.
> * **Assumptions on data-generating process and teacher network model:**
> We consider a teacher network with a low-dimensional hidden structure—a standard and widely used model for studying feature learning in neural networks (see [BAGJ21, MHPG+22, DLS22, BBSS22, AAM23, BBPV23, DKL+23,DNGL24, LOSW24], among others). While this model is admittedly simpler than real-world datasets, it offers a theoretically tractable framework that has proven essential for understanding key aspects of feature learning. Moreover, such simplified settings often serve as the foundation for more complex scenarios, including nonlinear feature structures [Nichani et. al 2023] and deeper architectures [Dandi et. al 2025].
> In addition, we note that in the quadratic setting the orthogonality of the target features can be assumed without loss of generality due to the spectral theorem — see our response to reviewer u5Uk. We will add a remark in the revision.
> * **Choice of student network:**
> Using student networks with the same activation function is an idealized but widely adopted starting point for theoretical analysis. Proof techniques for mismatched teacher-student settings [BBSS22, MHWSE23] typically build on those developed for the well-specified case [BAHJ21, DNGL23]. While the mismatched setting is certainly important for bridging theory and practice, it should naturally come after establishing a clear understanding of the well-specified model.
> * **General architectures, e.g.,  Transformers, and deeper networks:** The teacher-student framework we adopt has been extended in prior work to more complex architectures such as CNNs and Transformers [Arnaboldi et. al 2025] or deep MLPs  [Dandi et. al 2025], with the core techniques often relying on the well-specified setting we study. We reiterate that our work addresses a foundational open problem in this setting. As such, it lays important groundwork for future theoretical developments in more general architectures.
>
> ---
>
> ### Relevance and scope of our work:
> We respectfully disagree with the reviewer’s assessment regarding the narrow scope and limited relevance of our work. We address this concern below:
>
> * The main critique appears to stem from the *idealized assumptions* under which our results are derived. We note that most of these assumptions are **standard in the theoretical feature learning literature** and have been adopted in numerous NeurIPS papers (e.g., [BES+22, BBSS22, MZD+23, DNGL23, LOSW24, MTM+24], to list a few). Our goal is not to model all practical complexities, but to provide a *rigorous and analytically tractable framework* for understanding fundamental phenomena—specifically, smooth scaling-law behavior in the presence of emergent feature learning risk curves. To our knowledge, SGD dynamics in the extensive-width regime had not been theoretically understood prior to our work (and the concurrent work by Ren et al.), even under idealized conditions. We believe that our analysis (going beyond the "narrow-width" regime) represents significant theoretical progress in this direction.
>
> * These assumptions **simplify but do not trivialize** the problem. For example:
>
>   * The *teacher-student framework* is widely used to model low-dimensional structure in the data and underpins many foundational works.
>   * The use of *polynomial activations* is a common simplification (e.g., [LOSW24]), justified by the well-established fact that the lowest-degree harmonic component governs sample complexity in the feature learning regime. In particular, quadratic activations have received significant attention due to their analytic tractability and relevance in feature learning regime (see Martin et al. 2023, Maillard et al. 2024, Erba et al. 2025).
>   * The *matched activation* setting is another standard assumption that avoids technical complications from the link mismatch. Since the $IE = 2$ regime remained unresolved even in this setting, we believe it is both natural and necessary to begin with this baseline.
>
> In light of these points, we kindly ask the reviewer to reconsider their assessment of the *relevance and contribution* of our work to the NeurIPS community.
>
> ---
>
> ### Additional Experimental Results Demonstrating Broader Applicability
>
> To demonstrate the broader applicability of our theoretical results beyond the quadratic activation function, we conducted additional experiments using three alternative activation functions: ReLU, absolute value, and $x + H_{e_2}(x)$.  The results are summarized below. Due to the character limit, we kindly refer our reviewer  to see our response to reviewer u5Uk for the discussion.
>
> | Activation Function      | $\alpha = 0.8$  | $\alpha = 1.0$  | $\alpha = 1.5$  |
> | ------------------------ | --------------- | --------------- | --------------- |
> | $H_{e_2}$ (Theoretical)  | -0.75           | -1              | -1.34           |
> | $H_{e_2}$ (Experimental) | \[-0.83, -0.81] | \[-1.06, -1.04] | \[-1.36, -1.23] |
> | ReLU                     | \[-0.82, -0.72] | \[-1.02, -0.92] | \[-1.35, -1.27] |
> | Absolute Value           | \[-0.73, -0.71] | \[-0.99, -0.94] | \[-1.27, -1.23] |
> | $x + H_{e_2}(x)$         | \[-0.87, -0.84] | \[-1.11, -1.01] | \[-1.33, -1.31] |
>
>
>
>
> ---
>
> We would be happy to answer any follow-up questions in the discussion period.

---

> > ### Comment · Reviewer_s1U3 · 2025-08-05
> >
> > Thank you for your response and for the additional experiments.
> >
> > My main concern remains the lack of sufficient evidence for the practical relevance of this delicate refinement of existing theory.
> >
> > The rebuttal emphasizes the novelty of the analysis in the "extensive-width" regime (going beyond the "narrow-width" regime). However, the claim that the dynamics were not understood "even under idealized conditions" prior to this work should be carefully contextualized. A significant body of work, notably the Tensor Programs framework by Yang et al., has already made major theoretical advances in understanding feature learning dynamics beyond the narrow/lazy regime (in the infinite-width limit). Their work has a broader scope (e.g., arbitrary architectures) and has led to demonstrable practical impact. The current paper analyzes a highly specific setting (quadratic activation, shallow architecture for teacher and student with equal activation function). While this is a valid technical contribution, its practical significance relative to prior breakthroughs in feature learning theory is not made clear. For example, for a given DL model used in practice, it is unclear whether it belongs to the extensive-width regime or the muP regime for practitioners to find the relevant theory to seek guidance and insights.
> >
> > Ultimately, assessing the future impact of such a specialized theoretical result involves a degree of subjective judgment. I admit that I am not an expert in the literature on extensive-width limits specifically. It is possible that this work may lay the foundation for future, more impactful results, and perhaps a reviewer with deeper expertise in this niche will see that more clearly.
> >
> > From my perspective, however, the combination of extremely restrictive assumptions and the failure to convincingly present the paper's broader significance leads me to maintain my original score. I want to be clear that this is just one perspective. I trust the judgment of the other reviewers and the Area Chair in making the final decision, especially since assessing the long-term impact of such specific theoretical work is ultimately a subjective matter.

---

> ### Author Response · Authors · 2025-08-06
>
> We thank the reviewer for the follow-up comment.
> We would like to emphasize that our work is motivated by a purely theoretical perspective, and the setting we study lies at the boundary of what is currently under precise analytical control in deep learning theory. As highlighted in our previous response, a complete analysis of the He2 component plays a central role in understanding the training dynamics of widely-used activation functions such as ReLU (due to the harmonic decomposition of the loss landscape); we have also outlined how our framework plausibly extends to such settings (see our response to Reviewer 6WHe for additional details).
>
> While we respect the reviewer's practice-oriented perspective, we respectfully disagree with the assertion that our contribution lacks value due to the “lack of sufficient evidence for practical relevance.” By the same reasoning, foundational works such as the first few Tensor Programs (TP) papers should not be dismissed simply because they addressed only the kernel at random initialization. Moreover, the assumptions used in our analysis are widely accepted in the literature and consistent with other theoretical studies in this area.
>
> We would also like to clarify a few points regarding the reviewer’s comment that “*the Tensor Programs framework by Yang et al. has already made major theoretical advances in understanding feature learning dynamics beyond the narrow/lazy regime*”:
>
> - The TP framework establishes the existence of a feature-learning limit in a specific scaling regime where only the network width diverges, with fixed input dimension and training horizon. In contrast, our work analyzes an alternative high-dimensional regime where model width and/or training horizon scale with input dimension. This regime has attracted growing interest, and recent works have leveraged tools from random matrix theory to characterize SGD dynamics under idealized conditions [BES+22, BAGJ22]. We view the exploration of such alternative regimes as mathematically motivated and valuable.
>
> - More importantly, while the existence of a well-defined muP limit is a meaningful conceptual advance, it does not imply analytic tractability of the feature learning dynamics. Even in simple settings (e.g., shallow nonlinear networks with well-specified targets), the muP dynamics remain difficult to solve, and most prior results rely heavily on numerical simulation.
> Consequently, existing works on TP do not provide meaningful statistical rates (i.e., how many samples are required to learn a target function) and sharp computational guarantees (i.e., how many neurons are required to track the muP limit).
> To our knowledge, all scaling-law theory results using TP/DMFT are currently limited to linear networks [BAP24], where assumptions such as Gaussian data are also standard. In contrast, our work provides a sharp and rigorous understanding of feature learning in a nonlinear setting and explicitly characterizes the statistical and computational complexity, which we believe constitutes a meaningful step forward in this direction.
>
> We hope these clarifications are helpful and appreciate your time in engaging with our work.

---

### Note · Authors · 2025-08-13

We sincerely thank all reviewers and the Area Chair for their engagement, constructive feedback, and thoughtful evaluations. We especially appreciate the recognition of our technical novelties and our contribution to understanding how smooth power-law risk curves can emerge in high-dimensional feature-learning settings, where prior analyses typically yield staircase-like behaviour.

During the rebuttal, we:
* Outlined the technical challenges in analyzing quadratic dynamics in the extensive-width regime, and why existing techniques cannot be directly applied.
* Explained why our analysis is a key step toward better understanding of SGD dynamics for general activations such as ReLU, since $\mathrm{He}_2$ is the first non-trivial component that reveals the target function’s full support; we also provided a sketch and empirical evidence for extending our findings to activations with non-zero $\mathrm{He}_2$.
* Clarified that orthogonal teacher directions can be assumed without loss of generality, and discussed limitations for general covariance and mismatched activations.

For the updated manuscript, we will:
* Add the rebuttal experiments and sketch to explain how our analysis can be extended to general activations with non-zero $\mathrm{He}_2$ component.
* Include label noise in our statements, which adds no complexity to our theoretical analysis.

We believe these additions will further strengthen the paper and advance understanding of scaling law behaviour in gradient-based feature learning.

---

### Decision · Program_Chairs · 2025-09-17

**Decision:**

Accept (poster)

**Comment:**

The submission focuses on the study of SGD for a two-layers neural networks with quadratic activations and orthonormal first-layer features. The authors obtain as a byproduct some neural scaling laws. The paper received two positive reviews and a mildly negative one. Reviewers acknowledge the non-trivial theoretical contribution that justify empirical scaling laws by a robust and elegant analysis. During the discussion period further numerical evidences in support of the results have been provided. The main objection to the acceptance of the paper was raised by Reviewer ```s1U3```, who, although recognizing the quality of the submission, stressed the highly specialistic nature of the contribution and the very idealized setup, that might have a marginal impact beyond the small community of experts working on such idealized set-ups. Although I recognize the validity of many objections of Reviewer ```s1U3``` on the distance between the idealized setup and practitioners, I must recognise that this type of works has in fact a tradition in the NeurIPS tradition and can be of interest to a subcommunity of theoreticians working on simple models in high dimension. In fact, the study of the dynamics in two layer neural networks in the extensive width regime is a current challenge tackled by the theory and the proposed contribution might be significant in this sense. After reading the manuscript I lean therefore towards its acceptance.